# Understanding the Benefits of SimCLR Pre-Training in Two-Layer Convolutional Neural Networks

## Abstract

SimCLR is one of the most popular contrastive learning methods for vision tasks. It pre-trains deep neural networks based on a large amount of unlabeled data by teaching the model to distinguish between positive and negative pairs of augmented images. It is believed that SimCLR can pre-train a deep neural network to learn efficient representations that can lead to a better performance of future supervised fine-tuning. Despite its effectiveness, our theoretical understanding of the underlying mechanisms of SimCLR is still limited. In this paper, we theoretically introduce a case study of the SimCLR method. Specifically, we consider training a two-layer convolutional neural network (CNN) to learn a toy image data model. We show that, under certain conditions on the number of labeled data, SimCLR pre-training combined with supervised fine-tuning achieves almost optimal test loss. Notably, the label complexity for SimCLR pre-training is far less demanding compared to direct training on supervised data. Our analysis sheds light on the benefits of SimCLR in learning with fewer labels.

## 1 Introduction

In recent years, self-supervised learning has emerged as a promising machine learning paradigm, offering a way to learn meaningful representations from vast amounts of unlabeled data. Self-supervised learning is of vital importance because the success of supervised learning is dependent on the accessibility of a large number of carefully labeled data, while the high-quality labeled data is expensive and time-consuming to obtain. Self-supervised learning leverages a large amount of unlabeled data to pre-train the representations for the following supervised fine-tuning learning task without requiring more labeled data.

Major categories of self-supervised learning methods include contrastive learning (Oord et al., 2018; Chen et al., 2020; He et al., 2020) and generative self-supervised learning (Kingma & Welling, 2013; Goodfellow et al., 2014). Among the various self-supervised learning methods, SimCLR (Chen et al., 2020) algorithm has gained significant attention due to its simplicity and remarkable performance for vision tasks. SimCLR leverages the idea of contrastive learning, where representations are learned by maximizing agreement between differently augmented views of the same image while minimizing agreement between views of different images. Compared with purely supervised learning, this approach has demonstrated exceptional capabilities in capturing high-level semantic information and achieving state-of-the-art results on various downstream tasks.

While such a contrastive learning method has demonstrated great success from the empirical perspective, it remains relatively unclear how the pre-training scheme helps improve the performance of the fine-tuning. Some recent papers have been devoted to the theoretical understanding of contrastive learning (Saunshi et al., 2019; Tsai et al., 2020; Wen & Li, 2021). Saunshi et al. (2019) introduced a theoretical framework that contains latent classes and presented the generalization bound to demonstrate provable good performance and reduced sample complexity of downstream tasks, but this framework fails to explain the case of over-parameterization. Tsai et al. (2020) provided an information-theoretical framework based on mutual information to explain the good performance of self-supervised learning. However, the aforementioned papers focus on the setting where the hypothesis class has limited complexity, and cannot handle the setting where the number of model

parameters is larger than the sample size, which is more common in modern deep learning, especially for vision tasks. Wen & Li (2021) considered the over-parameterized setting and analyzed the feature learning process of contrastive learning and the dependence of learned features on data augmentations. However, Wen & Li (2021) only analyzed a very specific type of learning task solved by a slightly non-standard optimization algorithm. Therefore, current theoretical understanding of contrastive learning is still quite limited.

In this paper, how SimCLR pre-training method makes improvements in the fine-tuning training of a two-layer convolutional neural network (CNN) is studied. The case we focus on is a binary classification problem on a toy image data model, which has been studied in a series of recent works (Cao et al., 2022; Jelassi & Li, 2022; Kou et al., 2023b). Under certain conditions related to the number of labeled and unlabeled data and the signal-to-noise ratio (SNR), we study SimCLR-based pre-training followed by supervised fine-tuning, and establish convergence as well as generalization guarantees of the obtained two-layer CNN.

The contributions of this paper are summarized as follows.

- We consider using CNNs given by SimCLR pre-training and supervised fine-tuning to learn a certain type of signal-noise data studied in recent works. Under certain conditions on the amount of unlabeled data and labeled data, we establish training loss convergence guarantees as well as generalization guarantees for two-layer CNNs trained by SimCLR pre-training and supervised fine-tuning. Specifically, our results demonstrate that, although the training losses in the pre-training and fine-tuning are both highly non-convex, the training of the CNN will successfully minimize the training loss. Moreover, although we consider an over-parameterized setting where the CNN overfits the training data, our results demonstrate that the CNN will achieve a small test loss.

- The learning task we investigate is a standard toy data model that has been studied by many recent works (Cao et al., 2022; Jelassi & Li, 2022; Kou et al., 2023b). This enables an easy comparison between the theoretical guarantees of learning with SimCLR pre-training and those without SimCLR pre-training. In particular, Cao et al. (2022) showed that, direct supervised learning on the data model can achieve small test loss if and (almost) only if the condition $n \cdot \text{SNR}^q = \widetilde{\Omega}(1)$ [1] holds, where SNR is a notion of the signal-to-noise ratio, $n$ is the labeled sample size, and $q$ is a constant related to the activation function. In comparison, the label complexity for SimCLR pre-training followed by supervised fine-tuning is far less demanding: our results show that when the unlabeled sample size $n_0$ and labeled sample size $n$ satisfy $n_0 \cdot \text{SNR}^2 = \widetilde{\Omega}(1)$, $n = \widetilde{\Omega}(1)$, the obtained CNN can achieve small training and test losses. Clearly, our analysis demonstrates the advantage of SimCLR in reducing label complexity in learning tasks with low signal-to-noise ratio. Our result serves as a concrete example where SimCLR-based pre-training is provably helpful.

- In our theoretical analysis, we introduce many novel analysis tools that enable the study of the SimCLR algorithm. In particular, we establish a key result that, up to sufficiently many iterations, the SimCLR pre-training updates can be characterized by the power method based on a matrix defined by the pre-training data and their augmentations. Notably, although our analysis focuses on a very specific toy data model, we believe similar results on the connection between SimCLR and power method should hold for more general settings. Therefore, this result may be of independent interest. Moreover, all of our analysis of the SimCLR algorithm should also hold for the case where the data inputs are generated from Gaussian mixtures. Therefore, a side product of our analysis is the effectiveness guarantee of using SimCLR to learn Gaussian mixtures.

**Notation.** $\| \cdot \|_2$ denotes the $\ell^2$-norm. $\| \cdot \|_F$ denotes the Frobenius norm. $[n]$ refers to the set $\{1, 2, \ldots, n\}$. For two sequences $\{a_n\}$ and $\{b_n\}$, denote $a_n = O(b_n)$ if there exists some absolute constant $C > 0$ and $N > 0$ such that $|a_n| \leq C|b_n|$ holds for all $n \geq N$. Denote $a_n = \Omega(b_n)$ if there exist some absolute constant $C > 0$ and $N > 0$ such that $|a_n| \geq C|b_n|$ holds for all $n \geq N$. Denote $a_n = \Theta(b_n)$ if both $a_n = O(b_n)$ and $a_n = \Omega(b_n)$ hold. $\widetilde{O}(\cdot)$, $\widetilde{\Omega}(\cdot)$, $\widetilde{\Theta}(\cdot)$ are used to omit the logarithmic factors in these notations.

---

[1] Here the $\widetilde{\Omega}(\cdot)$ hides logarithmic factors.

## 2 RELATED WORK

**Self-supervised Learning.** Self-supervised learning has won great success in application and has covered many important fields of machine learning, for example, natural language processing (Mikolov et al., 2013; Devlin et al., 2018), and computer vision (Chen et al., 2020). As one of the important methods for vision tasks, contrastive learning began with learning the latent variable of the data (Carreira-Perpinan & Hinton, 2005). Cole et al. (2022) investigated the factors that improve the performance of contrastive learning, but the analysis was still from an empirical aspect. From a theoretical perspective, some works have also been done to understand contrastive learning. Wang & Isola (2020) identified alignment and uniformity as two important properties concerned with contrastive loss, and proved that contrastive loss optimizes these properties asymptotically. Information theory was also introduced to establish theoretical framework that explains why contrastive learning works (Tsai et al., 2020; Tian et al., 2020a). Shwartz Ziv & LeCun (2024) examined different self-supervised learning methods from the information-theoretic aspect and proposed a unified framework that includes them as information-theoretic learning problems. Tian et al. (2020b) proposed a framework for the theoretical understanding of SimCLR self-supervised learning method and demonstrated that the updates of SimCLR capture variations across data points. HaoChen et al. (2021) considered a spectral contrastive loss and performed spectral clustering on the population augmentation graph, but the applicability of this spectral contrastive loss is limited. Tan et al. (2024) extended the idea of HaoChen et al. (2021) to general loss functions by showing the equivalence between InfoNCE loss and spectral clustering. Furthermore, HaoChen et al. (2021) also extended this to more general settings, including multi-modal scenarios. However, the aforementioned papers focused on the analysis of contrastive learning and do not analyze how contrastive learning influences the performance of the following fine-tuning stage. Bansal et al. (2021) presented a new upper bound of the generalization gap of classifiers by performing self-supervised training to learn representations, followed by fitting a simple classifier such as linear classifier to the labels.

**Feature Learning Theory of Neural Networks.** There are a series of works that provide theoretical foundations for feature learning theory of neural networks. Frei et al. (2022) considered the benign overfitting phenomenon in two-layer neural networks with smoothed leaky ReLU activations when both model and learning dynamics are nonlinear. Cao et al. (2022) analyzed the benign overfitting that appeared in the supervised learning of two-layer convolutional neural networks, and showed arbitrary small training and test loss can be achieved under certain conditions on SNR, but the analysis was based on specified initialization distribution. Without requiring the smoothness of the activation function, Kou et al. (2023b) focused on benign overfitting of two-layer ReLU convolutional neural networks with label-flipping noise. They showed that, under mild conditions, the neural networks can achieve near-zero training loss and Bayes optimal test risk. Xu et al. (2023) demonstrated that benign overfitting and grokking provably appeared in the feature learning of two-layer ReLU neural networks trained by gradient descent on non-linearly separable data distribution. Meng et al. (2024) analyzed one category of XOR-type classification tasks with label-flipping noises, showing two-layer ReLU convolutional neural networks can achieve near Bayes-optimal accuracy. Kou et al. (2023a) investigated a semi-supervised learning method that combines pre-training with linear probing for two-layer neural networks, and found the semi-supervised approach achieves nearly zero test loss. However, how self-supervised learning improves the training of neural networks remains largely unexplored.

## 3 PROBLEM SETTING

This section presents the problem setup in this paper. We first introduce the data model considered in this paper, and then introduce the detailed setup for SimCLR pre-training and supervised fine-tuning respectively.

### 3.1 A DATA MODEL FOR THE CASE STUDY

In this paper, we consider a simple binary classification task. We consider a toy data model that has been studied in a series of recent works (Cao et al., 2022; Jelassi & Li, 2022; Kou et al., 2023b). This paper is motivated by Cao et al. (2022), which analyzed the performance of direct supervised

learning. To enable a direct comparison between SimCLR pre-training followed by fine-tuning and direct supervised learning, this paper adopts the same toy data model as Cao et al. (2022).

**Definition 3.1.** *Let $\boldsymbol{\mu} \in \mathbb{R}^d$ be a fixed vector. Each data point $(\mathbf{x}, y)$ is given in the format of $\mathbf{x} = [\mathbf{x}^{(1)\top}, \mathbf{x}^{(2)\top}]^\top \in \mathbb{R}^{2d}$. Assume the data is generated from the following distribution $\mathcal{D}$:*

1. *The label $y$ is generated as a Rademacher random variable with $y \in \{-1, 1\}$.*

2. *A noise vector $\boldsymbol{\xi}$ is generated from the Gaussian distribution $\mathcal{N}(\mathbf{0}, \sigma_p^2 \cdot (\mathbf{I} - \boldsymbol{\mu}\boldsymbol{\mu}^\top \cdot \|\boldsymbol{\mu}\|_2^{-2}))$.*

3. *One of the two patches $\mathbf{x}^{(1)}, \mathbf{x}^{(2)}$ generated and is assigned as $\mathbf{x}^{(1)} = y \cdot \boldsymbol{\mu}$ which represents the signal patch, and the other patch is assigned as $\boldsymbol{\xi}$ which represents the noise patch.*

As is commented in Cao et al. (2022), the data distribution in Definition 3.1 is motivated by image data, where the data input consists of multiple patches, and only some of the patches are directly related to its corresponding label. Therefore, this data model is particularly suitable to study SimCLR, which is originally proposed for vision tasks. The data input consists of several patches, among which some are signal patches and the rest are noise patches. Following the notation given in Cao et al. (2022), we define the signal-to-noise ratio (SNR) as $\mathrm{SNR} = \|\boldsymbol{\mu}\|/(\sigma_p\sqrt{d})$ since $\|\boldsymbol{\xi}\|_2 \approx \sigma_p\sqrt{d}$ when dimension $d$ is large.

### 3.2 SELF-SUPERVISED PRE-TRAINING WITH SIMCLR

We consider using SimCLR (Chen et al., 2020) to pre-train a simple linear CNN on unlabeled data. The linear CNN $\mathbf{F}(\mathbf{W}, \mathbf{x})$ with output of dimension $2m$ is defined as follows:

$$[\mathbf{F}(\mathbf{W}, \mathbf{x})]_r = \langle \mathbf{w}_r, \mathbf{x}^{(1)} \rangle + \langle \mathbf{w}_r, \mathbf{x}^{(2)} \rangle, \quad r \in [2m],$$

where $\mathbf{W} = (\mathbf{w}_1, \cdots, \mathbf{w}_{2m})^\top \in \mathbb{R}^{2m \times d}$, $\mathbf{w}_r \in \mathbb{R}^{d \times 1}$, $r \in [2m]$. The linear CNN model defined above is composed of a linear CNN layer with $2m$ convolution filters $\mathrm{LinearConv}(\cdot) : \mathbb{R}^{2d} \rightarrow \mathbb{R}^{2m \times 2}$ and a fixed linear projection head $\mathrm{ProjHead}(\cdot) : \mathbb{R}^{2m \times 2} \rightarrow \mathbb{R}^{2m}$ defined as follows:

$$[\mathrm{LinearConv}(\mathbf{W}, \mathbf{x})]_{r,p} := \langle \mathbf{w}_r, \mathbf{x}^{(p)} \rangle, \ r \in [2m], \ p \in [2], \ \text{ and } \ \mathrm{ProjHead}(\mathbf{Z}) := \mathbf{Z}[1\ 1]^\top.$$

Then it is clear that

$$\mathbf{F}(\mathbf{W}, \mathbf{x}) = \mathrm{ProjHead}[\mathrm{LinearConv}(\mathbf{W}, \mathbf{x})]. \tag{3.1}$$

Suppose that we are given an unlabeled dataset $S_{\mathrm{unlabeled}} = \{\mathbf{x}_1^{\mathrm{pre\text{-}training}}, \ldots, \mathbf{x}_{n_0}^{\mathrm{pre\text{-}training}}\}$, where $\mathbf{x}_i^{\mathrm{pre\text{-}training}}$, $i \in [n_0]$ are unlabeled data independently generated from distribution $\mathcal{D}$ in Definition 3.1. In SimCLR, we train the linear CNN model $\mathbf{F}(\mathbf{W}, \mathbf{x})$ as follows: For each data point $\mathbf{x}_i^{\mathrm{pre\text{-}training}}$, $i \in [n_0]$, we apply data augmentation to obtain an augmented data point $\widetilde{\mathbf{x}}_i^{\mathrm{pre\text{-}training}}$. We consider an ideal setting that $\widetilde{\mathbf{x}}_i^{\mathrm{pre\text{-}training}}$ is generated from $\mathbb{P}(\mathbf{x}|y = y_i)$. Then following Definition 3.1, it holds that $\widetilde{\mathbf{x}}_i^{\mathrm{pre\text{-}training}} = [\widetilde{\mathbf{x}}_i^{(1)\top}, \widetilde{\mathbf{x}}_i^{(2)\top}]^\top$, where one of $\widetilde{\mathbf{x}}_i^{(1)}, \widetilde{\mathbf{x}}_i^{(2)}$ is randomly assigned as $y_i \cdot \boldsymbol{\mu}$ while the other is assigned as $\widetilde{\boldsymbol{\xi}}_i \sim \mathcal{N}(\mathbf{0}, \sigma_p^2 \cdot (\mathbf{I} - \boldsymbol{\mu}\boldsymbol{\mu}^\top \cdot \|\boldsymbol{\mu}\|_2^{-2}))$. Based on $\mathbf{x}_i^{\mathrm{pre\text{-}training}}$ and $\widetilde{\mathbf{x}}_i^{\mathrm{pre\text{-}training}}$, $i \in [n_0]$, we define the following similarity scores

$$\mathrm{sim}_i = \left\langle \mathbf{F}(\mathbf{W}, \mathbf{x}_i^{\mathrm{pre\text{-}training}}), \mathbf{F}(\mathbf{W}, \widetilde{\mathbf{x}}_i^{\mathrm{pre\text{-}training}}) \right\rangle, \quad \mathrm{sim}_{i,i'} = \left\langle \mathbf{F}(\mathbf{W}, \mathbf{x}_i^{\mathrm{pre\text{-}training}}), \mathbf{F}(\mathbf{W}, \mathbf{x}_{i'}^{\mathrm{pre\text{-}training}}) \right\rangle,$$

for all $i, i' \in [n_0]$ with $i \neq i'$.

The convolution filters $\mathbf{w}_r \in \mathbb{R}^d$, $r \in [2m]$ in the SimCLR pre-training are initialized following Gaussian distribution, namely $\mathbf{w}_r^{(0)} \sim \mathcal{N}(\mathbf{0}, \sigma_0^2\mathbf{I})$, $r \in [2m]$. The loss function of the pre-training stage is defined as

$$L_{S_{\mathrm{unlabeled}}}(\mathbf{W}) = -\frac{1}{n_0} \sum_{i=1}^{n_0} \log\left(\frac{\exp(\mathrm{sim}_i/\tau)}{\exp(\mathrm{sim}_i/\tau) + \sum_{i' \neq i} \exp(\mathrm{sim}_{i,i'}/\tau)}\right),$$

where $\tau$ is a constant. $\tau$ is the temperature parameter in SimCLR (Chen et al., 2020). In the pre-training stage, gradient descent with learning rate $\eta$ is used to minimize the loss function $L(\mathbf{W})$.

## 3.3 SUPERVISED FINE-TUNING

Suppose that the labeled training dataset is given as $S = \{(\mathbf{x}_1^{\text{fine-tuning}}, y_1), \ldots, (\mathbf{x}_n^{\text{fine-tuning}}, y_n)\}$, where $n$ is the number of labeled data, and each data point $(\mathbf{x}_i^{\text{fine-tuning}}, y_i)$, $i \in [n]$ is generated from the distribution $\mathcal{D}$ in Definition 3.1. The two-layer convolutional neural network model is considered in the fine-tuning stage, namely $f(\mathbf{W}, \mathbf{x}) = F_{+1}(\mathbf{W}_{+1}, \mathbf{x}) - F_{-1}(\mathbf{W}_{-1}, \mathbf{x})$, where

$$F_j(\mathbf{W}_j, \mathbf{x}) = \frac{1}{m} \sum_{r=1}^{m} \left[ \sigma(\langle \mathbf{w}_{j,r}, \mathbf{x}^{(1)} \rangle) + \sigma(\langle \mathbf{w}_{j,r}, \mathbf{x}^{(2)} \rangle) \right], \quad j = \pm 1, \tag{3.2}$$

where $m$ is the number of filters in $F_{+1}$ and $F_{-1}$ respectively, $\sigma(z) = (\max\{0, z\})^q$ is the ReLU$^q$ activation function with $q > 2$.

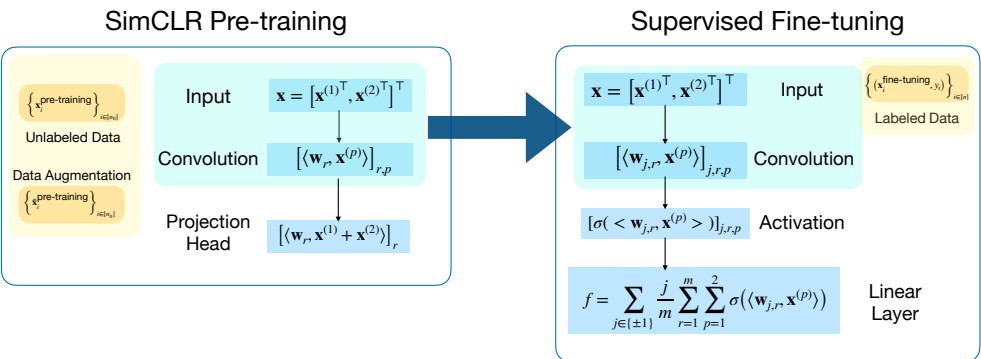

Figure 1: Illustration of the SimCLR pre-training and supervised fine-tuning stages.

In this paper, we consider the initialization $\mathbf{W}^{(0)}$ of the CNN (3.2) given by the result of SimCLR pre-training. For the $2m$ filters $\mathbf{w}_r^{(T_{\text{SimCLR}})}$, $r \in [2m]$ obtained in the pre-training stage, we randomly sample $m$ filters out of $2m$ filters and assign them to the initialization of filters in $F_{+1}$, and denote $\mathcal{M} \subseteq [2m]$ the collection of these filters with $|\mathcal{M}| = m$. Correspondingly, the rest $m$ filters $\mathbf{w}_r^{(T_{\text{SimCLR}})}$, $r \in [2m] \cap \mathcal{M}^c$ is assigned to the initialization of filters in $F_{-1}$. Therefore, the initialization of the supervised fine-tuning is given as

$$\{\mathbf{w}_{1,r}^{(0)}, r \in [m]\} = \{\mathbf{w}_r^{(T_{\text{SimCLR}})}, r \in \mathcal{M}\}, \quad \{\mathbf{w}_{-1,r}^{(0)}, r \in [m]\} = \{\mathbf{w}_r^{(T_{\text{SimCLR}})}, r \in \mathcal{M}^c\}.$$

Clearly, the above procedure is equivalent to the practical implementations of SimCLR, where after pre-training, we essentially remove the projection head part of the model and attach another classifier to perform supervised fine-tuning.

The training of this convolutional neural network is conducted by minimizing the empirical cross-entry loss function, namely

$$L_S(\boldsymbol{W}) = \frac{1}{n} \sum_{i=1}^{n} \ell[y_i \cdot f(\mathbf{W}, \mathbf{x}_i^{\text{fine-tuning}})],$$

where $S$ denotes the training dataset in the fine-tuning stage given by Definition 3.1, and $\ell(z) = \log(1 + \exp(-z))$. Based on Definition 3.1, the corresponding true loss is defined as $L_{\mathcal{D}}(\boldsymbol{W}) := \mathbb{E}_{(\mathbf{x}, y) \sim \mathcal{D}} \ell[y \cdot f(\mathbf{W}, \mathbf{x})]$.

In the fine-tuning stage, based on the gradient descent algorithm and the CNN structure defined in (3.2), the filters of the CNN $\mathbf{w}_{j,r}$, $j \in \{-1, +1\}, r \in [m]$ is trained according to the following gradient descent updating rules

$$\mathbf{w}_{j,r}^{(t+1)} = \mathbf{w}_{j,r}^{(t)} - \eta \cdot \nabla_{\mathbf{w}_{j,r}} L_S(\boldsymbol{W}^{(t)}). \tag{3.3}$$

The whole two-stage training procedure is depicted in Figure 1.

# 4 MAIN RESULT

In this section, we present the main learning guarantee of the two-layer CNN given by SimCLR pre-training and supervised fine-tuning. We first introduce several assumptions on the number of unlabeled training samples $n_0$, the number of labeled data samples $n$, the dimension $d$, the number of convolutional filters $m$, Gaussian initialization scale $\sigma_0$, and the learning rate $\eta$.

**Condition 4.1.** *Suppose that the following conditions hold for the pre-training and fine-tuning stage,*

*1. The number of unlabeled training samples $n_0$ satisfies $n_0 = \widetilde{\Omega}(\max\{\mathrm{SNR}^{-2}, 1\})$.*

*2. The number of labeled training samples $n$ satisfies $n = \widetilde{\Omega}(1)$.*

*3. The dimension $d$ is sufficiently large: $d \geq \widetilde{\Omega}(n^{\frac{-6}{q-2}}\mathrm{SNR}^{-\frac{6q}{q-2}} \cdot \max\{n_0^{-1}, \mathrm{SNR}^{-2}\} + n_0^4)$.*

*4. The number of convolutional filters $m$ satisfies $m = \Omega(\log(1/\delta))$.*

*5. Gaussian initialization scale $\sigma_0$ for SimCLR pre-training is sufficiently small:*

$$\sigma_0 \leq \widetilde{O}\big(\min\{1, d^{-1}n^{\frac{4}{q-2}}\mathrm{SNR}^{\frac{4q}{q-2}} \cdot \|\boldsymbol{\mu}\|_2^{-2}\} \cdot \min\{1, \mathrm{SNR}^{-1}, \mathrm{SNR}^{-2}\}\big).$$

*6. The learning rate satisfies that $\eta = \widetilde{O}\big(\min\big\{(\sigma_p^2 d)^{-1}, (\sigma_p\sqrt{d})^{-2}, \|\boldsymbol{\mu}\|_2^{-2}\big\}\big)$.*

While Condition 4.1 gives a long list of conditions, we remark that most of these assumptions are easy to satisfy. In fact, the first two conditions in Condition 4.1 are the key conditions in this paper. The condition on $d$ is essentially assuming that the learning happens in a sufficiently over-parameterized setting, which is common in a series of recent works (Chatterji & Long, 2020; Cao et al., 2021; 2022; Jelassi & Li, 2022; Kou et al., 2023b). The condition on the number of convolutional filters $m$ is mild as we only require $m = \widetilde{\Omega}(1)$. Finally, the conditions on the initialization scale $\sigma_0$ and the learning rate $\eta$ essentially just assumes that the optimization is appropriately set up, and can be achieved by simply implementing a small enough initialization scale and a small enough learning rate.

The following Theorem 4.2 summarizes the main result of this paper.

**Theorem 4.2.** *Under Condition 4.1, for any $\epsilon > 0$, if $n_0 \cdot \mathrm{SNR}^2 = \widetilde{\Omega}(1)$, then within $T_{\mathrm{SimCLR}} = \widetilde{\Omega}(\eta^{-1}\tau\|\boldsymbol{\mu}\|_2^{-2})$ iterations of pre-training and $T = \widetilde{\Theta}(\eta^{-1}m\sigma_0^{-(q-2)}\|\boldsymbol{\mu}\|_2^{-2} + \eta^{-1}\epsilon^{-1}m^3\|\boldsymbol{\mu}\|_2^{-2})$ iterations of fine-tuning, the obtained network in the fine-tuning stage satisfies that: there exists some $0 \leq t \leq T$, such that*

*1. The training loss converges to $\epsilon$, i.e., $L_S(\mathbf{W}^{(t)}) \leq \epsilon$.*

*2. The trained CNN achieves a small test loss: $L_{\mathcal{D}}(\mathbf{W}^{(t)}) \leq 6\epsilon + \exp(-\widetilde{\Omega}(n^2))$.*

Theorem 4.2 presents the convergence and generalization guarantees of the two-layer CNN trained by SimCLR pre-training with supervised fine-tuning. According to Theorem 4.2, training loss convergence and small generalization are guaranteed for the two-layer CNNs when $n_0 = \widetilde{\Omega}(\max\{\mathrm{SNR}^{-2}, 1\}$ and $n \geq \Omega(\log(1/\delta))$ together with some other conditions listed in Condition 4.1. Here we comment that the case where $\mathrm{SNR} = \Omega(1)$ is a very easy setting, as according to Cao et al. (2022, Theorem 4.3), small test loss can be achieved even if there is only $\widetilde{\Omega}(1)$ training data. Therefore, we see that Condition 4.1 essentially requires that $n_0 \cdot \mathrm{SNR}^2 = \widetilde{\Omega}(1)$ and $n = \widetilde{\Omega}(1)$.

To compare the results of SimCLR pre-training followed by supervised fine-tuning with direct supervised learning, we cite the following theoretical results for direct supervised learning from Cao et al. (2022).

**Theorem 4.3** (Theorems 4.3 and 4.4 in Cao et al. (2022), bounds of direct supervised learning)**.** *For any $\epsilon > 0$, let $T = \widetilde{\Theta}(\eta^{-1}m \cdot n\sigma_0^{-(q-2)} \cdot \max\{(\sigma_p\sqrt{d})^{-q}, \|\boldsymbol{\mu}\|_2^{-q}\} + \eta^{-1}\epsilon^{-1}nm^3 \cdot \max\{(\sigma_p\sqrt{d})^{-2}, \|\boldsymbol{\mu}\|_2^{-2}\})$. Under Condition 4.2 in Cao et al. (2022), the following results hold:*

*1. If $n \cdot \mathrm{SNR}^q = \widetilde{\Omega}(1)$, then with probability at least $1 - d^{-1}$, there exists $0 \leq t \leq T$ such that:*

(a) *The training loss converges to $\epsilon$, i.e., $L_S(\mathbf{w}^{(t)}) \leq \epsilon$.*

(b) *The trained CNN achieves a small test loss: $L_{\mathcal{D}}(\mathbf{w}^{(t)}) = 6\epsilon + \exp(-n^2)$.*

2. *If $n^{-1} \cdot \mathrm{SNR}^{-q} = \widetilde{\Omega}(1)$, then with probability at least $1 - d^{-1}$, there exists $0 \leq t \leq T$ such that:*

(a) *The training loss converges to $\epsilon$, i.e., $L_S(\mathbf{w}^{(t)}) \leq \epsilon$.*

(b) *The trained CNN has a constant order test loss: $L_{\mathcal{D}}(\mathbf{w}^{(t)}) = \Theta(1)$.*

Theorem 4.3 above gives an upper bound on the test loss under the condition $n \cdot \mathrm{SNR}^q = \widetilde{\Omega}(1)$, while also gives a lower bound on the test loss under the almost complementary condition $n^{-1} \cdot \mathrm{SNR}^{-q} = \widetilde{\Omega}(1)$. Note that we have the condition that the supervised sample size $n = \widetilde{\Omega}(1)$ in Cao et al. (2022, Condition 4.2). Therefore, Theorem 4.3 demonstrates that for direct supervised learning to achieve small test loss, it is necessary to have a labeled sample size at least $n = \widetilde{\Omega}(\max\{\mathrm{SNR}^{-q}, 1\})$. This result also indicates that the learning task is relatively challenging when $\mathrm{SNR} \ll 1$, as smaller SNR requires more labeled training data. Notably, Theorem 4.3 also shows that when $n^{-1} \cdot \mathrm{SNR}^{-q} = \widetilde{\Omega}(1)$, direct supervised learning with $n$ labeled data is guaranteed to result in constant level test loss.

In comparison, Theorem 4.2 demonstrates that for SimCLR pre-training combined with supervised fine-tuning, as long as the unlabeled sample size $n_0$ is sufficiently large ($n_0 = \widetilde{\Omega}(\max\{\mathrm{SNR}^{-2}, 1\})$), $n = \widetilde{\Omega}(1)$ labeled data suffice to lead to small test loss. As previously mentioned, direct supervised learning requires that the number of labeled training data satisfies $n = \widetilde{\Omega}(\max\{\mathrm{SNR}^{-q}, 1\})$. Comparing these results, we can conclude that direct supervised learning requires more label complexity to achieve small test loss, especially in challenging tasks with low signal-to-noise ratio. The clear difference between Theorem 4.2 and Theorem 4.3 demonstrates the effectiveness of SimCLR pre-training.

**Remark 4.4.** *To demonstrate the practical value of the theoretical results in this paper, experiments on both synthetic and real-world datasets are provided in Appendix A. Our theoretical results on the advantage of SimCLR pre-training and the results of direct supervised learning in Cao et al. (2022) together indicate that when the signal-to-noise ratio is low, SimCLR pre-training followed by supervised fine-tuning may require far less labeled data compared with direct supervised learning. The experiments in this paper present typical cases where SimCLR pre-training followed by supervised fine-tuning achieves a significantly smaller test loss, while direct supervised learning achieves a larger test loss under same label complexity.*

## 5 PROOF SKETCH

In this section, we discuss the key proof steps of Theorem 4.2. Our analysis heavily focuses on the pre-training of the linear CNN with SimCLR. Therefore, for the simplicity of the notation, we omit the superscripts of the data used for pre-training: we denote by $y_1, \ldots, y_{n_0}$ the (unseen) labels of the pre-training data, by $\boldsymbol{\xi}_1, \ldots, \boldsymbol{\xi}_{n_0}$ the noise patches in the data inputs, and denote by $\widetilde{\boldsymbol{\xi}}_1, \ldots, \widetilde{\boldsymbol{\xi}}_{n_0}$ the noise patches in the augmented data inputs. On the other hand, the noise patches in the labeled data inputs are denoted as $\boldsymbol{\xi}_1^{\text{fine-tuning}}, \ldots, \boldsymbol{\xi}_n^{\text{fine-tuning}}$. We also introduce the following notations:

$$\mathbf{z}_i = y_i \cdot \boldsymbol{\mu} + \boldsymbol{\xi}_i, \quad \widetilde{\mathbf{z}}_i = y_i \cdot \boldsymbol{\mu} + \widetilde{\boldsymbol{\xi}}_i, \quad i \in [n_0].$$

The above notation is motivated by the observation that the linear CNN we consider in the pre-training stage is essentially a function of the summation of the two patches of the data input. Further notice that $\mathbf{z}_i, \widetilde{\mathbf{z}}_i, i \in [n_0]$ defined above are essentially Gaussian mixture data, and hence our proof is essentially based on an analysis of the performance of SimCLR in learning Gaussian mixtures.

**A characterization of SimCLR pre-training by power method.** Our proof for SimCLR is based on a key observation that the SimCLR updates of each CNN filter is very similar to those of a power method based on a matrix defined by the data. Specifically, we have the following lemma.

**Lemma 5.1.** *For any $M > 0$, suppose that $n_0 = \widetilde{\Omega}(\max\{\mathrm{SNR}^{-2}, 1\})$, and*

$$\sigma_0 \leq \widetilde{O}\big(\min\{1, d^{-1} M^{-\frac{4}{q-2}} n^{\frac{4}{q-2}} \mathrm{SNR}^{\frac{4q}{q-2}} \cdot \|\boldsymbol{\mu}\|_2^{-2}\} \cdot \min\{1, \mathrm{SNR}^{-1}, \mathrm{SNR}^{-2}\}\big).$$

*Let $\mathbf{A} = \frac{\eta}{n_0^2 \tau} \sum_{i=1}^{n_0} \sum_{i' \neq i} (\mathbf{z}_i \widetilde{\mathbf{z}}_i^\top + \widetilde{\mathbf{z}}_i \mathbf{z}_i^\top - \mathbf{z}_i \mathbf{z}_{i'}^\top - \mathbf{z}_{i'} \mathbf{z}_i^\top)$. Then for any $T_{\mathrm{SimCLR}}$ satisfying*

$$[1 + (1 - \mathcal{E}_{\mathrm{SimCLR}}) \|\mathbf{A}\|_2]^{T_{\mathrm{SimCLR}}} = \widetilde{O}\big( \max\{M^{\frac{1}{q-2}} n^{-\frac{1}{q-2}} \mathrm{SNR}^{-\frac{q}{q-2}}\}\big),$$

*with $\mathcal{E}_{\mathrm{SimCLR}} = \widetilde{O}(\max\{\mathrm{SNR}^{-1} n_0^{-1/2}, n_0^{-1}\})$ as specified in Lemma 5.2, we have for $t = 0, 1, \ldots T_{\mathrm{SimCLR}}$, the iterates of SimCLR satisfy*

$$\mathbf{w}_r^{(t+1)} = \mathbf{w}_r^{(t)} + (\mathbf{A} + \mathbf{\Xi}^{(t)}) \mathbf{w}_r^{(t)},$$

*where $\mathbf{\Xi}^{(0)}, \ldots, \mathbf{\Xi}^{(T_{\mathrm{SimCLR}})} \in \mathbb{R}^{d \times d}$ are matrices whose columns and rows are in the subspaces $\mathrm{span}\{\boldsymbol{\mu}, \boldsymbol{\xi}_1, \ldots, \boldsymbol{\xi}_{n_0}, \widetilde{\boldsymbol{\xi}}_1, \ldots, \widetilde{\boldsymbol{\xi}}_{n_0}\}$ and $\mathrm{span}\{\boldsymbol{\mu}^\top, \boldsymbol{\xi}_1^\top, \ldots, \boldsymbol{\xi}_{n_0}^\top, \widetilde{\boldsymbol{\xi}}_1^\top, \ldots, \widetilde{\boldsymbol{\xi}}_{n_0}^\top\}$ respectively, and*

$$\|\mathbf{\Xi}^{(t)}\|_2 \leq \sigma_0 \cdot \|\mathbf{A}\|_2,$$

*for all $t = 0, \ldots, T_{\mathrm{SimCLR}}$, where*

$$\mathbf{\Xi}^{(t)} = -\frac{\eta}{n_0^2 \tau} \sum_{i=1}^{n_0} \sum_{i' \neq i} \left( \frac{n_0 \cdot \exp(\mathrm{sim}_{i,i'}^{(t)}/\tau)}{\exp(\mathrm{sim}_i^{(t)}/\tau) + \sum_{i'' \neq i} \exp(\mathrm{sim}_{i,i'}^{(t)}/\tau)} - 1 \right) (\mathbf{z}_i \mathbf{z}_{i'}^\top + \mathbf{z}_{i'} \mathbf{z}_i^\top - \mathbf{z}_i \widetilde{\mathbf{z}}_i^\top - \widetilde{\mathbf{z}}_i \mathbf{z}_i^\top).$$

Lemma 5.1 gives an accurate characterization on how the CNN filters are updated during the Sim-CLR pre-training. In particular, when the initialization scale $\sigma_0$ is small, Lemma 5.1 implies that each convolutional filter is approximately updated according to the formula $\mathbf{w}_r^{(t+1)} = (\mathbf{I} + \mathbf{A}) \mathbf{w}_r^{(t)}$, which is essentially a power method in learning the leading eigenvector of the matrix $(\mathbf{I} + \mathbf{A})$, which is also the leading eigenvector of the matrix $\mathbf{A}$.

**Spectral analysis of the matrix A.** According to Lemma 5.1, SimCLR may approximately align the CNN filters along the leading direction of the matrix $\mathbf{A}$, and the convergence rate depends on the eigenvalue gap between the largest eigenvalue and the second largest eigenvalue. Motivated by this, we give the following lemma on the spectral decomposition of $\mathbf{A}$.

**Lemma 5.2.** *Let $\mathbf{A}$ be the matrix defined in Lemma 5.1, and let $\lambda_i$, $\mathbf{v}_i$, $i \in [d]$ be the eigenvalues and eigenvectors of $\mathbf{A}$ respectively, where $\lambda_i$, $i = 1, \ldots, d$ are in decreasing order. Suppose that $d \geq n_0$, $n_0 \cdot \mathrm{SNR}^2 = \widetilde{\Omega}(1)$, and Condition 4.1 hold. Then there exists*

$$\mathcal{E}_{\mathrm{SimCLR}} = \widetilde{O}(\max\{\mathrm{SNR}^{-1} n_0^{-1/2}, n_0^{-1}\}) = o(1),$$

*such that the following results hold:*

- *The first eigenvalue of $\mathbf{A}$ is significantly larger than the rest:*

$$(1 - \mathcal{E}_{\mathrm{SimCLR}}) \cdot \frac{2\eta}{\tau} \|\boldsymbol{\mu}\|_2^2 \leq \lambda_1 \leq (1 + \mathcal{E}_{\mathrm{SimCLR}}) \cdot \frac{2\eta}{\tau} \|\boldsymbol{\mu}\|_2^2, \quad \max_{i \geq 2} \lambda_i \leq \frac{\eta}{\tau} \cdot \|\boldsymbol{\mu}\|_2^2 \mathcal{E}_{\mathrm{SimCLR}},$$

- *The leading eigenvector of $\mathbf{A}$ aligns well with $\boldsymbol{\mu}$: Denote by $\mathbf{P}_{\boldsymbol{\mu}}^\perp = \mathbf{I} - \boldsymbol{\mu}\boldsymbol{\mu}^\top / \|\boldsymbol{\mu}\|_2^2$ the projection matrix onto $\mathrm{span}\{\boldsymbol{\mu}\}^\perp$. Then it holds that*

$$|\langle \mathbf{v}_1, \boldsymbol{\mu} \rangle| \geq (1 - \mathcal{E}_{\mathrm{SimCLR}}^2) \cdot \|\boldsymbol{\mu}\|_2, \qquad \|\mathbf{P}_{\boldsymbol{\mu}}^\perp \mathbf{v}_1\|_2 \leq \mathcal{E}_{\mathrm{SimCLR}}.$$

Lemma 5.2 gives a tight estimation on the eigenvalues and eigenvectors of the matrix $\mathbf{A}$, with a focus on the gap between the leading eigenvalue and the rest, and the relation between the leading eigenvector and the signal vector $\boldsymbol{\mu}$. According to Lemma 5.2, we can see that if the unlabeled sample size $n_0$ is sufficiently large, the leading eigenvalue and leading eigenvector will both be controlled by the signal $\boldsymbol{\mu}$, indicating that SimCLR can help enhance signal learning.

**Signal learning guarantee of SimCLR pre-training.** Based on Lemmas 5.1 and 5.2, we can derive the following key theorem on the signal learning guarantee of SimCLR pre-training.

**Theorem 5.3.** *Let $\mathbf{A}$ be defined in Lemma 5.1. For any $M > 0$, suppose that $n_0 = \widetilde{\Omega}(\max\{\mathrm{SNR}^{-2}, 1\})$, $\sigma_0 \leq \widetilde{O}(\min\{n_0^{-3}, n_0^{-3} \mathrm{SNR}^{-2}, M^{-\frac{1}{q-2}} \cdot n^{\frac{1}{q-2}} \mathrm{SNR}^{\frac{q}{q-2}}\})$, $d \geq \widetilde{\Omega}(M^{\frac{6}{q-2}} \cdot n^{\frac{-6}{q-2}} \mathrm{SNR}^{-\frac{6q}{q-2}} \cdot \max\{n_0^{-1}, \mathrm{SNR}^{-2}\})$. Then with*

$$T_{\mathrm{SimCLR}} = \left\lceil \frac{\log[288 M^{\frac{1}{q-2}} \cdot \log(1/\sigma_0)^{\frac{1}{q-2}} \cdot \sqrt{\log(dn) \cdot \log(md)}] - \log[n^{\frac{1}{q-2}} \mathrm{SNR}^{\frac{q}{q-2}}]}{\log[1 + (1 - \mathcal{E}_{\mathrm{SimCLR}}) \cdot \|\mathbf{A}\|_2]} \right\rceil$$

*iterations, SimCLR gives CNN weights* $\mathbf{w}_r^{(T_{\mathrm{SimCLR}})}$, $r \in [2m]$ *that can be decomposed as*

$$\mathbf{w}_r^{(T_{\mathrm{SimCLR}})} = \mathbf{w}_r^{\perp} + \gamma_r \cdot \boldsymbol{\mu}/\|\boldsymbol{\mu}\|_2^2 + \sum_{i=1}^{n} \rho_{r,i} \cdot \boldsymbol{\xi}_i^{\mathrm{fine-tuning}}/\|\boldsymbol{\xi}_i^{\mathrm{fine-tuning}}\|_2^2,$$

*where* $\mathbf{w}_r^{\perp}$ *is perpendicular to* $\boldsymbol{\mu}$ *and* $\boldsymbol{\xi}_1^{\mathrm{fine-tuning}}, \ldots, \boldsymbol{\xi}_n^{\mathrm{fine-tuning}}$. *Moreover, it holds that*

- *There exist disjoint index sets* $\mathcal{I}^+, \mathcal{I}^- \subseteq [2m]$ *with* $|\mathcal{I}^+| = |\mathcal{I}^-| = 2m/5$ *such that*

$$\frac{\min_{r \in \mathcal{I}^+} \gamma_r^{q-2}/\log(2/\gamma_r)}{\max_{r \in [2m], i \in [n]} |\rho_{r,i}|^{q-2}} \geq \frac{M}{n\mathrm{SNR}^2}, \qquad \frac{\min_{r \in \mathcal{I}^-} (-\gamma_r)^{q-2}/\log(-2/\gamma_r)}{\max_{r \in [2m], i \in [n]} |\rho_{r,i}|^{q-2}} \geq \frac{M}{n\mathrm{SNR}^2}.$$

- *All* $\mathbf{w}_r^{(T_{\mathrm{SimCLR}})}$, $r \in [2m]$ *are bounded:*

$$\max_{r \in [2m]} \|\mathbf{w}_r^{\perp}\|_2 \leq \frac{1}{n}, \quad \max_{r \in [2m]} |\gamma_r| \leq \frac{\mathrm{SNR}^{2/q-2}}{16m^{2/q-2}n_0}, \quad \max_{r \in [2m], i \in [n]} |\rho_{r,i}| \leq \frac{\mathrm{SNR}^{2/q-2}}{16m^{2/q-2}n_0}.$$

In Theorem 5.3, the decomposition

$$\mathbf{w}_r^{(T_{\mathrm{SimCLR}})} = \mathbf{w}_r^{\perp} + \gamma_r \cdot \boldsymbol{\mu}/\|\boldsymbol{\mu}\|_2^2 + \sum_{i=1}^{n} \rho_{r,i} \cdot \boldsymbol{\xi}_i^{\mathrm{fine\text{-}tuning}}/\|\boldsymbol{\xi}_i^{\mathrm{fine\text{-}tuning}}\|_2^2$$

is inspired by the "signal-noise decomposition" proposed in Cao et al. (2022), where the authors have demonstrated that such a decomposition can be very helpful for the analysis of supervised fine-tuning. Theorem 5.3 demonstrates that there exist at least $O(m)$ filters whose "signal coefficients" $\gamma_r$ are relatively large compared with the "noise coefficients" $\rho_{r,i}$ of *all* the $2m$ filters. This result makes good preparation for our analysis of the downstream task. Notably, as we have discussed, Theorem 5.3 can also serve as a theoretical guarantee for the setting of SimCLR pre-training on Gaussian mixture data, and hence we believe that the result of Theorem 5.3 and its proof may be of independent interest.

**Analysis of supervised fine-tuning.** We can now analyze the supervised fine-tuning of the nonlinear CNN where the initial CNN weights are given by SimCLR pre-training. We remind the readers that the nonlinear CNN model has second layer weights fixed as $+1/m$ and $-1/m$, and we define $F_{+1}(\mathbf{W}_{+1}, \mathbf{x})$, $F_{-1}(\mathbf{W}_{-1}, \mathbf{x})$ in (3.2) so that the CNN can be written as $f(\mathbf{W}, \mathbf{x}) = F_{+1}(\mathbf{W}_{+1}, \mathbf{x}) - F_{-1}(\mathbf{W}_{-1}, \mathbf{x})$. With filters $\mathbf{w}_r^{(T_{\mathrm{SimCLR}})}$, $r \in [2m]$ obtained by SimCLR pre-training, we randomly sample $m$ filters of them and assign them to the initialization of filters in $F_{+1}$, and we denote $\mathcal{M} \subseteq [2m]$ the collection of these filters with $|\mathcal{M}| = m$. In addition, the rest $m$ filters $\mathbf{w}_r^{(T_{\mathrm{SimCLR}})}$, $r \in [2m] \cap \mathcal{M}^c$ are randomly assigned to the initialization of filters in $F_{-1}$. Based on this random assignment procedure, we directly have the following lemma.

**Lemma 5.4.** *For any index sets* $\mathcal{I}^+, \mathcal{I}^- \subseteq [2m]$ *with* $|\mathcal{I}^+| = |\mathcal{I}^-| = 2m/5$, *with probability at least* $1 - 2^{-(2m/5-1)}$, *there exist* $r_+ \in \mathcal{I}^+$ *and* $r_- \in \mathcal{I}^-$, *such that* $r_+ \in \mathcal{M}$ *and* $r_- \in \mathcal{M}^c$.

Lemma 5.4 is a straightforward result on random sampling. Following this result, we can see that with high probability, there exists a filter in $\mathcal{I}^+$ whose second layer parameter is assigned as $+1$ for fine-tuning, and there also exists a filter in $\mathcal{I}^-$ whose second layer parameter is assigned as $-1$ for fine-tuning. With this result, we further give the following main theorem on the learning guarantees in the supervised fine-tuning stage.

**Theorem 5.5.** *Suppose that the supervised fine-tuning starts with initialization* $\mathbf{W}_+^{(0)}$ *and* $\mathbf{W}_-^{(0)}$ *where the convolution filters have decomposition*

$$\mathbf{w}_{j,r}^{(0)} = \mathbf{w}_{j,r}^{\perp} + j \cdot \gamma_{j,r} \cdot \boldsymbol{\mu}/\|\boldsymbol{\mu}\|_2^2 + \sum_{i=1}^{n} \rho_{j,r,i} \cdot \boldsymbol{\xi}_i^{\mathrm{fine-tuning}}/\|\boldsymbol{\xi}_i^{\mathrm{fine-tuning}}\|_2^2$$

*for* $j \in \{\pm 1\}$ *and* $r \in [m]$. *Moreover, suppose that the coefficients* $\gamma_{j,r}$'s *and* $\rho_{j,r,i}$'s *satisfy the following properties:*

- *There exist* $r_+, r_- \in [m]$ *such that*

$$\frac{\gamma_{+1,r_+}^{q-2}/\log(2/\gamma_{+1,r_+})}{\max_{j,r,i} |\rho_{j,r,i}|^{q-2}} = \Omega\left(\frac{\log(d)}{n\mathrm{SNR}^2}\right), \qquad \frac{\gamma_{-1,r_-}^{q-2}/\log(2/\gamma_{-1,r_-})}{\max_{j,r,i} |\rho_{j,r,i}|^{q-2}} = \Omega\left(\frac{\log(d)}{n\mathrm{SNR}^2}\right).$$

- *All* $\mathbf{w}_r^{(T_{\mathrm{SimCLR}})}$, $r \in [2m]$ *are bounded:*

$$\max_{j,r} \|\mathbf{w}_{j,r}^\perp\|_2 \leq \frac{1}{n}, \ \max_{j,r} |\gamma_{j,r}| \leq \widetilde{O}\left(\frac{\mathrm{SNR}^{2/q-2}}{m^{2/q-2}}\right), \ \max_{j,r,i} |\rho_{j,r,i}| \leq \widetilde{O}\left(\frac{\mathrm{SNR}^{2/q-2}}{m^{2/q-2}}\right).$$

*Let* $\gamma_0 = \min\{\gamma_{+1,r_+}, \gamma_{-1,r_-}\}$. *For any* $\epsilon > 0$, *let* $T = \widetilde{\Theta}(\eta^{-1} m \gamma_0^{-(q-2)} \|\boldsymbol{\mu}\|_2^{-2} + \eta^{-1}\epsilon^{-1}m^3\|\boldsymbol{\mu}\|_2^{-2})$, *then if* $\eta = \widetilde{O}(\min\{(\sigma_p^2 d)^{-1}, (\sigma_p\sqrt{d})^{-2}, \|\boldsymbol{\mu}\|_2^{-2}\})$, *with probability at least* $1 - d^{-1}$, *there exists* $0 \leq t \leq T$ *such that:*

1. *The training loss converges to* $\epsilon$, *i.e.,* $L_S(\mathbf{W}^{(t)}) \leq \epsilon$.

2. *The trained CNN achieves a small test loss:* $L_{\mathcal{D}}(\mathbf{W}^{(t)}) \leq 6\epsilon + \exp(-\widetilde{\Omega}(n^2))$.

Theorem 5.5 starts with an assumption on the properties of the "signal-noise decompositions" of the initial weights for fine-tuning. This analysis is inspired by Cao et al. (2022) where the signal-noise decomposition is proposed. However, compared with Cao et al. (2022) where the analysis focuses on training starting from random Gaussian initialization, Theorem 5.5 is in fact more general – $\mathbf{w}_{j,r}^{(0)}$'s are not necessarily randomly generated. In fact, by direct calculations, we can verify that if $\mathbf{w}_{j,r}^{(0)}$'s are all randomly generated from Gaussian distribution, then verifying the conditions in Theorem 5.5 will recover the condition that $n \cdot \mathrm{SNR}^q = \widetilde{\Omega}(1)$ in Cao et al. (2022) which guarantees benign overfitting. Therefore, Theorem 5.5 covers the result of benign overfitting in Cao et al. (2022).

Now it is clear that combining Theorem 5.3, Lemma 5.4 and Theorem 5.5 will immediately lead to Theorem 4.2. Therefore, our proof is finished.

## 6 CONCLUSION

In this paper, a case study on the benefits of SimCLR pre-training method for supervised fine-tuning is investigated. Based on a toy image data model for binary classification problems, we theoretically analyze how SimCLR pre-training based on unlabeled data benefits fine-tuning in training two-layer over-parameterized CNNs. Under mild conditions on the amount of labeled and unlabeled data and the signal-to-noise ratio (SNR), the training loss convergence and small test loss are guaranteed, while direct supervised learning requires more label complexity to achieve small training and test losses. Our work demonstrates the provable advantage of SimCLR pre-training in fine-tuning stage, which reduces label complexity to achieve a small test loss.

This paper focuses on the benefits of the popular SimCLR pre-training method for fine-tuning training. Apart from the SimCLR method for vision tasks, other contrastive learning (or self-supervised learning) methods could also be investigated. A more general question could be: How does the pre-training of representations influence the performance of fine-tuning in the over-parameterized models? Various fine-tuning training processes could also be analyzed, including single-task supervised learning or multi-task learning. Future works could explore the aforementioned directions.

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

# A  EXPERIMENTS

This section presents the synthetic-data experiments and real-world data experiments to back up theoretical results and demonstrate the practical value of our theory.

**Synthetic-data experiments.** The synthetic dataset is generated following the data model in Definition 3.1 with the dimension $d = 400$, $\sigma_p = 2$, $\|\boldsymbol{\mu}\|_2 = 10$. The training of the two-layer CNN model follows the two-stage (SimCLR pre-training followed by supervised fine-tuning) training procedure as depicted in Figure 1. Here, the number of filters is set as $m = 40$ with $\text{ReLU}^3$ activation function. The number of unlabeled data in the SimCLR pre-training stage is $n_0 = 250$ and the number of labeled data in the fine-tuning stage is $n = 40$. The test loss is calculated using $400$ test data points.

In parallel, we also conduct the training of the two-layer CNN model (3.2) through direct-supervised learning on $n = 40$ labeled data for comparison, and all the conditions of the labeled dataset are same as its SimCLR pre-training counterpart. The results on synthetic data experiments are presented in Figure 2.

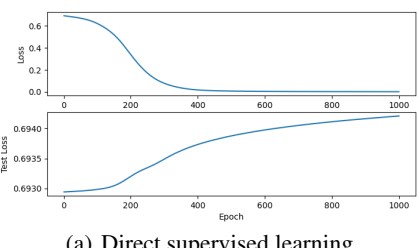
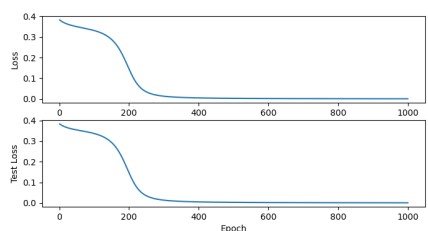

(a) Direct supervised learning      (b) SimCLR pre-training with supervised fine-tuning

Figure 2: Synthetic-data experiments: Under same conditions on label complexity, SimCLR pre-training combined with supervised fine-tuning ($n_0 = 250, n = 40$) achieves much smaller test loss than direct supervised learning ($n = 40$).

**Real-data experiments on MNIST dataset.** Real-world data experiments on MNIST dataset (LeCun et al., 1998) are also conducted. Following the data model in Definition 3.1, a binary classification problem is considered, where the signal $\boldsymbol{\mu}$ is originated from MNIST dataset with the size of $28 \times 28$. In the data model, the dimension is set as $d = 28 \cdot 28 = 784$, and $\sigma_p = 200$. Figure 3 presents the signals and dataset in the real-data experiments.

We train a two-layer CNN model with the number of filters $m = 16$, and the activation function is $\text{ReLU}^3$, and follows the two-stage training procedure of the CNN as depicted in the Figure 1. The training losses and the test losses of SimCLR pre-training followed by supervised fine-tuning are compared with its direct supervised learning counterpart. The comparison is made under the same label complexity condition where the number of unlabeled data in the SimCLR pre-training stage is $n_0 = 200$ and the number of labeled data in the fine-tuning stage as well as in the direct supervised learning counterpart is $n = 40$. The test losses are calculated using $400$ test data points. The results are presented in Figure 4. Figure 4 essentially presents an extreme case where SimCLR pre-training with supervised fine-tuning and direct supervised learning result in significantly different performances on noisy MNIST images despite of the same label complexity.

In the following part, we conduct large-scale experiments on both synthetic-data and real-data (MNIST) datasets. The basic data settings are the same as above experiments.

**Large-scale synthetic-data experiments.** The synthetic dataset is generated following the data model in Definition 3.1 with the dimension $d = 400$, $\sigma_p = 4$, $\|\boldsymbol{\mu}\|_2 = 10$. The training of the two-layer CNN model follows the two-stage (SimCLR pre-training followed by supervised fine-tuning) training procedure as depicted in Figure 1. Here, the number of filters is set as $m = 40$ with $\text{ReLU}^3$ activation function. The number of unlabeled data in the SimCLR pre-training stage is $n_0 = 100000$ and the number of labeled data in the fine-tuning stage is $n = 250$. The test accuracy is calculated using $400$ test data points. In parallel, we also conduct the training of the two-layer CNN model (3.2) through direct-supervised learning on $n = 40$ labeled data for comparison, and all the conditions of

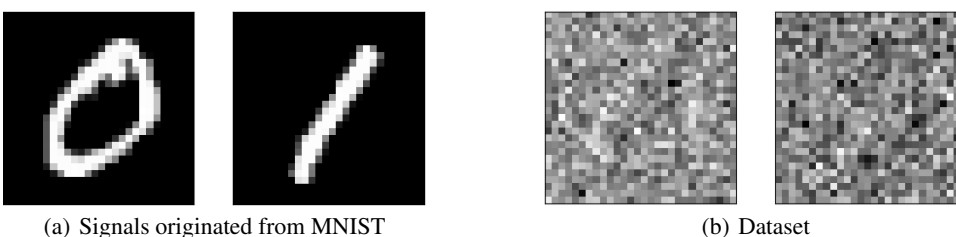

(a) Signals originated from MNIST                    (b) Dataset

Figure 3: Signals and the dataset in real-data experiments. The signals are originated from MNIST dataset. Following the data model in Definition 3.1, noise is added to the signals and obtain the dataset used in the training.

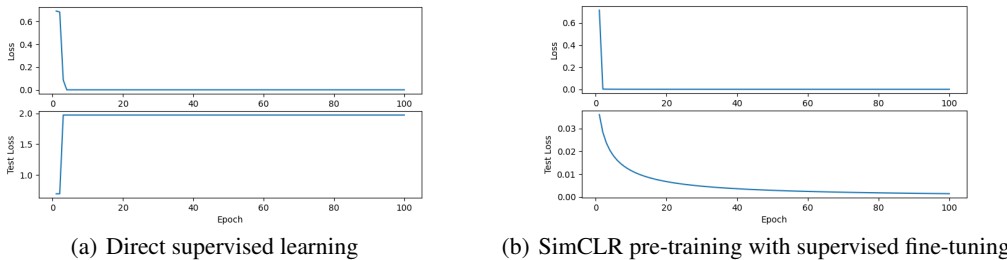

(a) Direct supervised learning          (b) SimCLR pre-training with supervised fine-tuning

Figure 4: Real-data experiments: Under same conditions on label complexity ($n = 40$), SimCLR pre-training combined with supervised fine-tuning ($n_0 = 200, n = 40$) achieves much smaller test loss than direct supervised learning.

the labeled dataset are same as its SimCLR pre-training counterpart. The results on synthetic data experiments are presented in Figure 5.

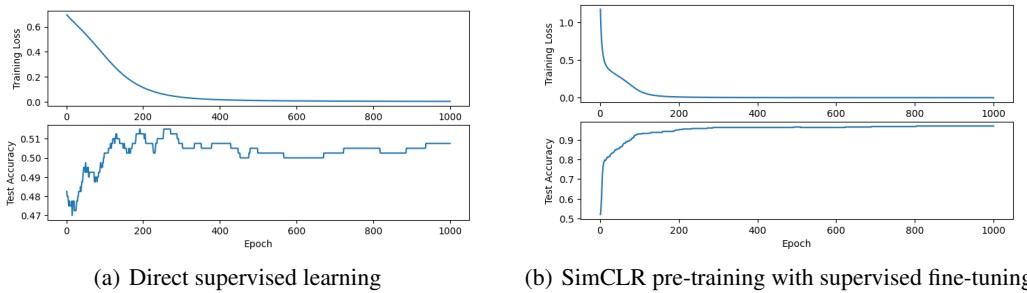

(a) Direct supervised learning          (b) SimCLR pre-training with supervised fine-tuning

Figure 5: Large-scale synthetic-data experiments: Under same conditions on label complexity, Sim-CLR pre-training combined with supervised fine-tuning ($n_0 = 100000, n = 250$) achieves much higher test accuracy than direct supervised learning ($n = 250$).

**Large-scale real-data experiments on MNIST dataset.** Large-scale real-world data experiments on MNIST dataset (LeCun et al., 1998) are also conducted. Following the data model in Definition 3.1, a binary classification problem is considered, where the signal $\boldsymbol{\mu}$ is originated from MNIST dataset with the size of $28 \times 28$. In the data model, the dimension is set as $d = 28 \cdot 28 = 784$, and the scale of the noise $\sigma_p = 2$. All signals originated from MNSIT dataset are normalized to have same norm $\|\boldsymbol{\mu}\|_2 = 20$. We train a two-layer CNN model with the number of filters $m = 16$, and the activation function is $\mathrm{ReLU}^3$, and follows the two-stage training procedure of the CNN as depicted in the Figure 1. The training losses and the test accuracies of SimCLR pre-training followed by supervised fine-tuning are compared with its direct supervised learning counterpart. The comparison is made under the same label complexity condition where the number of unlabeled data in the

SimCLR pre-training stage is $n_0 = 13800$ and the number of labeled data in the fine-tuning stage as well as in the direct supervised learning counterpart is $n = 40$. The test accuracy is calculated using 400 test data points. The results are presented in Figure 6. Figure 6 essentially presents an extreme case where SimCLR pre-training with supervised fine-tuning and direct supervised learning result in significantly different performances on noisy MNIST images despite of the same label complexity.

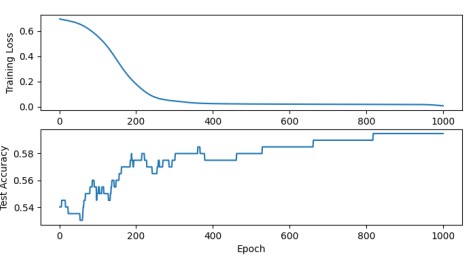 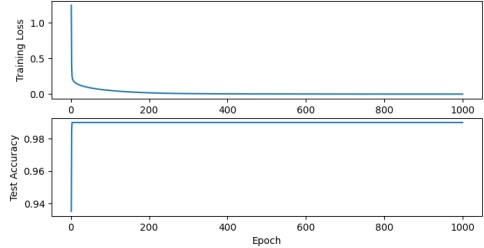

(a) Direct supervised learning      (b) SimCLR pre-training with supervised fine-tuning

Figure 6: Large-scale real-data experiments: Under same conditions on label complexity ($n = 40$), SimCLR pre-training combined with supervised fine-tuning ($n_0 = 13800, n = 40$) achieves much higher test accuracy than direct supervised learning.

We also conduct several groups of real-data experiments of SimCLR pre-training and followed by supervised fine-tuning on different values of signal-to-noise ratio (SNR), $n_0, n$, and compare their performance (test accuracies) on the supervised fine-tuning stage. These groups of experiments are conducted to see how the test accuracies of the fine-tuning stage vary with different values of $n_0, \mathrm{SNR}, n$. The corresponding comparison of results for $n_0, \mathrm{SNR}, n$ are presented in Figures 7, 8 and 9 respectively.

- The unlabeled data size $n_0$: Figure 7 presents the performance of experiments with different size of unlabeled pre-training data $n_0$. While all other conditions remain the same, experiments with larger size of unlabeled pre-training data $n_0$ achieves a better test accuracy.

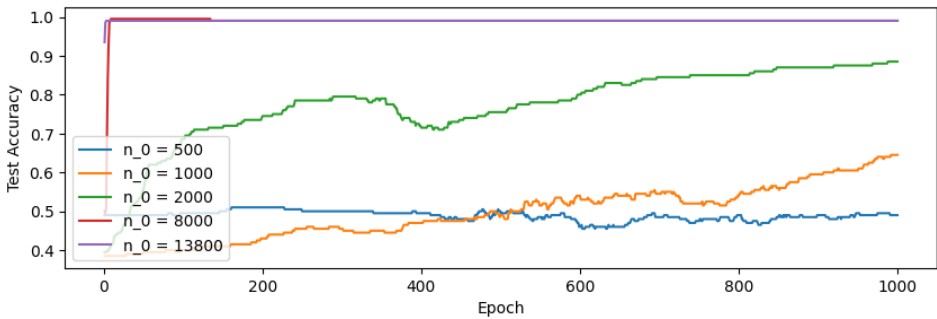

Figure 7: Training performance comparison for different size of unlabeled pre-training data $n_0$: the test accuracies in the supervised fine-tuning stage for the experiments with different unlabeled data size $n_0$ ($n_0 = 500, 1000, 2000, 8000, 13800$). All experiments are conducted with same labeled data size $n = 40$ for supervised learning and same SNR.

- Signal-to-noise ratio (SNR): Figure 8 shows that under same labeled data and unlabeled data size, for the experiments with smaller SNR, the training performance is worser. For experiment with smaller SNR, it requires more (labeled or unlabeled) data to achieve a good test performance.

- The labeled sample size $n$: Figure 9 shows that, in the SimCLR pre-training followed by supervised fine-tuning, given a satisfactory number of unlabeled data $n_0$, the condition on the size of

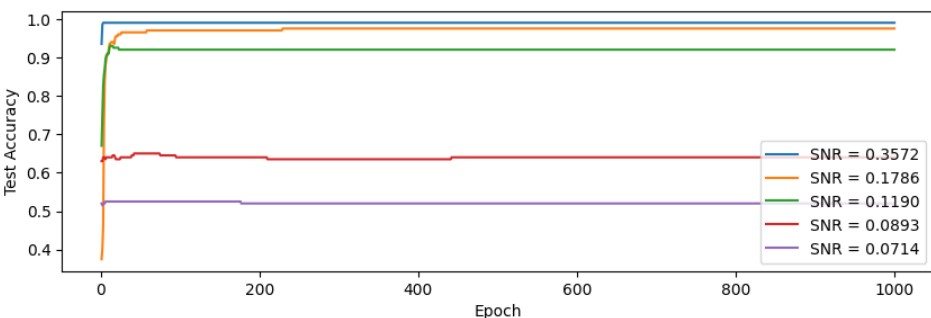

Figure 8: Training performance comparison for different SNR: the test accuracies in the supervised fine-tuning stage for the experiments with different SNR. All experiments are conducted with $n_0 = 13800$ unlabeled pre-training data and $n = 40$ labeled supervised learning data, and same scale of signal $\|\boldsymbol{\mu}\|_2 = 20$. Different noise scale $\sigma_p$ are selected and this leads to different SNR.

labeled data $n$ to achieve a high test accuracy is mild. This is accordance with the condition and the theoretical results presented in Condition 4.1 and Theorem 4.2.

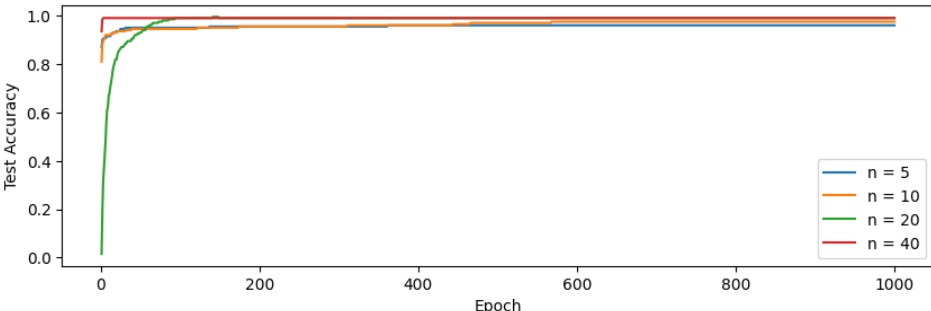

Figure 9: Training performance comparison for different size of labeled data $n$: the test accuracies in the supervised fine-tuning stage for the experiments with different labeled data size $n$ ($n = 5, 10, 20, 40$). All experiments are conducted with $n_0 = 13800$ unlabeled data for pre-training and same SNR (same signal scale $\|\boldsymbol{\mu}\|_2 = 20$ and noise $\sigma_p = 4$).

**Summary of experiment results.** The experiments above present typical cases where SimCLR pre-training followed by supervised fine-tuning achieves a much smaller test loss, while direct supervised learning achieves a larger test loss under the same label complexity. Both synthetic and real-world experiments match our theoretical results and demonstrate that SimCLR pre-training could relax the requirement of label complexity to achieve a small test loss.

In the following part, we analyze that whether the result that SimCLR pre-training advantages the supervised fine-tuning theoretically proved in this paper on CNNs also holds for other models by empirical experiments.

**Real-data experiments on simple Vision Transformers (ViT).** Real-data experiments based on images of digit $0$ and $1$ in the MNIST dataset (LeCun et al., 1998) are conducted on simple Vision Transformers. Following the data model in Definition 3.1, we consider the case where clean MNIST images are treated as "signal patches" and are hidden among other "noise patches". Therefore, the dimension for signal and noise patches is set as $d = 28 \cdot 28 = 784$ according to the size of MNIST images, and we set the standard deviation of the Gaussian noises to be $\sigma_p = 5$. All images from MNIST dataset are normalized to have same norm $\|\boldsymbol{\mu}\|_2 = 20$.

The training procedure involves first pre-training on unlabeled data to minimize the SimCLR loss, and then the model initialized by SimCLR pre-training is fine-tuned on labeled data to minimize the

cross-entropy loss. The model $f(\mathbf{X}) : \mathbb{R}^{d \times p} \to \mathbb{R}^m$ is defined as follows:

$$f(\mathbf{X}) = \mathbf{V}\mathbf{X} \cdot \text{Softmax}\big((\mathbf{K}\mathbf{X})^\top (\mathbf{Q}\mathbf{X})\big) \cdot \mathbf{1}_p,$$

where the input $\mathbf{X} \in \mathbb{R}^{d \times p}$, $p$ is number of patches, the input is given as $\mathbf{X} = (\mathbf{x}^{(1)}, \ldots, \mathbf{x}^{(p)}) \in \mathbb{R}^{d \times p}$ where $\mathbf{x}^{(i)} \in \mathbb{R}^{d \times 1}$, $i = 1, \ldots, p$ are images patches, and only one of the patches is a normalized MNIST image, while the other patches are Gaussian noises (corresponding to the noise patch in our data model in Definition 3.1). $\text{Softmax}(\cdot)$ refers to the column-wise Softmax function. The parameter matrices $\mathbf{V} \in \mathbb{R}^{m \times d}$, $\mathbf{K}, \mathbf{Q} \in \mathbb{R}^{k \times d}$, vector $\mathbf{1}_p = (1, \ldots, 1)^\top \in \mathbb{R}^{p \times 1}$ is a constant vector of all ones. In the pre-training stage, $f(\mathbf{X})$ and its augmented pair $f(\widetilde{\mathbf{X}})$ is trained under the SimCLR loss, where $\widetilde{\mathbf{X}}$ refers to the augmented data pair of $\mathbf{X}$ following the data augmentation in Section 3.2.

In the follow-up supervised fine-tuning stage, the model $g(\mathbf{X}) : \mathbb{R}^{d \times p} \to \mathbb{R}$ is fine-tuned by the cross-entropy loss $\ell(z) = \log(1 + \exp(-z))$, where $g(\mathbf{X})$ is derived based on the $f(\mathbf{X})$ initialized by the pre-training stage,

$$g(\mathbf{X}) = \mathbf{a}_m^\top \cdot \sigma(f(\mathbf{X})), \tag{A.1}$$

where $\mathbf{a}_m = (a_1, \ldots, a_m)^\top \in \mathbb{R}^m$ is a constant vector with entry $a_i, i \in [m]$ random sampled from $\{-1, 1\}$ with probability $1/2$, $\sigma(\cdot)$ is the ReLU$^q$ activation function with $q > 2$. Here, we select $m = 40$, $k = 2, p = 2$ in the model setting.

The training losses and the test accuracies of SimCLR pre-training followed by supervised fine-tuning are compared with its direct supervised learning counterpart on simple ViTs defined in (A.1). The comparison is made under the same number of labeled data and same other conditions, where the number of unlabeled data in the SimCLR pre-training stage is $n_0 = 128000$ and the number of labeled data in the fine-tuning stage as well as in the direct supervised learning counterpart is $n = 64$. The test accuracy is calculated using $400$ test data points.

The results are presented in Figure 10. Figure 10 essentially presents an extreme case where with the same labeled dataset, SimCLR pre-training followed by supervised fine-tuning achieves a significantly better test performance than direct supervised learning on the simple ViT models. It demonstrates that the phenomenon that SimCLR pre-training advantages the fine-tuning theoretically proved in this paper on two-layer CNNs can also be observed in empirical experiments on other models such as Vision Transformers.

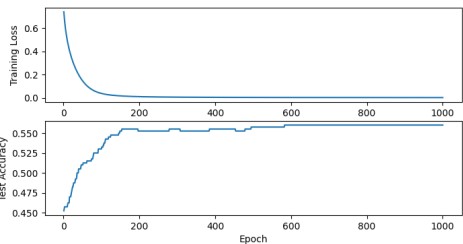
(a) Direct supervised learning

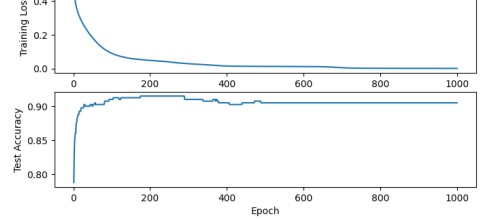
(b) SimCLR pre-training with supervised fine-tuning

Figure 10: Real-data experiments on simple ViTs: the training losses and test accuracies comparison in the supervised learning stage for direct supervised learning and SimCLR pre-training with supervised fine-tuning. Under same conditions on label complexity ($n = 64$), SimCLR pre-training combined with supervised fine-tuning ($n_0 = 128000, n = 64$) achieves much higher test accuracy than direct supervised learning on simple ViTs.

# B   PROOFS FOR SIMCLR PRE-TRAINING

## B.1   PROOFS OF LEMMAS IN SECTION 5

### B.1.1   PROOF OF LEMMA 5.1

In this section, the following Lemma B.1 is introduced to prove Lemma 5.1.

**Lemma B.1.** *For any $\widetilde{\delta} > 0$, with probability at least $1 - \widetilde{\delta}$, a union bound for $\|\boldsymbol{\xi}_i\|_2^2$ and $\|\widetilde{\boldsymbol{\xi}}_i\|_2^2$, $i \in [n_0]$ is*

$$d\sigma_p^2 - C_2\sigma_p^2\sqrt{d\log(4n_0/\widetilde{\delta})} \leq \|\boldsymbol{\xi}_i\|_2^2 \leq d\sigma_p^2 + C_2\sigma_p^2\sqrt{d\log(4n_0/\widetilde{\delta})}$$

$$d\sigma_p^2 - C_2\sigma_p^2\sqrt{d\log(4n_0/\widetilde{\delta})} \leq \left\|\widetilde{\boldsymbol{\xi}}_i\right\|_2^2 \leq d\sigma_p^2 + C_2\sigma_p^2\sqrt{d\log(4n_0/\widetilde{\delta})}$$

*where $C_2$ is an absolute constant that does not depend on other variables.*

Based on the Lemma B.1, the proof of Lemma 5.1 is presented as follows.

*Proof of Lemma 5.1.* By direct calculation, we have

$$\nabla_{\mathbf{w}_r} L_{S_{\text{unlabeled}}}(\mathbf{W}) = \frac{1}{n_0\tau}\sum_{i=1}^{n_0}\text{softmax}_i \cdot \sum_{i'\neq i}\exp(\text{sim}_{i,i'}/\tau - \text{sim}_i/\tau) \cdot (\nabla_{\mathbf{w}_r}\text{sim}_{i,i'} - \nabla_{\mathbf{w}_r}\text{sim}_i),$$

where

$$\text{softmax}_i = \frac{\exp(\text{sim}_i/\tau)}{\exp(\text{sim}_i/\tau) + \sum_{i'\neq i}\exp(\text{sim}_{i,i'}/\tau)},$$

$$\nabla_{\mathbf{w}_r}\text{sim}_i = (\mathbf{z}_i\widetilde{\mathbf{z}}_i^\top + \widetilde{\mathbf{z}}_i\mathbf{z}_i^\top) \cdot \mathbf{w}_r,$$

$$\nabla_{\mathbf{w}_r}\text{sim}_{i,i'} = (\mathbf{z}_i\mathbf{z}_{i'}^\top + \mathbf{z}_{i'}\mathbf{z}_i^\top) \cdot \mathbf{w}_r.$$

Reorganize terms then gives

$$\nabla_{\mathbf{w}_r} L_{S_{\text{unlabeled}}}(\mathbf{W}) = \frac{1}{n_0\tau}\sum_{i=1}^{n_0}\sum_{i'\neq i}\text{softmax}_i \cdot \exp(\text{sim}_{i,i'}/\tau - \text{sim}_i/\tau) \cdot (\nabla_{\mathbf{w}_r}\text{sim}_{i,i'} - \nabla_{\mathbf{w}_r}\text{sim}_i)$$

$$= \frac{1}{n_0\tau}\sum_{i=1}^{n_0}\sum_{i'\neq i}\left(\frac{\exp(\text{sim}_{i,i'}/\tau)}{\exp(\text{sim}_i/\tau) + \sum_{i''\neq i}\exp(\text{sim}_{i,i'}/\tau)}\right) \cdot (\mathbf{z}_i\mathbf{z}_{i'}^\top + \mathbf{z}_{i'}\mathbf{z}_i^\top - \mathbf{z}_i\widetilde{\mathbf{z}}_i^\top - \widetilde{\mathbf{z}}_i\mathbf{z}_i^\top)\mathbf{w}_r.$$

Now define

$$\boldsymbol{\Xi}^{(t)} = -\frac{\eta}{n_0^2\tau} \cdot \sum_{i=1}^{n_0}\sum_{i'\neq i}\left(\frac{n_0 \cdot \exp(\text{sim}_{i,i'}^{(t)}/\tau)}{\exp(\text{sim}_i^{(t)}/\tau) + \sum_{i''\neq i}\exp(\text{sim}_{i,i'}^{(t)}/\tau)} - 1\right) \cdot (\mathbf{z}_i\mathbf{z}_{i'}^\top + \mathbf{z}_{i'}\mathbf{z}_i^\top - \mathbf{z}_i\widetilde{\mathbf{z}}_i^\top - \widetilde{\mathbf{z}}_i\mathbf{z}_i^\top).$$

Then by gradient descent update rule, we have

$$\mathbf{w}_r^{(t+1)} = \mathbf{w}_r^{(t)} + (\mathbf{A} + \boldsymbol{\Xi}^{(t)})\mathbf{w}_r^{(t)}.$$

Moreover, by the definition of $\mathbf{W}$ and the fact that $\mathbf{A}$, $\boldsymbol{\Xi}^{(t)}$ are both symmetric matrices, we also have

$$\mathbf{W}^{(t+1)} = \mathbf{W}^{(t)} + \mathbf{W}^{(t)}(\mathbf{A} + \boldsymbol{\Xi}^{(t)}). \tag{B.1}$$

By the definition of $T_{\text{SimCLR}}$ and the assumption that $\mathcal{E}_{\text{SimCLR}} \leq 1/2$, for $t \leq T_{\text{SimCLR}}$ we have

$$\left[1 + (1 - \mathcal{E}_{\text{SimCLR}}) \cdot \|\mathbf{A}\|_2\right]^t \leq \max\left\{288M^{\frac{1}{q-2}} \cdot \frac{\log(2/\sigma_0)^{\frac{1}{q-2}} \cdot \sqrt{\log(dn)} \cdot \sqrt{\log(md)}}{n^{\frac{1}{q-2}}\text{SNR}^{\frac{q}{q-2}}}, 2\right\}.$$

This inequality further implies that

$$\left[1 + (1 + \sigma_0) \cdot \|\mathbf{A}\|_2\right]^t \leq \left[1 + (1 - \mathcal{E}_{\text{SimCLR}}) \cdot \|\mathbf{A}\|_2\right]^{\frac{1+\sigma_0}{1-\mathcal{E}_{\text{SimCLR}}}t}$$

$$\leq \left[1 + (1 - \mathcal{E}_{\text{SimCLR}}) \cdot \|\mathbf{A}\|_2\right]^{2t}$$

$$\leq \max\left\{288M^{\frac{1}{q-2}} \cdot \frac{\log(2/\sigma_0)^{\frac{1}{q-2}} \cdot \sqrt{\log(dn)} \cdot \sqrt{\log(md)}}{n^{\frac{1}{q-2}}\text{SNR}^{\frac{q}{q-2}}}, 2\right\}^2 \tag{B.2}$$

for all $t = 0, 1, \ldots, T_{\mathrm{SimCLR}}$, where the first inequality follows by the fact that $1 + a \le (1+b)^{a/b}$ for all $a > b > 0$, and the second inequality follows by the assumption that $\mathcal{E}_{\mathrm{SimCLR}} \le 1/4$ and $\sigma_0 \le 1/4$.

In the following, we utilize (B.2) to prove the upper bound $\|\mathbf{\Xi}^{(t)}\|_2 \le \sigma_0 \cdot \|\mathbf{A}\|_2$ by induction. To do so, we first check the bound at $t = 0$. We have

$$\|\mathbf{\Xi}^{(0)}\|_2 \le \frac{\eta}{\tau} \max_{i \ne i'} \left| \frac{n_0 \cdot \exp(\mathrm{sim}_{i,i'}^{(0)}/\tau)}{\exp(\mathrm{sim}_i^{(0)}/\tau) + \sum_{i'' \ne i} \exp(\mathrm{sim}_{i,i'}^{(0)}/\tau)} - 1 \right| \cdot \|\mathbf{z}_i \mathbf{z}_{i'}^\top + \mathbf{z}_{i'} \mathbf{z}_i^\top - \mathbf{z}_i \widetilde{\mathbf{z}}_i^\top - \widetilde{\mathbf{z}}_i \mathbf{z}_i^\top\|_2$$

$$\le \frac{2\eta}{\tau} \max_{i \ne i'} \left| \frac{n_0 \cdot \exp(\mathrm{sim}_{i,i'}^{(0)}/\tau)}{\exp(\mathrm{sim}_i^{(0)}/\tau) + \sum_{i'' \ne i} \exp(\mathrm{sim}_{i,i'}^{(0)}/\tau)} - 1 \right| \cdot (\|\mathbf{z}_i\|_2 \|\mathbf{z}_{i'}\|_2 + \|\mathbf{z}_i\|_2 \|\widetilde{\mathbf{z}}_i\|_2)$$

$$\le \frac{8\eta}{\tau} \cdot (\|\boldsymbol{\mu}\|_2^2 + 2\sigma_p^2 d) \cdot \max_{i \ne i'} \left| \frac{n_0 \cdot \exp(\mathrm{sim}_{i,i'}^{(0)}/\tau)}{\exp(\mathrm{sim}_i^{(0)}/\tau) + \sum_{i'' \ne i} \exp(\mathrm{sim}_{i,i'}^{(0)}/\tau)} - 1 \right|, \qquad \text{(B.3)}$$

where the last inequality follows by Lemma B.1, which implies that $\|\boldsymbol{\xi}_i\|_2^2, \|\widetilde{\boldsymbol{\xi}}_i\|_2^2 \le 2\sigma_p^2 d$ with probability at least $1 - d^{-3}$. Moreover, by definition, we have

$$|\mathrm{sim}_i^{(0)}| = |\langle \mathbf{W}^{(0)} \mathbf{z}_i, \mathbf{W}^{(0)} \widetilde{\mathbf{z}}_i \rangle| \le \|\mathbf{W}^{(0)}\|_2^2 \cdot \|\mathbf{z}_i\|_2 \cdot \|\widetilde{\mathbf{z}}_i\|_2 \le 2\|\mathbf{W}^{(0)}\|_2^2 \cdot (\|\boldsymbol{\mu}\|_2^2 + 2\sigma_p^2 d),$$

for all $i \in [n_0]$. Since $\mathbf{W}^{(0)}$ is a random matrix whose entries are independently generated from $\mathcal{N}(0, \sigma_0^2)$, with probability at least $1 - d^{-3}$, we have

$$\|\mathbf{W}^{(0)}\|_2^2 \le c_1 \cdot \sigma_0^2 \cdot (d + n_0) \le 2c_1 \cdot \sigma_0^2 d,$$

where $c_1$ is an absolute constant. Therefore, we have

$$|\mathrm{sim}_i^{(0)}| \le c_2 \cdot \sigma_0^2 d \cdot (\|\boldsymbol{\mu}\|_2^2 + 2\sigma_p^2 d)$$

for all $i \in [n_0]$, where $c_2$ is an absolute constant. With exactly the same proof, we also have

$$|\mathrm{sim}_{i,i'}^{(0)}| \le 2\|\mathbf{W}^{(0)}\|_2^2 \cdot (\|\boldsymbol{\mu}\|_2^2 + 2\sigma_p^2 d) \le c_3 \cdot \sigma_0^2 d \cdot (\|\boldsymbol{\mu}\|_2^2 + 2\sigma_p^2 d)$$

for all $i, i' \in [n]$ with $i \ne i'$, where $c_3$ is an absolute constant. Now by the assumption that $\sigma_0 \le O(\tau^{1/2} \cdot d^{-1/2} \cdot \min\{\|\boldsymbol{\mu}\|_2^{-1}, \sigma_p^{-1} d^{-1/2}\})$, we have $|\mathrm{sim}_i^{(0)}|, |\mathrm{sim}_{i,i'}^{(0)}| \le \tau/4$. Since $|\exp(z) - 1| \le 2|z|$ for all $z \le 1$, we have

$$|\exp(\mathrm{sim}_i^{(0)}/\tau) - 1| \le 2|\mathrm{sim}_i^{(0)}/\tau| \le c_4 \cdot \sigma_0^2 d \cdot (\|\boldsymbol{\mu}\|_2^2 + 2\sigma_p^2 d) \le 1/2, \qquad \text{(B.4)}$$

$$|\exp(\mathrm{sim}_{i,i'}^{(0)}/\tau) - 1| \le 2|\mathrm{sim}_{i,i'}^{(0)}/\tau| \le c_5 \cdot \sigma_0^2 d \cdot (\|\boldsymbol{\mu}\|_2^2 + 2\sigma_p^2 d) \le 1/2 \qquad \text{(B.5)}$$

for all $i, i' \in [n]$ with $i \ne i'$, where $c_4, c_5$ are absolute constants. Therefore, for all $i, i' \in [n]$ with $i \ne i'$, we have

$$\left| \frac{n_0 \cdot \exp(\mathrm{sim}_{i,i'}^{(0)}/\tau)}{\exp(\mathrm{sim}_i^{(0)}/\tau) + \sum_{i'' \ne i} \exp(\mathrm{sim}_{i,i'}^{(0)}/\tau)} - 1 \right|$$

$$\le \left| \frac{n_0 \cdot [\exp(\mathrm{sim}_{i,i'}^{(0)}/\tau) - 1]}{\exp(\mathrm{sim}_i^{(0)}/\tau) + \sum_{i'' \ne i} \exp(\mathrm{sim}_{i,i'}^{(0)}/\tau)} \right| + \left| \frac{n_0}{\exp(\mathrm{sim}_i^{(0)}/\tau) + \sum_{i'' \ne i} \exp(\mathrm{sim}_{i,i'}^{(0)}/\tau)} - 1 \right|$$

$$\le \left| \frac{n_0 \cdot [\exp(\mathrm{sim}_{i,i'}^{(0)}/\tau) - 1]}{\exp(\mathrm{sim}_i^{(0)}/\tau) + \sum_{i'' \ne i} \exp(\mathrm{sim}_{i,i'}^{(0)}/\tau)} \right| + \left| \frac{1 - \exp(\mathrm{sim}_i^{(0)}/\tau) + \sum_{i'' \ne i}[1 - \exp(\mathrm{sim}_{i,i'}^{(0)}/\tau)]}{\exp(\mathrm{sim}_i^{(0)}/\tau) + \sum_{i'' \ne i} \exp(\mathrm{sim}_{i,i'}^{(0)}/\tau)} \right|$$

$$\le \left| \frac{n_0 \cdot [\exp(\mathrm{sim}_{i,i'}^{(0)}/\tau) - 1]}{n_0/2} \right| + \left| \frac{1 - \exp(\mathrm{sim}_i^{(0)}/\tau) + \sum_{i'' \ne i}[1 - \exp(\mathrm{sim}_{i,i'}^{(0)}/\tau)]}{n_0/2} \right|$$

$$\le c_6 \cdot \sigma_0^2 d \cdot (\|\boldsymbol{\mu}\|_2^2 + 2\sigma_p^2 d),$$

where $c_6$ is an absolute constant, and the first inequality above follows the triangle inequality, the third inequality applies (B.4) and (B.5) to the denominators, and the last inequality applies (B.4) and (B.5) to the numerators. Plugging the bound above into (B.3) then gives

$$\|\boldsymbol{\Xi}^{(0)}\|_2 \leq \frac{c_7 \eta}{\tau} \cdot \sigma_0^2 d \cdot (\|\boldsymbol{\mu}\|_2^2 + 2\sigma_p^2 d)^2 \leq \sigma_0 \cdot \frac{\eta}{\tau} \cdot \|\boldsymbol{\mu}\|_2^2$$

$$\leq \sigma_0 \cdot (1 - \mathcal{E}_{\text{SimCLR}}) \cdot \frac{2\eta}{\tau} \cdot \|\boldsymbol{\mu}\|_2^2 \leq \sigma_0 \cdot \|\mathbf{A}\|_2, \tag{B.6}$$

where $c_7$ is an absolute constant, and the second inequality follows by the assumption that $\sigma_0 \leq O(d^{-1} \cdot \|\boldsymbol{\mu}\|_2^2 \cdot \min\{\|\boldsymbol{\mu}\|_2^{-4}, (\sigma_p^2 d)^{-2}\})$, the third inequality is by the assumption that $\mathcal{E}_{\text{SimCLR}} \leq 1/2$, and the last inequality is by $(1 - \mathcal{E}_{\text{SimCLR}}) \cdot \frac{2\eta}{\tau} \cdot \|\boldsymbol{\mu}\|_2^2 \leq \|\mathbf{A}\|_2$ in Lemma 5.2. Now let us suppose that there exists $t_0 \leq T_{\text{SimCLR}} - 1$ such that for $t = 0, \ldots, t_0$, it holds that

$$\|\boldsymbol{\Xi}^{(t)}\|_2 \leq \sigma_0 \cdot \|\mathbf{A}\|_2.$$

Then we have

$$\|\mathbf{A} + \boldsymbol{\Xi}^{(t)}\|_2 \leq (1 + \sigma_0) \cdot \|\mathbf{A}\|_2,$$

for all $t = 0, \ldots, t_0$. Then by (B.1), we have

$$\|\mathbf{W}^{(t_0+1)}\|_2 \leq \|\mathbf{W}^{(t_0)}\|_2 \cdot \|\mathbf{I} + \mathbf{A} + \boldsymbol{\Xi}^{(t_0)}\|_2$$

$$\leq \|\mathbf{W}^{(t_0)}\|_2 \cdot (1 + \|\mathbf{A}\|_2 + \|\boldsymbol{\Xi}^{(t_0)}\|_2)$$

$$\leq (1 + (1 + \sigma_0) \cdot \|\mathbf{A}\|_2) \cdot \|\mathbf{W}^{(t_0)}\|_2$$

$$\leq \cdots$$

$$\leq (1 + (1 + \sigma_0) \cdot \|\mathbf{A}\|_2)^{t_0+1} \cdot \|\mathbf{W}^{(0)}\|_2$$

$$\leq \max\left\{ 288 M^{\frac{1}{q-2}} \cdot \frac{\log(2/\sigma_0)^{\frac{1}{q-2}} \cdot \sqrt{\log(dn)} \cdot \sqrt{\log(md)}}{n^{\frac{1}{q-2}} \text{SNR}^{\frac{q}{q-2}}}, 2 \right\}^2 \cdot O(\sigma_0 \sqrt{d}), \tag{B.7}$$

where the last inequality follows by (B.2). The rest of the proof follows almost the same derivation as the bound for $\|\boldsymbol{\Xi}^{(0)}\|_2$. We have

$$\|\boldsymbol{\Xi}^{(t_0+1)}\|_2 \leq \frac{\eta}{\tau} \max_{i \neq i'} \left| \frac{n_0 \cdot \exp(\text{sim}_{i,i'}^{(t_0+1)}/\tau)}{\exp(\text{sim}_i^{(t_0+1)}/\tau) + \sum_{i'' \neq i} \exp(\text{sim}_{i,i'}^{(t_0+1)}/\tau)} - 1 \right| \cdot \|\mathbf{z}_i \mathbf{z}_{i'}^\top + \mathbf{z}_{i'} \mathbf{z}_i^\top - \mathbf{z}_i \widetilde{\mathbf{z}}_i^\top - \widetilde{\mathbf{z}}_i \mathbf{z}_i^\top\|_2$$

$$\leq \frac{2\eta}{\tau} \max_{i \neq i'} \left| \frac{n_0 \cdot \exp(\text{sim}_{i,i'}^{(t_0+1)}/\tau)}{\exp(\text{sim}_i^{(t_0+1)}/\tau) + \sum_{i'' \neq i} \exp(\text{sim}_{i,i'}^{(t_0+1)}/\tau)} - 1 \right| \cdot (\|\mathbf{z}_i\|_2 \|\mathbf{z}_{i'}\|_2 + \|\mathbf{z}_i\|_2 \|\widetilde{\mathbf{z}}_i\|_2)$$

$$\leq \frac{8\eta}{\tau} \cdot (\|\boldsymbol{\mu}\|_2^2 + 2\sigma_p^2 d) \cdot \max_{i \neq i'} \left| \frac{n_0 \cdot \exp(\text{sim}_{i,i'}^{(t_0+1)}/\tau)}{\exp(\text{sim}_i^{(t_0+1)}/\tau) + \sum_{i'' \neq i} \exp(\text{sim}_{i,i'}^{(t_0+1)}/\tau)} - 1 \right|, \tag{B.8}$$

where the last inequality follows by Lemma B.1, which implies that $\|\boldsymbol{\xi}_i\|_2^2, \|\widetilde{\boldsymbol{\xi}}_i\|_2^2 \leq 2\sigma_p^2 d$ with probability at least $1 - d^{-3}$. Moreover, by definition, we have

$$|\text{sim}_i^{(t_0+1)}| = |\langle \mathbf{W}^{(t_0+1)} \mathbf{z}_i, \mathbf{W}^{(t_0+1)} \widetilde{\mathbf{z}}_i \rangle| \leq \|\mathbf{W}^{(t_0+1)}\|_2^2 \cdot \|\mathbf{z}_i\|_2 \cdot \|\widetilde{\mathbf{z}}_i\|_2 \leq 2\|\mathbf{W}^{(t_0+1)}\|_2^2 \cdot (\|\boldsymbol{\mu}\|_2^2 + 2\sigma_p^2 d)$$

for all $i \in [n_0]$. By (B.7), we have

$$\|\mathbf{W}^{(t_0+1)}\|_2^2 \leq \max\left\{ 288 M^{\frac{1}{q-2}} \cdot \frac{\log(2/\sigma_0)^{\frac{1}{q-2}} \cdot \sqrt{\log(dn)} \cdot \sqrt{\log(md)}}{n^{\frac{1}{q-2}} \text{SNR}^{\frac{q}{q-2}}}, 2 \right\}^4 \cdot O(\sigma_0^2 d).$$

Therefore, we have

$$|\text{sim}_i^{(t_0+1)}| \leq \max\left\{ 288 M^{\frac{1}{q-2}} \cdot \frac{\log(2/\sigma_0)^{\frac{1}{q-2}} \cdot \sqrt{\log(dn)} \cdot \sqrt{\log(md)}}{n^{\frac{1}{q-2}} \text{SNR}^{\frac{q}{q-2}}}, 2 \right\}^4 \cdot O(\sigma_0^2 d \cdot (\|\boldsymbol{\mu}\|_2^2 + 2\sigma_p^2 d))$$

for all $i \in [n_0]$. With exactly the same proof, we also have

$$|\mathrm{sim}_{i,i'}^{(t_0+1)}| \leq \max\left\{288M^{\frac{1}{q-2}} \cdot \frac{\log(2/\sigma_0)^{\frac{1}{q-2}} \cdot \sqrt{\log(dn)} \cdot \sqrt{\log(md)}}{n^{\frac{1}{q-2}}\mathrm{SNR}^{\frac{q}{q-2}}}, 2\right\}^4 \cdot O(\sigma_0^2 d \cdot (\|\boldsymbol{\mu}\|_2^2 + 2\sigma_p^2 d))$$

for all $i, i' \in [n]$ with $i \neq i'$. Now by the assumption that $\sigma_0 = \min\left\{288M^{\frac{1}{q-2}} \cdot \right.$

$\left.\frac{\log(2/\sigma_0)^{\frac{1}{q-2}} \cdot \sqrt{\log(dn)} \cdot \sqrt{\log(md)}}{n^{\frac{1}{q-2}}\mathrm{SNR}^{\frac{q}{q-2}}}, 2\right\}^{-2} \cdot O(\tau^{1/2} \cdot d^{-1/2} \cdot \min\{\|\boldsymbol{\mu}\|_2^{-1}, \sigma_p^{-1}d^{-1/2}\})$, we have

$|\mathrm{sim}_i^{(t_0+1)}|, |\mathrm{sim}_{i,i'}^{(t_0+1)}| \leq \tau/4$. Since $|\exp(z) - 1| \leq 2|z|$ for all $z \leq 1$, we have

$$\begin{aligned} |\exp(\mathrm{sim}_i^{(t_0+1)}/\tau) - 1| &\leq 2|\mathrm{sim}_i^{(t_0+1)}/\tau| \\ &\leq \max\left\{288M^{\frac{1}{q-2}} \cdot \frac{\log(2/\sigma_0)^{\frac{1}{q-2}} \cdot \sqrt{\log(dn)} \cdot \sqrt{\log(md)}}{n^{\frac{1}{q-2}}\mathrm{SNR}^{\frac{q}{q-2}}}, 2\right\}^4 \cdot O(\sigma_0^2 d \cdot (\|\boldsymbol{\mu}\|_2^2 + 2\sigma_p^2 d)) \\ &\leq 1/2, \end{aligned} \tag{B.9}$$

and

$$\begin{aligned} |\exp(\mathrm{sim}_{i,i'}^{(t_0+1)}/\tau) - 1| &\leq 2|\mathrm{sim}_{i,i'}^{(t_0+1)}/\tau| \\ &\leq \max\left\{288M^{\frac{1}{q-2}} \cdot \frac{\log(2/\sigma_0)^{\frac{1}{q-2}} \cdot \sqrt{\log(dn)} \cdot \sqrt{\log(md)}}{n^{\frac{1}{q-2}}\mathrm{SNR}^{\frac{q}{q-2}}}, 2\right\}^4 \cdot O(\sigma_0^2 d \cdot (\|\boldsymbol{\mu}\|_2^2 + 2\sigma_p^2 d)) \\ &\leq 1/2 \end{aligned} \tag{B.10}$$

for all $i, i' \in [n]$ with $i \neq i'$. Therefore, for all $i, i' \in [n]$ with $i \neq i'$, we have

$$\begin{aligned} &\left|\frac{n_0 \cdot \exp(\mathrm{sim}_{i,i'}^{(t_0+1)}/\tau)}{\exp(\mathrm{sim}_i^{(t_0+1)}/\tau) + \sum_{i'' \neq i}\exp(\mathrm{sim}_{i,i'}^{(t_0+1)}/\tau)} - 1\right| \\ &\leq \left|\frac{n_0 \cdot [\exp(\mathrm{sim}_{i,i'}^{(t_0+1)}/\tau) - 1]}{\exp(\mathrm{sim}_i^{(t_0+1)}/\tau) + \sum_{i'' \neq i}\exp(\mathrm{sim}_{i,i'}^{(t_0+1)}/\tau)}\right| + \left|\frac{n_0}{\exp(\mathrm{sim}_i^{(t_0+1)}/\tau) + \sum_{i'' \neq i}\exp(\mathrm{sim}_{i,i'}^{(t_0+1)}/\tau)} - 1\right| \\ &\leq \left|\frac{n_0 \cdot [\exp(\mathrm{sim}_{i,i'}^{(t_0+1)}/\tau) - 1]}{\exp(\mathrm{sim}_i^{(t_0+1)}/\tau) + \sum_{i'' \neq i}\exp(\mathrm{sim}_{i,i'}^{(t_0+1)}/\tau)}\right| + \left|\frac{1 - \exp(\mathrm{sim}_i^{(t_0+1)}/\tau) + \sum_{i'' \neq i}[1 - \exp(\mathrm{sim}_{i,i'}^{(0)}/\tau)]}{\exp(\mathrm{sim}_i^{(t_0+1)}/\tau) + \sum_{i'' \neq i}\exp(\mathrm{sim}_{i,i'}^{(t_0+1)}/\tau)}\right| \\ &\leq \left|\frac{n_0 \cdot [\exp(\mathrm{sim}_{i,i'}^{(t_0+1)}/\tau) - 1]}{n_0/2}\right| + \left|\frac{1 - \exp(\mathrm{sim}_i^{(t_0+1)}/\tau) + \sum_{i'' \neq i}[1 - \exp(\mathrm{sim}_{i,i'}^{(t_0+1)}/\tau)]}{n_0/2}\right| \\ &\leq \max\left\{288M^{\frac{1}{q-2}} \cdot \frac{\log(2/\sigma_0)^{\frac{1}{q-2}} \cdot \sqrt{\log(dn)} \cdot \sqrt{\log(md)}}{n^{\frac{1}{q-2}}\mathrm{SNR}^{\frac{q}{q-2}}}, 2\right\}^4 \cdot O(\sigma_0^2 d \cdot (\|\boldsymbol{\mu}\|_2^2 + 2\sigma_p^2 d)), \end{aligned}$$

where the first inequality follows by triangle inequality, the third inequality applies (B.9) and (B.10) to the denominators, and the last inequality applies (B.9) and (B.10) to the numerators. Plugging the bound above into (B.8) then gives

$$\begin{aligned} \|\boldsymbol{\Xi}^{(t_0+1)}\|_2 &\leq \max\left\{288M^{\frac{1}{q-2}} \cdot \frac{\log(2/\sigma_0)^{\frac{1}{q-2}} \cdot \sqrt{\log(dn)} \cdot \sqrt{\log(md)}}{n^{\frac{1}{q-2}}\mathrm{SNR}^{\frac{q}{q-2}}}, 2\right\}^4 \cdot \frac{\eta}{\tau} \cdot O(\sigma_0^2 d \cdot (\|\boldsymbol{\mu}\|_2^2 + 2\sigma_p^2 d)^2) \\ &\leq \sigma_0 \cdot \frac{\eta}{\tau} \cdot \|\boldsymbol{\mu}\|_2^2 \leq \sigma_0 \cdot (1 - \mathcal{E}_{\mathrm{SimCLR}}) \cdot \frac{2\eta}{\tau} \cdot \|\boldsymbol{\mu}\|_2^2 \leq \sigma_0 \cdot \|\mathbf{A}\|_2, \end{aligned}$$

where the second inequality follows by the assumption that $\sigma_0 \leq \min\left\{288M^{\frac{1}{q-2}} \cdot \right.$

$\left.\frac{\log(2/\sigma_0)^{\frac{1}{q-2}} \cdot \sqrt{\log(dn)} \cdot \sqrt{\log(md)}}{n^{\frac{1}{q-2}}\mathrm{SNR}^{\frac{q}{q-2}}}, 2\right\}^{-4} \cdot O(d^{-1} \cdot \min\{\|\boldsymbol{\mu}\|_2^{-4}, (\sigma_p^2 d)^{-2}\} \cdot \|\boldsymbol{\mu}\|_2^2)$, the third inequality

is by the assumption that $\mathcal{E}_{\text{SimCLR}} \leq 1/2$, and the last inequality is by $(1 - \mathcal{E}_{\text{SimCLR}}) \cdot \frac{2\eta}{\tau} \cdot \|\boldsymbol{\mu}\|_2^2 \leq \|\mathbf{A}\|_2$ in Lemma 5.2. Therefore, by induction, we conclude that

$$\|\mathbf{\Xi}^{(t)}\|_2 \leq \sigma_0 \cdot \|\mathbf{A}\|_2$$

for all $t = 0, \ldots, T_{\text{SimCLR}}$. This finishes the proof. $\qquad\square$

### B.1.2 Proof of Lemma 5.2

In this section, the following Lemma B.2 is introduced to prove Lemma 5.2.

**Lemma B.2.** *Let* $\mathbf{A}$ *be the matrix defined in Lemma 5.1, and define* $\mathbf{A}_0 = \frac{2\eta}{n_0^2\tau}\left[n_0^2 - (\sum_{i=1}^{n_0} y_i)^2\right]\boldsymbol{\mu}\boldsymbol{\mu}^\top$, $\mathbf{\Delta} = \mathbf{A} - \mathbf{A}_0$. *Then it holds that*

$$\|\mathbf{\Delta}\|_2 \leq \left[\widetilde{O}(\text{SNR}^{-1} \cdot n_0^{-1}) + \widetilde{O}(\text{SNR}^{-1} \cdot \frac{1}{\sqrt{n_0}}) + \widetilde{O}(\text{SNR}^{-2} \cdot n_0^{-1})\right.$$

$$\left. + \widetilde{O}(\text{SNR}^{-2}) \cdot \max\left\{\sqrt{\frac{\log(9/\widetilde{\delta})}{dn_0}}, \frac{\log(9/\widetilde{\delta})}{n_0}\right\}\right]\frac{\eta}{\tau}\|\boldsymbol{\mu}\|_2^2.$$

Based on the Lemma B.2, we give the following proof of Lemma 5.2.

*Proof of Lemma 5.2.* Consider $\widehat{\mathbf{\Sigma}} = \mathbf{A}_0 = \frac{2\eta}{n_0^2\tau}\left[n_0^2 - (\sum_{i=1}^{n_0} y_i)^2\right]\boldsymbol{\mu}\boldsymbol{\mu}^\top$, $\mathbf{\Sigma} = \mathbf{A}$, and denote $\lambda_1 \geq \cdots \geq \lambda_d, \widehat{\lambda}_1 \geq \cdots \geq \widehat{\lambda}_d, \widetilde{\lambda}_1 \geq \cdots \geq \widetilde{\lambda}_d$ the eigenvalues of matrix $\mathbf{\Sigma}, \widehat{\mathbf{\Sigma}}$, and $\mathbf{\Delta} = \mathbf{\Sigma} - \widehat{\mathbf{\Sigma}}$ respectively. By Lemma B.2, we have

$$\|\mathbf{\Delta}\|_2 \leq \left[\widetilde{O}(\text{SNR}^{-1} \cdot n_0^{-1}) + \widetilde{O}(\text{SNR}^{-1} \cdot \frac{1}{\sqrt{n_0}}) + \widetilde{O}(\text{SNR}^{-2} \cdot n_0^{-1}) + c_3 \cdot \text{SNR}^{-2} \cdot \frac{\log(9/\delta)}{n_0}\right] \cdot \frac{\eta}{\tau}\|\boldsymbol{\mu}\|_2^2.$$

Denote

$$U = \widetilde{O}(\text{SNR}^{-1} \cdot n_0^{-1}) + \widetilde{O}(\text{SNR}^{-1} \cdot \frac{1}{\sqrt{n_0}}) + \widetilde{O}(\text{SNR}^{-2} \cdot n_0^{-1}) + c_3 \cdot \text{SNR}^{-2} \cdot \frac{\log(9/\delta)}{n_0},$$

then by assumption we have $U = \widetilde{O}(\text{SNR}^{-1} \cdot n_0^{-1/2})$ and $U \leq 1/2$. Then we have

$$\lambda_1 \geq \|A_0\|_2 - \|\mathbf{\Delta}\|_2$$

$$= \frac{2\eta}{n_0^2\tau}\left[n_0^2 - \left(\sum_{i=1}^{n_0} y_i\right)^2\right] \cdot \|\boldsymbol{\mu}\|_2^2 - \|\mathbf{\Delta}\|_2$$

$$\geq \frac{2\eta}{\tau} \cdot \left(1 - \frac{2\log(2/\widetilde{\delta})}{n_0}\right) \cdot \|\boldsymbol{\mu}\|_2^2 - \|\mathbf{\Delta}\|_2$$

$$\geq \left(1 - \frac{2\log(2/\widetilde{\delta})}{n_0} - \frac{U}{2}\right) \cdot \frac{2\eta}{\tau}\|\boldsymbol{\mu}\|_2^2, \tag{B.11}$$

where the second inequality is by $|\sum_{i=1}^{n_0} y_i| \leq \sqrt{2n_0\log(2/\widetilde{\delta})}$ in Lemma B.5, and the last inequality is by Lemma B.2. This proves the lower bound of $\lambda_1$. Similarly, for the upper bound, we have that

$$\lambda_1 \leq \|A_0\|_2 + \|\mathbf{\Delta}\|_2 \leq \frac{2\eta}{\tau}\|\boldsymbol{\mu}\|_2^2 + \|\mathbf{\Delta}\|_2 = (2 + U) \cdot \frac{\eta}{\tau} \cdot \|\boldsymbol{\mu}\|_2^2, \tag{B.12}$$

where the second inequality is by $\widehat{\lambda}_1 = \frac{2\eta}{n_0^2\tau}\left[n_0^2 - (\sum_{i=1}^{n_0} y_i)^2\right]\|\boldsymbol{\mu}\|_2^2 \leq \frac{2\eta}{\tau}\|\boldsymbol{\mu}\|_2^2$ and the upper bound of $\|\mathbf{\Delta}\|_2$ in Lemma B.2.

By the variant of Davis-Kahan Theorem (Theorem 1 in Yu et al. (2015)), we have that

$$
\sin\theta(\widehat{\mathbf{v}}_1, \mathbf{v}_1) \leq \frac{\|\mathbf{\Sigma} - \widehat{\mathbf{\Sigma}}\|_2}{|\widehat{\lambda}_2 - \lambda_1|}
$$

$$
\overset{(i)}{=} \frac{\|\mathbf{\Sigma} - \widehat{\mathbf{\Sigma}}\|_2}{|\lambda_1|}
$$

$$
\overset{(ii)}{\leq} \frac{\|\mathbf{\Delta}\|_2}{\frac{\eta}{\tau}\|\boldsymbol{\mu}\|_2^2 - \|\mathbf{\Delta}\|_2}
$$

$$
= \frac{1}{\frac{\eta}{\tau}\|\boldsymbol{\mu}\|_2^2/\|\mathbf{\Delta}\|_2 - 1}
$$

$$
\overset{(iii)}{\leq} \frac{U}{1 - U}, \tag{B.13}
$$

where $(i)$ is by $\widehat{\lambda}_2 = 0$ based on the definition of $\widehat{\mathbf{\Sigma}}$, and $(ii)$ is by the lower bound of $\lambda_1$ in (B.11), and $(iii)$ is by the upper bound of $\|\mathbf{\Delta}\|_2$ in Lemma B.2. Since $\widehat{\mathbf{v}}_1 = \boldsymbol{\mu}$ by the definition of $\mathbf{A}_0$, denote $\bar{\boldsymbol{\mu}} = (1/\|\boldsymbol{\mu}\|)\boldsymbol{\mu}$, it follows that

$$
\langle \mathbf{v}_1, \bar{\boldsymbol{\mu}} \rangle = \cos\theta(\boldsymbol{\mu}, \mathbf{v}_1) \geq 1 - \sin^2\theta(\boldsymbol{\mu}, \mathbf{v}_1) \geq \frac{1 - 2U}{(1 - U)^2}, \tag{B.14}
$$

where without loss of generality, we assume $\langle \mathbf{v}_1, \bar{\boldsymbol{\mu}} \rangle > 0$, and the second inequality is by (B.13). Therefore,

$$
\langle \mathbf{v}_1, \boldsymbol{\mu} \rangle = \|\boldsymbol{\mu}\|_2 \langle \mathbf{v}_1, \bar{\boldsymbol{\mu}} \rangle \geq \frac{1 - 2U}{(1 - U)^2}\|\boldsymbol{\mu}\|_2. \tag{B.15}
$$

Moreover, by definition, we also have

$$
\|\mathbf{P}_{\boldsymbol{\mu}}^{\perp}\mathbf{v}_1\|_2^2 = \left\| \mathbf{v}_1 - \langle \mathbf{v}_1, \bar{\boldsymbol{\mu}} \rangle \cdot \bar{\boldsymbol{\mu}} \right\|_2^2
$$

$$
= 1 - 2\langle \mathbf{v}_1, \bar{\boldsymbol{\mu}} \rangle^2 + \langle \mathbf{v}_1, \bar{\boldsymbol{\mu}} \rangle^2
$$

$$
= 1 - \langle \mathbf{v}_1, \bar{\boldsymbol{\mu}} \rangle^2
$$

$$
\leq 1 - \left[ \frac{1 - 2U}{(1 - U)^2} \right]^2
$$

$$
= \frac{U^2(U^2 - 4U + 2)}{(1 - U)^4}
$$

$$
\leq 32U^2,
$$

where the first inequality follows by (B.14), and the second inequality follows by the assumption that $0 \leq U \leq 1/2$. Therefore, we have

$$
\|\mathbf{P}_{\boldsymbol{\mu}}^{\perp}\mathbf{v}_1\|_2 \leq 4\sqrt{2} \cdot U. \tag{B.16}
$$

Finally, we prove the upper bound of $\max_{i \geq 2} \lambda_i$. Since we have

$$
\|\mathbf{A} - \lambda_1 \mathbf{v}_1 \mathbf{v}_1^{\top}\|_2 \leq \|\mathbf{A} - \mathbf{A}_0\|_2 + \|\mathbf{A}_0 - \lambda_1 \mathbf{v}_1 \mathbf{v}_1^{\top}\|_2
$$

$$
= \|\mathbf{\Delta}\|_2 + \|\widehat{\lambda}_1 \bar{\boldsymbol{\mu}} \bar{\boldsymbol{\mu}}^{\top} - \widehat{\lambda}_1 \bar{\boldsymbol{\mu}} \mathbf{v}_1^{\top} + \widehat{\lambda}_1 \bar{\boldsymbol{\mu}} \mathbf{v}_1^{\top} - \widehat{\lambda}_1 \mathbf{v}_1 \mathbf{v}_1^{\top} + \widehat{\lambda}_1 \mathbf{v}_1 \mathbf{v}_1^{\top} - \lambda_1 \mathbf{v}_1 \mathbf{v}_1^{\top}\|_2
$$

$$
\leq \|\mathbf{\Delta}\|_2 + 2\widehat{\lambda}_1 \|\bar{\boldsymbol{\mu}} - \mathbf{v}_1\|_2 + \|\mathbf{\Delta}\|_2
$$

$$
\overset{(i)}{\leq} 2U\frac{\eta}{\tau}\|\boldsymbol{\mu}\|_2^2 + \frac{4\eta}{\tau}\|\boldsymbol{\mu}\|_2^2 \frac{\sqrt{2}U}{1 - U}
$$

$$
= \left( 2U + \frac{4\sqrt{2}U}{1 - U} \right) \frac{\eta}{\tau}\|\boldsymbol{\mu}\|_2^2, \tag{B.17}
$$

where $(i)$ is by $\|\bar{\boldsymbol{\mu}} - \mathbf{v}_1\|_2^2 = \|\bar{\boldsymbol{\mu}}\|_2^2 + \|\mathbf{v}_1\|_2^2 - 2\langle \mathbf{v}_1, \bar{\boldsymbol{\mu}} \rangle = 2(1 - \langle \mathbf{v}_1, \bar{\boldsymbol{\mu}} \rangle) \leq 2[1 - \frac{1 - 2U}{(1 - U)^2}] = 2U^2/(1 - U)^2$ based on (B.14) and $\widehat{\lambda}_1 = \frac{2\eta}{n_0^2\tau}\left[ n_0^2 - (\sum_{i=1}^{n_0} y_i)^2 \right]\|\boldsymbol{\mu}\|_2^2 \leq \frac{2\eta}{\tau}\|\boldsymbol{\mu}\|_2^2$, and the bound of $\|\mathbf{\Delta}\|_2$ in Lemma B.2.

Now, combining the conclusions in (B.11), (B.12), (B.15), (B.16), and (B.17), we see that by setting $\mathcal{E}_{\text{SimCLR}} = \Theta(U + n_0^{-1}\log(2/\widetilde{\delta})) = \widetilde{O}(\max\{\text{SNR}^{-1}n_0^{-1/2}, n_0^{-1}\})$, all the conclusions in Lemma 5.2 hold. $\qquad\square$

### B.1.3 PROOF OF THEOREM 5.3

In this section, the following Lemmas B.3 and B.4 are introduced to prove Theorem 5.3.

**Lemma B.3.** *Let $\mathbf{A}$ be the matrix defined in Lemma 5.1, and let $\lambda_i$, $\mathbf{v}_i$, $i \in [d]$ be the eigenvalues and eigenvectors of $\mathbf{A}$ respectively. Suppose that $d \geq \Omega(\log(2mn_0/\delta))$, $m = \Omega(\log(1/\delta))$. Then with probability at least $1 - \delta$, it holds that*

$$|\langle \mathbf{w}_r^{(0)}, \mathbf{v}_i \rangle| \leq \sqrt{2 \log(16m/\delta)} \cdot \sigma_0$$

*for all $r \in [2m]$ and all $i \in [d]$. Moreover, there exist disjoint index sets $\mathcal{I}^+, \mathcal{I}^- \subseteq [2m]$ with $|\mathcal{I}^+| = |\mathcal{I}^-| = 2m/5$ such that*

$$\langle \mathbf{w}_r^{(0)}, \mathbf{v}_1 \rangle \geq \sigma_0/2 \text{ for all } r \in \mathcal{I}^+, \qquad \langle \mathbf{w}_r^{(0)}, \mathbf{v}_1 \rangle \leq -\sigma_0/2 \text{ for all } r \in \mathcal{I}^-.$$

**Lemma B.4.** *Let $\mathbf{A}$ be the matrix defined in Lemma 5.1, and let $\lambda_i$, $\mathbf{v}_i$, $i \in [d]$ be the eigenvalues and eigenvectors of $\mathbf{A}$ respectively. Let $\mathbf{\Xi}$ be any symmetric matrix with $\|\mathbf{\Xi}\|_2 \leq \sigma_0 \cdot \|\mathbf{A}\|_2$, and let $\widetilde{\lambda}_i$, $\widehat{\mathbf{v}}_i$, $i \in [d]$ be the eigenvalues and eigenvectors of $\mathbf{A} + \mathbf{\Xi}$ respectively. Suppose that $\mathcal{E}_{\mathrm{SimCLR}} \leq 1/4$ and $\sigma_0 \leq 1/4$. Then the following results hold:*

- $|\lambda_i - \widetilde{\lambda}_i| \leq \sigma_0 \cdot \|\mathbf{A}\|_2$, $i \in [n]$.

- $|\langle \mathbf{v}_1, \widetilde{\mathbf{v}}_1 \rangle| \geq 1 - 4\sigma_0^2$.

- $|\langle \mathbf{v}_1, \widetilde{\mathbf{v}}_i \rangle| \leq 4\sigma_0$, $i \geq 2$.

- *Let $\mathbf{P}_{\mathbf{v}_1}^\perp = \mathbf{I} - \mathbf{v}_1 \mathbf{v}_1^\top$, then $\|\mathbf{P}_{\mathbf{v}_1}^\perp \widetilde{\mathbf{v}}_1\|_2 \leq 4\sigma_0$.*

Therefore, based on the Lemmas B.3, B.4, 5.1, and 5.2, the proof of Theorem 5.3 is presented as follows.

*Proof of Theorem 5.3.* Denote

$$\mathcal{X}_{\mathrm{col}} = \mathrm{span}\{\boldsymbol{\mu}, \boldsymbol{\xi}_1, \ldots, \boldsymbol{\xi}_{n_0}, \widetilde{\boldsymbol{\xi}}_1, \ldots, \widetilde{\boldsymbol{\xi}}_{n_0}\}, \qquad \mathcal{X}_{\mathrm{row}} = \mathrm{span}\{\boldsymbol{\mu}^\top, \boldsymbol{\xi}_1^\top, \ldots, \boldsymbol{\xi}_{n_0}^\top, \widetilde{\boldsymbol{\xi}}_1^\top, \ldots, \widetilde{\boldsymbol{\xi}}_{n_0}^\top\}.$$

Moreover, let $\mathbf{e}_1, \ldots, \mathbf{e}_{2n_0+1}$ be a set of orthogonal bases in $\mathcal{X}_{\mathrm{col}}$, and let $\bar{\boldsymbol{\mu}} = \boldsymbol{\mu}/\|\boldsymbol{\mu}\|_2$, $\mathbf{P}_{\mathcal{X}} = \sum_{i=1}^{2n_0+1} \mathbf{e}_i \mathbf{e}_i^\top$, $\mathbf{P}_{\mathcal{X}, \mathbf{v}_1}^\perp = \sum_{i=1}^{2n_0+1} \mathbf{e}_i \mathbf{e}_i^\top - \mathbf{v}_1 \mathbf{v}_1^\top$.

By Lemma 5.1, for $t = 0, \ldots, T_{\mathrm{SimCLR}}$, we have

$$\mathbf{w}_r^{(t+1)} = \mathbf{w}_r^{(t)} + (\mathbf{A} + \mathbf{\Xi}^{(t)}) \mathbf{w}_r^{(t)}, \text{ with } \|\mathbf{\Xi}^{(t)}\|_2 \leq \sigma_0 \cdot \|\mathbf{A}\|_2,$$

where the columns and rows of $\mathbf{\Xi}^{(t)}$ are in $\mathcal{X}_{\mathrm{col}}$ and $\mathcal{X}_{\mathrm{row}}$ respectively. By definition, it is clear that the columns and rows of $\mathbf{A}$ are also in $\mathcal{X}_{\mathrm{col}}$ and $\mathcal{X}_{\mathrm{row}}$ respectively. Therefore, we see that the rank of $\mathbf{A} + \mathbf{\Xi}^{(t)}$ is at most $2n_0 + 1$. Denote by $\lambda_1^{(t)}, \ldots, \lambda_{2n_0+1}^{(t)}$ and $\mathbf{v}_1^{(t)}, \ldots, \mathbf{v}_1^{(2n_0+1)}$ the first $2n_0 + 1$ eigenvalues and eigenvectors of $\mathbf{A} + \mathbf{\Xi}^{(t)}$ respectively. Since $\mathbf{v}_1^{(t)}$ and $-\mathbf{v}_1^{(t)}$ are both the first eigenvector of $\mathbf{A} + \mathbf{\Xi}^{(t)}$, without loss of generality, we can assume that $\langle \mathbf{v}_1^{(t)}, \mathbf{v}_1 \rangle \geq 0$ for all $t \geq 0$.

By Lemma B.3, there exist disjoint index sets $\mathcal{I}^+, \mathcal{I}^- \subseteq [2m]$ with $|\mathcal{I}^+| = |\mathcal{I}^-| = 2m/5$ such that

$$\langle \mathbf{w}_r^{(0)}, \mathbf{v}_1 \rangle \geq \sigma_0/2 \text{ for all } r \in \mathcal{I}^+, \qquad \langle \mathbf{w}_r^{(0)}, \mathbf{v}_1 \rangle \leq -\sigma_0/2 \text{ for all } r \in \mathcal{I}^-. \tag{B.18}$$

Note that $\|\mathbf{P}_{\mathcal{X}, \mathbf{v}_1}^\perp \mathbf{w}_r^{(0)}\|_2$ is essentially the Euclidean norm of a $(2n_0)$-dimensional Gaussian random vector with independent entries from $\mathcal{N}(0, \sigma_0^2)$. Therefore, by Bernstein's inequality, with probability at least $1 - d^{-2}$, we have

$$n_0 \sigma_0^2 \leq \|\mathbf{P}_{\mathcal{X}, \mathbf{v}_1}^\perp \mathbf{w}_r^{(0)}\|_2^2 \leq 4n_0 \sigma_0^2$$

for all $r \in [2m]$. Therefore, we have

$$\sigma_0 \cdot \sqrt{n_0} \leq \|\mathbf{P}_{\mathcal{X}, \mathbf{v}_1}^\perp \mathbf{w}_r^{(0)}\|_2 \leq 2\sigma_0 \cdot \sqrt{n_0}. \tag{B.19}$$

Similarly, with probability at least $1 - d^{-2}$, we also have

$$\|\mathbf{P}_{\mathcal{X}}^{\top}\mathbf{w}_r^{(0)}\|_2 \leq 4\sigma_0 \cdot \sqrt{n_0}. \tag{B.20}$$

In the following, we use induction to prove the following results:

$$\langle \mathbf{v}_1, \mathbf{w}_r^{(t)} \rangle \geq 0, \ r \in \mathcal{I}^+, \tag{B.21}$$

$$\langle \mathbf{v}_1, \mathbf{w}_r^{(t)} \rangle \geq \sqrt{\sigma_0} \cdot \|\mathbf{P}_{\mathcal{X}, \mathbf{v}_1}^{\perp}\mathbf{w}_r^{(t)}\|_2, \ r \in \mathcal{I}^+, \tag{B.22}$$

$$\|\mathbf{P}_{\mathcal{X}, \mathbf{v}_1}^{\perp}\mathbf{w}_r^{(t)}\|_2 \geq \sqrt{\sigma_0} \cdot \langle \mathbf{v}_1, \mathbf{w}_r^{(t)} \rangle, \ r \in [2m]. \tag{B.23}$$

We first check that (B.21), (B.22) and (B.23) hold for $t = 0$. We see that (B.21) directly follows by (B.18). By (B.18), we have

$$\langle \mathbf{v}_1, \mathbf{w}_r^{(0)} \rangle \geq \sigma_0/2 \geq 2\sigma_0^{3/2} \cdot \sqrt{n_0} \geq \sqrt{\sigma_0} \cdot \|\mathbf{P}_{\mathcal{X}, \mathbf{v}_1}^{\perp}\mathbf{w}_r^{(0)}\|_2$$

for all $r \in \mathcal{I}^+$, where the second inequality follows by the assumption that $\sigma_0 \leq 1/(16n_0)$, and the third inequality follows by (B.19). Similarly, we have

$$\|\mathbf{P}_{\mathcal{X}, \mathbf{v}_1}^{\perp}\mathbf{w}_r^{(0)}\|_2 \geq \sigma_0 \cdot \sqrt{n_0} \geq 4\sigma_0^{3/2} \cdot \sqrt{n_0} \geq \sqrt{\sigma_0} \cdot \|\mathbf{P}_{\mathcal{X}}\mathbf{w}_r^{(0)}\|_2 \geq \sqrt{\sigma_0} \cdot \langle \mathbf{v}_1, \mathbf{w}_r^{(0)} \rangle$$

for all $r \in [2m]$, where the first inequality follows by (B.19), the second inequality follows by the assumption that $\sigma_0 \leq 1/16$, and the third inequality follows by (B.20). Thus, we have verified all the induction hypotheses at $t = 0$.

Now suppose that (B.21), (B.22) and (B.23) hold for all $t = 0, 1, \ldots, t_0$, where $t_0 \leq T_{\text{SimCLR}} - 1$. Then by Lemma 5.1, for $t = 0, \ldots, T_{\text{SimCLR}}$, we have

$$\mathbf{w}_r^{(t+1)} = \mathbf{w}_r^{(t)} + (\mathbf{A} + \boldsymbol{\Xi}^{(t)})\mathbf{w}_r^{(t)}.$$

Then for $t = 0, \ldots, t_0$ and $r \in \mathcal{I}^+$, we have

$$\langle \mathbf{w}_r^{(t+1)}, \mathbf{v}_1 \rangle = \langle \mathbf{w}_r^{(t)}, \mathbf{v}_1 \rangle + \mathbf{v}_1^{\top}(\mathbf{A} + \boldsymbol{\Xi}^{(t)})\mathbf{w}_r^{(t)}$$

$$= \langle \mathbf{w}_r^{(t)}, \mathbf{v}_1 \rangle + \sum_{i=1}^{2n_0+1} \lambda_i^{(t)}\mathbf{v}_1^{\top}\mathbf{v}_i^{(t)}\mathbf{v}_i^{(t)\top}\mathbf{w}_r^{(t)}$$

$$= \langle \mathbf{w}_r^{(t)}, \mathbf{v}_1 \rangle + \lambda_1^{(t)}\mathbf{v}_1^{\top}\mathbf{v}_1^{(t)}\mathbf{v}_1^{(t)\top}\mathbf{w}_r^{(t)} + \sum_{i=2}^{2n_0+1} \lambda_i^{(t)}\mathbf{v}_1^{\top}\mathbf{v}_i^{(t)}\mathbf{v}_i^{(t)\top}\mathbf{w}_r^{(t)}$$

$$= \langle \mathbf{w}_r^{(t)}, \mathbf{v}_1 \rangle + \lambda_1^{(t)}\mathbf{v}_1^{\top}\mathbf{v}_1^{(t)}\mathbf{v}_1^{(t)\top}\mathbf{P}_{\mathcal{X}}\mathbf{w}_r^{(t)} + \sum_{i=2}^{2n_0+1} \lambda_i^{(t)}\mathbf{v}_1^{\top}\mathbf{v}_i^{(t)}\mathbf{v}_i^{(t)\top}\mathbf{P}_{\mathcal{X}}\mathbf{w}_r^{(t)}$$

$$= \langle \mathbf{w}_r^{(t)}, \mathbf{v}_1 \rangle + \lambda_1^{(t)}\mathbf{v}_1^{\top}\mathbf{v}_1^{(t)}\mathbf{v}_1^{(t)\top}(\mathbf{P}_{\mathcal{X}, \mathbf{v}_1}^{\perp} + \mathbf{v}_1\mathbf{v}_1^{\top})\mathbf{w}_r^{(t)}$$

$$\quad + \sum_{i=2}^{2n_0+1} \lambda_i^{(t)}\mathbf{v}_1^{\top}\mathbf{v}_i^{(t)}\mathbf{v}_i^{(t)\top}(\mathbf{P}_{\mathcal{X}, \mathbf{v}_1}^{\perp} + \mathbf{v}_1\mathbf{v}_1^{\top})\mathbf{w}_r^{(t)}$$

$$= (1 + \lambda_1) \cdot \langle \mathbf{w}_r^{(t)}, \mathbf{v}_1 \rangle + I_1 + I_2 + I_3 + I_4, \tag{B.24}$$

where the fourth equality follows by the fact that $\mathbf{v}_i^{t_0} \in \mathcal{X}_{\text{col}}$, and

$$I_1 = \lambda_1^{(t)}\mathbf{v}_1^{\top}\mathbf{v}_1^{(t)}\mathbf{v}_1^{(t)\top}\mathbf{v}_1\mathbf{v}_1^{\top}\mathbf{w}_r^{(t)} - \lambda_1 \cdot \langle \mathbf{w}_r^{(t)}, \mathbf{v}_1 \rangle,$$

$$I_2 = \lambda_1^{(t)}\mathbf{v}_1^{\top}\mathbf{v}_1^{(t)}\mathbf{v}_1^{(t)\top}\mathbf{P}_{\mathcal{X}, \mathbf{v}_1}^{\perp}\mathbf{w}_r^{(t)},$$

$$I_3 = \sum_{i=2}^{2n_0+1} \lambda_i^{(t)}\mathbf{v}_1^{\top}\mathbf{v}_i^{(t)}\mathbf{v}_i^{(t)\top}\mathbf{v}_1\mathbf{v}_1^{\top}\mathbf{w}_r^{(t)},$$

$$I_4 = \sum_{i=2}^{2n_0+1} \lambda_i^{(t)}\mathbf{v}_1^{\top}\mathbf{v}_i^{(t)}\mathbf{v}_i^{(t)\top}\mathbf{P}_{\mathcal{X}, \mathbf{v}_1}^{\perp}\mathbf{w}_r^{(t)}.$$

We give upper bounds of $|I_1|$, $|I_2|$, $|I_3|$ and $|I_4|$. By the induction hypothesis that $\mathbf{v}_1^\top \mathbf{w}_r^{(t)} \geq 0$, we have

$$
\begin{aligned}
I_1 &\geq (1 - \sigma_0) \cdot \lambda_1 \cdot (1 - 4\sigma_0^2)^2 \cdot \langle \mathbf{w}_r^{(t)}, \mathbf{v}_1 \rangle - \lambda_1 \cdot \langle \mathbf{w}_r^{(t)}, \mathbf{v}_1 \rangle \\
&\geq (1 - 2\sigma_0) \cdot \lambda_1 \cdot \langle \mathbf{w}_r^{(t)}, \mathbf{v}_1 \rangle - \lambda_1 \cdot \langle \mathbf{w}_r^{(t)}, \mathbf{v}_1 \rangle \\
&\geq -2\sigma_0 \lambda_1 \cdot \langle \mathbf{w}_r^{(t)}, \mathbf{v}_1 \rangle,
\end{aligned}
$$

where the first inequality follows by Lemma B.4. Similarly, we also

$$
I_1 \leq (1 + \sigma_0) \cdot \lambda_1 \cdot \langle \mathbf{w}_r^{(t)}, \mathbf{v}_1 \rangle - \lambda_1 \cdot \langle \mathbf{w}_r^{(t)}, \mathbf{v}_1 \rangle \leq \sigma_0 \lambda_1 \cdot \langle \mathbf{w}_r^{(t)}, \mathbf{v}_1 \rangle.
$$

Therefore, we conclude that

$$
|I_1| \leq 2\sigma_0 \lambda_1 \cdot \langle \mathbf{w}_r^{(t)}, \mathbf{v}_1 \rangle. \tag{B.25}
$$

Let $\mathbf{P}_{\mathbf{v}_1}^\perp = \mathbf{I} - \mathbf{v}_1 \mathbf{v}_1^\top$. Then for $I_2$, by the property of project matrices, we have $\mathbf{P}_{\mathcal{X}, \mathbf{v}_1}^\perp = \mathbf{P}_{\mathbf{v}_1}^\perp \mathbf{P}_{\mathcal{X}, \mathbf{v}_1}^\perp$, and therefore

$$
\begin{aligned}
|I_2| &= |\lambda_1^{(t)} \mathbf{v}_1^\top \mathbf{v}_1^{(t)} (\mathbf{P}_{\mathbf{v}_1}^\perp \mathbf{v}_1^{(t)})^\top \mathbf{P}_{\mathcal{X}, \mathbf{v}_1}^\perp \mathbf{w}_r^{(t)}| \\
&\leq (1 + \sigma_0) \lambda_1 \cdot 4\sigma_0 \cdot \|\mathbf{P}_{\mathcal{X}, \mathbf{v}_1}^\perp \mathbf{w}_r^{(t)}\|_2 \\
&\leq 8\sigma_0 \lambda_1 \cdot \|\mathbf{P}_{\mathcal{X}, \mathbf{v}_1}^\perp \mathbf{w}_r^{(t)}\|_2 \\
&\leq 8\sqrt{\sigma_0} \cdot \lambda_1 \cdot \langle \mathbf{w}_r^{(t)}, \mathbf{v}_1 \rangle, \tag{B.26}
\end{aligned}
$$

where the first inequality follows by Lemma B.4, and the third inequality follows by the induction hypothesis. For $I_3$, we have

$$
\begin{aligned}
|I_3| &\leq \sum_{i=2}^{2n_0+1} |\lambda_i^{(t)}| \cdot |\mathbf{v}_1^\top \mathbf{v}_i^{(t)}|^2 \cdot \mathbf{v}_1^\top \mathbf{w}_r^{(t)} \\
&\leq 2n_0 \cdot \left( \frac{\mathcal{E}_{\mathrm{SimCLR}}}{2(1 - \mathcal{E}_{\mathrm{SimCLR}})} + \sigma_0 \right) \cdot \lambda_1 \cdot 16\sigma_0^2 \cdot \langle \mathbf{w}_r^{(t)}, \mathbf{v}_1 \rangle \\
&\leq n_0 \sigma_0^2 \lambda_1 \cdot \langle \mathbf{w}_r^{(t)}, \mathbf{v}_1 \rangle, \tag{B.27}
\end{aligned}
$$

where in the second inequality we use Lemmas 5.2 and B.4, and the last inequality follows by the assumption that $\sigma_0, \mathcal{E}_{\mathrm{SimCLR}} \leq 1/64$. Finally for $I_4$, we have

$$
\begin{aligned}
|I_4| &\leq \left\| \sum_{i=2}^{2n_0+1} \lambda_i^{(t)} \cdot \langle \mathbf{v}_1, \mathbf{v}_i^{(t)} \rangle \cdot \mathbf{v}_i^{(t)\top} \right\|_2 \cdot \|\mathbf{P}_{\mathcal{X}, \mathbf{v}_1}^\perp \mathbf{w}_r^{(t)}\|_2 \\
&\leq \sqrt{\sum_{i=2}^{2n_0+1} \lambda_i^{(t)2} \cdot \langle \mathbf{v}_1, \mathbf{v}_i^{(t)} \rangle^2} \cdot \|\mathbf{P}_{\mathcal{X}, \mathbf{v}_1}^\perp \mathbf{w}_r^{(t)}\|_2 \\
&\leq \sqrt{2n_0} \cdot \left( \frac{\mathcal{E}_{\mathrm{SimCLR}}}{2(1 - \mathcal{E}_{\mathrm{SimCLR}})} + \sigma_0 \right) \cdot \lambda_1 \cdot 4\sigma_0 \cdot \|\mathbf{P}_{\mathcal{X}, \mathbf{v}_1}^\perp \mathbf{w}_r^{(t)}\|_2 \\
&\leq \sqrt{n_0} \cdot \sigma_0 \lambda_1 \cdot \|\mathbf{P}_{\mathcal{X}, \mathbf{v}_1}^\perp \mathbf{w}_r^{(t)}\|_2 \\
&\leq \sqrt{n_0 \sigma_0} \cdot \lambda_1 \cdot \langle \mathbf{w}_r^{(t)}, \mathbf{v}_1 \rangle, \tag{B.28}
\end{aligned}
$$

where the second inequality follows by the fact that $\mathbf{v}_i^{(t)}$, $i = 2, \ldots, 2n_0 + 1$ are mutually orthogonal unit vectors, the third inequality follows by Lemmas 5.2 and B.4, the fourth inequality follows by the assumption that $\sigma_0, \mathcal{E}_{\mathrm{SimCLR}} \leq 1/64$, and the fifth inequality follows by the induction hypothesis.

Now plugging (B.25), (B.26), (B.27) and (B.28) into (B.24), we obtain

$$
[1 + (1 - 2\sqrt{n_0 \sigma_0})\lambda_1] \cdot \langle \mathbf{w}_r^{(t)}, \mathbf{v}_1 \rangle \leq \langle \mathbf{w}_r^{(t+1)}, \mathbf{v}_1 \rangle \leq [1 + (1 + 2\sqrt{n_0 \sigma_0})\lambda_1] \cdot \langle \mathbf{w}_r^{(t)}, \mathbf{v}_1 \rangle \quad \text{(B.29)}
$$

for all $t = 0, \ldots, t_0$ and $r \in \mathcal{I}^+$, where we use the assumption that $\sigma_0 \leq 1/(8n_0^2)$ and $\sigma_0 \leq 1/512$.

Moreover, for $t = 0, \ldots, t_0$ and $r \in [2m]$, we also have

$$
\begin{aligned}
\mathbf{P}_{\mathcal{X}, \mathbf{v}_1}^{\perp} \mathbf{w}_r^{(t+1)} &= \mathbf{P}_{\mathcal{X}, \mathbf{v}_1}^{\perp} \mathbf{w}_r^{(t)} + \mathbf{P}_{\mathcal{X}, \mathbf{v}_1}^{\perp} (\mathbf{A} + \mathbf{\Xi}^{(t)}) \mathbf{w}_r^{(t)} \\
&= \mathbf{P}_{\mathcal{X}, \mathbf{v}_1}^{\perp} \mathbf{w}_r^{(t)} + \lambda_1^{(t)} \cdot \mathbf{P}_{\mathcal{X}, \mathbf{v}_1}^{\perp} \mathbf{v}_1^{(t)} \mathbf{v}_1^{(t)\top} \mathbf{w}_r^{(t)} + \mathbf{P}_{\mathcal{X}, \mathbf{v}_1}^{\perp} \left( \sum_{i=2}^{2n_0+1} \lambda_i^{(t)} \mathbf{v}_i^{(t)} \mathbf{v}_i^{(t)\top} \right) \mathbf{w}_r^{(t)} \\
&= \mathbf{P}_{\mathcal{X}, \mathbf{v}_1}^{\perp} \mathbf{w}_r^{(t)} + I_5 + I_6 + I_7 + I_8,
\end{aligned}
\tag{B.30}
$$

where

$$
\begin{aligned}
I_5 &= \lambda_1^{(t)} \cdot \mathbf{P}_{\mathcal{X}, \mathbf{v}_1}^{\perp} \mathbf{v}_1^{(t)} \mathbf{v}_1^{(t)\top} \mathbf{P}_{\mathcal{X}, \mathbf{v}_1}^{\perp} \mathbf{w}_r^{(t)}, \\
I_6 &= \lambda_1^{(t)} \cdot \mathbf{P}_{\mathcal{X}, \mathbf{v}_1}^{\perp} \mathbf{v}_1^{(t)} \mathbf{v}_1^{(t)\top} \mathbf{v}_1 \mathbf{v}_1^{\top} \mathbf{w}_r^{(t)}, \\
I_7 &= \mathbf{P}_{\mathcal{X}, \mathbf{v}_1}^{\perp} \left( \sum_{i=2}^{2n_0+1} \lambda_i^{(t)} \mathbf{v}_i^{(t)} \mathbf{v}_i^{(t)\top} \right) \mathbf{P}_{\mathcal{X}, \mathbf{v}_1}^{\perp} \mathbf{w}_r^{(t)}, \\
I_8 &= \mathbf{P}_{\mathcal{X}, \mathbf{v}_1}^{\perp} \left( \sum_{i=2}^{2n_0+1} \lambda_i^{(t)} \mathbf{v}_i^{(t)} \mathbf{v}_i^{(t)\top} \right) \mathbf{v}_1 \mathbf{v}_1^{\top} \mathbf{w}_r^{(t)}.
\end{aligned}
$$

For $I_5$, we have

$$
\begin{aligned}
|I_5| &\leq (1 + \sigma_0) \lambda_1 \cdot 16\sigma_0^2 \cdot \| \mathbf{P}_{\mathcal{X}, \mathbf{v}_1}^{\perp} \mathbf{w}_r^{(t)} \|_2 \\
&\leq 32 \lambda_1 \sigma_0^2 \cdot \| \mathbf{P}_{\mathcal{X}, \mathbf{v}_1}^{\perp} \mathbf{w}_r^{(t)} \|_2,
\end{aligned}
\tag{B.31}
$$

where the first inequality follows by Lemma B.4. For $I_6$, we have

$$
\begin{aligned}
|I_6| &\leq (1 + \sigma_0) \lambda_1 \cdot 4\sigma_0 \cdot \langle \mathbf{w}_r^{(t)}, \mathbf{v}_1 \rangle \\
&\leq 8 \lambda_1 \sigma_0 \cdot \langle \mathbf{w}_r^{(t)}, \mathbf{v}_1 \rangle \\
&\leq 8 \lambda_1 \sqrt{\sigma_0} \cdot \| \mathbf{P}_{\mathcal{X}, \mathbf{v}_1}^{\perp} \mathbf{w}_r^{(t)} \|_2,
\end{aligned}
\tag{B.32}
$$

where the first inequality follows by Lemma B.4, and the third inequality follows by the induction hypothesis. For $I_7$, we have

$$
\begin{aligned}
|I_7| &\leq \max_{i \geq 2} |\lambda_i^{(t)}| \cdot \| \mathbf{P}_{\mathcal{X}, \mathbf{v}_1}^{\perp} \mathbf{w}_r^{(t)} \|_2 \\
&\leq \left( \frac{\mathcal{E}_{\mathrm{SimCLR}}}{2(1 - \mathcal{E}_{\mathrm{SimCLR}})} + \sigma_0 \right) \cdot \lambda_1 \cdot \| \mathbf{P}_{\mathcal{X}, \mathbf{v}_1}^{\perp} \mathbf{w}_r^{(t)} \|_2 \\
&\leq \frac{2}{3} \cdot \mathcal{E}_{\mathrm{SimCLR}} \cdot \lambda_1 \cdot \| \mathbf{P}_{\mathcal{X}, \mathbf{v}_1}^{\perp} \mathbf{w}_r^{(t)} \|_2,
\end{aligned}
\tag{B.33}
$$

where the second inequality follows by Lemmas 5.2 and B.4, and the third inequality follows by the assumption that $\mathcal{E}_{\mathrm{SimCLR}} \leq 1/16$ and $\sigma_0 \leq \mathcal{E}_{\mathrm{SimCLR}}/16$. For $I_8$, we have

$$
\begin{aligned}
|I_8| &\leq \left\| \sum_{i=2}^{2n_0+1} \lambda_i^{(t)} \cdot \langle \mathbf{v}_1, \mathbf{v}_i^{(t)} \rangle \cdot \mathbf{v}_i^{(t)} \right\|_2 \cdot \langle \mathbf{w}_r^{(t)}, \mathbf{v}_1 \rangle \\
&\leq \sqrt{\sum_{i=2}^{2n_0+1} \lambda_i^{(t)2} \cdot \langle \mathbf{v}_1, \mathbf{v}_i^{(t)} \rangle^2} \cdot \langle \mathbf{w}_r^{(t)}, \mathbf{v}_1 \rangle \\
&\leq \sqrt{2n_0} \cdot \left( \frac{\mathcal{E}_{\mathrm{SimCLR}}}{2(1 - \mathcal{E}_{\mathrm{SimCLR}})} + \sigma_0 \right) \cdot \lambda_1 \cdot 4\sigma_0 \cdot \langle \mathbf{w}_r^{(t)}, \mathbf{v}_1 \rangle \\
&\leq \sqrt{n_0} \cdot \lambda_1 \sigma_0 \cdot \langle \mathbf{w}_r^{(t)}, \mathbf{v}_1 \rangle \\
&\leq \sqrt{n_0 \sigma_0} \cdot \lambda_1 \cdot \| \mathbf{P}_{\mathcal{X}, \mathbf{v}_1}^{\perp} \mathbf{w}_r^{(t)} \|_2,
\end{aligned}
\tag{B.34}
$$

where the second inequality follows by the fact that $\mathbf{v}_i^{(t)}, i = 2, \ldots, 2n_0+1$ are mutually orthogonal unit vectors, the third inequality follows by Lemmas 5.2 and B.4, the fourth inequality follows by the

assumption that $\sigma_0, \mathcal{E}_{\text{SimCLR}} \leq 1/64$, and the fifth inequality follows by the induction hypothesis. Now combining (B.30), (B.31), (B.32), (B.33), and (B.34) gives

$$\left(1 - \frac{5}{6}\mathcal{E}_{\text{SimCLR}}\lambda_1\right) \cdot \|\mathbf{P}^{\perp}_{\mathcal{X},\mathbf{v}_1}\mathbf{w}_r^{(t)}\|_2 \leq \|\mathbf{P}^{\perp}_{\mathcal{X},\mathbf{v}_1}\mathbf{w}_r^{(t+1)}\|_2 \leq \left(1 + \frac{5}{6}\mathcal{E}_{\text{SimCLR}}\lambda_1\right) \cdot \|\mathbf{P}^{\perp}_{\mathcal{X},\mathbf{v}_1}\mathbf{w}_r^{(t)}\|_2$$
(B.35)

for all $t = 0, \ldots, t_0$ and $r \in [2m]$, where we use the assumption that $\sigma_0 \leq \mathcal{E}^2_{\text{SimCLR}}/(64n_0)$ and $\sigma_0 \leq \mathcal{E}_{\text{SimCLR}}/16$.

Now by (B.29) and (B.35), we have

$$\frac{\langle \mathbf{w}_r^{(t_0+1)}, \mathbf{v}_1 \rangle}{\|\mathbf{P}^{\perp}_{\mathcal{X},\mathbf{v}_1}\mathbf{w}_r^{(t_0+1)}\|_2} \geq \frac{1 + (1 - 2\sqrt{n_0\sigma_0})\lambda_1}{1 + (5/6) \cdot \mathcal{E}_{\text{SimCLR}} \cdot \lambda_1} \cdot \frac{\langle \mathbf{w}_r^{(t_0)}, \mathbf{v}_1 \rangle}{\|\mathbf{P}^{\perp}_{\mathcal{X},\mathbf{v}_1}\mathbf{w}_r^{(t_0)}\|_2} \geq \frac{\langle \mathbf{w}_r^{(t_0)}, \mathbf{v}_1 \rangle}{\|\mathbf{P}^{\perp}_{\mathcal{X},\mathbf{v}_1}\mathbf{w}_r^{(t_0)}\|_2} \geq \sigma_0,$$

for all $r \in \mathcal{I}^+$, where the second inequality follows by the assumption that $\sigma_0 \leq \mathcal{E}^2_{\text{SimCLR}}/(64n_0)$. This verifies the induction hypothesis (B.22) at $t = t_0 + 1$. Moreover, by (B.29) and (B.35), we also have

$$\frac{\langle \mathbf{w}_r^{(t_0+1)}, \mathbf{v}_1 \rangle}{\|\mathbf{P}^{\perp}_{\mathcal{X},\mathbf{v}_1}\mathbf{w}_r^{(t_0+1)}\|_2} \leq \frac{\|\mathbf{P}_{\mathcal{X}}\mathbf{w}_r^{(t_0+1)}\|_2}{\|\mathbf{P}^{\perp}_{\mathcal{X},\mathbf{v}_1}\mathbf{w}_r^{(t_0+1)}\|_2}$$

$$\leq \frac{[1 + (1 + \sigma_0)\lambda_1] \cdot \|\mathbf{P}_{\mathcal{X}}\mathbf{w}_r^{(t_0)}\|_2}{(1 - 5\mathcal{E}_{\text{SimCLR}}\lambda_1/6) \cdot \|\mathbf{P}^{\perp}_{\mathcal{X},\mathbf{v}_1}\mathbf{w}_r^{(t_0)}\|_2}$$

$$\leq [1 + (1 + \mathcal{E}_{\text{SimCLR}})\lambda_1] \cdot \frac{\|\mathbf{P}_{\mathcal{X}}\mathbf{w}_r^{(t_0)}\|_2}{\|\mathbf{P}^{\perp}_{\mathcal{X},\mathbf{v}_1}\mathbf{w}_r^{(t_0)}\|_2}$$

$$\leq [1 + (1 + \mathcal{E}_{\text{SimCLR}})\lambda_1]^{t_0} \cdot \frac{\|\mathbf{P}_{\mathcal{X}}\mathbf{w}_r^{(0)}\|_2}{\|\mathbf{P}^{\perp}_{\mathcal{X},\mathbf{v}_1}\mathbf{w}_r^{(0)}\|_2}$$

$$\leq [1 + (1 - \mathcal{E}_{\text{SimCLR}})\lambda_1]^{\frac{1+\mathcal{E}_{\text{SimCLR}}}{1-\mathcal{E}_{\text{SimCLR}}} \cdot t_0} \cdot \frac{\|\mathbf{P}_{\mathcal{X}}\mathbf{w}_r^{(0)}\|_2}{\|\mathbf{P}^{\perp}_{\mathcal{X},\mathbf{v}_1}\mathbf{w}_r^{(0)}\|_2}$$

$$\leq [1 + (1 + \mathcal{E}_{\text{SimCLR}})\lambda_1]^{2T_{\text{SimCLR}}} \cdot \frac{\|\mathbf{P}_{\mathcal{X}}\mathbf{w}_r^{(0)}\|_2}{\|\mathbf{P}^{\perp}_{\mathcal{X},\mathbf{v}_1}\mathbf{w}_r^{(0)}\|_2}$$

for all $r \in [2m]$, where the second inequality follows by Lemma B.4, the third inequality follows by the assumption that $\sigma_0 \leq \mathcal{E}_{\text{SimCLR}}/64$, and the fifth inequality follows by the fact that $1 + a \leq (1 + b)^{a/b}$ for all $a > b > 0$. Now by the definition of $T_{\text{SimCLR}}$, we know that

$$[1 + (1 - \mathcal{E}_{\text{SimCLR}})\lambda_1]^{T_{\text{SimCLR}}} \leq \max\left\{288M^{\frac{1}{q-2}} \cdot \frac{\log(2/\sigma_0)^{\frac{1}{q-2}} \cdot \sqrt{\log(dn)} \cdot \sqrt{\log(md)}}{n^{\frac{1}{q-2}}\text{SNR}^{\frac{q}{q-2}}}, 2\right\}.$$

Therefore, we have

$$\frac{\langle \mathbf{w}_r^{(t_0+1)}, \mathbf{v}_1 \rangle}{\|\mathbf{P}^{\perp}_{\mathcal{X},\mathbf{v}_1}\mathbf{w}_r^{(t_0+1)}\|_2} \leq \max\left\{288M^{\frac{1}{q-2}} \cdot \frac{\log(2/\sigma_0)^{\frac{1}{q-2}} \cdot \sqrt{\log(dn)} \cdot \sqrt{\log(md)}}{n^{\frac{1}{q-2}}\text{SNR}^{\frac{q}{q-2}}}, 2\right\} \cdot \frac{\|\mathbf{P}_{\mathcal{X}}\mathbf{w}_r^{(0)}\|_2}{\|\mathbf{P}^{\perp}_{\mathcal{X},\mathbf{v}_1}\mathbf{w}_r^{(0)}\|_2}$$

$$\leq \max\left\{288M^{\frac{1}{q-2}} \cdot \frac{\log(2/\sigma_0)^{\frac{1}{q-2}} \cdot \sqrt{\log(dn)} \cdot \sqrt{\log(md)}}{n^{\frac{1}{q-2}}\text{SNR}^{\frac{q}{q-2}}}, 2\right\} \cdot \frac{4\sigma_0 \cdot \sqrt{n_0}}{\sigma_0 \cdot \sqrt{n_0}}$$

$$\leq \sigma_0^{-1}$$

for all $r \in [2m]$, where the second inequality follows by (B.19) and (B.20), and the last inequality follows by the assumption that $\sigma_0 \leq \widetilde{O}(M^{-\frac{1}{q-2}} \cdot n^{\frac{1}{q-2}}\text{SNR}^{\frac{q}{q-2}})$. This verifies the induction hypothesis (B.23) at $t = t_0 + 1$.

Based on the discussion above, by induction, we conclude that (B.21), (B.22), (B.23), (B.29), and (B.35) hold for all $t = 0, \ldots, T_{\text{SimCLR}}$. In other words, we can conclude that:

$$[1 + (1 - 2\sqrt{n_0\sigma_0})\lambda_1] \cdot \langle \mathbf{w}_r^{(t)}, \mathbf{v}_1 \rangle \leq \langle \mathbf{w}_r^{(t+1)}, \mathbf{v}_1 \rangle \leq [1 + (1 + 2\sqrt{n_0\sigma_0})\lambda_1] \cdot \langle \mathbf{w}_r^{(t)}, \mathbf{v}_1 \rangle \quad \text{(B.36)}$$

for all $t = 0, \ldots, T_{\mathrm{SimCLR}}$ and $r \in \mathcal{I}^+$, and

$$\left(1 - \frac{5}{6}\mathcal{E}_{\mathrm{SimCLR}}\lambda_1\right) \cdot \|\mathbf{P}^{\perp}_{\mathcal{X},\mathbf{v}_1}\mathbf{w}_r^{(t)}\|_2 \le \|\mathbf{P}^{\perp}_{\mathcal{X},\mathbf{v}_1}\mathbf{w}_r^{(t+1)}\|_2 \le \left(1 + \frac{5}{6}\mathcal{E}_{\mathrm{SimCLR}}\lambda_1\right) \cdot \|\mathbf{P}^{\perp}_{\mathcal{X},\mathbf{v}_1}\mathbf{w}_r^{(t)}\|_2 \tag{B.37}$$

for all $t = 0, \ldots, T_{\mathrm{SimCLR}}$ and $r \in [2m]$. Moreover, by Lemma 5.1 and the fact that the columns and rows of $\mathbf{A}, \mathbf{\Xi}^{(t)}$ are in $\mathcal{X}_{\mathrm{col}}$ and $\mathcal{X}_{\mathrm{row}}$ respectively, we also have

$$\mathbf{P}_{\mathcal{X}}\mathbf{w}_r^{(t+1)} = [\mathbf{I} + \mathbf{A} + \mathbf{\Xi}^{(t)}]\mathbf{P}_{\mathcal{X}}\mathbf{w}_r^{(t)}$$

for all $t = 0, \ldots, T_{\mathrm{SimCLR}}$ and all $r \in [2m]$. Therefore, by Lemmas 5.2 and B.4, we have

$$\|\mathbf{P}_{\mathcal{X}}\mathbf{w}_r^{(t+1)}\|_2 \le (1 + (1 + \sigma_0)\lambda_1) \cdot \|\mathbf{P}_{\mathcal{X}}\mathbf{w}_r^{(t)}\|_2 \tag{B.38}$$

for all $t = 0, \ldots, T_{\mathrm{SimCLR}}$ and all $r \in [2m]$. By (B.18) and (B.36), we have

$$\langle \mathbf{w}_r^{(T_{\mathrm{SimCLR}})}, \mathbf{v}_1 \rangle \ge [1 + (1 - 2\sqrt{n_0\sigma_0})\lambda_1]^{T_{\mathrm{SimCLR}}} \cdot \langle \mathbf{w}_r^{(0)}, \mathbf{v}_1 \rangle$$
$$\ge [1 + (1 - 2\sqrt{n_0\sigma_0})\lambda_1]^{T_{\mathrm{SimCLR}}} \cdot \sigma_0/2 \tag{B.39}$$

for all $r \in \mathcal{I}^+$. Moreover, by (B.19) and (B.37), we also have

$$\|\mathbf{P}^{\perp}_{\mathcal{X},\mathbf{v}_1}\mathbf{w}_r^{(T_{\mathrm{SimCLR}})}\|_2 \le \left(1 + \frac{5}{6}\mathcal{E}_{\mathrm{SimCLR}}\lambda_1\right)^{T_{\mathrm{SimCLR}}} \cdot \|\mathbf{P}^{\perp}_{\mathcal{X},\mathbf{v}_1}\mathbf{w}_r^{(t)}\|_2$$
$$\le \left(1 + \frac{5}{6}\mathcal{E}_{\mathrm{SimCLR}}\lambda_1\right)^{T_{\mathrm{SimCLR}}} \cdot 2\sigma_0 \cdot \sqrt{n_0} \tag{B.40}$$

for all $r \in [2m]$. In addition, by (B.38), it holds that

$$\|\mathbf{P}_{\mathcal{X}}\mathbf{w}_r^{(T_{\mathrm{SimCLR}})}\|_2 \le \|\mathbf{P}_{\mathcal{X}}\mathbf{w}_r^{(0)}\|_2 \le 4(1 + (1 + \sigma_0)\lambda_1)^{T_{\mathrm{SimCLR}}} \cdot \sigma_0 \cdot \sqrt{n_0} \tag{B.41}$$

for all $t = 0, \ldots, T_{\mathrm{SimCLR}}$ and all $r \in [2m]$, where the last inequality above is by (B.20).

Now denote $\bar{\boldsymbol{\mu}} = \boldsymbol{\mu}/\|\boldsymbol{\mu}\|_2$, and $\mathbf{P}^{\perp}_{\boldsymbol{\mu}} = \mathbf{I} - \bar{\boldsymbol{\mu}}\bar{\boldsymbol{\mu}}^{\top}$. Then for all $r \in \mathcal{I}^+$, we have

$$\langle \mathbf{w}_r^{(T_{\mathrm{SimCLR}})}, \bar{\boldsymbol{\mu}} \rangle = \langle \mathbf{w}_r^{(T_{\mathrm{SimCLR}})}, \mathbf{P}_{\mathcal{X}}\bar{\boldsymbol{\mu}} \rangle$$
$$= \langle \mathbf{w}_r^{(T_{\mathrm{SimCLR}})}, (\mathbf{P}^{\perp}_{\mathcal{X},\mathbf{v}_1} + \mathbf{v}_1\mathbf{v}_1^{\top})\bar{\boldsymbol{\mu}} \rangle$$
$$= \langle \mathbf{w}_r^{(T_{\mathrm{SimCLR}})}, (\mathbf{I} - \mathbf{P}_{\mathcal{X}})\bar{\boldsymbol{\mu}} \rangle + \langle \mathbf{w}_r^{(T_{\mathrm{SimCLR}})}, \mathbf{P}^{\perp}_{\mathcal{X},\mathbf{v}_1}\bar{\boldsymbol{\mu}} \rangle + \langle \mathbf{w}_r^{(T_{\mathrm{SimCLR}})}, \mathbf{v}_1 \rangle \cdot \langle \mathbf{v}_1, \bar{\boldsymbol{\mu}} \rangle$$
$$= \langle \mathbf{w}_r^{(T_{\mathrm{SimCLR}})}, \mathbf{P}^{\perp}_{\mathcal{X},\mathbf{v}_1}\bar{\boldsymbol{\mu}} \rangle + \langle \mathbf{w}_r^{(T_{\mathrm{SimCLR}})}, \mathbf{v}_1 \rangle \cdot \langle \mathbf{v}_1, \bar{\boldsymbol{\mu}} \rangle$$
$$\ge -\|\mathbf{P}^{\perp}_{\mathcal{X},\mathbf{v}_1}\mathbf{w}_r^{(T_{\mathrm{SimCLR}})}\|_2 + \langle \mathbf{w}_r^{(T_{\mathrm{SimCLR}})}, \mathbf{v}_1 \rangle \cdot (1 - \mathcal{E}_{\mathrm{SimCLR}}), \tag{B.42}$$

where the first inequality follows by Lemma 5.2. Moreover, we also have

$$\frac{[1 + (1 - 2\sqrt{n_0\sigma_0})\lambda_1]^{T_{\mathrm{SimCLR}}} \cdot \sigma_0/2}{2\sigma_0\sqrt{n_0} \cdot (1 + 5\mathcal{E}_{\mathrm{SimCLR}}\lambda_1/6)^{T_{\mathrm{SimCLR}}}} \ge \frac{1}{4\sqrt{n_0}} \cdot [1 + (1 - \mathcal{E}_{\mathrm{SimCLR}}) \cdot \lambda_1]^{T_{\mathrm{SimCLR}}}$$
$$\ge \frac{1}{4\sqrt{n_0}} \cdot \max\left\{288M^{\frac{1}{q-2}} \cdot \frac{\log(2/\sigma_0)^{\frac{1}{q-2}} \cdot \sqrt{\log(dn)} \cdot \sqrt{\log(md)}}{n^{\frac{1}{q-2}}\mathrm{SNR}^{\frac{q}{q-2}}}, 2\right\}$$
$$\ge \frac{1}{4\sigma_0\sqrt{n_0}}$$
$$\ge 2,$$

where the third inequality is by the $\sigma_0 \le \widetilde{O}(M^{-\frac{1}{q-2}} \cdot n^{\frac{1}{q-2}}\mathrm{SNR}^{\frac{q}{q-2}})$, and the last inequality is by $\sigma_0 \le 1/(64n_0)$. Therefore, by (B.39) and (B.40), we have

$$\|\mathbf{P}^{\perp}_{\mathcal{X},\mathbf{v}_1}\mathbf{w}_r^{(T_{\mathrm{SimCLR}})}\|_2 \le \langle \mathbf{w}_r^{(T_{\mathrm{SimCLR}})}, \mathbf{v}_1 \rangle/4,$$

and hence by (B.42), we have

$$\langle \mathbf{w}_r^{(T_{\mathrm{SimCLR}})}, \boldsymbol{\mu} \rangle = \langle \mathbf{w}_r^{(T_{\mathrm{SimCLR}})}, \bar{\boldsymbol{\mu}} \rangle \cdot \|\boldsymbol{\mu}\|_2$$
$$\ge \langle \mathbf{w}_r^{(T_{\mathrm{SimCLR}})}, \mathbf{v}_1 \rangle \cdot \|\boldsymbol{\mu}\|_2/4$$
$$\ge [1 + (1 - 2\sqrt{n_0\sigma_0})\lambda_1]^{T_{\mathrm{SimCLR}}} \cdot \sigma_0 \cdot \|\boldsymbol{\mu}\|_2/8, \tag{B.43}$$

for all $r \in \mathcal{I}^+$, where the last inequality follows by (B.39). In addition, since the update only happens in the subspace $\mathcal{X}_{\mathrm{col}}$ which is spanned by the data, we have $(\mathbf{I} - \mathbf{P}_{\mathcal{X}})\mathbf{w}_r^{(T_{\mathrm{SimCLR}})} = (\mathbf{I} - \mathbf{P}_{\mathcal{X}})\mathbf{w}_r^{(0)}$. Moreover, we have

$$
\begin{aligned}
\|\mathbf{P}_{\boldsymbol{\mu}}^{\perp}\mathbf{w}_r^{(T_{\mathrm{SimCLR}})}\|_2 &= \|(\mathbf{I} - \mathbf{P}_{\mathcal{X}} + \mathbf{P}_{\mathcal{X},\mathbf{v}_1}^{\perp} + \mathbf{v}_1\mathbf{v}_1^{\top} - \bar{\boldsymbol{\mu}}\bar{\boldsymbol{\mu}}^{\top})\mathbf{w}_r^{(T_{\mathrm{SimCLR}})}\|_2 \\
&\leq \|(\mathbf{I} - \mathbf{P}_{\mathcal{X}})\mathbf{w}_r^{(T_{\mathrm{SimCLR}})}\|_2 + \|(\mathbf{P}_{\mathcal{X},\mathbf{v}_1}^{\perp} + \mathbf{v}_1\mathbf{v}_1^{\top} - \bar{\boldsymbol{\mu}}\bar{\boldsymbol{\mu}}^{\top})\mathbf{w}_r^{(T_{\mathrm{SimCLR}})}\|_2 \\
&\leq \|(\mathbf{I} - \mathbf{P}_{\mathcal{X}})\mathbf{w}_r^{(0)}\|_2 + \|\mathbf{P}_{\mathcal{X},\mathbf{v}_1}^{\perp}\mathbf{w}_r^{(T_{\mathrm{SimCLR}})}\|_2 + \|(\mathbf{v}_1\mathbf{v}_1^{\top} - \bar{\boldsymbol{\mu}}\bar{\boldsymbol{\mu}}^{\top})\mathbf{w}_r^{(T_{\mathrm{SimCLR}})}\|_2,
\end{aligned}
\tag{B.44}
$$

where the last inequality follows by the fact that $(\mathbf{I} - \mathbf{P}_{\mathcal{X}})\mathbf{w}_r^{(0)}$ is a centered spherical Gaussian random vector with standard deviation bounded by $\sigma_0$, and by Gaussian tail bound, with probability at least $1 - d^{-2}$,

$$
\|(\mathbf{I} - \mathbf{P}_{\mathcal{X}})\mathbf{w}_r^{(0)}\|_3 \leq 4\sigma_0 \cdot \sqrt{d\log(md)}, \tag{B.45}
$$

for all $r \in [2m]$. Moreover, we have

$$
\begin{aligned}
\|(\mathbf{v}_1\mathbf{v}_1^{\top} - \bar{\boldsymbol{\mu}}\bar{\boldsymbol{\mu}}^{\top})\mathbf{w}_r^{(T_{\mathrm{SimCLR}})}\|_2 &\leq \|(\mathbf{v}_1 - \bar{\boldsymbol{\mu}})\mathbf{v}_1^{\top}\mathbf{w}_r^{(T_{\mathrm{SimCLR}})}\|_2 + \|\bar{\boldsymbol{\mu}}(\mathbf{v}_1 - \bar{\boldsymbol{\mu}})^{\top}\mathbf{w}_r^{(T_{\mathrm{SimCLR}})}\|_2 \\
&= \langle \mathbf{w}_r^{(T_{\mathrm{SimCLR}})}, \mathbf{v}_1\rangle \cdot \|\mathbf{v}_1 - \bar{\boldsymbol{\mu}}\|_2 + |(\mathbf{v}_1 - \bar{\boldsymbol{\mu}})^{\top}\mathbf{w}_r^{(T_{\mathrm{SimCLR}})}| \\
&\leq \langle \mathbf{w}_r^{(T_{\mathrm{SimCLR}})}, \mathbf{v}_1\rangle \cdot \|\mathbf{v}_1 - \bar{\boldsymbol{\mu}}\|_2 + |(\mathbf{v}_1 - \bar{\boldsymbol{\mu}})^{\top}\mathbf{v}_1 \cdot \mathbf{v}_1^{\top}\mathbf{w}_r^{(T_{\mathrm{SimCLR}})}| \\
&\quad + |(\mathbf{v}_1 - \bar{\boldsymbol{\mu}})^{\top}\mathbf{P}_{\mathcal{X},\mathbf{v}_1}^{\perp}\mathbf{w}_r^{(T_{\mathrm{SimCLR}})}| \\
&\leq \langle \mathbf{w}_r^{(T_{\mathrm{SimCLR}})}, \mathbf{v}_1\rangle \cdot \|\mathbf{v}_1 - \bar{\boldsymbol{\mu}}\|_2 + \langle \mathbf{w}_r^{(T_{\mathrm{SimCLR}})}, \mathbf{v}_1\rangle \cdot |(\mathbf{v}_1 - \bar{\boldsymbol{\mu}})^{\top}\mathbf{v}_1| \\
&\quad + \|\mathbf{v}_1 - \bar{\boldsymbol{\mu}}\|_2 \cdot \|\mathbf{P}_{\mathcal{X},\mathbf{v}_1}^{\perp}\mathbf{w}_r^{(T_{\mathrm{SimCLR}})}\|_2 \\
&\leq (\sqrt{2} \cdot \mathcal{E}_{\mathrm{SimCLR}} + \mathcal{E}_{\mathrm{SimCLR}}^2) \cdot \langle \mathbf{w}_r^{(T_{\mathrm{SimCLR}})}, \mathbf{v}_1\rangle \\
&\quad + \sqrt{2} \cdot \mathcal{E}_{\mathrm{SimCLR}} \cdot \|\mathbf{P}_{\mathcal{X},\mathbf{v}_1}^{\perp}\mathbf{w}_r^{(T_{\mathrm{SimCLR}})}\|_2 \\
&\leq 2\mathcal{E}_{\mathrm{SimCLR}} \cdot \langle \mathbf{w}_r^{(T_{\mathrm{SimCLR}})}, \mathbf{v}_1\rangle + \|\mathbf{P}_{\mathcal{X},\mathbf{v}_1}^{\perp}\mathbf{w}_r^{(T_{\mathrm{SimCLR}})}\|_2 \\
&\leq 2\mathcal{E}_{\mathrm{SimCLR}} \cdot \|\mathbf{P}_{\mathcal{X}}\mathbf{w}_r^{(T_{\mathrm{SimCLR}})}\|_2 + \|\mathbf{P}_{\mathcal{X},\mathbf{v}_1}^{\perp}\mathbf{w}_r^{(T_{\mathrm{SimCLR}})}\|_2.
\end{aligned}
\tag{B.46}
$$

Plugging (B.46) and (B.45) into (B.44) gives

$$
\|\mathbf{P}_{\boldsymbol{\mu}}^{\perp}\mathbf{w}_r^{(T_{\mathrm{SimCLR}})}\|_2 \leq 4\sigma_0 \cdot \sqrt{d\log(md)} + 2\mathcal{E}_{\mathrm{SimCLR}} \cdot \|\mathbf{P}_{\mathcal{X}}\mathbf{w}_r^{(T_{\mathrm{SimCLR}})}\|_2 + 2\|\mathbf{P}_{\mathcal{X},\mathbf{v}_1}^{\perp}\mathbf{w}_r^{(T_{\mathrm{SimCLR}})}\|_2.
$$

Therefore, we have that for $r \in \mathcal{I}^+$ and $r' \in [2m]$,

$$
\begin{aligned}
\frac{\langle \mathbf{w}_r^{(T_{\mathrm{SimCLR}})}, \boldsymbol{\mu}\rangle}{\|\mathbf{P}_{\boldsymbol{\mu}}^{\perp}\mathbf{w}_{r'}^{(T_{\mathrm{SimCLR}})}\|_2} &\geq \frac{\langle \mathbf{w}_r^{(T_{\mathrm{SimCLR}})}, \boldsymbol{\mu}\rangle}{4\sigma_0 \cdot \sqrt{d\log(md)} + 2\mathcal{E}_{\mathrm{SimCLR}} \cdot \|\mathbf{P}_{\mathcal{X}}\mathbf{w}_{r'}^{(T_{\mathrm{SimCLR}})}\|_2 + 2\|\mathbf{P}_{\mathcal{X},\mathbf{v}_1}^{\perp}\mathbf{w}_{r'}^{(T_{\mathrm{SimCLR}})}\|_2} \\
&\geq \frac{1}{3} \cdot \min\left\{\frac{\langle \mathbf{w}_r^{(T_{\mathrm{SimCLR}})}, \boldsymbol{\mu}\rangle}{4\sigma_0 \cdot \sqrt{d\log(md)}}, \frac{\langle \mathbf{w}_r^{(T_{\mathrm{SimCLR}})}, \boldsymbol{\mu}\rangle}{2\mathcal{E}_{\mathrm{SimCLR}} \cdot \|\mathbf{P}_{\mathcal{X}}\mathbf{w}_{r'}^{(T_{\mathrm{SimCLR}})}\|_2}, \frac{\langle \mathbf{w}_r^{(T_{\mathrm{SimCLR}})}, \boldsymbol{\mu}\rangle}{2\|\mathbf{P}_{\mathcal{X},\mathbf{v}_1}^{\perp}\mathbf{w}_{r'}^{(T_{\mathrm{SimCLR}})}\|_2}\right\}
\end{aligned}
$$

By (B.39), we have

$$
\begin{aligned}
\frac{\langle \mathbf{w}_r^{(T_{\mathrm{SimCLR}})}, \boldsymbol{\mu}\rangle}{4\sigma_0 \cdot \sqrt{d\log(md)}} &\geq \frac{[1 + (1 - 2\sqrt{n_0\sigma_0})\lambda_1]^{T_{\mathrm{SimCLR}}} \cdot \|\boldsymbol{\mu}\|_2}{4\sqrt{d\log(md)}} \\
&\geq \frac{[1 + (1 - \mathcal{E}_{\mathrm{SimCLR}})\lambda_1]^{T_{\mathrm{SimCLR}}} \cdot \|\boldsymbol{\mu}\|_2 \cdot \sigma_p}{4\sigma_p\sqrt{d\log(md)}} \\
&= \frac{[1 + (1 - \mathcal{E}_{\mathrm{SimCLR}})\lambda_1]^{T_{\mathrm{SimCLR}}} \cdot \mathrm{SNR} \cdot \sigma_p}{4\sqrt{\log(md)}} \\
&\geq 72C^{\frac{1}{q-2}}\sigma_p \cdot \frac{\log(2/\sigma_0)^{\frac{1}{q-2}} \cdot \sqrt{\log(dn)}}{n^{\frac{1}{q-2}}\mathrm{SNR}^{\frac{2}{q-2}}},
\end{aligned}
$$

where the first inequality is by the assumption that $\sigma_0 \leq \mathcal{E}_{\text{SimCLR}}/(4n_0)$, and the second inequality is by the definition of $T_{\text{SimCLR}}$, which implies that

$$
[1 + (1 - \mathcal{E}_{\text{SimCLR}})\lambda_1]^{T_{\text{SimCLR}}} \geq 288M^{\frac{1}{q-2}} \cdot \frac{\log(2/\sigma_0)^{\frac{1}{q-2}} \cdot \sqrt{\log(dn)} \cdot \sqrt{\log(md)}}{n^{\frac{1}{q-2}}\text{SNR}^{\frac{q}{q-2}}}.
$$

By (B.41) and (B.43), we have

$$
\begin{aligned}
\frac{\langle \mathbf{w}_r^{(T_{\text{SimCLR}})}, \boldsymbol{\mu}\rangle}{2\mathcal{E}_{\text{SimCLR}} \cdot \|\mathbf{P}_{\mathcal{X}}\mathbf{w}_{r'}^{(T_{\text{SimCLR}})}\|_2} &\geq \frac{[1 + (1 - 2\sqrt{n_0\sigma_0})\lambda_1]^{T_{\text{SimCLR}}} \cdot \sigma_0 \cdot \|\boldsymbol{\mu}\|_2/8}{8\mathcal{E}_{\text{SimCLR}} \cdot (1 + (1+\sigma_0)\lambda_1)^{T_{\text{SimCLR}}} \cdot \sigma_0 \cdot \sqrt{n_0}} \\
&= \frac{[1 + (1 - 2\sqrt{n_0\sigma_0})\lambda_1]^{T_{\text{SimCLR}}} \cdot \|\boldsymbol{\mu}\|_2}{64\mathcal{E}_{\text{SimCLR}} \cdot (1 + (1+\sigma_0)\lambda_1)^{T_{\text{SimCLR}}} \cdot \sqrt{n_0}} \\
&\geq \frac{\|\boldsymbol{\mu}\|_2}{64\mathcal{E}_{\text{SimCLR}} \cdot (1 + (1+\sigma_0)\lambda_1)^{T_{\text{SimCLR}}} \cdot \sqrt{n_0}} \\
&\geq \frac{\|\boldsymbol{\mu}\|_2}{64\mathcal{E}_{\text{SimCLR}} \cdot (1 + (1 - \mathcal{E}_{\text{SimCLR}})\lambda_1)^{\frac{1+\sigma_0}{1-\mathcal{E}_{\text{SimCLR}}} \cdot T_{\text{SimCLR}}} \cdot \sqrt{n_0}} \\
&\geq \frac{\|\boldsymbol{\mu}\|_2}{64\mathcal{E}_{\text{SimCLR}} \cdot (1 + (1 - \mathcal{E}_{\text{SimCLR}})\lambda_1)^{2T_{\text{SimCLR}}} \cdot \sqrt{n_0}},
\end{aligned}
$$

where the second inequality follows by the assumption that $\sigma_0 \leq 1/(4n_0)$, the third inequality follows by the fact that $1 + a \leq (1 + b)^{a/b}$ for all $a > b > 0$, and the fourth inequality follows by the assumption that $\sigma_0, \mathcal{E}_{\text{SimCLR}} \leq 1/4$. Now by the definition of $T_{\text{SimCLR}}$, we know that

$$
[1 + (1 - \mathcal{E}_{\text{SimCLR}})\lambda_1]^{T_{\text{SimCLR}}} \leq \max\left\{288M^{\frac{1}{q-2}} \cdot \frac{\log(2/\sigma_0)^{\frac{1}{q-2}} \cdot \sqrt{\log(dn)} \cdot \sqrt{\log(md)}}{n^{\frac{1}{q-2}}\text{SNR}^{\frac{q}{q-2}}}, 2\right\}.
$$

Therefore, by the assumption that

$$
\mathcal{E}_{\text{SimCLR}} \leq \frac{\sqrt{d} \cdot n^{\frac{3}{q-2}}\text{SNR}^{\frac{3q}{q-2}}}{64 \cdot 288^2 \cdot 72 \cdot \sqrt{n_0} \cdot M^{\frac{3}{q-2}} \cdot \log(2/\sigma_0)^{\frac{3}{q-2}} \cdot \log(dn)^{\frac{3}{2}} \cdot \log(md)},
$$

we have

$$
\begin{aligned}
\frac{\langle \mathbf{w}_r^{(T_{\text{SimCLR}})}, \boldsymbol{\mu}\rangle}{2\mathcal{E}_{\text{SimCLR}} \cdot \|\mathbf{P}_{\mathcal{X}}\mathbf{w}_{r'}^{(T_{\text{SimCLR}})}\|_2} &\geq \frac{\|\boldsymbol{\mu}\|_2}{64\mathcal{E}_{\text{SimCLR}}(1 + (1 - \mathcal{E}_{\text{SimCLR}})\lambda_1)^{2T_{\text{SimCLR}}}\sqrt{n_0}} \\
&= \frac{\text{SNR} \cdot \sigma_p \cdot \sqrt{d}}{64\mathcal{E}_{\text{SimCLR}}(1 + (1 - \mathcal{E}_{\text{SimCLR}})\lambda_1)^{2T_{\text{SimCLR}}}\sqrt{n_0}} \\
&\geq 72M^{\frac{1}{q-2}}\sigma_p \cdot \frac{\log(2/\sigma_0)^{\frac{1}{q-2}} \cdot \sqrt{\log(dn)}}{n^{\frac{1}{q-2}}\text{SNR}^{\frac{2}{q-2}}}.
\end{aligned}
$$

Finally, by (B.39) and (B.40), we also have

$$
\begin{aligned}
\frac{\langle \mathbf{w}_r^{(T_{\text{SimCLR}})}, \boldsymbol{\mu}\rangle}{2\|\mathbf{P}_{\mathcal{X},\mathbf{v}_1}^{\perp}\mathbf{w}_{r'}^{(T_{\text{SimCLR}})}\|_2} &\geq \frac{[1 + (1 - 2\sqrt{n_0\sigma_0})\lambda_1]^{T_{\text{SimCLR}}} \cdot \|\boldsymbol{\mu}\|_2}{8\sqrt{n_0} \cdot \left(1 + \frac{5}{6}\mathcal{E}_{\text{SimCLR}}\lambda_1\right)^{T_{\text{SimCLR}}}} \\
&\geq \frac{[1 + (1 - \mathcal{E}_{\text{SimCLR}})\lambda_1]^{T_{\text{SimCLR}}} \cdot \|\boldsymbol{\mu}\|_2}{8\sqrt{n_0}} \\
&= \frac{[1 + (1 - \mathcal{E}_{\text{SimCLR}})\lambda_1]^{T_{\text{SimCLR}}} \cdot \text{SNR} \cdot \sigma_p\sqrt{d}}{8\sqrt{n_0}} \\
&\geq 72M^{\frac{1}{q-2}}\sigma_p \cdot \frac{\log(2/\sigma_0)^{\frac{1}{q-2}} \cdot \sqrt{\log(dn)}}{n^{\frac{1}{q-2}}\text{SNR}^{\frac{2}{q-2}}},
\end{aligned}
$$

where the second inequality follows by the assumption that $\sigma_0 \leq \mathcal{E}_{\mathrm{SimCLR}}^2/(64n_0)$, and the last inequality follows by the choice of $T_{\mathrm{SimCLR}}$ which implies that

$$[1 + (1 - \mathcal{E}_{\mathrm{SimCLR}})\lambda_1]^{T_{\mathrm{SimCLR}} \cdot} \geq 288M^{\frac{1}{q-2}} \cdot \frac{\log(2/\sigma_0)^{\frac{1}{q-2}} \cdot \sqrt{\log(dn)} \cdot \sqrt{\log(md)}}{n^{\frac{1}{q-2}}\mathrm{SNR}^{\frac{q}{q-2}}}$$

$$\geq 576M^{\frac{1}{q-2}} \cdot \frac{\sqrt{n_0} \cdot \log(2/\sigma_0)^{\frac{1}{q-2}} \cdot \sqrt{\log(dn)}}{n^{\frac{1}{q-2}}\mathrm{SNR}^{\frac{q}{q-2}} \cdot \sqrt{d}},$$

where the second inequality follows by the assumption that $d \geq 4n_0$. Therefore, we conclude that for $r \in \mathcal{I}^+$ and $r' \in [2m]$, we have

$$\frac{\langle \mathbf{w}_r^{(T_{\mathrm{SimCLR}})}, \boldsymbol{\mu} \rangle}{\|(\mathbf{I} - \boldsymbol{\mu}\boldsymbol{\mu}^\top/\|\boldsymbol{\mu}\|_2^2)\mathbf{w}_{r'}^{(T_{\mathrm{SimCLR}})}\|_2} \geq 24M^{\frac{1}{q-2}}\sigma_p \cdot \frac{\log(2/\sigma_0)^{\frac{1}{q-2}} \cdot \sqrt{\log(dn)}}{n^{\frac{1}{q-2}}\mathrm{SNR}^{\frac{2}{q-2}}}, \qquad (\text{B.47})$$

Now for any $r' \in [2m]$, consider the following decomposition for $\mathbf{w}_{r'}^{(T_{\mathrm{SimCLR}})}$:

$$\mathbf{w}_{r'}^{(T_{\mathrm{SimCLR}})} = \mathbf{w}_{r'}^\perp + \gamma_{r'} \cdot \boldsymbol{\mu}/\|\boldsymbol{\mu}\|_2^2 + \sum_{i=1}^n \rho_{r',i} \cdot \boldsymbol{\xi}_i^{\text{fine-tuning}}/\|\boldsymbol{\xi}_i^{\text{fine-tuning}}\|_2^2,$$

where $\mathbf{w}_{r'}^\perp$ is perpendicular to $\boldsymbol{\mu}$ and $\boldsymbol{\xi}_1^{\text{fine-tuning}}, \ldots, \boldsymbol{\xi}_n^{\text{fine-tuning}}$. Then we directly have

$$\gamma_r = \langle \mathbf{w}_r^{(T_{\mathrm{SimCLR}})}, \boldsymbol{\mu} \rangle \qquad (\text{B.48})$$

for all $r \in \mathcal{I}^+$. Note that $\mathbf{w}_{r'}^{(T_{\mathrm{SimCLR}})}$ is independent of $\boldsymbol{\xi}_i^{\text{fine-tuning}}$, $i \in [n]$. For any $i \in [n]$, considering the randomness of $\boldsymbol{\xi}_i^{\text{fine-tuning}}$, we see that $\langle \mathbf{w}_{r'}^{(T_{\mathrm{SimCLR}})}, \boldsymbol{\xi}_i^{\text{fine-tuning}} \rangle$ is a Gaussian random variable with mean zero and standard deviation $\sigma_p \cdot \|\mathbf{w}_{r'}^{(T_{\mathrm{SimCLR}})}\|_2$. Therefore, by Gaussian tail bound and union bound, with probability at least $1 - d^{-2}$, we have

$$|\langle \mathbf{w}_{r'}^{(T_{\mathrm{SimCLR}})}, \boldsymbol{\xi}_i^{\text{fine-tuning}} \rangle| \leq 8\sigma_p \cdot \|\mathbf{w}_{r'}^{(T_{\mathrm{SimCLR}})}\|_2 \cdot \sqrt{\log(dn)}.$$

Now denote $\mathbf{E} = [\boldsymbol{\xi}_1, \ldots, \boldsymbol{\xi}_n]$, $\mathbf{D} = \mathrm{diag}(\|\boldsymbol{\xi}_1^{\text{fine-tuning}}\|_2^{-2}, \|\boldsymbol{\xi}_2^{\text{fine-tuning}}\|_2^{-2}, \ldots, \|\boldsymbol{\xi}_n^{\text{fine-tuning}}\|_2^{-2})$, $\boldsymbol{\rho}_{r'} = [\rho_{r',1}, \ldots, \rho_{r',n}]^\top$. Then we have

$$\|\mathbf{w}_{r'}^{(T_{\mathrm{SimCLR}})\top}\mathbf{E}\|_\infty = \|\mathbf{w}_{r'}^{(T_{\mathrm{SimCLR}})\top}(\mathbf{I} - \boldsymbol{\mu}\boldsymbol{\mu}^\top/\|\boldsymbol{\mu}\|_2^2)\mathbf{E}\|_\infty$$

$$\leq 8\sigma_p \cdot \|(\mathbf{I} - \boldsymbol{\mu}\boldsymbol{\mu}^\top/\|\boldsymbol{\mu}\|_2^2)\mathbf{w}_{r'}^{(T_{\mathrm{SimCLR}})}\|_2 \cdot \sqrt{\log(dn)}.$$

By definition, we have

$$\langle \mathbf{w}_{r'}^{(T_{\mathrm{SimCLR}})}, \boldsymbol{\xi}_{i_0} \rangle = \sum_{i=1}^n \rho_{r',i} \cdot \langle \boldsymbol{\xi}_i, \boldsymbol{\xi}_{i_0} \rangle/\|\boldsymbol{\xi}_i\|_2^2,$$

and

$$\mathbf{E}\mathbf{w}_{r'}^{(T_{\mathrm{SimCLR}})} = \mathbf{E}^\top\mathbf{E}\mathbf{D}\boldsymbol{\rho}_{r'}.$$

Moreover, we have

$$\left|[(\mathbf{E}^\top\mathbf{E})^{-1}]_{ij}\right| = \frac{1}{\sigma_p^2 d} \cdot \left|[(\mathbf{I} + \sigma_p^{-2}d^{-1} \cdot \mathbf{E}^\top\mathbf{E} - \mathbf{I}))^{-1}]_{ij}\right|$$

$$= \frac{1}{\sigma_p^2 d} \cdot \left|\left[\mathbf{I} + \sum_{k=1}^\infty (\mathbf{I} - \sigma_p^{-2}d^{-1} \cdot \mathbf{E}^\top\mathbf{E})^k\right]_{ij}\right|$$

$$\leq \begin{cases} 3/(2\sigma_p^2 d), & \text{if } i = j, \\ 1/(2n\sigma_p^2 d), & \text{if } i \neq j, \end{cases} \qquad (\text{B.49})$$

where the last inequality follows by the fact that $(\mathbf{I} - \sigma_p^{-2}d^{-1} \cdot \mathbf{E}^\top\mathbf{E})^k$ is an $n \times n$ matrix whose entries are bounded by $\widetilde{O}(d^{-1/2})$ according to Lemma C.8. Therefore, we have

$$
\begin{aligned}
\|\boldsymbol{\rho}_{r'}\|_\infty &\leq \|\mathbf{D}^{-1}(\mathbf{E}^\top\mathbf{E})^{-1}\mathbf{E}\mathbf{w}_{r'}^{(T_{\mathrm{SimCLR}})}\|_\infty \\
&\leq \|\mathbf{D}^{-1}\|_\infty \cdot \|(\mathbf{E}^\top\mathbf{E})^{-1}\|_\infty \cdot \|\mathbf{E}\mathbf{w}_{r'}^{(T_{\mathrm{SimCLR}})}\|_\infty \\
&\leq \frac{3}{2}\sigma_p^2 d \cdot \frac{2}{\sigma_p^2 d} \cdot 8\sigma_p \cdot \|(\mathbf{I} - \boldsymbol{\mu}\boldsymbol{\mu}^\top/\|\boldsymbol{\mu}\|_2^2)\mathbf{w}_{r'}^{(T_{\mathrm{SimCLR}})}\|_2 \cdot \sqrt{\log(dn)} \\
&= 24\sigma_p \cdot \|(\mathbf{I} - \boldsymbol{\mu}\boldsymbol{\mu}^\top/\|\boldsymbol{\mu}\|_2^2)\mathbf{w}_{r'}^{(T_{\mathrm{SimCLR}})}\|_2 \cdot \sqrt{\log(dn)} \quad\quad\quad\text{(B.50)}
\end{aligned}
$$

for all $r' \in [2m]$, where the third inequality follows by (B.49) and Lemma C.8. Therefore, combining (B.47), (B.48) and (B.50), we have

$$
\frac{\gamma_r}{|\rho_{r',i}|} \geq \frac{\langle \mathbf{w}_r^{(T_{\mathrm{SimCLR}})}, \boldsymbol{\mu}\rangle}{24\sigma_p \cdot \|(\mathbf{I} - \boldsymbol{\mu}\boldsymbol{\mu}^\top/\|\boldsymbol{\mu}\|_2^2)\mathbf{w}_{r'}^{(T_{\mathrm{SimCLR}})}\|_2 \cdot \sqrt{\log(dn)}} \geq \frac{M^{\frac{1}{q-2}}\log(1/\sigma_0)^{\frac{1}{q-2}}}{n^{\frac{1}{q-2}}\mathrm{SNR}^{\frac{2}{q-2}}}
$$

for all $r \in \mathcal{I}^+$ and all $r' \in [2m]$. By (B.39), it is clear that $\gamma_r \geq 2\sigma_0$ for all $r \in \mathcal{I}^+$. This further implies that

$$
\frac{\gamma_r/\log(2/\gamma_r)^{\frac{1}{q-2}}}{|\rho_{r',i}|} \geq \frac{M^{\frac{1}{q-2}}}{n^{\frac{1}{q-2}}\mathrm{SNR}^{\frac{2}{q-2}}}
$$

for all $r \in \mathcal{I}^+$ and all $r' \in [2m]$. With exactly the same proof, we can also show that

$$
\frac{-\gamma_r/\log(-2/\gamma_r)^{\frac{1}{q-2}}}{|\rho_{r',i}|} \geq \frac{M^{\frac{1}{q-2}}\log(1/\sigma_0)^{\frac{1}{q-2}}}{n^{\frac{1}{q-2}}\mathrm{SNR}^{\frac{2}{q-2}}}
$$

for all $r \in \mathcal{I}^-$ and all $r' \in [2m]$.

Finally, for all $r \in [2m]$, by Lemma 5.1, we have

$$
\mathbf{w}_r^{(t+1)} = \mathbf{w}_r^{(t)} + (\mathbf{A} + \boldsymbol{\Xi}^{(t)})\mathbf{w}_r^{(t)}
$$

for $t = 0, \ldots, T_{\mathrm{SimCLR}}$. Therefore, we have

$$
\begin{aligned}
\|\mathbf{w}_r^\perp\|_2 &\leq \|\mathbf{w}_r^{(T_{\mathrm{SimCLR}})}\|_2 \\
&\leq (1 + (1 + \sigma_0) \cdot \lambda_1)^{T_{\mathrm{SimCLR}}} \cdot \|\mathbf{w}_r^{(0)}\|_2 \\
&\leq (1 + (1 + \sigma_0) \cdot \lambda_1)^{T_{\mathrm{SimCLR}}} \cdot 2\sigma_0 \cdot \sqrt{d} \\
&\leq (1 + (1 - \mathcal{E}_{\mathrm{SimCLR}}) \cdot \lambda_1)^{\frac{1+\sigma_0}{1-\mathcal{E}_{\mathrm{SimCLR}}}T_{\mathrm{SimCLR}}} \cdot 2\sigma_0 \cdot \sqrt{d} \\
&\leq (1 + (1 - \mathcal{E}_{\mathrm{SimCLR}}) \cdot \lambda_1)^{2T_{\mathrm{SimCLR}}} \cdot 2\sigma_0 \cdot \sqrt{d}
\end{aligned}
$$

where the fourth inequality follows by the assumption that $\sigma_0, \mathcal{E}_{\mathrm{SimCLR}} \leq 1/4$. Now by the definition of $T_{\mathrm{SimCLR}}$, we know that

$$
[1 + (1 - \mathcal{E}_{\mathrm{SimCLR}})\lambda_1]^{T_{\mathrm{SimCLR}}} \leq \max\left\{288M^{\frac{1}{q-2}} \cdot \frac{\log(2/\sigma_0)^{\frac{1}{q-2}} \cdot \sqrt{\log(dn)} \cdot \sqrt{\log(md)}}{n^{\frac{1}{q-2}}\mathrm{SNR}^{\frac{q}{q-2}}}, 2\right\}.
$$

Therefore, we have

$$
\|\mathbf{w}_r^\perp\|_2 \leq 2\sigma_0 \cdot \sqrt{d} \cdot \max\left\{288M^{\frac{1}{q-2}} \cdot \frac{\log(2/\sigma_0)^{\frac{1}{q-2}} \cdot \sqrt{\log(dn)} \cdot \sqrt{\log(md)}}{n^{\frac{1}{q-2}}\mathrm{SNR}^{\frac{q}{q-2}}}, 2\right\} \leq \frac{1}{n},
$$

where we implement the assumption that $\sigma_0 \leq d^{-1/2}n^{-1}/4$ and

$$
\sigma_0 \leq \frac{d^{-1/2}n^{-1} \cdot n^{\frac{1}{q-2}}\mathrm{SNR}^{\frac{q}{q-2}}}{576M^{\frac{1}{q-2}}\log(2/\sigma_0)^{\frac{1}{q-2}} \cdot \sqrt{\log(dn)} \cdot \sqrt{\log(md)}}.
$$

This finishes the proof. □

## B.2 PROOFS OF LEMMAS IN APPENDIX B.1.1

### B.2.1 PROOF OF LEMMA B.1

*Proof of Lemma B.1.* Since $\boldsymbol{\xi}_i, i \in [n_0]$ i.i.d follows $\mathcal{N}(\mathbf{0}, \sigma_p^2 \cdot (\mathbf{I} - \boldsymbol{\mu}\boldsymbol{\mu}^\top \cdot \|\boldsymbol{\mu}\|_2^{-2}))$, thus $\|\boldsymbol{\xi}_i\|_2^2$ is sub-exponential random variable with

$$\left\| \|\boldsymbol{\xi}_i\|_2^2 \right\|_{\psi_1} \leq \overline{C}_2 \sigma_p^2,$$

where $\overline{C}_2$ is an absolute constant. By Bernstein inequality, with the probability of at least $1 - \widetilde{\delta}/(2n_0)$ we have that

$$-\frac{\overline{C}_2}{\sqrt{c}} \sigma_p^2 \sqrt{d \log(4n_0/\widetilde{\delta})} \leq \|\boldsymbol{\xi}_i\|_2^2 - d\sigma_p^2 \leq \frac{\overline{C}_2}{\sqrt{c}} \sigma_p^2 \sqrt{d \log(4n_0/\widetilde{\delta})},$$

where $c$ is also an absolute constant, and it is equivalent to

$$d\sigma_p^2 - C_2 \sigma_p^2 \sqrt{d \log(4n_0/\widetilde{\delta})} \leq \|\boldsymbol{\xi}_i\|_2^2 \leq d\sigma_p^2 + C_2 \sigma_p^2 \sqrt{d \log(4n_0/\widetilde{\delta})},$$

where $C_2$ is an absolute constant that does not depend on other variables. Similarly, we could obtain that with the probability of at least $1 - \widetilde{\delta}/(2n_0)$,

$$d\sigma_p^2 - C_2 \sigma_p^2 \sqrt{d \log(4n_0/\widetilde{\delta})} \leq \|\widetilde{\boldsymbol{\xi}}_i\|_2^2 \leq d\sigma_p^2 + C_2 \sigma_p^2 \sqrt{d \log(4n_0/\widetilde{\delta})},$$

Apply a union bound for $\|\boldsymbol{\xi}_i\|_2^2, \|\widetilde{\boldsymbol{\xi}}_i\|_2^2, i \in [n_0]$ finishes the proof of this lemma. $\qquad \square$

## B.3 PROOFS OF LEMMAS IN APPENDIX B.1.2

### B.3.1 PROOF OF LEMMA B.2

In this section, the following Lemma B.5, B.6 and B.7 are introduced to prove Lemma B.2.

**Lemma B.5.** *Suppose that $\widetilde{\delta} > 0$, then with probability at least $1 - \widetilde{\delta}$,*

$$|\sum_{i=1}^{n_0} \frac{1}{n_0} y_i| \leq \sqrt{\frac{2}{n_0} \log(2/\widetilde{\delta})}$$

*Proof of Lemma B.5.* Since $y_i, i \in [n_0]$ independent and identically follow Bernoulli distribution, then by Hoeffding inequality, with the probability of at least $1 - \widetilde{\delta}$, we have

$$|\sum_{i=1}^{n_0} \frac{1}{n_0} y_i| \leq \sqrt{\frac{2}{n_0} \log(2/\widetilde{\delta})}$$

$\qquad \square$

**Lemma B.6.** *For any $\widetilde{\delta} > 0$, with probability at least $1 - \widetilde{\delta}$, it holds that*

$$dn_0 \sigma_p^2 - C_1 n_0 \sigma_p^2 \sqrt{d \log(2/\widetilde{\delta})} \leq \left\| \sum_{i=1}^{n_0} \boldsymbol{\xi}_i \right\|_2^2 \leq dn_0 \sigma_p^2 + C_1 n_0 \sigma_p^2 \sqrt{d \log(2/\widetilde{\delta})}$$

$$dn_0 \sigma_p^2 - C_1 n_0 \sigma_p^2 \sqrt{d \log(2/\widetilde{\delta})} \leq \left\| \sum_{i=1}^{n_0} y_i \boldsymbol{\xi}_i \right\|_2^2 \leq dn_0 \sigma_p^2 + C_1 n_0 \sigma_p^2 \sqrt{d \log(2/\widetilde{\delta})}$$

$$dn_0 \sigma_p^2 - C_1 n_0 \sigma_p^2 \sqrt{d \log(2/\widetilde{\delta})} \leq \left\| \sum_{i=1}^{n_0} y_i \widetilde{\boldsymbol{\xi}}_i \right\|_2^2 \leq dn_0 \sigma_p^2 + C_1 n_0 \sigma_p^2 \sqrt{d \log(2/\widetilde{\delta})}$$

*where $C_1$ is an absolute constant that does not depend on other variables.*

*Proof of Lemma B.6.* Since $\boldsymbol{\xi}_i, i \in [n_0]$ i.i.d follows $\mathcal{N}(\mathbf{0}, \sigma_p^2 \cdot (\mathbf{I} - \boldsymbol{\mu}\boldsymbol{\mu}^\top \cdot \|\boldsymbol{\mu}\|_2^{-2}))$, therefore $\sum_{i=1}^{n_0} \boldsymbol{\xi}_i \sim \mathcal{N}(\mathbf{0}, n_0\sigma_p^2 \cdot (\mathbf{I} - \boldsymbol{\mu}\boldsymbol{\mu}^\top \cdot \|\boldsymbol{\mu}\|_2^{-2}))$, thus $\|\sum_{i=1}^{n_0} \boldsymbol{\xi}_i\|_2^2$ is sub-exponential random variable with

$$\left\| \left\| \sum_{i=1}^{n_0} \boldsymbol{\xi}_i \right\|_2^2 \right\|_{\psi_1} \leq \overline{C}_1 \sigma_p^2 n_0,$$

where $\overline{C}_1$ is an absolute constant. By Bernstein inequality, with the probability of at least $1 - \widetilde{\delta}$ we have

$$-\frac{\overline{C}_1}{\sqrt{c}} n_0 \sigma_p^2 \sqrt{d\log(2/\widetilde{\delta})} \leq \left\| \sum_{i=1}^{n_0} \boldsymbol{\xi}_i \right\|_2^2 - dn_0\sigma_p^2 \leq \frac{\overline{C}_1}{\sqrt{c}} n_0 \sigma_p^2 \sqrt{d\log(2/\widetilde{\delta})},$$

where $c$ is also an absolute constant, and it is equivalent to

$$dn_0\sigma_p^2 - C_1 n_0 \sigma_p^2 \sqrt{d\log(2/\widetilde{\delta})} \leq \left\| \sum_{i=1}^{n_0} \boldsymbol{\xi}_i \right\|_2^2 \leq dn_0\sigma_p^2 + C_1 n_0 \sigma_p^2 \sqrt{d\log(2/\widetilde{\delta})},$$

where $C_1$ is an absolute constant, which does not depend on other variables. Notice that similar results could be proved for $\|\sum_{i=1}^{n_0} y_i \boldsymbol{\xi}_i\|_2^2$ and $\left\|\sum_{i=1}^{n_0} y_i \widetilde{\boldsymbol{\xi}}_i\right\|_2^2$. $\qquad\square$

**Lemma B.7.** *For any $\widetilde{\delta} > 0$, with probability at least $1 - \widetilde{\delta}$, it holds that*

$$\left\| \frac{1}{n_0} \sum_{i=1}^{n_0} (\boldsymbol{\xi}_i \widetilde{\boldsymbol{\xi}}_i^\top + \widetilde{\boldsymbol{\xi}}_i \boldsymbol{\xi}_i^\top) \right\|_2 \leq C_3 \sigma_p^2 \cdot \max\left\{ \sqrt{\frac{d\log(9/\widetilde{\delta})}{n_0}}, \frac{d\log(9/\widetilde{\delta})}{n_0} \right\},$$

*where $C_3$ is an absolute constant.*

*Proof of Lemma B.7.* Within this proof, we denote function

$$\mathbf{M} = \frac{1}{n_0} \sum_{i=1}^{n_0} (\boldsymbol{\xi}_i \widetilde{\boldsymbol{\xi}}_i^\top + \widetilde{\boldsymbol{\xi}}_i \boldsymbol{\xi}_i^\top),$$

and

$$g(\mathbf{a}) = \mathbf{a}^\top \mathbf{M} \mathbf{a},$$

for all $\mathbf{a} \in \mathbb{R}^d$. By Lemma 5.2 in Vershynin (2010), there exists a $1/4$-net $\mathcal{N}$ covering the $d$-dimensional unit sphere $\mathbb{S}^{d-1}$ with $|\mathcal{N}| \leq 9^d$. Then for any $\mathbf{a} \in \mathbb{S}^{d-1}$, there exists $\widehat{\mathbf{a}} \in \mathcal{N} \subseteq \mathbb{S}^{d-1}$ such that $\|\widehat{\mathbf{a}} - \mathbf{a}\|_2 \leq 1/4$.

Now for any fixed $\widehat{\mathbf{a}}_0 \in \mathcal{N}$, with direct calculation we have

$$g(\widehat{\mathbf{a}}_0) = \frac{2}{n_0} \sum_{i=1}^{n_0} \langle \widehat{\mathbf{a}}_0, \boldsymbol{\xi}_i \rangle \cdot \langle \widehat{\mathbf{a}}_0, \widetilde{\boldsymbol{\xi}}_i \rangle.$$

Since $\|\widehat{\mathbf{a}}\|_2 = 1$, $\langle \widehat{\mathbf{a}}_0, \boldsymbol{\xi}_i \rangle, \langle \widehat{\mathbf{a}}_0, \widetilde{\boldsymbol{\xi}}_i \rangle$ are independent $\mathcal{N}(0, \sigma_p)$ random variables, $i = 1, \ldots, n_0$. Therefore, by Lemma 5.14 in Vershynin (2010), $\langle \widehat{\mathbf{a}}_0, \boldsymbol{\xi}_i \rangle \cdot \langle \widehat{\mathbf{a}}_0, \widetilde{\boldsymbol{\xi}}_i \rangle$ is sub-exponential with

$$\|\langle \widehat{\mathbf{a}}_0, \boldsymbol{\xi}_i \rangle \cdot \langle \widehat{\mathbf{a}}_0, \widetilde{\boldsymbol{\xi}}_i \rangle\|_{\psi_1} \leq c_1 \cdot \sigma_p^2,$$

where $c_1$ is an absolute constant. Then by Bernstein-type inequality (Proposition 5.16 in Vershynin (2010)), with probability at least $1 - 9^{-d}\widetilde{\delta}$, we have

$$|g(\widehat{\mathbf{a}}_0)| = \left| \frac{2}{n_0} \sum_{i=1}^{n_0} \langle \widehat{\mathbf{a}}_0, \boldsymbol{\xi}_i \rangle \cdot \langle \widehat{\mathbf{a}}_0, \widetilde{\boldsymbol{\xi}}_i \rangle \right| \leq 2c_1 \sigma_p^2 \cdot \max\left\{ \sqrt{\frac{\log(9^d/\widetilde{\delta})}{n_0}}, \frac{\log(9^d/\widetilde{\delta})}{n_0} \right\}$$

$$\leq 2c_1 \sigma_p^2 \cdot \max\left\{ \sqrt{\frac{d\log(9/\widetilde{\delta})}{n_0}}, \frac{d\log(9/\widetilde{\delta})}{n_0} \right\}.$$

Since the above conclusion holds for arbitrary $\widehat{\mathbf{a}}_0 \in \mathcal{N}$, by union bound, with probability at least $1 - \widetilde{\delta}$ we have

$$|g(\widehat{\mathbf{a}})| \leq 2c_1\sigma_p^2 \cdot \max\left\{\sqrt{\frac{\log(9^d/\widetilde{\delta})}{n_0}}, \frac{\log(9^d/\widetilde{\delta})}{n_0}\right\} \leq 2c_1\sigma_p^2 \cdot \max\left\{\sqrt{\frac{d\log(9/\widetilde{\delta})}{n_0}}, \frac{d\log(9/\widetilde{\delta})}{n_0}\right\}$$

for all $\widehat{\mathbf{a}} \in \mathcal{N}$. Now for any $\mathbf{a} \in \mathbb{S}^{d-1}$, there exists $\widehat{\mathbf{a}} \in \mathcal{N}$ such that $\|\widehat{\mathbf{a}} - \mathbf{a}\|_2 \leq 1/4$, and hence

$$\begin{aligned}
|g(\mathbf{a})| &\leq |g(\widehat{\mathbf{a}})| + |g(\mathbf{a}) - g(\widehat{\mathbf{a}})| \\
&= |g(\widehat{\mathbf{a}})| + |\mathbf{a}^\top \mathbf{M}\mathbf{a} - \widehat{\mathbf{a}}^\top \mathbf{M}\widehat{\mathbf{a}}| \\
&\leq 2c_1\sigma_p^2 \cdot \max\left\{\sqrt{\frac{d\log(9/\widetilde{\delta})}{n_0}}, \frac{d\log(9/\widetilde{\delta})}{n_0}\right\} + |\mathbf{a}^\top \mathbf{M}\mathbf{a} - \mathbf{a}^\top \mathbf{M}\widehat{\mathbf{a}}| + |\mathbf{a}^\top \mathbf{M}\widehat{\mathbf{a}} - \widehat{\mathbf{a}}^\top \mathbf{M}\widehat{\mathbf{a}}| \\
&\leq 2c_1\sigma_p^2 \cdot \max\left\{\sqrt{\frac{d\log(9/\widetilde{\delta})}{n_0}}, \frac{d\log(9/\widetilde{\delta})}{n_0}\right\} + |\mathbf{a}^\top \mathbf{M}(\mathbf{a} - \widehat{\mathbf{a}})| + |(\mathbf{a} - \widehat{\mathbf{a}})^\top \mathbf{M}\widehat{\mathbf{a}}|
\end{aligned}$$

By Cauchy-Schwarz inequality, we have

$$\begin{aligned}
|\mathbf{a}^\top \mathbf{M}(\mathbf{a} - \widehat{\mathbf{a}})| &\leq \sqrt{\mathbf{a}^\top \mathbf{M}\mathbf{a}} \cdot \sqrt{(\mathbf{a} - \widehat{\mathbf{a}})^\top \mathbf{M}(\mathbf{a} - \widehat{\mathbf{a}})} = \sqrt{g(\mathbf{a})} \cdot \|\mathbf{a} - \widehat{\mathbf{a}}\|_2 \cdot \sqrt{g(\mathbf{a} - \widehat{\mathbf{a}})}, \\
|(\mathbf{a} - \widehat{\mathbf{a}})^\top \mathbf{M}\widehat{\mathbf{a}}| &\leq \sqrt{\widehat{\mathbf{a}}^\top \mathbf{M}\widehat{\mathbf{a}}} \cdot \sqrt{(\mathbf{a} - \widehat{\mathbf{a}})^\top \mathbf{M}(\mathbf{a} - \widehat{\mathbf{a}})} = \sqrt{g(\widehat{\mathbf{a}})} \cdot \|\mathbf{a} - \widehat{\mathbf{a}}\|_2 \cdot \sqrt{g(\mathbf{a} - \widehat{\mathbf{a}})}.
\end{aligned}$$

Therefore, we further have

$$\begin{aligned}
|g(\mathbf{a})| &\leq 2c_1\sigma_p^2 \cdot \max\left\{\sqrt{\frac{d\log(9/\widetilde{\delta})}{n_0}}, \frac{d\log(9/\widetilde{\delta})}{n_0}\right\} + \sqrt{g(\mathbf{a})} \cdot \|\mathbf{a} - \widehat{\mathbf{a}}\|_2 \cdot \sqrt{g(\mathbf{a} - \widehat{\mathbf{a}})} \\
&\quad + \sqrt{g(\widehat{\mathbf{a}})} \cdot \|\mathbf{a} - \widehat{\mathbf{a}}\|_2 \cdot \sqrt{g(\mathbf{a} - \widehat{\mathbf{a}})} \\
&\leq 2c_1\sigma_p^2 \cdot \max\left\{\sqrt{\frac{d\log(9/\widetilde{\delta})}{n_0}}, \frac{d\log(9/\widetilde{\delta})}{n_0}\right\} + \frac{1}{4} \cdot \sup_\mathbf{a} g(\mathbf{a}) + \frac{1}{4} \cdot \sup_\mathbf{a} g(\mathbf{a}) \\
&= 2c_1\sigma_p^2 \cdot \max\left\{\sqrt{\frac{d\log(9/\widetilde{\delta})}{n_0}}, \frac{d\log(9/\widetilde{\delta})}{n_0}\right\} + \frac{1}{2} \cdot \sup_\mathbf{a} g(\mathbf{a})
\end{aligned}$$

for all $\mathbf{a} \in \mathbb{S}^{d-1}$. Taking a supremum then gives

$$\sup_\mathbf{a} |g(\mathbf{a})| \leq 2c_1\sigma_p^2 \cdot \max\left\{\sqrt{\frac{d\log(9/\widetilde{\delta})}{n_0}}, \frac{d\log(9/\widetilde{\delta})}{n_0}\right\} + \frac{1}{2} \cdot \sup_\mathbf{a} g(\mathbf{a}).$$

Therefore, we conclude that

$$\sup_\mathbf{a} |g(\mathbf{a})| \leq 4c_1\sigma_p^2 \cdot \max\left\{\sqrt{\frac{d\log(9/\widetilde{\delta})}{n_0}}, \frac{d\log(9/\widetilde{\delta})}{n_0}\right\}.$$

This finishes the proof. $\qquad\square$

Based on Lemmas B.5, B.6 and B.7, the following Lemma B.2 is proved.

*Proof of Lemma B.2.* The matrix $\mathbf{A}$ defined in Lemma 5.1 can be written and simplified in the following way.

$$\mathbf{A} = -\frac{\eta}{n_0^2\tau}\sum_{i=1}^{n_0}\sum_{i'\neq i}(\mathbf{z}_i\mathbf{z}_{i'}^\top + \mathbf{z}_{i'}\mathbf{z}_i^\top - \mathbf{z}_i\widetilde{\mathbf{z}}_i^\top - \widetilde{\mathbf{z}}_i\mathbf{z}_i^\top)$$

$$= -\frac{\eta}{n_0^2\tau}\sum_{i=1}^{n_0}\sum_{i'\neq i}\Big[2(y_iy_{i'}-1)\boldsymbol{\mu}\boldsymbol{\mu}^\top + y_i(\boldsymbol{\mu}\boldsymbol{\xi}_{i'}^\top + \boldsymbol{\xi}_{i'}\boldsymbol{\mu}^\top) + y_{i'}(\boldsymbol{\mu}\boldsymbol{\xi}_i^\top + \boldsymbol{\xi}_i\boldsymbol{\mu}^\top)$$

$$- y_i(\boldsymbol{\mu}\widetilde{\boldsymbol{\xi}}_i^\top + \widetilde{\boldsymbol{\xi}}_i\boldsymbol{\mu}^\top + \boldsymbol{\mu}\boldsymbol{\xi}_i^\top + \boldsymbol{\xi}_i\boldsymbol{\mu}^\top) + \boldsymbol{\xi}_i\boldsymbol{\xi}_{i'}^\top + \boldsymbol{\xi}_{i'}\boldsymbol{\xi}_i^\top - \boldsymbol{\xi}_i\widetilde{\boldsymbol{\xi}}_i^\top - \widetilde{\boldsymbol{\xi}}_i\boldsymbol{\xi}_i^\top\Big]$$

$$= -\frac{\eta}{n_0^2\tau}\Bigg\{\Bigg[\sum_{i=1}^{n_0}\sum_{i'\neq i}2(y_iy_{i'}-1)\Bigg]\boldsymbol{\mu}\boldsymbol{\mu}^\top + 2(\sum_{i=1}^{n_0}y_i)\boldsymbol{\mu}(\sum_{i'=1}^{n_0}\boldsymbol{\xi}_{i'})^\top + 2(\sum_{i=1}^{n_0}y_i)(\sum_{i'=1}^{n_0}\boldsymbol{\xi}_{i'})\boldsymbol{\mu}^\top$$

$$- (n_0-1)\sum_{i=1}^{n_0}(y_i\boldsymbol{\mu}\widetilde{\boldsymbol{\xi}}_i^\top + y_i\widetilde{\boldsymbol{\xi}}_i\boldsymbol{\mu}^\top) - (n_0+1)\sum_{i=1}^{n_0}(y_i\boldsymbol{\mu}\boldsymbol{\xi}_i^\top + y_i\boldsymbol{\xi}_i\boldsymbol{\mu}^\top)$$

$$+ \sum_{i=1}^{n_0}\sum_{i'\neq i}(\boldsymbol{\xi}_i\boldsymbol{\xi}_{i'}^\top + \boldsymbol{\xi}_{i'}\boldsymbol{\xi}_i^\top) - \sum_{i=1}^{n_0}(n_0-1)\boldsymbol{\xi}_i\widetilde{\boldsymbol{\xi}}_i^\top - \sum_{i=1}^{n_0}(n_0-1)\widetilde{\boldsymbol{\xi}}_i\boldsymbol{\xi}_i^\top\Bigg\}$$

$$= -\frac{\eta}{n_0^2\tau}\Bigg\{\Bigg[2(\sum_{i=1}^{n_0}y_i)^2 - 2n_0^2\Bigg]\boldsymbol{\mu}\boldsymbol{\mu}^\top + 2(\sum_{i=1}^{n_0}y_i)\boldsymbol{\mu}(\sum_{i'=1}^{n_0}\boldsymbol{\xi}_{i'})^\top + 2(\sum_{i=1}^{n_0}y_i)(\sum_{i'=1}^{n_0}\boldsymbol{\xi}_{i'})\boldsymbol{\mu}^\top$$

$$- (n_0-1)\Big[\boldsymbol{\mu}(\sum_{i=1}^{n_0}y_i\widetilde{\boldsymbol{\xi}}_i)^\top + (\sum_{i=1}^{n_0}y_i\widetilde{\boldsymbol{\xi}}_i)\boldsymbol{\mu}^\top\Big] - (n_0+1)\Big[\boldsymbol{\mu}(\sum_{i=1}^{n_0}y_i\boldsymbol{\xi}_i)^\top + (\sum_{i=1}^{n_0}y_i\boldsymbol{\xi}_i)\boldsymbol{\mu}^\top\Big]$$

$$+ 2(\sum_{i=1}^{n_0}\boldsymbol{\xi}_i)(\sum_{i=1}^{n_0}\boldsymbol{\xi}_i)^\top - 2\sum_{i=1}^{n_0}\boldsymbol{\xi}_i\boldsymbol{\xi}_i^\top - \sum_{i=1}^{n_0}(n_0-1)\boldsymbol{\xi}_i\widetilde{\boldsymbol{\xi}}_i^\top - \sum_{i=1}^{n_0}(n_0-1)\widetilde{\boldsymbol{\xi}}_i\boldsymbol{\xi}_i^\top\Bigg\}.$$

Then by definition, we have

$$\boldsymbol{\Delta} = -\frac{\eta}{n_0^2\tau}[\boldsymbol{\Delta}_1 - \boldsymbol{\Delta}_2 - \boldsymbol{\Delta}_3 + \boldsymbol{\Delta}_4 - \boldsymbol{\Delta}_5 - \boldsymbol{\Delta}_6],$$

where

$$\boldsymbol{\Delta}_1 = 2(\sum_{i=1}^{n_0}y_i)\boldsymbol{\mu}(\sum_{i'=1}^{n_0}\boldsymbol{\xi}_{i'})^\top + 2(\sum_{i=1}^{n_0}y_i)(\sum_{i'=1}^{n_0}\boldsymbol{\xi}_{i'})\boldsymbol{\mu}^\top,$$

$$\boldsymbol{\Delta}_2 = (n_0-1)[\boldsymbol{\mu}(\sum_{i=1}^{n_0}y_i\widetilde{\boldsymbol{\xi}}_i)^\top + (\sum_{i=1}^{n_0}y_i\widetilde{\boldsymbol{\xi}}_i)\boldsymbol{\mu}^\top],$$

$$\boldsymbol{\Delta}_3 = (n_0+1)[\boldsymbol{\mu}(\sum_{i=1}^{n_0}y_i\boldsymbol{\xi}_i)^\top + (\sum_{i=1}^{n_0}y_i\boldsymbol{\xi}_i)\boldsymbol{\mu}^\top],$$

$$\boldsymbol{\Delta}_4 = 2(\sum_{i=1}^{n_0}\boldsymbol{\xi}_i)(\sum_{i=1}^{n_0}\boldsymbol{\xi}_i)^\top,$$

$$\boldsymbol{\Delta}_5 = 2\sum_{i=1}^{n_0}\boldsymbol{\xi}_i\boldsymbol{\xi}_i^\top,$$

$$\boldsymbol{\Delta}_6 = (n_0-1)\sum_{i=1}^{n_0}\boldsymbol{\xi}_i\widetilde{\boldsymbol{\xi}}_i^\top + (n_0-1)\sum_{i=1}^{n_0}\widetilde{\boldsymbol{\xi}}_i\boldsymbol{\xi}_i^\top.$$

Thus, $\|\boldsymbol{\Delta}_1\|_2$ can be bounded as follows,

$$\|\boldsymbol{\Delta}_1\|_2 \leq 4\left|\sum_{i=1}^{n_0}y_i\right| \cdot \|\boldsymbol{\mu}\|_2 \cdot \left\|\sum_{i'=1}^{n_0}\boldsymbol{\xi}_{i'}\right\|_2$$

$$\leq 4\sqrt{2n_0} \cdot \|\boldsymbol{\mu}\|_2 \cdot \widetilde{O}(\sigma_p\sqrt{n_0 d}), \tag{B.51}$$

where the second inequality is by $|\sum_{i=1}^{n_0} y_i| \leq \sqrt{2n_0 \log(2/\widetilde{\delta})}$ in Lemma B.5, and $\|\sum_{i=1}^{n_0} \boldsymbol{\xi}_i\|_2^2 \leq dn_0\sigma_p^2 + c_1 n_0 \sigma_p^2 \sqrt{d \log(2/\widetilde{\delta})}$ in Lemma B.6. Also, $\boldsymbol{\Delta}_2, \boldsymbol{\Delta}_3, \boldsymbol{\Delta}_4$ are handled similarly as $\boldsymbol{\Delta}_1$, namely

$$
\begin{aligned}
\|\boldsymbol{\Delta}_2\|_2 \leq & 2(n_0 - 1) \cdot \|\boldsymbol{\mu}\|_2 \cdot \left\| \sum_{i=1}^{n_0} y_i \widetilde{\boldsymbol{\xi}}_i \right\|_2 \\
\leq & 2(n_0 - 1) \cdot \|\boldsymbol{\mu}\|_2 \cdot \widetilde{O}(\sigma_p \sqrt{n_0 d}),
\end{aligned}
\tag{B.52}
$$

where the second inequality is by $\left\| \sum_{i=1}^{n_0} y_i \widetilde{\boldsymbol{\xi}}_i \right\|_2^2 \leq dn_0\sigma_p^2 + c_1 n_0 \sigma_p^2 \sqrt{d \log(2/\widetilde{\delta})}$ in Lemma B.6.

$$
\begin{aligned}
\|\boldsymbol{\Delta}_3\|_2 \leq & 2(n_0 + 1) \cdot \|\boldsymbol{\mu}\|_2 \cdot \left\| \sum_{i=1}^{n_0} y_i \boldsymbol{\xi}_i \right\|_2 \\
\leq & 2(n_0 + 1) \cdot \|\boldsymbol{\mu}\|_2 \cdot \widetilde{O}(\sigma_p \sqrt{n_0 d}),
\end{aligned}
\tag{B.53}
$$

where the second inequality is by $\|\sum_{i=1}^{n_0} y_i \boldsymbol{\xi}_i\|_2^2 \leq dn_0\sigma_p^2 + c_1 n_0 \sigma_p^2 \sqrt{d \log(2/\widetilde{\delta})}$ in Lemma B.6.

$$
\begin{aligned}
\|\boldsymbol{\Delta}_4\|_2 = & \left\| 2(\sum_{i=1}^{n_0} \boldsymbol{\xi}_i)(\sum_{i=1}^{n_0} \boldsymbol{\xi}_i)^\top \right\|_2 \\
\leq & 2 \cdot \left\| \sum_{i=1}^{n_0} \boldsymbol{\xi}_i \right\|_2^2 \\
\leq & 2\widetilde{O}(\sigma_p^2 n_0 d),
\end{aligned}
\tag{B.54}
$$

where the second inequality is by $\|\sum_{i=1}^{n_0} \boldsymbol{\xi}_i\|_2^2 \leq dn_0\sigma_p^2 + c_1 n_0 \sigma_p^2 \sqrt{d \log(2/\widetilde{\delta})}$ in Lemma B.6.

For $\boldsymbol{\Delta}_5$, by Theorem 5.39 in Vershynin (2010), with probability at least $1 - \widetilde{\delta}$, we have that

$$
\begin{aligned}
\|\boldsymbol{\Delta}_5\|_2 = & 2 \left\| \sum_{i=1}^{n_0} \boldsymbol{\xi}_i \boldsymbol{\xi}_i^\top \right\|_2 \\
\leq & 2\sigma_p^2 \left( \sqrt{d} + c_4\sqrt{n_0} + \frac{1}{\sqrt{c_4}} \cdot \sqrt{\log(2/\widetilde{\delta})} \right)^2 \leq 6\sigma_p^2 \left( d + c_4^2 n_0 + \frac{1}{c_4} \cdot \log(2/\widetilde{\delta}) \right),
\end{aligned}
\tag{B.55}
$$

where $c_4$ is an absolute constant. The bound for $\|\boldsymbol{\Delta}_6\|_2$ is already proved in Lemma B.7. Therefore, by (B.51), (B.52), (B.53), (B.54), (B.55), and Lemma B.7, set $\widetilde{\delta} = \delta/6$, with probability at least

$1 - \delta$, we have

$$\|\boldsymbol{\Delta}\|_2 = \|\boldsymbol{\Sigma} - \widehat{\boldsymbol{\Sigma}}\|_2 \leq \frac{\eta}{n_0^2 \tau} \left[ \|\boldsymbol{\Delta}_1\|_2 + \|\boldsymbol{\Delta}_2\|_2 + \|\boldsymbol{\Delta}_3\|_2 + \|\boldsymbol{\Delta}_4\|_2 + \|\boldsymbol{\Delta}_5\|_2 + \|\boldsymbol{\Delta}_6\|_2 \right]$$

$$\leq \frac{\eta}{n_0^2 \tau} \left[ 4\|\boldsymbol{\mu}\|_2 \sqrt{2n_0} \cdot \widetilde{O}(\sigma_p \sqrt{n_0 d}) + 2(n_0 - 1)\|\boldsymbol{\mu}\|_2 \cdot \widetilde{O}(\sigma_p \sqrt{n_0 d}) \right.$$

$$+ 2(n_0 + 1)\|\boldsymbol{\mu}\|_2 \cdot \widetilde{O}(\sigma_p \sqrt{n_0 d}) + 2\widetilde{O}(\sigma_p^2 n_0 d) + 6\sigma_p^2 \left( d + c_4^2 n_0 + \frac{1}{\sqrt{c_4}^2} \cdot \log(2/\widetilde{\delta}) \right)$$

$$\left. + c_3 n_0^2 \sigma_p^2 \cdot \max \left\{ \sqrt{\frac{d \log(9/\widetilde{\delta})}{n_0}}, \frac{d \log(9/\widetilde{\delta})}{n_0} \right\} \right]$$

$$\leq \left[ 4\sqrt{2}\|\boldsymbol{\mu}\|_2^{-1} \cdot \widetilde{O}(\sigma_p \sqrt{d} n_0^{-1}) + 2\|\boldsymbol{\mu}\|_2^{-1} \cdot \widetilde{O}(\sigma_p \sqrt{d} \frac{1}{\sqrt{n_0}}) \right.$$

$$+ 2\|\boldsymbol{\mu}\|_2^{-1} \cdot \widetilde{O}(\sigma_p \sqrt{d} \frac{1}{\sqrt{n_0}}) + 2\|\boldsymbol{\mu}\|_2^{-2} \cdot \widetilde{O}(\sigma_p^2 d n_0^{-1}) + \|\boldsymbol{\mu}\|_2^{-2} \cdot \widetilde{O}(\sigma_p^2 d n_0^{-1})$$

$$\left. + c_3 \sigma_p^2 d \|\boldsymbol{\mu}\|_2^{-2} \cdot \max \left\{ \sqrt{\frac{\log(9/\widetilde{\delta})}{d n_0}}, \frac{\log(9/\widetilde{\delta})}{n_0} \right\} \right] \frac{\eta}{\tau} \|\boldsymbol{\mu}\|_2^2,$$

$$\leq \left[ \widetilde{O}(\mathrm{SNR}^{-1} \cdot n_0^{-1}) + \widetilde{O}(\mathrm{SNR}^{-1} \cdot \frac{1}{\sqrt{n_0}}) + \widetilde{O}(\mathrm{SNR}^{-2} \cdot n_0^{-1}) \right.$$

$$\left. + c_3 \cdot \mathrm{SNR}^{-2} \cdot \max \left\{ \sqrt{\frac{\log(9/\widetilde{\delta})}{d n_0}}, \frac{\log(9/\widetilde{\delta})}{n_0} \right\} \right] \frac{\eta}{\tau} \|\boldsymbol{\mu}\|_2^2,$$

where $\mathrm{SNR} = \|\boldsymbol{\mu}\|_2/(\sigma_p \sqrt{d})$, and $\frac{\eta}{\tau}\|\boldsymbol{\mu}\|_2^2$ is the lower bound of $\widehat{\lambda}_1$ proved in Lemma 5.2. $\qquad\square$

### B.4 Proofs of lemmas in Appendix B.1.3

#### B.4.1 Proof of Lemma B.3

With a proof similar to Lemma B.3 in Cao et al. (2022), we have the following Lemma B.3. Although the proof is almost the same as in Cao et al. (2022), since the results are presented in different forms, for self-consistency, we still present the proof of this Lemma B.3.

*Proof of Lemma B.3.* Since $\mathbf{v}$ is a unit vector, for each $r \in [2m]$, $j \cdot \langle \mathbf{w}_r^{(0)}, \mathbf{v}_j \rangle$ is a Gaussian random variable with mean zero and variance $\sigma_0^2$. Therefore, by Gaussian tail bound and union bound, with probability at least $1 - \delta/2$,

$$|\langle \mathbf{w}_r^{(0)}, \mathbf{v}_j \rangle| \leq \sqrt{2 \log(16 m n_0/\delta)} \cdot \sigma_0 \tag{B.56}$$

for all $r \in [2m]$ and $j \in [d]$. This proves the first part of the result. For the second part of the result, we note that $\mathbb{P}(\langle \mathbf{w}_r^{(0)}, \mathbf{v}_1 \rangle \geq \sigma_0/2) = \mathbb{P}_{Z \sim \mathcal{N}(0,1)}(Z \geq 1/2) \geq 0.3$ is an absolute constant. Therefore, binary random variables $\mathbb{1}\{\langle \mathbf{w}_r^{(0)}, \mathbf{v}_1 \rangle \geq \sigma_0/2\}$, $r \in [2m]$ are independent $\mathrm{Bernoulli}(p)$ random variables with constant $0.3 \leq p \leq 0.5$. By Hoeffding's inequality, with probability at least $1 - \delta/4$,

$$\left| \sum_{r=1}^{2m} \mathbb{1}\{\langle \mathbf{w}_r^{(0)}, \mathbf{v}_1 \rangle \geq \sigma_0/2\} - 2mp \right| \leq \sqrt{2m \log(8/\delta)}.$$

Therefore, with probability at least $1 - \delta/4$,

$$\sum_{r=1}^{2m} \mathbb{1}\{\langle \mathbf{w}_r^{(0)}, \mathbf{v}_1 \rangle \geq \sigma_0/2\} \geq 2mp - \sqrt{2m \log(8/\delta)} \geq 2m/5,$$

where the last inequality holds by the assumption that $m = \widetilde{\Omega}(1)$. The inequality above implies that there exist distinct $r_1^+, \ldots, r_{2m/5}^+ \in [2m]$ such that $\langle \mathbf{w}_r^{(0)}, \mathbf{v}_1 \rangle \geq \sigma_0/2$ for all $r \in \{r_1^+, \ldots, r_{2m/5}^+\}$.

With exactly the same proof, we also have that, with probability at least $1 - \delta/4$, there exist distinct $r_1^-, \ldots, r_{2m/5}^- \in [2m]$ such that $\langle \mathbf{w}_r^{(0)}, \mathbf{v}_1 \rangle \leq -\sigma_0/2$ for all $r \in \{r_1^-, \ldots, r_{2m/5}^-\}$. It is also clear that as long as the sets $\{r_1^+, \ldots, r_{2m/5}^+\}$ and $\{r_1^-, \ldots, r_{2m/5}^-\}$ exist, they must be disjoint. Therefore, applying a union bound finishes the proof. □

### B.4.2  PROOF OF LEMMA B.4

*Proof of Lemma B.4.* The first conclusion that $|\lambda_i - \widetilde{\lambda}_i| \leq \sigma_0 \cdot \|\mathbf{A}\|_2$, $i \in [n]$ directly follows by Weyl's theorem and the assumption that $\mathbf{\Xi}$ is symmetric and $\|\mathbf{\Xi}\|_2 \leq \sigma_0 \cdot \|\mathbf{A}\|_2$.

For the second result, we have

$$\sin\theta(\widetilde{\mathbf{v}}_1, \mathbf{v}_1) \leq \frac{\|\mathbf{A} - (\mathbf{A} + \mathbf{\Xi})\|_2}{|\widetilde{\lambda}_2 - \lambda_1|}$$

$$\leq \frac{\sigma_0 \cdot \lambda_1}{(1 - \sigma_0)\lambda_1 - \lambda_2}$$

$$\leq \frac{\sigma_0 \cdot \lambda_1}{(1 - \sigma_0)\lambda_1 - \frac{\mathcal{E}_{\mathrm{SimCLR}}}{2(1 - \mathcal{E}_{\mathrm{SimCLR}})} \cdot \lambda_1}$$

$$\leq 2\sigma_0,$$

where the first inequality follows by the variant of Davis-Kahan Theorem (Theorem 1 in Yu et al. (2015)), the second inequality follows by the first conclusion of is lemma (which has been proved above) and the assumption that $\|\mathbf{\Xi}\|_2 \leq \sigma_0 \cdot \lambda_1$, the third inequality follows by Lemma 5.2, and the fourth inequality follows by the assumption that $\mathcal{E}_{\mathrm{SimCLR}} \leq 1/4$ and $\sigma_0 \leq 1/4$. Then we have

$$|\langle \mathbf{v}_1, \widetilde{\mathbf{v}}_1 \rangle| = \sqrt{1 - \sin^2\theta(\widetilde{\mathbf{v}}_1, \mathbf{v}_1)} \geq \sqrt{1 - 4\sigma_0^2} \geq 1 - 4\sigma_0^2.$$

This proves the second conclusion. For the last result, for $i = 2, \ldots, d$, since $\widetilde{\mathbf{v}}_i$ is perpendicular to $\widetilde{\mathbf{v}}_1$, we have

$$|\langle \mathbf{v}_1, \widetilde{\mathbf{v}}_i \rangle| = |\langle \mathbf{v}_1, (\mathbf{I} - \widetilde{\mathbf{v}}_1 \widetilde{\mathbf{v}}_1^\top + \widetilde{\mathbf{v}}_1 \widetilde{\mathbf{v}}_1^\top) \widetilde{\mathbf{v}}_i \rangle|$$

$$= |\langle \mathbf{v}_1, (\mathbf{I} - \widetilde{\mathbf{v}}_1 \widetilde{\mathbf{v}}_1^\top) \widetilde{\mathbf{v}}_i \rangle|$$

$$\leq \|(\mathbf{I} - \widetilde{\mathbf{v}}_1 \widetilde{\mathbf{v}}_1^\top) \mathbf{v}_1\|_2$$

$$= \sqrt{1 - \langle \mathbf{v}_1, \widetilde{\mathbf{v}}_1 \rangle^2}$$

$$\leq \sqrt{1 - (1 - 4\sigma_0^2)^2}$$

$$= \sqrt{8\sigma_0^2 - 16\sigma_0^4}$$

$$\leq \sqrt{8\sigma_0^2}$$

$$\leq 4\sigma_0.$$

To prove the last inequality, we have

$$\|(\mathbf{I} - \mathbf{v}_1 \mathbf{v}_1^\top) \widetilde{\mathbf{v}}_1\|_2 = \sqrt{\|\widetilde{\mathbf{v}}_1 - \langle \widetilde{\mathbf{v}}_1, \mathbf{v}_1 \rangle \cdot \mathbf{v}_1\|_2^2}$$

$$= \sqrt{1 - 2\langle \widetilde{\mathbf{v}}_1, \mathbf{v}_1 \rangle^2 + \langle \widetilde{\mathbf{v}}_1, \mathbf{v}_1 \rangle^2}$$

$$= \sqrt{1 - \langle \widetilde{\mathbf{v}}_1, \mathbf{v}_1 \rangle^2}$$

$$\leq \sqrt{1 - (1 - 4\sigma_0^2)^2}$$

$$\leq \sqrt{8\sigma_0^2}$$

$$\leq 4\sigma_0.$$

This finishes the proof. □

## C    Proofs for supervised fine-tuning

In this section, the training process of the fine-tuning stage is investigated. We first present the basic setting and the decomposition of coefficients.

The initialization $\mathbf{W}^{(0)}$ of the fine-tuning stage is derived by the pre-training stage, and will be fine-tuned in the following stage based on the CNN model (3.2). It can be directly decomposed as

$$\overline{\mathbf{w}}_{jr}^{(0)} = \mathbf{w}_{jr}^{\perp} + j \cdot \gamma_{jr}^{(0)} \cdot \|\boldsymbol{\mu}\|_2^{-2} \cdot \boldsymbol{\mu} + \sum_{i=1}^{n} \rho_{jri}^{(0)} \cdot \|\boldsymbol{\xi}_i\|_2^{-2} \cdot \boldsymbol{\xi}_i, \tag{C.1}$$

where $\mathbf{w}_{jr}^{\perp}$ is a component of $\overline{\mathbf{w}}_{jr}^{(0)}$ perpendicular with $\langle \mathbf{w}_{jr}^{\perp}, \boldsymbol{\mu} \rangle = 0$, $\langle \mathbf{w}_{jr}^{\perp}, \boldsymbol{\xi}_i \rangle = 0, i \in [n]$, and we have $\max_{j,r} \|\mathbf{w}_{jr}^{\perp}\|_2 \le 1/n$ by Theorem 5.3.

In the fine-tuning stage, based on the gradient descent algorithm and the CNN structure defined in (3.2), the updating rules of $\mathbf{w}_{j,r}, j \in \{-1, +1\}, r \in [m]$ is given as

$$\mathbf{w}_{j,r}^{(t+1)} = \mathbf{w}_{j,r}^{(t)} - \eta \cdot \nabla_{\mathbf{w}_{j,r}} L_S(\boldsymbol{W}^{(t)})$$

$$= \mathbf{w}_{j,r}^{(t)} - \frac{\eta}{nm} \sum_{i=1}^{n} \ell_i'^{(t)} \cdot \sigma'(\langle \mathbf{w}_{j,r}^{(t)}, \boldsymbol{\xi}_i \rangle) \cdot j y_i \boldsymbol{\xi}_i - \frac{\eta}{nm} \sum_{i=1}^{n} \ell_i'^{(t)} \cdot \sigma'(\langle \mathbf{w}_{j,r}^{(t)}, y_i \boldsymbol{\mu} \rangle) \cdot j \boldsymbol{\mu},$$

where $\ell_i'^{(t)} = \ell'[y_i \cdot f(\mathbf{W}^{(t)}, \mathbf{x}_i)]$.

The convolution filters $\mathbf{w}_{j,r}^{(t)}$, $r \in [m]$, $j \in \{+1, -1\}$ can be decomposed into the following format. There exist unique coefficient $\gamma_{j,r}^{(t)}$ and $\rho_{j,r,i}^{(t)}$ such that

$$\mathbf{w}_{j,r}^{(t)} = \mathbf{w}_{jr}^{\perp} + j \cdot \gamma_{j,r}^{(t)} \cdot \|\boldsymbol{\mu}\|_2^{-2} \cdot \boldsymbol{\mu} + \sum_{i=1}^{n} \rho_{j,r,i}^{(t)} \cdot \|\boldsymbol{\xi}_i\|_2^{-2} \cdot \boldsymbol{\xi}_i, \ \ t \ge 0. \tag{C.2}$$

If further decompose $\rho_{j,r,i}^{(t)}$ into $\overline{\rho}_{j,r,i}^{(t)} := \rho_{j,r,i}^{(t)} \mathbb{1}(\rho_{j,r,i}^{(t)} \ge 0)$, $\underline{\rho}_{j,r,i}^{(t)} := \rho_{j,r,i}^{(t)} \mathbb{1}(\rho_{j,r,i}^{(t)} \le 0)$, then the decomposition of $\mathbf{w}_{j,r}^{(t)}$ can be converted into

$$\mathbf{w}_{j,r}^{(t)} = \mathbf{w}_{jr}^{\perp} + j \cdot \gamma_{j,r}^{(t)} \cdot \|\boldsymbol{\mu}\|_2^{-2} \cdot \boldsymbol{\mu} + \sum_{i=1}^{n} \overline{\rho}_{j,r,i}^{(t)} \cdot \|\boldsymbol{\xi}_i\|_2^{-2} \cdot \boldsymbol{\xi}_i + \sum_{i=1}^{n} \underline{\rho}_{j,r,i}^{(t)} \cdot \|\boldsymbol{\xi}_i\|_2^{-2} \cdot \boldsymbol{\xi}_i, \ \ t \ge 0. \tag{C.3}$$

Based on the updating rules (3.3) and decomposition (C.3) of $\mathbf{w}_{j,r}^{(t)}$, the updating rules of coefficients $\gamma_{j,r}^{(t)}, \overline{\rho}_{j,r,i}^{(t)}, \underline{\rho}_{j,r,i}^{(t)}$ are as follows

$$\gamma_{j,r}^{(0)}, \overline{\rho}_{j,r,i}^{(0)}, \underline{\rho}_{j,r,i}^{(0)} \neq 0, \tag{C.4}$$

$$\gamma_{j,r}^{(t+1)} = \gamma_{j,r}^{(t)} - \frac{\eta}{nm} \cdot \sum_{i=1}^{n} \ell_i'^{(t)} \cdot \sigma'(\langle \mathbf{w}_{j,r}^{(t)}, y_i \cdot \boldsymbol{\mu} \rangle) \cdot \|\boldsymbol{\mu}\|_2^2, \tag{C.5}$$

$$\overline{\rho}_{j,r,i}^{(t+1)} = \overline{\rho}_{j,r,i}^{(t)} - \frac{\eta}{nm} \cdot \ell_i'^{(t)} \cdot \sigma'(\langle \mathbf{w}_{j,r}^{(t)}, \boldsymbol{\xi}_i \rangle) \cdot \|\boldsymbol{\xi}_i\|_2^2 \cdot \mathbb{1}(y_i = j), \tag{C.6}$$

$$\underline{\rho}_{j,r,i}^{(t+1)} = \underline{\rho}_{j,r,i}^{(t)} + \frac{\eta}{nm} \cdot \ell_i'^{(t)} \cdot \sigma'(\langle \mathbf{w}_{j,r}^{(t)}, \boldsymbol{\xi}_i \rangle) \cdot \|\boldsymbol{\xi}_i\|_2^2 \cdot \mathbb{1}(y_i = -j). \tag{C.7}$$

It follows that by substitute $\langle \mathbf{w}_{j,r}^{(t)}, y_i \cdot \boldsymbol{\mu} \rangle$, $\langle \mathbf{w}_{j,r}^{(t)}, \boldsymbol{\xi}_i \rangle$ in (C.5)-(C.7), we have,

$$\gamma_{j,r}^{(0)}, \overline{\rho}_{j,r,i}^{(0)}, \underline{\rho}_{j,r,i}^{(0)} \neq 0,$$

$$\gamma_{j,r}^{(t+1)} = \gamma_{j,r}^{(t)} - \frac{\eta}{nm} \cdot \sum_{i=1}^{n} \ell_i'^{(t)} \cdot \sigma'(jy_i \cdot \gamma_{j,r}^{(t)}) \cdot \|\boldsymbol{\mu}\|_2^2,$$

$$\rho_{j,r,i}^{(t+1)} = \rho_{j,r,i}^{(t)} - \frac{\eta}{nm} \cdot \ell_i'^{(t)} \cdot \sigma'(\sum_{i'=1}^{n} \rho_{j,r,i'}^{(t)} \frac{\langle \boldsymbol{\xi}_{i'}, \boldsymbol{\xi}_i \rangle}{\|\boldsymbol{\xi}_{i'}\|_2^2}) \cdot \|\boldsymbol{\xi}_i\|_2^2 ;$$

$$\overline{\rho}_{j,r,i}^{(t+1)} = \overline{\rho}_{j,r,i}^{(t)} - \frac{\eta}{nm} \cdot \ell_i'^{(t)} \cdot \sigma'(\sum_{i'=1}^{n} \overline{\rho}_{j,r,i'}^{(t)} \frac{\langle \boldsymbol{\xi}_{i'}, \boldsymbol{\xi}_i \rangle}{\|\boldsymbol{\xi}_{i'}\|_2^2} + \sum_{i'=1}^{n} \underline{\rho}_{j,r,i'}^{(t)} \frac{\langle \boldsymbol{\xi}_{i'}, \boldsymbol{\xi}_i \rangle}{\|\boldsymbol{\xi}_{i'}\|_2^2}) \cdot \|\boldsymbol{\xi}_i\|_2^2 \cdot \mathbb{1}(y_i = j),$$

$$\underline{\rho}_{j,r,i}^{(t+1)} = \underline{\rho}_{j,r,i}^{(t)} + \frac{\eta}{nm} \cdot \ell_i'^{(t)} \cdot \sigma'(\sum_{i'=1}^{n} \overline{\rho}_{j,r,i'}^{(t)} \frac{\langle \boldsymbol{\xi}_{i'}, \boldsymbol{\xi}_i \rangle}{\|\boldsymbol{\xi}_{i'}\|_2^2} + \sum_{i'=1}^{n} \underline{\rho}_{j,r,i'}^{(t)} \frac{\langle \boldsymbol{\xi}_{i'}, \boldsymbol{\xi}_i \rangle}{\|\boldsymbol{\xi}_{i'}\|_2^2}) \cdot \|\boldsymbol{\xi}_i\|_2^2 \cdot \mathbb{1}(y_i = -j).$$

$$(C.8)$$

The coefficients initialization (C.4) is determined by the pre-training stage, which is given in (C.1), while the one-step updating rules for the coefficients are not influenced by the initialization.

Denote $T^* = \eta^{-1}\text{poly}(\epsilon^{-1}, \|\boldsymbol{\mu}\|_2^{-1}, d^{-1}\sigma_p^{-2}, n, m, d)$ the maximum admissible iterations.

Based on the result of pre-training stage in Theorem 5.3, it is easy to verify that the following assumptions hold.

**Assumption C.1** (Assumptions on the scale of initialization). *Assume the following equations hold,*

$$-4m^{\frac{1}{q}}\log(T^*) \leq -C_0 \leq \gamma_{j,r}^{(0)} \leq 4m^{\frac{1}{q}}\log(T^*)$$

$$0 \leq \overline{\rho}_{j,r,i}^{(0)} \leq 4m^{\frac{1}{q}}\log(T^*)$$

$$0 \geq \underline{\rho}_{j,r,i}^{(0)} \geq -64nm^{\frac{1}{q}}\sqrt{\frac{\log(4n^2/\delta)}{d}} \cdot \log(T^*)$$

*for all $r \in [m]$, $j \in \{\pm 1\}$ and all $i \in [n]$, where $C_0$ is constant such that $0 \leq C_0 \leq 4m^{\frac{1}{q}}\log(T^*)$.*

**Assumption C.2** (Assumptions on the initialization of $\gamma$). *There exists at least one index $r_1 \in [m]$ such that $\gamma_{1,r_1}^{(0)} \geq \gamma_0$, and there exists at least one index $r_2 \in [m]$ such that $\gamma_{-1,r_2}^{(0)} \geq \gamma_0$. Furthermore, we require that*

$$\max_{j,r}\{0, (-\gamma_{j,r}^{(0)})^q\} \ll 4m^{\frac{1}{q}}\log(T^*) \tag{C.9}$$

### C.1 PROOF OF THEOREM 5.5

In this section, in order to prove Theorem 5.5, the following Lemma C.3-C.6 are introduced to analyze the signal learning in the fine-tuning stage. Two stages of the signal learning as well as the population loss are analyzed here.

#### C.1.1 FIRST STAGE OF SIGNAL LEARNING

**Lemma C.3.** *Under the same conditions as Theorem 5.5, in particular if the SNR satisfies that*

$$\text{SNR}^2 \geq \frac{4\log(2/\gamma_0)8^q\rho_0^{q-2}}{C_1 n\gamma_0^{q-2}} \tag{C.10}$$

*where $C_1 = O(1)$ is a positive constant, there exists time*

$$T_1 = \frac{\log(2/\gamma_0)8m}{C_1\eta q\gamma_0^{q-2}\|\boldsymbol{\mu}\|_2^2}$$

*such that*

$$\max_r \gamma_{j,r}^{(T_1)} \geq 2, \text{ for } j \in \{\pm 1\}. \tag{C.11}$$

$$|\rho_{j,r,i}^{(t)}| \leq 2\rho_0, \text{ for all } j \in \{\pm 1\}, r \in [m], i \in [n], 0 \leq t \leq T_1. \tag{C.12}$$

*where $\rho_0 = \max_{j,r,i}|\rho_{j,r,i}^{(0)}|$, $\gamma_0$ is as defined in Assumption C.3.*

### C.1.2 SECOND STAGE OF SIGNAL LEARNING

Based on the result of First Stage (Section C.1.1) in Lemma C.3, at the beginning of the second stage, we have the following properties,

- $\max_r \gamma_{j,r}^{(T_1)} \geq 2$, $j \in \{\pm 1\}$.

- $\max_{j,r,i} |\rho_{j,r,i}^{(T_1)}| \leq 2\rho_0$.

The learned feature $\gamma_{j,r}^{(t)}$ will not get worse, i.e., for $t \geq T_1$, we have that $\gamma_{j,r}^{(t+1)} \geq \gamma_{j,r}^{(t)}$, and therefore $\max_r \gamma_{j,r}^{(t)} \geq 2$. Now we choose $\mathbf{W}^*$ as follows:

$$\mathbf{w}_{j,r}^* = \mathbf{w}_{jr}^\perp + 2qm\log(2q/\epsilon)\cdot j \cdot \frac{\boldsymbol{\mu}}{\|\boldsymbol{\mu}\|_2^2}, \quad j \in \{+1,-1\}, \ r \in [m]. \tag{C.13}$$

Based on the above definition of $\mathbf{W}^*$, we have the following Lemma C.4.

**Lemma C.4.** *Under the same conditions as Theorem 5.5, we have that* $\|\mathbf{W}^{(T_1)} - \mathbf{W}^*\|_F \leq \widetilde{O}(m^{3/2}\|\boldsymbol{\mu}\|_2^{-1}) + O(nm\rho_0(\sigma_p\sqrt{d})^{-1})$.

**Lemma C.5.** *Under the same conditions as Theorem 5.5, let* $T = T_1 + \left\lfloor \frac{\|\mathbf{W}^{(T_1)} - \mathbf{W}^*\|_F^2}{2\eta\epsilon} \right\rfloor = T_1 + \widetilde{O}(m^3\eta^{-1}\epsilon^{-1}\|\boldsymbol{\mu}\|_2^{-2})$. *Then we have* $\max_{j,r,i} |\rho_{j,r,i}^{(t)}| \leq 4\rho_0$ *for all* $T_1 \leq t \leq T$. *Besides,*

$$\frac{1}{t - T_1 + 1}\sum_{s=T_1}^{t} L_S(\mathbf{W}^{(s)}) \leq \frac{\|\mathbf{W}^{(T_1)} - \mathbf{W}^*\|_F^2}{(2q-1)\eta(t - T_1 + 1)} + \frac{\epsilon}{2q-1}$$

*for all* $T_1 \leq t \leq T$, *and we can find an iteration with training loss smaller than* $\epsilon$ *within* $T$ *iterations.*

### C.1.3 POPULATION LOSS

In this section, the bound of the test loss is presented. For a new data point $(\mathbf{x}, y)$ drawn from the same distribution as training data generated from. Without loss of generality, we assume that the data point has the following structure: the first patch is the signal patch and the second patch is the noise patch, i.e., $\mathbf{x} = [y\boldsymbol{\mu}, \boldsymbol{\xi}]$.

**Lemma C.6.** *Let* $T$ *the same as defined in Lemma C.5 in Second Stage (Section C.1.2). Under the same conditions as Theorem 5.5, for any* $0 \leq t \leq T$ *with* $L_S(\mathbf{W}^{(t)}) \leq \frac{1}{4}$, *it holds that* $L_{\mathcal{D}}(\mathbf{W}^{(t)}) \leq 6 \cdot L_S(\mathbf{W}^{(t)}) + \exp(-\widetilde{\Omega}(n^2))$.

Then, based on the above lemmas, we provide a simplified version of the proof for Theorem 5.5.

*Proof of Theorem 5.5.* For the first result in Theorem 5.5, based on the result of the pre-training stage in Theorem 5.3, we have that the conditions of Lemma C.3 hold. The result of First Stage signal learning in Lemma C.3 hold. Then, we could define $\mathbf{W}^*$ as (C.13), and by Lemma C.4, we have

$$\|\mathbf{W}^{(T_1)} - \mathbf{W}^*\|_F \leq \widetilde{O}(m^{3/2}\|\boldsymbol{\mu}\|_2^{-1}) + O(nm\rho_0(\sigma_p\sqrt{d})^{-1}).$$

It follows that for any $\epsilon > 0$, choose $T = T_1 + \widetilde{O}(m^3\eta^{-1}\epsilon^{-1}\|\boldsymbol{\mu}\|_2^{-2})$, by Lemma C.5, we have that

$$\frac{1}{T - T_1 + 1}\sum_{s=T_1}^{T} L_S(\mathbf{W}^{(s)}) \leq \frac{\|\mathbf{W}^{(T_1)} - \mathbf{W}^*\|_F^2}{(2q-1)\eta(T - T_1 + 1)} + \frac{\epsilon}{2q-1} \leq \frac{3\epsilon}{2q-1} < \epsilon.$$

Therefore, there exists some $T_1 \leq t \leq T$ with $L_S(\mathbf{W}^{(t)}) \leq \epsilon$. This completes the proof of the first result. Then combine this with Lemma C.6, the second result of Theorem 5.5 is given by

$$L_{\mathcal{D}}(\mathbf{W}^{(t)}) \leq 6 \cdot L_S(\mathbf{W}^{(t)}) + \exp(-\widetilde{\Omega}(n^2)).$$

$\square$

## C.2 PROOF OF LEMMAS IN SECTION C.1

### C.2.1 PROOF OF LEMMA C.3

To prove Lemma C.3, we first introduce the following Lemma C.7, C.8, C.9 and C.10.

**Lemma C.7.** *Suppose that $\delta > 0$ and $n \geq \Omega(\log(1/\delta))$ Then with probability at least $1 - \delta$,*

$$|\{i \in [n] : y_i = 1\}|, \ |\{i \in [n] : y_i = -1\}| \geq n/4.$$

The following Lemma C.8 provides an estimate of the norm of $\boldsymbol{\xi}_i$ and a bound of their inner products between each other.

**Lemma C.8.** *Suppose that $\delta > 0$ and $d = \Omega(\log(4n/\delta))$. Then with probability at least $1 - \delta$,*

$$\sigma_p^2 d/2 \leq \|\boldsymbol{\xi}_i\|_2^2 \leq 3\sigma_p^2 d/2,$$
$$|\langle \boldsymbol{\xi}_i, \boldsymbol{\xi}_{i'} \rangle| \leq 2\sigma_p^2 \cdot \sqrt{d \log(4n^2/\delta)},$$

*for all $i, i' \in [n]$.*

**Lemma C.9.** *Under Condition 4.1, suppose (C.16), (C.17) and (C.18) hold at iteration $t$. Then*

$$\langle \mathbf{w}_{j,r}^{(t)}, y_i \boldsymbol{\mu} \rangle \leq \max_{j,r}\{0, -\gamma_{j,r}^{(0)}\},$$
$$\langle \mathbf{w}_{j,r}^{(t)}, \boldsymbol{\xi}_i \rangle \leq 32nm^{\frac{1}{q}}\sqrt{\frac{\log(4n^2/\delta)}{d}} \cdot \log(T^*),$$

*for all $r \in [m]$ and $j \neq y_i$. Since by Assumption C.1, $\max_{j,r}\{-\gamma_{j,r}^{(0)}\} \leq C_0$, we further have that $F_j(\mathbf{W}_j^{(t)}, \mathbf{x}_i) = O(1)$.*

**Lemma C.10.** *Under Condition 4.1, suppose (C.16), (C.17) and (C.18) hold at iteration $t$. Then*

$$\langle \mathbf{w}_{j,r}^{(t)}, y_i \boldsymbol{\mu} \rangle = \gamma_{j,r}^{(t)},$$
$$\langle \mathbf{w}_{j,r}^{(t)}, \boldsymbol{\xi}_i \rangle \leq \overline{\rho}_{j,r,i}^{(t)} + 32nm^{\frac{1}{q}}\sqrt{\frac{\log(4n^2/\delta)}{d}} \cdot \log(T^*)$$

*for all $r \in [m]$, $j = y$ and $i \in [n]$. If $\max_{j,r,i}\{\gamma_{j,r}^{(t)}, \overline{\rho}_{j,r,i}^{(t)}\} = O(1)$, we further have that $F_j(\mathbf{W}_j^{(t)}, \mathbf{x}_i) = O(1)$.*

Based on the above Lemma C.7, C.8, C.9 and C.10, we could prove the Lemma C.3 now.

*Proof of Lemma C.3.* Let

$$T_1^+ = \frac{1}{\frac{2\eta q}{nm}\sigma_p^2 d \cdot 8^{q-1}\rho_0^{q-2}}. \tag{C.14}$$

We first prove the second conclusion (C.12). Define $\Psi^{(t)} = \max_{j,r,i}|\rho_{j,r,i}^{(t)}| = \max_{j,r,i}\{\overline{\rho}_{j,r,i}^{(t)}, -\underline{\rho}_{j,r,i}^{(t)}\}$. We use induction to show that

$$\Psi^{(t)} \leq 2\rho_0 \tag{C.15}$$

for all $0 \le t \le T_1^+$. By definition, clearly we have $\Psi^{(0)} = \rho_0$. Now suppose that there exists some $\widetilde{T} \le T_1^+$ such that (C.15) holds for $0 < t \le \widetilde{T} - 1$. Then by (C.8) we have

$$\Psi^{(t+1)} \le \Psi^{(t)} + \max_{j,r,i} \left\{ \frac{\eta}{nm} \cdot |\ell_i'^{(t)}| \cdot \sigma'\left( \sum_{i'=1}^n \Psi^{(t)} \cdot \frac{|\langle \boldsymbol{\xi}_{i'}, \boldsymbol{\xi}_i \rangle|}{\|\boldsymbol{\xi}_{i'}\|_2^2} + \sum_{i'=1}^n \Psi^{(t)} \cdot \frac{|\langle \boldsymbol{\xi}_{i'}, \boldsymbol{\xi}_i \rangle|}{\|\boldsymbol{\xi}_{i'}\|_2^2} \right) \cdot \|\boldsymbol{\xi}_i\|_2^2 \right\}$$

$$\le \Psi^{(t)} + \max_{j,r,i} \left\{ \frac{\eta}{nm} \cdot \sigma'\left( 2 \cdot \sum_{i'=1}^n \Psi^{(t)} \cdot \frac{|\langle \boldsymbol{\xi}_{i'}, \boldsymbol{\xi}_i \rangle|}{\|\boldsymbol{\xi}_{i'}\|_2^2} \right) \cdot \|\boldsymbol{\xi}_i\|_2^2 \right\}$$

$$= \Psi^{(t)} + \max_{j,r,i} \left\{ \frac{\eta}{nm} \cdot \sigma'\left( 2\Psi^{(t)} + 2 \cdot \sum_{i' \ne i}^n \Psi^{(t)} \cdot \frac{|\langle \boldsymbol{\xi}_{i'}, \boldsymbol{\xi}_i \rangle|}{\|\boldsymbol{\xi}_{i'}\|_2^2} \right) \cdot \|\boldsymbol{\xi}_i\|_2^2 \right\}$$

$$\le \Psi^{(t)} + \frac{\eta q}{nm} \cdot \left[ \left( 2 + \frac{4n\sigma_p^2 \cdot \sqrt{d \log(4n^2/\delta)}}{\sigma_p^2 d/2} \right) \cdot \Psi^{(t)} \right]^{q-1} \cdot 2\sigma_p^2 d$$

$$\le \Psi^{(t)} + \frac{\eta q}{nm} \cdot \left( 4\Psi^{(t)} \right)^{q-1} \cdot 2\sigma_p^2 d$$

$$\le \Psi^{(t)} + \frac{\eta q}{nm} \cdot \left( 8\rho_0 \right)^{q-1} \cdot 2\sigma_p^2 d,$$

where the second inequality is by $|\ell_i'^{(t)}| \le 1$, the third inequality is due to Lemma C.8, the fourth inequality follows by the condition that $d \ge 16n^2 \log(4n^2/\delta)$ in Condition 4.1, and the last inequality follows by the induction hypothesis (C.15). Taking a telescoping sum over $t = 0, 1, \ldots, \widetilde{T} - 1$ then gives

$$\Psi^{(\widetilde{T})} \le \Psi^{(0)} + \widetilde{T} \frac{\eta q}{nm} \cdot \left( 8\rho_0 \right)^{q-1} \cdot 2\sigma_p^2 d$$

$$\le \rho_0 + T_1^+ \frac{\eta q}{nm} \cdot \left( 8\rho_0 \right)^{q-1} \cdot 2\sigma_p^2 d$$

$$\le 2\rho_0,$$

where the second inequality follows by $\widetilde{T} \le T_1^+$ in our induction hypothesis. Therefore, by induction, we prove that $\Psi^{(t)} \le 2\rho_0$ for all $t \le T_1^+$.

To prove the first conclusion (C.11), without loss of generality, consider $j = 1$ first (similar ideas for the proof of $j = -1$). Denote by $T_{1,1}$ the last time for $t$ in $[0, T_1^+]$ satisfying that $\max_r \gamma_{1,r}^{(t)} \le 2$. Then for $t \le T_{1,1}$, $\max_{j,r,i}\{|\rho_{j,r,i}^{(t)}|\} = O(\rho_0 \sigma_p^2 d) = O(1)$ and $\max_r \gamma_{1,r}^{(t)} \le 2$. Therefore, by Lemma C.9 and C.10, we know that $F_{-1}(\mathbf{W}_{-1}^{(t)}, \mathbf{x}_i), F_{+1}(\mathbf{W}_{+1}^{(t)}, \mathbf{x}_i) = O(1)$ for all $i$ with $y_i = 1$. Thus, there exists a positive constant $C_1$ such that $-\ell_i'^{(t)} \ge C_1$ for all $i$ with $y_i = 1$.

Since (C.11) focuses on the $\max_r \gamma_{1,r}^{(t)}$, we only need to consider the training dynamic of $\max_r \gamma_{1,r}^{(t)}$, which is positive at time $t = 0$ by Assumption C.2. By (C.8), for positive $\gamma_{1,r}^{(t)}$ and $t \le T_{1,1}$ we have

$$\gamma_{1,r}^{(t+1)} = \gamma_{1,r}^{(t)} - \frac{\eta}{nm} \cdot \sum_{i=1}^n \ell_i'^{(t)} \cdot \sigma'(y_i \cdot \gamma_{1,r}^{(t)}) \cdot \|\boldsymbol{\mu}\|_2^2$$

$$\ge \gamma_{1,r}^{(t)} + \frac{C_1 \eta}{nm} \cdot \sum_{y_i=1} \sigma'(\gamma_{1,r}^{(t)}) \cdot \|\boldsymbol{\mu}\|_2^2.$$

Denote $A^{(t)} = \max_r \gamma_{1,r}^{(t)}$, $\gamma_0$ is defined in Assumption C.2 with $\max_r \gamma_{1,r}^{(0)} \geq \gamma_0 \geq 0$. Then we have

$$
\begin{aligned}
A^{(t+1)} &\geq A^{(t)} + \frac{C_1 \eta}{nm} \cdot \sum_{y_i=1} \sigma'(A^{(t)}) \cdot \|\boldsymbol{\mu}\|_2^2 \\
&\geq A^{(t)} + \frac{C_1 \eta q \|\boldsymbol{\mu}\|_2^2}{4m} \big(A^{(t)}\big)^{q-1} \\
&\geq \left[ 1 + \frac{C_1 \eta q \|\boldsymbol{\mu}\|_2^2}{4m} \big(A^{(0)}\big)^{q-2} \right] A^{(t)} \\
&\geq \left( 1 + \frac{C_1 \eta q \gamma_0^{q-2} \|\boldsymbol{\mu}\|_2^2}{4m} \right) A^{(t)},
\end{aligned}
$$

where the second inequality is by the lower bound on the number of positive data in Lemma C.7 , the third inequality is due to the fact that $A^{(t)}$ is an increasing sequence, and the last inequality follows by $A^{(0)} = \max_r \langle \mathbf{w}_{1,r}^{(0)}, \boldsymbol{\mu} \rangle \geq \gamma_0$. Therefore, the sequence $A^{(t)}$ will exponentially grow and we have that

$$
A^{(t)} \geq A^{(0)} \left( 1 + \frac{C_1 \eta q \gamma_0^{q-2} \|\boldsymbol{\mu}\|_2^2}{4m} \right)^t \geq A^{(0)} \exp \left( \frac{C_1 \eta q \gamma_0^{q-2} \|\boldsymbol{\mu}\|_2^2}{8m} t \right) \geq \gamma_0 \exp \left( \frac{C_1 \eta q \gamma_0^{q-2} \|\boldsymbol{\mu}\|_2^2}{8m} t \right),
$$

where the second inequality is due to the fact that $1 + z \geq \exp(z/2)$ for $z \leq 2$ and our condition of $\eta \leq O(mq^{-1} \gamma_0^{-(q-2)} \|\boldsymbol{\mu}\|_2^{-2})$ in Condition 4.1, and the last inequality follows by $A^{(0)} = \max_r \gamma_{1,r}^{(0)}$. Therefore, $A^{(t)} = \max_r \gamma_{1,r}^{(t)}$ will reach 2 within

$$
T_1 = \frac{\log(2/\gamma_0) 8m}{C_1 \eta q \gamma_0^{q-2} \|\boldsymbol{\mu}\|_2^2}
$$

iterations.

We can next verify the value of $T_1$ and $T_1^+$ follow the following relationship

$$
T_1 = \frac{\log(2/\gamma_0) 8m}{C_1 \eta q \gamma_0^{q-2} \|\boldsymbol{\mu}\|_2^2} \leq \frac{1}{\frac{4 \eta q}{nm} \sigma_p^2 d \cdot 8^{q-1} \rho_0^{q-2}} = T_1^+/2,
$$

where the inequality holds due to our SNR condition in (C.10). Therefore, by the definition of $T_{1,1}$, we have $T_{1,1} \leq T_1 \leq T_1^+/2$, where we use the non-decreasing property of $\gamma$. The proof for $j = -1$ is similar, and we can prove that $\max_r \gamma_{-1,r}^{(T_{1,-1})} \geq 2$ while $T_{1,-1} \leq T_1 \leq T_1^+/2$, which completes the proof. $\qquad\square$

### C.2.2 Proof of Lemma C.4

Before proving the Lemma C.4, we first show the following Proposition C.11, which shows that the coefficients $\gamma_{j,r}^{(t)}, \overline{\rho}_{j,r,i}^{(t)}, \underline{\rho}_{-j,r,i}^{(t)}$ will stay a reasonable scale during the training period $0 < t < T^*$.

**Proposition C.11.** *Under Condition 4.1, which indicates that $16n \sqrt{\frac{\log(4n^2/\delta)}{d}} \leq 0.5$, if Assumption C.1 holds, then for $0 \leq t \leq T^*$, we have that*

$$
-4m^{\frac{1}{q}} \log(T^*) \leq \gamma_{j,r}^{(0)} \leq \gamma_{j,r}^{(t)} \leq 4m^{\frac{1}{q}} \log(T^*), \tag{C.16}
$$

$$
0 \leq \overline{\rho}_{j,r,i}^{(t)} \leq 4m^{\frac{1}{q}} \log(T^*), \tag{C.17}
$$

$$
0 \geq \underline{\rho}_{-j,r,i}^{(t)} \geq -64nm^{\frac{1}{q}} \sqrt{\frac{\log(4n^2/\delta)}{d}} \cdot \log(T^*) \geq -4m^{\frac{1}{q}} \log(T^*), \tag{C.18}
$$

*for all $r \in [m]$, $j \in \{\pm 1\}$ and $i \in [n]$.*

Then, based on Proposition C.11 and Lemma C.3, we could prove the following Lemma C.4.

*Proof of Lemma C.4.* We have

$$
\begin{aligned}
\|\mathbf{W}^{(T_1)} - \mathbf{W}^*\|_F &\leq \sum_{j,r} \frac{|\gamma_{j,r}^{(T_1)}|}{\|\boldsymbol{\mu}\|_2} + \sum_{j,r,i} \frac{|\overline{\rho}_{j,r,i}^{(T_1)}|}{\|\boldsymbol{\xi}_i\|_2} + \sum_{j,r,i} \frac{|\underline{\rho}_{j,r,i}^{(T_1)}|}{\|\boldsymbol{\xi}_i\|_2} + O(m^{3/2}\log(1/\epsilon))\|\boldsymbol{\mu}\|_2^{-1} \\
&\leq O(m\|\boldsymbol{\mu}\|^{-1}) + O(nm\rho_0(\sigma_p\sqrt{d})^{-1}) + O(m^{3/2}\log(1/\epsilon))\|\boldsymbol{\mu}\|_2^{-1} \\
&\leq \widetilde{O}(m^{3/2}\|\boldsymbol{\mu}\|_2^{-1}) + O(nm\rho_0(\sigma_p\sqrt{d})^{-1}),
\end{aligned}
$$

where the first inequality is by our decomposition of $\mathbf{W}^{(T_1)}$ and the definition of $\mathbf{W}^*$, the second inequality is by Proposition C.11 and Lemma C.3. $\qquad\square$

### C.2.3 PROOF OF LEMMA C.5

In this section, Lemma C.12 is presented first, then Lemma C.13 and C.14 are proved before finally proving Lemma C.5. Based on Proposition C.11, the following Lemma C.12 introduces some important properties of the training loss function for $0 \leq t \leq T^*$.

**Lemma C.12.** *Under Condition 4.1, for $0 \leq t \leq T^*$, the following result holds,*

$$
\|\nabla L_S(\mathbf{W}^{(t)})\|_F^2 \leq O(\max\{\|\boldsymbol{\mu}\|_2^2, \sigma_p^2 d\})L_S(\mathbf{W}^{(t)}).
$$

**Lemma C.13.** *Under the same conditions as Theorem 5.5, we have that $y_i\langle\nabla f(\mathbf{W}^{(t)}, \mathbf{x}_i), \mathbf{W}^*\rangle \geq q^2 2^q \log(2q/\epsilon)$ for all $i \in [n]$ and $T_1 \leq t \leq T^*$.*

*Proof of Lemma C.13.* Recall that $f(\mathbf{W}^{(t)}, \mathbf{x}_i) = (1/m)\sum_{j,r} j \cdot \left[\sigma(\langle\mathbf{w}_{j,r}^{(t)}, y_i \cdot \boldsymbol{\mu}\rangle) + \sigma(\langle\mathbf{w}_{j,r}^{(t)}, \boldsymbol{\xi}_i\rangle)\right]$ and the definition of $\mathbf{W}^*$ in (C.13), we have

$$
\begin{aligned}
y_i\langle\nabla f(\mathbf{W}^{(t)}, \mathbf{x}_i), \mathbf{W}^*\rangle &= \frac{1}{m}\sum_{j,r} \sigma'(\langle\mathbf{w}_{j,r}^{(t)}, y_i\boldsymbol{\mu}\rangle)\langle\boldsymbol{\mu}, j\mathbf{w}_{j,r}^*\rangle + \frac{1}{m}\sum_{j,r} \sigma'(\langle\mathbf{w}_{j,r}^{(t)}, \boldsymbol{\xi}_i\rangle)\langle y_i\boldsymbol{\xi}_i, j\mathbf{w}_{j,r}^*\rangle \\
&= \frac{1}{m}\sum_{j,r} \sigma'(\langle\mathbf{w}_{j,r}^{(t)}, y_i\boldsymbol{\mu}\rangle)2qm\log(2q/\epsilon) \qquad\qquad\text{(C.19)}
\end{aligned}
$$

where the second equality holds because $\langle\boldsymbol{\mu}, j\mathbf{w}_{j,r}^*\rangle = 2qm\log(2q/\epsilon)$, $\langle y_i\boldsymbol{\xi}_i, j\mathbf{w}_{j,r}^*\rangle = 0$ by $\langle\boldsymbol{\mu}, j\mathbf{w}_{jr}^\perp\rangle = 0$, $\langle y_i\boldsymbol{\xi}_i, j\mathbf{w}_{jr}^\perp\rangle = 0$ in the definition of $\mathbf{W}^*$ (C.13).

Next we will give a bound for the inner-product term in (C.19). By Lemma C.10 and the initialization and non-decreasing property of $\gamma_{j,r}^{(t)}$ in Second Stage (Section C.1.2), we have that for $j = y_i$

$$
\max_r\langle\mathbf{w}_{j,r}^{(t)}, y_i\boldsymbol{\mu}\rangle = \max_r \gamma_{j,r}^{(t)} \geq 2. \qquad\qquad\text{(C.20)}
$$

Plug (C.20) into (C.19) can we obtain

$$
y_i\langle\nabla f(\mathbf{W}^{(t)}, \mathbf{x}_i), \mathbf{W}^*\rangle \geq q^2 2^q \log(2q/\epsilon)
$$

This completes the proof. $\qquad\square$

**Lemma C.14.** *Under the same conditions as Theorem 5.5, we have that*

$$
\|\mathbf{W}^{(t)} - \mathbf{W}^*\|_F^2 - \|\mathbf{W}^{(t+1)} - \mathbf{W}^*\|_F^2 \geq (2q-1)\eta L_S(\mathbf{W}^{(t)}) - \eta\epsilon
$$

*for all $T_1 \leq t \leq T^*$.*

*Proof of Lemma C.14.* Here we assume the neural network is $q$ homogeneous, namely $\langle \nabla f(\mathbf{W}^{(t)}, \mathbf{x}_i), \mathbf{W}^{(t)} \rangle = q f(\mathbf{W}^{(t)}, \mathbf{x}_i)$, thus we have

$$
\|\mathbf{W}^{(t)} - \mathbf{W}^*\|_F^2 - \|\mathbf{W}^{(t+1)} - \mathbf{W}^*\|_F^2
$$

$$
= 2\eta \langle \nabla L_S(\mathbf{W}^{(t)}), \mathbf{W}^{(t)} - \mathbf{W}^* \rangle - \eta^2 \|\nabla L_S(\mathbf{W}^{(t)})\|_F^2
$$

$$
= \frac{2\eta}{n} \sum_{i=1}^n \ell_i'^{(t)} [q y_i f(\mathbf{W}^{(t)}, \mathbf{x}_i) - y_i \langle \nabla f(\mathbf{W}^{(t)}, \mathbf{x}_i), \mathbf{W}^* \rangle] - \eta^2 \|\nabla L_S(\mathbf{W}^{(t)})\|_F^2
$$

$$
\geq \frac{2\eta}{n} \sum_{i=1}^n \ell_i'^{(t)} [q y_i f(\mathbf{W}^{(t)}, \mathbf{x}_i) - q^2 2^q \log(2q/\epsilon)] - \eta^2 \|\nabla L_S(\mathbf{W}^{(t)})\|_F^2
$$

$$
\geq \frac{2q\eta}{n} \sum_{i=1}^n [\ell(y_i f(\mathbf{W}^{(t)}, \mathbf{x}_i)) - \ell(q 2^q \log(2q/\epsilon))] - \eta^2 \|\nabla L_S(\mathbf{W}^{(t)})\|_F^2
$$

$$
\geq \frac{2q\eta}{n} \sum_{i=1}^n [\ell(y_i f(\mathbf{W}^{(t)}, \mathbf{x}_i)) - \epsilon/(2q)] - \eta^2 \|\nabla L_S(\mathbf{W}^{(t)})\|_F^2
$$

$$
\geq (2q-1)\eta L_S(\mathbf{W}^{(t)}) - \eta\epsilon,
$$

where the first inequality is by Lemma C.13, the second and third inequality is due to the convexity of the cross entropy function and the property of loss function, and the last inequality is by Lemma C.12 and by $\eta \leq O\left( \min\{\|\boldsymbol{\mu}\|_2^{-2}, (\sigma_p^2 \sqrt{d})^{-2}\} \right)$ in Condition 4.1. □

Based on the above lemmas, the proof of Lemma C.5 is presented as follows.

*Proof of Lemma C.5.* By Lemma C.14, for any $t \in [T_1, T]$, we have that for $s \leq t$

$$
\|\mathbf{W}^{(s)} - \mathbf{W}^*\|_F^2 - \|\mathbf{W}^{(s+1)} - \mathbf{W}^*\|_F^2 \geq (2q-1)\eta L_S(\mathbf{W}^{(s)}) - \eta\epsilon
$$

holds. Taking a summation, we obtain that

$$
\sum_{s=T_1}^t L_S(\mathbf{W}^{(s)}) \leq \frac{\|\mathbf{W}^{(T_1)} - \mathbf{W}^*\|_F^2 + \eta\epsilon(t - T_1 + 1)}{(2q-1)\eta} \tag{C.21}
$$

for all $T_1 \leq t \leq T$. Dividing $(t - T_1 + 1)$ on both side of (C.21) gives that

$$
\frac{1}{t - T_1 + 1} \sum_{s=T_1}^t L_S(\mathbf{W}^{(s)}) \leq \frac{\|\mathbf{W}^{(T_1)} - \mathbf{W}^*\|_F^2}{(2q-1)\eta(t - T_1 + 1)} + \frac{\epsilon}{2q-1}.
$$

Then we can take $t = T$ where $T = T_1 + \left\lfloor \frac{\|\mathbf{W}^{(T_1)} - \mathbf{W}^*\|_F^2}{2\eta\epsilon} \right\rfloor$ and have that

$$
\frac{1}{T - T_1 + 1} \sum_{s=T_1}^T L_S(\mathbf{W}^{(s)}) \leq \frac{\|\mathbf{W}^{(T_1)} - \mathbf{W}^*\|_F^2}{(2q-1)\eta(T - T_1 + 1)} + \frac{\epsilon}{2q-1} \leq \frac{3\epsilon}{2q-1} < \epsilon,
$$

where we use the fact that $q > 2$ and the choice of $T$. Since the mean is smaller than $\epsilon$, we can conclude that there exist $T_1 \leq t \leq T$ such that $L_S(\mathbf{W}^{(t)}) < \epsilon$.

Secondly, we will prove that $\max_{j,r,i} |\rho_{j,r,i}^{(t)}| \leq 4\rho_0$ for all $t \in [T_1, T]$. Plugging $T = T_1 + \left\lfloor \frac{\|\mathbf{W}^{(T_1)} - \mathbf{W}^*\|_F^2}{2\eta\epsilon} \right\rfloor$ into (C.21) gives that

$$
\sum_{s=T_1}^T L_S(\mathbf{W}^{(s)}) \leq \frac{2\|\mathbf{W}^{(T_1)} - \mathbf{W}^*\|_F^2}{(2q-1)\eta} = \widetilde{O}(\eta^{-1} m^3 \|\boldsymbol{\mu}\|_2^{-2}) + O(\eta^{-1} n^2 m^2 \rho_0^2 (\sigma_p \sqrt{d})^{-2}), \tag{C.22}
$$

where the inequality is due to $\|\mathbf{W}^{(T_1)} - \mathbf{W}^*\|_F \leq \widetilde{O}(m^{3/2} \|\boldsymbol{\mu}\|_2^{-1}) + O(n m \rho_0 (\sigma_p \sqrt{d})^{-1})$ in Lemma C.4. Define $\Psi^{(t)} = \max_{j,r,i} |\rho_{j,r,i}^{(t)}|$. We will use induction to prove $\Psi^{(t)} \leq 4\rho_0$ for all

$t \in [T_1, T]$. At $t = T_1$, by the properties at the beginning of Second Stage (Section C.1.2), we have $\Psi^{(T_1)} \leq 2\rho_0$. Now suppose that there exists $\widetilde{T} \in [T_1, T]$ such that $\Psi^{(t)} \leq 4\rho_0$ for all $t \in [T_1, \widetilde{T} - 1]$. Then we prove it also holds for $t = \widetilde{T}$: For $t \in [T_1, \widetilde{T} - 1]$, by (C.8), we have

$$\Psi^{(t+1)} \leq \Psi^{(t)} + \max_{j,r,i} \left\{ \frac{\eta}{nm} \cdot |\ell_i'^{(t)}| \cdot \sigma'\left( 2 \sum_{i'=1}^{n} \Psi^{(t)} \cdot \frac{|\langle \boldsymbol{\xi}_{i'}, \boldsymbol{\xi}_i \rangle|}{\|\boldsymbol{\xi}_{i'}\|_2^2} \right) \cdot \|\boldsymbol{\xi}_{i'}\|_2^2 \right\}$$

$$= \Psi^{(t)} + \max_{j,r,i} \left\{ \frac{\eta}{nm} \cdot |\ell_i'^{(t)}| \cdot \sigma'\left( 2\Psi^{(t)} + 2 \sum_{i' \neq i}^{n} \Psi^{(t)} \cdot \frac{|\langle \boldsymbol{\xi}_{i'}, \boldsymbol{\xi}_i \rangle|}{\|\boldsymbol{\xi}_{i'}\|_2^2} \right) \cdot \|\boldsymbol{\xi}_{i'}\|_2^2 \right\}$$

$$\leq \Psi^{(t)} + \frac{\eta q}{nm} \cdot \max_i |\ell_i'^{(t)}| \cdot \left[ \left( 2 + \frac{4n\sigma_p^2 \cdot \sqrt{d \log(4n^2/\delta)}}{\sigma_p^2 d/2} \right) \cdot \Psi^{(t)} \right]^{q-1} \cdot 2\sigma_p^2 d$$

$$\leq \Psi^{(t)} + \frac{\eta q}{nm} \cdot \max_i |\ell_i'^{(t)}| \cdot \left( 4 \cdot \Psi^{(t)} \right)^{q-1} \cdot 2\sigma_p^2 d,$$

where the second inequality is due to Lemma C.8, and the last inequality follows by the assumption that $d \geq 1024n^2 \log(4n^2/\delta)$ in Condition 4.1. Taking a telescoping sum over $t = T_1, \ldots, \widetilde{T} - 1$, we have that

$$\Psi^{(T)} \overset{(i)}{\leq} \Psi^{(T_1)} + \frac{\eta q}{nm} \sum_{s=T_1}^{\widetilde{T}-1} \max_i |\ell_i'^{(s)}| \widetilde{O}(\sigma_p^2 d) \cdot (2\rho_0)^{q-1}$$

$$\overset{(ii)}{\leq} \Psi^{(T_1)} + \frac{\eta q}{nm} 4^{q-1} 2^q \sigma_p^2 d(2\rho_0)^{q-1} \sum_{s=T_1}^{\widetilde{T}-1} \max_i \ell_i^{(s)}$$

$$\overset{(iii)}{\leq} \Psi^{(T_1)} + \eta q m^{-1} 4^{q-1} 2^q \sigma_p^2 d(2\rho_0)^{q-1} \sum_{s=T_1}^{\widetilde{T}-1} L_S(\mathbf{W}^{(s)})$$

$$\overset{(iv)}{\leq} \Psi^{(T_1)} + \widetilde{O}(qm^2 4^{q-1} 2^q \mathrm{SNR}^{-2}) \cdot (2\rho_0)^{q-1}$$

$$\leq 2\rho_0 + \widetilde{O}(qm^2 4^{q-1} 2^q (2\rho_0)^{q-2} \mathrm{SNR}^{-2}) \cdot 2\rho_0$$

$$\overset{(v)}{\leq} 2\rho_0 + \rho_0 + \rho_0$$

$$= 4\rho_0,$$

where (i) is by out induction hypothesis that $\Psi^{(t)} \leq 4\rho_0$ for $t \in [T_1, \widetilde{T} - 1]$, (ii) is by $|\ell'| \leq \ell$, (iii) is by $\max_i \ell_i^{(s)} \leq \sum_i \ell_i^{(s)} = nL_S(\mathbf{W}^{(s)})$, (iv) is due to $\sum_{s=T_1}^{\widetilde{T}-1} L_S(\mathbf{W}^{(s)}) \leq \sum_{s=T_1}^{T} L_S(\mathbf{W}^{(s)}) = \widetilde{O}(\eta^{-1}m^3\|\boldsymbol{\mu}\|_2^{-2}) + O(\eta^{-1}n^2 m^2 \rho_0^2 (\sigma_p \sqrt{d})^{-2})$ in (C.22), (v) is by the condition for SNR : $\mathrm{SNR}^2 \geq \widetilde{\Omega}(2qm^2 4^{q-1} 2^q (2\rho_0)^{q-2}) = \widetilde{\Omega}(2qm^2 16^{q-1} \rho_0^{q-2})$ and $\rho_0 \leq O((\frac{1}{8}qn^2 m)^{-\frac{1}{q}})$ obtained from Theorem 5.3. This completes the induction. $\qquad \square$

### C.2.4   PROOF OF LEMMA C.6

We first present the following Lemma C.15, which shows the bound of $\langle \mathbf{w}_{j,r}^{(t)}, \boldsymbol{\xi}_i \rangle$.

**Lemma C.15.** *Under Condition 4.1, suppose* (C.16), (C.17) *and* (C.18) *hold at iteration t. Then*

$$\underline{\rho}_{j,r,i}^{(t)} - 32nm^{\frac{1}{q}}\sqrt{\frac{\log(4n^2/\delta)}{d}} \cdot \log(T^*) \leq \langle \mathbf{w}_{j,r}^{(t)}, \boldsymbol{\xi}_i \rangle \leq \underline{\rho}_{j,r,i}^{(t)} + 32nm^{\frac{1}{q}}\sqrt{\frac{\log(4n^2/\delta)}{d}} \cdot \log(T^*), \ j \neq y_i,$$

$$\overline{\rho}_{j,r,i}^{(t)} - 32nm^{\frac{1}{q}}\sqrt{\frac{\log(4n^2/\delta)}{d}} \cdot \log(T^*) \leq \langle \mathbf{w}_{j,r}^{(t)}, \boldsymbol{\xi}_i \rangle \leq \overline{\rho}_{j,r,i}^{(t)} + 32nm^{\frac{1}{q}}\sqrt{\frac{\log(4n^2/\delta)}{d}} \cdot \log(T^*), \ j = y_i$$

*for all* $r \in [m]$, $j \in \{\pm1\}$ *and* $i \in [n]$.

We then prove the following two lemmas before proving Lemma C.6.

**Lemma C.16.** *Under the same conditions as Theorem 5.5, we have that $\max_{j,r} |\langle \mathbf{w}_{j,r}^{(t)}, \boldsymbol{\xi}_i \rangle| \leq 1/2$ for all $0 \leq t \leq T$, where $T$ is defined in Lemma C.5 in Second Stage (Section C.1.2).*

*Proof of Lemma C.16.* We can get the upper bound of the inner products between the parameter and the noise as follows:

$$
\begin{aligned}
|\langle \mathbf{w}_{j,r}^{(t)}, \boldsymbol{\xi}_i \rangle| &\overset{(i)}{\leq} |\rho_{j,r,i}^{(t)}| + 8n\sqrt{\frac{\log(4n^2/\delta)}{d}} \cdot 4m^{\frac{1}{q}} \log(T^*) \\
&\overset{(ii)}{\leq} 4\rho_0 + 8n\sqrt{\frac{\log(4n^2/\delta)}{d}} \cdot 4m^{\frac{1}{q}} \log(T^*) \\
&\overset{(iii)}{\leq} 1/2
\end{aligned}
$$

for all $j \in \{\pm 1\}$, $r \in [m]$ and $i \in [n]$, where (i) is by Lemma C.15, (ii) is by $\max_{j,r,i} |\rho_{j,r,i}^{(t)}| \leq 4\rho_0$ in Lemma C.5, and (iii) is due to the condition $8n\sqrt{\frac{\log(4n^2/\delta)}{d}} \cdot 4m^{\frac{1}{q}} \log(T^*) \leq 1/4$ in Condition 4.1 and the result $\rho_0 \leq 1/16$ in Theorem 5.3. $\qquad\square$

The following Lemma C.17 provides the upper bound for $\max_{j,r} |\langle \mathbf{w}_{j,r}^{(t)}, \boldsymbol{\xi} \rangle|$, where $\boldsymbol{\xi}$ is from the test population.

**Lemma C.17.** *Under the same conditions as Theorem 5.5, with probability at least $1 - 4mT \cdot \exp(-\Omega(n^2))$, we have that $\max_{j,r} |\langle \mathbf{w}_{j,r}^{(t)}, \boldsymbol{\xi} \rangle| \leq 1/2$ for all $0 \leq t \leq T$.*

*Proof of Lemma C.17.* Define $\widetilde{\mathbf{w}}_{j,r}^{(t)} = \mathbf{w}_{j,r}^{(t)} - j \cdot \gamma_{j,r}^{(t)} \cdot \frac{\boldsymbol{\mu}}{\|\boldsymbol{\mu}\|_2^2}$, then we have $\langle \widetilde{\mathbf{w}}_{j,r}^{(t)}, \boldsymbol{\xi} \rangle = \langle \mathbf{w}_{j,r}^{(t)}, \boldsymbol{\xi} \rangle$. Since $\widetilde{\mathbf{w}}_{j,r}^{(t)} = \mathbf{w}_{jr}^{\perp} + \sum_{i=1}^{n} \rho_{j,r,i}^{(t)} \cdot \|\boldsymbol{\xi}_i\|_2^{-2} \cdot \boldsymbol{\xi}_i$, we have

$$
\|\widetilde{\mathbf{w}}_{j,r}^{(t)}\|_2 \leq \|\mathbf{w}_{jr}^{\perp}\|_2 + 4n\rho_0 \frac{2}{\sigma_p \sqrt{d}} = \|\mathbf{w}_{jr}^{\perp}\|_2 + \widetilde{O}(\frac{n\rho_0}{\sigma_p \sqrt{d}}), \tag{C.23}
$$

where the inequality is due to the bound for $\rho_{j,r,i}^{(t)}$ and $\|\boldsymbol{\xi}_i\|_2$.

By (C.23), $\max_{j,r} \|\widetilde{\mathbf{w}}_{j,r}^{(t)}\|_2 \leq \max_{j,r} \|\mathbf{w}_{jr}^{\perp}\|_2 + \widetilde{C}_2 \frac{n\rho_0}{\sigma_p \sqrt{d}}$, where $\widetilde{C}_2 = \widetilde{O}(1)$. Clearly $\langle \widetilde{\mathbf{w}}_{j,r}^{(t)}, \boldsymbol{\xi} \rangle$ is a Gaussian distribution with mean zero and standard deviation smaller than $\max_{j,r} \|\mathbf{w}_{jr}^{\perp}\|_2 + \widetilde{C}_2 \frac{n\rho_0}{\sqrt{d}}$. Therefore, the probability is bounded by

$$
\begin{aligned}
\mathbb{P}\big(|\langle \widetilde{\mathbf{w}}_{j,r}^{(t)}, \boldsymbol{\xi} \rangle| \geq 1/2\big) &\leq 2\exp\left(-\frac{1}{8\big[\frac{\widetilde{C}_2^2 n^2 \rho_0^2}{d} + 2\frac{\widetilde{C}_2 n\rho_0}{\sqrt{d}} \max_{j,r} \|\mathbf{w}_{jr}^{\perp}\|_2 + (\max_{j,r} \|\mathbf{w}_{jr}^{\perp}\|_2)^2\big]}\right) \\
&\leq 2\exp\left(-\frac{1}{8\big[\frac{\widetilde{C}_2^2 n^2 \rho_0^2}{d} + 2\frac{\widetilde{C}_2 \rho_0}{\sqrt{d}} + n^{-2}\big]}\right) \\
&\leq 2\exp\big(-\Omega(n^2)\big),
\end{aligned}
$$

where the second inequality is by the assumption $\max_{j,r} \|\mathbf{w}_{jr}^{\perp}\|_2 \leq 1/n$, the third inequality is by $\rho_0 \leq O(\sqrt{d}/n^2)$ in Theorem 5.3. Applying a union bound over $j, r, t$ completes the proof. $\qquad\square$

Based on Lemmas C.16 and C.17, we now prove Lemma C.6.

*Proof of Lemma C.6.* Let event $\mathcal{E}$ to be the event that Lemma C.17 holds. Then we can divide $L_{\mathcal{D}}(\mathbf{W}^{(t)})$ into two parts:

$$
\mathbb{E}\big[\ell\big(yf(\mathbf{W}^{(t)}, \mathbf{x})\big)\big] = \underbrace{\mathbb{E}[\mathbb{1}(\mathcal{E})\ell\big(yf(\mathbf{W}^{(t)}, \mathbf{x})\big)]}_{I_1} + \underbrace{\mathbb{E}[\mathbb{1}(\mathcal{E}^c)\ell\big(yf(\mathbf{W}^{(t)}, \mathbf{x})\big)]}_{I_2}. \tag{C.24}
$$

In the following analysis, we bound $I_1$ and $I_2$ respectively.

**Bounding $I_1$:** Denote $I_j = \{i | y_i = j\}$, $j = \pm 1$. Since we have

$$L_S(\mathbf{W}^{(t)}) = \frac{1}{n} \left[ \sum_{i' \in I_+} \ell\big(y_{i'} f(\mathbf{W}^{(t)}, \mathbf{x}_{i'})\big) + \sum_{i' \in I_-} \ell\big(y_{i'} f(\mathbf{W}^{(t)}, \mathbf{x}_{i'})\big) \right] \leq \frac{1}{4},$$

thus, $\sum_{i' \in I_j} \ell\big(y_{i'} f(\mathbf{W}^{(t)}, \mathbf{x}_{i'})\big) \leq \frac{1}{4}$, $j = \pm 1$. It follows that for $j = \pm 1$, we have

$$\frac{1}{|I_j|} \sum_{i' \in I_j} \ell\big(y_{i'} f(\mathbf{W}^{(t)}, \mathbf{x}_{i'})\big) \leq \frac{n}{|I_j|} \frac{1}{4} \leq 1$$

where the last inequality is by Lemma C.7. Therefore, there must exist one $(\mathbf{x}_i, y_i)$ with $y = y_i \in I_{j_0}$ such that
$\ell\big(y_i f(\mathbf{W}^{(t)}, \mathbf{x}_i)\big) \leq \frac{1}{|I_{j_0}|} \sum_{i' \in I_{j_0}} \ell\big(y_{i'} f(\mathbf{W}^{(t)}, \mathbf{x}_{i'})\big) \leq 1$, which implies that $y_i f(\mathbf{W}^{(t)}, \mathbf{x}_i) \geq 0$.
Therefore, we have that

$$\exp(-y_i f(\mathbf{W}^{(t)}, \mathbf{x}_i)) \overset{(i)}{\leq} 2\log\big(1 + \exp(-y_i f(\mathbf{W}^{(t)}, \mathbf{x}_i))\big) = 2\ell\big(y_i f(\mathbf{W}^{(t)}, \mathbf{x}_i)\big) \leq 2L_S(\mathbf{W}^{(t)}), \tag{C.25}$$

where (i) is by $z \leq 2\log(1 + z)$, for $z \leq 1$ and here we have $\exp(-y_i f(\mathbf{W}^{(t)}, \mathbf{x}_i)) \leq 1$. If event $\mathcal{E}$ holds, we have that

$$|yf(\mathbf{W}^{(t)}, \mathbf{x}) - y_i f(\mathbf{W}^{(t)}, \mathbf{x}_i)| \leq \frac{1}{m} \sum_{j,r} \sigma(\langle \mathbf{w}_{j,r}^{(t)}, \boldsymbol{\xi}_i \rangle) + \frac{1}{m} \sum_{j,r} \sigma(\langle \mathbf{w}_{j,r}^{(t)}, \boldsymbol{\xi} \rangle)$$

$$\leq \frac{1}{m} \sum_{j,r} \sigma(1/2) + \frac{1}{m} \sum_{j,r} \sigma(1/2)$$

$$\leq 1, \tag{C.26}$$

where the second inequality is by $\max_{j,r} |\langle \mathbf{w}_{j,r}^{(t)}, \boldsymbol{\xi} \rangle| \leq 1/2$ in Lemma C.17 and $\max_{j,r} |\langle \mathbf{w}_{j,r}^{(t)}, \boldsymbol{\xi}_i \rangle| \leq 1/2$ in Lemma C.16. Thus, we have that

$$I_1 \leq \mathbb{E}[\mathbb{1}(\mathcal{E}) \exp(-yf(\mathbf{W}^{(t)}, \mathbf{x}))]$$

$$\leq e \cdot \mathbb{E}[\mathbb{1}(\mathcal{E}) \exp(-y_i f(\mathbf{W}^{(t)}, \mathbf{x}_i))]$$

$$\leq 2e \cdot \mathbb{E}[\mathbb{1}(\mathcal{E}) L_S(\mathbf{W}^{(t)})],$$

where the first inequality is by the property of cross-entropy loss that $\ell(z) \leq \exp(-z)$ for all $z$, the second inequality is by $-yf(\mathbf{W}^{(t)}, \mathbf{x}) \leq 1 - y_i f(\mathbf{W}^{(t)}, \mathbf{x}_i)$ in (C.26), and the third inequality is by $\exp(-y_i f(\mathbf{W}^{(t)}, \mathbf{x}_i)) \leq 2L_S(\mathbf{W}^{(t)})$ in (C.25). Dropping the event in the expectation gives $I_1 \leq 6L_S(\mathbf{W}^{(t)})$.

**Bounding $I_2$:** Next we bound the second term $I_2$. We choose an arbitrary training data $(\mathbf{x}_{i'}, y_{i'})$ such that $y_{i'} = y$. Then we have

$$\ell\big(yf(\mathbf{W}^{(t)}, \mathbf{x})\big) = \log(1 + \exp(-yF_+(\mathbf{W}_+^{(t)}, \mathbf{x}) + yF_-(\mathbf{W}_-^{(t)}, \mathbf{x})))$$

$$\leq \log(1 + \exp(F_{-y}(\mathbf{W}_{-y}^{(t)}, \mathbf{x})))$$

$$\leq 1 + F_{-y}(\mathbf{W}_{-y}^{(t)}, \mathbf{x})$$

$$= 1 + \frac{1}{m} \sum_{j=-y, r \in [m]} \sigma(\langle \mathbf{w}_{j,r}^{(t)}, y\boldsymbol{\mu} \rangle) + \frac{1}{m} \sum_{j=-y, r \in [m]} \sigma(\langle \mathbf{w}_{j,r}^{(t)}, \boldsymbol{\xi} \rangle)$$

$$\leq 1 + 4m^{\frac{1}{q}} \log(T^*) + \frac{1}{m} \sum_{j=-y, r \in [m]} \sigma(\langle \mathbf{w}_{j,r}^{(t)}, \boldsymbol{\xi} \rangle)$$

$$\leq 1 + 4m^{\frac{1}{q}} \log(T^*) + \widetilde{O}((n\rho_0 \sigma_p^{-1} d^{-\frac{1}{2}})^q)) \|\boldsymbol{\xi}\|_2^q, \tag{C.27}$$

where the first inequality is due to $F_y(\mathbf{W}^{(t)}, \mathbf{x}) \geq 0$, the second inequality is by the property of cross-entropy loss, i.e., $\log(1 + \exp(z)) \leq 1 + z$ for all $z \geq 0$, the third inequality is by Lemma C.9 and (C.9) in Assumption C.2, i.e., $\frac{1}{m} \sum_{j=-y,r\in[m]} \sigma(\langle \mathbf{w}_{j,r}^{(t)}, y\boldsymbol{\mu}\rangle) \leq \frac{1}{m} \sum_{j=-y,r\in[m]} \sigma(-\gamma_{j,r}^{(t)}) \leq \frac{1}{m} \sum_{j=-y,r\in[m]} \sigma(-\gamma_{j,r}^{(0)}) \leq \max_{j,r}\{0, (-\gamma_{j,r}^{(0)})^q\} \ll 4m^{\frac{1}{q}} \log(T^*)$, and the last inequality is by (C.23), we have $\langle \widetilde{\mathbf{w}}_{j,r}^{(t)}, \boldsymbol{\xi}\rangle = \langle \mathbf{w}_{j,r}^{(t)}, \boldsymbol{\xi}\rangle \leq \|\widetilde{\mathbf{w}}_{j,r}^{(t)}\|_2 \cdot \|\boldsymbol{\xi}\|_2 \leq \widetilde{O}(n\rho_0\sigma_p^{-1}d^{-\frac{1}{2}})\|\boldsymbol{\xi}\|_2$. Then we further have that

$$
\begin{aligned}
I_2 &\leq \sqrt{\mathbb{E}[\mathbb{1}(\mathcal{E}^c)]} \cdot \sqrt{\mathbb{E}\left[\ell\big(yf(\mathbf{W}^{(t)}, \mathbf{x})\big)^2\right]} \\
&\leq \sqrt{\mathbb{P}(\mathcal{E}^c)} \cdot \sqrt{[1 + 4m^{\frac{1}{q}}\log(T^*)]^2 + \widetilde{O}(n^{2q}\rho_0^{2q}\sigma_p^{-2q}d^{-q})\mathbb{E}[\|\boldsymbol{\xi}\|_2^{2q}]} \\
&\leq \exp[-\widetilde{\Omega}(n^2) + \mathrm{polylog}(n)] \\
&\leq \exp(-\widetilde{\Omega}(n^2)),
\end{aligned}
$$

where the first inequality is by Cauchy-Schwartz inequality, the second inequality is by (C.27), the third inequality is by Lemma C.17, the definition of $\boldsymbol{\xi}$, and the result $\rho_0 \leq 1/16$ in Theorem 5.3.

Plugging the bounds of $I_1$, $I_2$ into (C.24) completes the proof. $\qquad\square$

## C.3 PROOF OF LEMMAS IN SECTION C.2

In this section, we prove the lemmas used in the proof of Section C.2. These lemmas are mainly concerned with the properties of data and the basic properties of the coefficients $\gamma_{j,r}^{(t)}, \overline{\rho}_{j,r,i}^{(t)}, \underline{\rho}_{j,r,i}^{(t)}$.

We first prove the following Lemmas C.7 and C.8, which are related to the data distribution.

*Proof of Lemma C.7.* Since $y_i$ follow Rademache distribution, then by Hoeffding's inequality, with probability at least $1 - \delta/2$,

$$
\Big| \sum_{i=1}^n \mathbb{1}\{y_i = 1\} - \frac{n}{2} \Big| \leq \sqrt{2n\log(4/\delta)}.
$$

By our assumption $n \geq \Omega(\log(1/\delta))$, it follows that

$$
|\{i \in [n] : y_i = 1\}| = \sum_{i=1}^n \mathbb{1}\{y_i = 1\} \geq \frac{n}{2} - \sqrt{2n\log(4/\delta)} \geq \frac{n}{4}.
$$

Same result could be obtained for $|\{i \in [n] : y_i = -1\}|$. Apply a union bound finishes the proof of this lemma. $\qquad\square$

*Proof of Lemma C.8.* Since $\boldsymbol{\xi}_i, i \in [n]$ i.i.d follows $\mathcal{N}(\mathbf{0}, \sigma_p^2 \cdot (\mathbf{I} - \boldsymbol{\mu}\boldsymbol{\mu}^\top \cdot \|\boldsymbol{\mu}\|_2^{-2}))$, the proof follows exactly same proof as Lemma B.1. By Bernstein inequality, with the probability of at least $1 - \delta/(2n)$, we have that

$$
d\sigma_p^2 - C\sigma_p^2\sqrt{d\log(4/\delta)} \leq \|\boldsymbol{\xi}_i\|_2^2 \leq d\sigma_p^2 + C\sigma_p^2\sqrt{d\log(4n/\delta)},
$$

where $C$ is an absolute constant that does not depend on other variables. By assumption $d = \Omega(\log(4n/\delta))$, it follows that

$$
\sigma_p^2 d/2 \leq \|\boldsymbol{\xi}_i\|_2^2 \leq 3\sigma_p^2 d/2
$$

For the second result, by Bernstein inequality, for all $i, i' \in [n]$ with $i \neq i'$, with the probability of at least $1 - \delta/(2n^2)$, we have that

$$
|\langle \boldsymbol{\xi}_i, \boldsymbol{\xi}_{i'}\rangle| \leq 2\sigma_p^2 \cdot \sqrt{d\log(4n^2/\delta)}.
$$

Apply a union bound for finishes the proof of this lemma. $\qquad\square$

We then prove a series of lemmas that will be used in the proof of Proposition C.11 by induction.

**Lemma C.18.** *For any $t \geq 0$, it holds that $\langle \mathbf{w}_{j,r}^{(t)}, \boldsymbol{\mu} \rangle = j \cdot \gamma_{j,r}^{(t)}$ for all $r \in [m], j \in \{\pm 1\}$.*

*Proof of Lemma C.18.* For any $t \geq 0$, we have that

$$\langle \mathbf{w}_{j,r}^{(t)}, \boldsymbol{\mu} \rangle = j \cdot \gamma_{j,r}^{(t)} + \sum_{i'=1}^{n} \overline{\rho}_{j,r,i'}^{(t)} \|\boldsymbol{\xi}_{i'}\|_2^{-2} \cdot \langle \boldsymbol{\xi}_{i'}, \boldsymbol{\mu} \rangle + \sum_{i'=1}^{n} \underline{\rho}_{j,r,i'}^{(t)} \|\boldsymbol{\xi}_{i'}\|_2^{-2} \cdot \langle \boldsymbol{\xi}_{i'}, \boldsymbol{\mu} \rangle$$

$$= j \cdot \gamma_{j,r}^{(t)},$$

where the equality is by our orthogonal assumption of $\boldsymbol{\xi}$ and $\boldsymbol{\mu}$. $\qquad\square$

*Proof of Lemma C.15.* For $j \neq y_i$, we have $\overline{\rho}_{j,r,i}^{(t)} = 0$ and

$$\langle \mathbf{w}_{j,r}^{(t)}, \boldsymbol{\xi}_i \rangle = \sum_{i'=1}^{n} \overline{\rho}_{j,r,i'}^{(t)} \|\boldsymbol{\xi}_{i'}\|_2^{-2} \cdot \langle \boldsymbol{\xi}_{i'}, \boldsymbol{\xi}_i \rangle + \sum_{i'=1}^{n} \underline{\rho}_{j,r,i'}^{(t)} \|\boldsymbol{\xi}_{i'}\|_2^{-2} \cdot \langle \boldsymbol{\xi}_{i'}, \boldsymbol{\xi}_i \rangle$$

$$\leq 4\sqrt{\frac{\log(4n^2/\delta)}{d}} \sum_{i' \neq i} |\overline{\rho}_{j,r,i'}^{(t)}| + 4\sqrt{\frac{\log(4n^2/\delta)}{d}} \sum_{i' \neq i} |\underline{\rho}_{j,r,i'}^{(t)}| + \underline{\rho}_{j,r,i}^{(t)}$$

$$\leq \underline{\rho}_{j,r,i}^{(t)} + 32nm^{\frac{1}{q}}\sqrt{\frac{\log(4n^2/\delta)}{d}} \cdot \log(T^*),$$

where the second inequality is by Lemma C.8 and the last inequality is by $|\overline{\rho}_{j,r,i'}^{(t)}|, |\underline{\rho}_{j,r,i'}^{(t)}| \leq 4m^{\frac{1}{q}} \log(T^*)$ in (C.17).

For $y_i = j$, we have that $\underline{\rho}_{j,r,i}^{(t)} = 0$ and

$$\langle \mathbf{w}_{j,r}^{(t)}, \boldsymbol{\xi}_i \rangle = \sum_{i'=1}^{n} \overline{\rho}_{j,r,i'}^{(t)} \|\boldsymbol{\xi}_{i'}\|_2^{-2} \cdot \langle \boldsymbol{\xi}_{i'}, \boldsymbol{\xi}_i \rangle + \sum_{i'=1}^{n} \underline{\rho}_{j,r,i'}^{(t)} \|\boldsymbol{\xi}_{i'}\|_2^{-2} \cdot \langle \boldsymbol{\xi}_{i'}, \boldsymbol{\xi}_i \rangle$$

$$\leq \overline{\rho}_{j,r,i}^{(t)} + 4\sqrt{\frac{\log(4n^2/\delta)}{d}} \sum_{i' \neq i} |\overline{\rho}_{j,r,i'}^{(t)}| + 4\sqrt{\frac{\log(4n^2/\delta)}{d}} \sum_{i' \neq i} |\underline{\rho}_{j,r,i'}^{(t)}|$$

$$\leq \overline{\rho}_{j,r,i}^{(t)} + 32nm^{\frac{1}{q}}\sqrt{\frac{\log(4n^2/\delta)}{d}} \log(T^*),$$

where the first inequality is by Lemma C.7 and the second inequality is by $|\overline{\rho}_{j,r,i'}^{(t)}|, |\underline{\rho}_{j,r,i'}^{(t)}| \leq 4m^{\frac{1}{q}} \log(T^*)$ in (C.17). Similarly, we can show that $\langle \mathbf{w}_{j,r}^{(t)}, \boldsymbol{\xi}_i \rangle \geq \underline{\rho}_{j,r,i}^{(t)} - 32nm^{\frac{1}{q}}\sqrt{\log(4n^2/\delta)/d} \cdot \log(T^*)$ and $\langle \mathbf{w}_{j,r}^{(t)}, \boldsymbol{\xi}_i \rangle \geq \overline{\rho}_{j,r,i}^{(t)} - 32nm^{\frac{1}{q}}\sqrt{\log(4n^2/\delta)/d} \cdot \log(T^*)$. This completes the proof. $\quad\square$

*Proof of Lemma C.9.* For $j \neq y_i$, by Lemma C.18, we have that

$$\langle \mathbf{w}_{j,r}^{(t)}, y_i \boldsymbol{\mu} \rangle = y_i \cdot j \cdot \gamma_{j,r}^{(t)} = -\gamma_{j,r}^{(t)} \leq \begin{cases} 0, \text{ if } \gamma_{j,r}^{(t)} \geq 0 \\ -\gamma_{j,r}^{(0)}, \text{ otherwise} \end{cases}. \tag{C.28}$$

Also, we have

$$\langle \mathbf{w}_{j,r}^{(t)}, \boldsymbol{\xi}_i \rangle \leq \underline{\rho}_{j,r,i}^{(t)} + 32nm^{\frac{1}{q}}\sqrt{\frac{\log(4n^2/\delta)}{d}} \cdot \log(T^*) \leq 32nm^{\frac{1}{q}}\sqrt{\frac{\log(4n^2/\delta)}{d}} \cdot \log(T^*), \tag{C.29}$$

where the first inequality is by Lemma C.15 and the second inequality is due to $\underline{\rho}_{j,r,i}^{(t)} \leq 0$. Thus, we can get that

$$F_j(\mathbf{W}_j^{(t)}, \mathbf{x}_i) = \frac{1}{m} \sum_{r=1}^{m} [\sigma(\langle \mathbf{w}_{j,r}^{(t)}, -j \cdot \boldsymbol{\mu} \rangle) + \sigma(\langle \mathbf{w}_{j,r}^{(t)}, \boldsymbol{\xi}_i \rangle)]$$

$$\leq 2 \cdot 2^q \max_{j,r} \left\{ -\gamma_{j,r}^{(t)}, 32nm^{\frac{1}{q}}\sqrt{\frac{\log(4n^2/\delta)}{d}} \cdot \log(T^*) \right\}^q$$

$$= O(1),$$

where the first inequality is by (C.28), (C.29) and the last line is by Condition 4.1 which implies $128nm^{\frac{1}{q}}\sqrt{\frac{\log(4n^2/\delta)}{d}} \cdot \log(T^*) \leq 1$ and $-\gamma_{j,r}^{(t)} \leq \max_{j,r}\{0, -\gamma_{j,r}^{(0)}\} \leq C_0$ in Assumption C.1. $\qquad\square$

*Proof of Lemma C.10.* For $j = y_i$, we have that

$$\langle \mathbf{w}_{j,r}^{(t)}, y_i\boldsymbol{\mu}\rangle = \gamma_{j,r}^{(t)}, \tag{C.30}$$

where the equality is by Lemma C.18. We also have that

$$\langle \mathbf{w}_{j,r}^{(t)}, \boldsymbol{\xi}_i\rangle \leq \overline{\rho}_{j,r,i}^{(t)} + 32nm^{\frac{1}{q}}\sqrt{\frac{\log(4n^2/\delta)}{d}} \cdot \log(T^*), \tag{C.31}$$

where the inequality is by Lemma C.15. If $\max\{\gamma_{j,r}^{(t)}, \overline{\rho}_{j,r,i}^{(t)}\} = O(1)$, we have the following bound

$$F_j(\mathbf{W}_j^{(t)}, \mathbf{x}_i) = \frac{1}{m}\sum_{r=1}^m [\sigma(\langle \mathbf{w}_{j,r}^{(t)}, j \cdot \boldsymbol{\mu}\rangle) + \sigma(\langle \mathbf{w}_{j,r}^{(t)}, \boldsymbol{\xi}_i\rangle)]$$

$$\leq 2 \cdot 3^q \max_{j,r,i}\left\{\gamma_{j,r}^{(t)}, \overline{\rho}_{j,r,i}^{(t)}, 32nm^{\frac{1}{q}}\sqrt{\frac{\log(4n^2/\delta)}{d}} \cdot \log(T^*)\right\}^q$$

$$= O(1),$$

where the first inequality is by (C.30), (C.31), and the last line is by $\max_{j,r,i}\{\gamma_{j,r}^{(t)}, \overline{\rho}_{j,r,i}^{(t)}\} = O(1)$ and Condition 4.1 which implies $128nm^{\frac{1}{q}}\sqrt{\frac{\log(4n^2/\delta)}{d}} \cdot \log(T^*) \leq 1$. $\qquad\square$

Now, we prove the Proposition C.11 by induction.

*Proof of Proposition C.11.* By Assumption C.1, the results in Proposition C.11 hold at $t = 0$. Suppose that there exists $\widetilde{T} \leq T^*$ such that the results in Proposition C.11 hold for all time $0 \leq t \leq \widetilde{T} - 1$, we aim to prove Proposition C.11 also hold for $t = \widetilde{T}$.

1. Proof of (C.18) holds for $t = \widetilde{T}$, i.e., $\underline{\rho}_{j,r,i}^{(t)} \geq -64nm^{\frac{1}{q}}\sqrt{\frac{\log(4n^2/\delta)}{d}} \cdot \log(T^*)$ for $t = \widetilde{T}$, $r \in [m]$, $j \in \{\pm 1\}$ and $i \in [n]$:

   Notice that $\underline{\rho}_{j,r,i}^{(t)} = 0$, for $j = y_i$. Therefore, we only need to consider the case that $j \neq y_i$. When $-64nm^{\frac{1}{q}}\sqrt{\frac{\log(4n^2/\delta)}{d}} \cdot \log(T^*) \leq \underline{\rho}_{j,r,i}^{(\widetilde{T}-1)} \leq -32nm^{\frac{1}{q}}\sqrt{\frac{\log(4n^2/\delta)}{d}} \cdot \log(T^*)$, by Lemma C.15 we have that

   $$\langle \mathbf{w}_{j,r}^{(\widetilde{T}-1)}, \boldsymbol{\xi}_i\rangle \leq \underline{\rho}_{j,r,i}^{(\widetilde{T}-1)} + 32nm^{\frac{1}{q}}\sqrt{\frac{\log(4n^2/\delta)}{d}} \cdot \log(T^*) \leq 0,$$

   and thus by (C.7),

   $$\underline{\rho}_{j,r,i}^{(\widetilde{T})} = \underline{\rho}_{j,r,i}^{(\widetilde{T}-1)} + \frac{\eta}{nm} \cdot \ell_i'^{(\widetilde{T}-1)} \cdot \sigma'(\langle \mathbf{w}_{j,r}^{(\widetilde{T}-1)}, \boldsymbol{\xi}_i\rangle) \cdot \mathbb{1}(y_i = -j)\|\boldsymbol{\xi}_i\|_2^2$$

   $$= \underline{\rho}_{j,r,i}^{(\widetilde{T}-1)}$$

   $$\geq -64nm^{\frac{1}{q}}\sqrt{\frac{\log(4n^2/\delta)}{d}} \cdot \log(T^*),$$

   where the last inequality is by induction hypothesis. When $-64nm^{\frac{1}{q}}\sqrt{\frac{\log(4n^2/\delta)}{d}} \cdot \log(T^*) \leq -32nm^{\frac{1}{q}}\sqrt{\frac{\log(4n^2/\delta)}{d}} \cdot \log(T^*) \leq \underline{\rho}_{j,r,i}^{(\widetilde{T}-1)} \leq 0$, by Lemma C.15 we have that

   $$\langle \mathbf{w}_{j,r}^{(\widetilde{T}-1)}, \boldsymbol{\xi}_i\rangle \leq \underline{\rho}_{j,r,i}^{(\widetilde{T}-1)} + 32nm^{\frac{1}{q}}\sqrt{\frac{\log(4n^2/\delta)}{d}} \cdot \log(T^*) \leq 32nm^{\frac{1}{q}}\sqrt{\frac{\log(4n^2/\delta)}{d}} \cdot \log(T^*), \tag{C.32}$$

thus we have that

$$\rho_{j,r,i}^{(\widetilde{T})} = \rho_{j,r,i}^{(\widetilde{T}-1)} + \frac{\eta}{nm} \cdot \ell_i'^{(\widetilde{T}-1)} \cdot \sigma'(\langle \mathbf{w}_{j,r}^{(T-1)}, \boldsymbol{\xi}_i \rangle) \cdot \mathbb{1}(y_i = -j)\|\boldsymbol{\xi}_i\|_2^2$$

$$\geq -32nm^{\frac{1}{q}}\sqrt{\frac{\log(4n^2/\delta)}{d}} \cdot \log(T^*) - \frac{\eta}{nm}\frac{3\sigma_p^2 d}{2}\sigma'\left(32nm^{\frac{1}{q}}\sqrt{\frac{\log(4n^2/\delta)}{d}} \cdot \log(T^*)\right)$$

$$\geq -32nm^{\frac{1}{q}}\sqrt{\frac{\log(4n^2/\delta)}{d}} \cdot \log(T^*) - \frac{\eta}{nm}\frac{3\sigma_p^2 d}{2}q\left(32nm^{\frac{1}{q}}\sqrt{\frac{\log(4n^2/\delta)}{d}} \cdot \log(T^*)\right)$$

$$\geq -64nm^{\frac{1}{q}}\sqrt{\frac{\log(4n^2/\delta)}{d}} \cdot \log(T^*),$$

where we use $\ell_i'^{(\widetilde{T}-1)} \geq -1$ and $\|\boldsymbol{\xi}_i\|_2 \leq \frac{3}{2}\sigma_p^2 d$, and (C.32) in the first inequality, the second inequality is by $32nm^{\frac{1}{q}}\sqrt{\frac{\log(4n^2/\delta)}{d}} \cdot \log(T^*) \leq 1$ in Condition 4.1 , and the third inequality is by $\eta = O\big(nm/(q\sigma_p^2 d)\big)$ in Condition 4.1.

2. The proof of upper bound of $\overline{\rho}_{j,r,i}^{(t)}$ in (C.17) holds for $t = \widetilde{T}$: We have

$$|\ell_i'^{(t)}| = \frac{1}{1 + \exp\{y_i \cdot [F_{+1}(\mathbf{W}_{+1}^{(t)}, \mathbf{x}_i) - F_{-1}(\mathbf{W}_{-1}^{(t)}, \mathbf{x}_i)]\}}$$

$$\leq \exp\{-y_i \cdot [F_{+1}(\mathbf{W}_{+1}^{(t)}, \mathbf{x}_i) - F_{-1}(\mathbf{W}_{-1}^{(t)}, \mathbf{x}_i)]\}$$

$$\leq \exp\{-F_{y_i}(\mathbf{W}_{y_i}^{(t)}, \mathbf{x}_i) + \widetilde{C}_0\}$$

$$\leq \exp\{-\frac{1}{m}\sum_{r'=1}^{m}[\sigma(\langle \mathbf{w}_{y_i,r'}^{(t)}, y_i \cdot \boldsymbol{\mu} \rangle) + \sigma(\langle \mathbf{w}_{y_i,r'}^{(t)}, \boldsymbol{\xi}_i \rangle)] + \widetilde{C}_0\}. \tag{C.33}$$

where the second inequality is due to Lemma C.9 holds, there exists constant $\widetilde{C}_0$ such that $F_j(\mathbf{W}_j^{(t)}, \mathbf{x}_i) \leq \widetilde{C}_0, j = -y_i$, and the third inequality is by the definition of $F_{y_i}$. Moreover, recall the update rule of $\gamma_{j,r}^{(t)}$ and $\overline{\rho}_{j,r,i}^{(t)}$ in (C.6) and (C.7),

$$\gamma_{j,r}^{(t+1)} = \gamma_{j,r}^{(t)} - \frac{\eta}{nm} \cdot \sum_{i=1}^{n} \ell_i'^{(t)} \cdot \sigma'(\langle \mathbf{w}_{j,r}^{(t)}, y_i \cdot \boldsymbol{\mu} \rangle)\|\boldsymbol{\mu}\|_2^2,$$

$$\overline{\rho}_{j,r,i}^{(t+1)} = \overline{\rho}_{j,r,i}^{(t)} - \frac{\eta}{nm} \cdot \ell_i'^{(t)} \cdot \sigma'(\langle \mathbf{w}_{j,r}^{(t)}, \boldsymbol{\xi}_i \rangle) \cdot \mathbb{1}(y_i = j)\|\boldsymbol{\xi}_i\|_2^2.$$

Assume there exists $\overline{\rho}_{j,r,i}^{(t)} > 2m^{\frac{1}{q}}\log(T^*)$ for some $t \in [0, T^*]$, if this does not hold, $\overline{\rho}_{j,r,i}^{(t)} \leq 2m^{\frac{1}{q}}\log(T^*) \leq 4m^{\frac{1}{q}}\log(T^*)$ holds for all $t \in [0, T^*]$, which indicates (C.17) holds. Thus, denote $t_{j,r,i}$ to be the last time $t < T^*$ that $\overline{\rho}_{j,r,i}^{(t)} \leq 2m^{\frac{1}{q}}\log(T^*)$. Then we have that

$$\overline{\rho}_{j,r,i}^{(\widetilde{T})} = \overline{\rho}_{j,r,i}^{(t_{j,r,i})} \underbrace{- \frac{\eta}{nm} \cdot \ell_i'^{(t_{j,r,i})} \cdot \sigma'(\langle \mathbf{w}_{j,r}^{(t_{j,r,i})}, \boldsymbol{\xi}_i \rangle) \cdot \mathbb{1}(y_i = j)\|\boldsymbol{\xi}_i\|_2^2}_{I_1}$$

$$\underbrace{- \sum_{t_{j,r,i} < t < \widetilde{T}} \frac{\eta}{nm} \cdot \ell_i'^{(t)} \cdot \sigma'(\langle \mathbf{w}_{j,r}^{(t)}, \boldsymbol{\xi}_i \rangle) \cdot \mathbb{1}(y_i = j)\|\boldsymbol{\xi}_i\|_2^2}_{I_2}. \tag{C.34}$$

We first bound $I_1$ as follows,

$$|I_1| \leq qn^{-1}m^{-1}\eta\left(\overline{\rho}_{j,r,i}^{(t_{j,r,i})} + 32nm^{\frac{1}{q}}\sqrt{\frac{\log(4n^2/\delta)}{d}} \cdot \log(T^*)\right)^{q-1}\frac{3}{2}\sigma_p^2 d$$

$$\leq q2^q n^{-1}m^{-1}\eta[4m^{\frac{1}{q}}\log(T^*)]^{q-1}\sigma_p^2 d$$

$$\leq m^{\frac{1}{q}}\log(T^*),$$

where the first inequality is by Lemmas C.8 and C.15, the second inequality is by $32nm^{\frac{1}{q}}\sqrt{\frac{\log(4n^2/\delta)}{d}} \cdot \log(T^*) \leq 2m^{\frac{1}{q}}\log(T^*)$ and $\overline{\rho}_{j,r,i}^{(t)} \leq 2m^{\frac{1}{q}}\log(T^*)$, the last inequality is by $\eta \leq nm/\{6q[4m^{\frac{1}{q}}\log(T^*)]^{q-2}\sigma_p^2 d\}$ in Condition 4.1.

We then give a bound for $I_2$. For $t_{j,r,i} < t < \widetilde{T}$ and $y_i = j$, we can lower bound $\langle \mathbf{w}_{j,r}^{(t)}, \boldsymbol{\xi}_i \rangle$ as follows,

$$\langle \mathbf{w}_{j,r}^{(t)}, \boldsymbol{\xi}_i \rangle \geq \overline{\rho}_{j,r,i}^{(t)} - 32nm^{\frac{1}{q}}\sqrt{\frac{\log(4n^2/\delta)}{d}} \cdot \log(T^*)$$

$$\geq 2m^{\frac{1}{q}}\log(T^*) - 32nm^{\frac{1}{q}}\sqrt{\frac{\log(4n^2/\delta)}{d}} \cdot \log(T^*)$$

$$\geq m^{\frac{1}{q}}\log(T^*),$$

where the first inequality is by Lemma C.15, the second inequality is by $\overline{\rho}_{j,r,i}^{(t)} > 2m^{\frac{1}{q}}\log(T^*)$ due to the definition of $t_{j,r,i}$, the last inequality is by $32nm^{\frac{1}{q}}\sqrt{\frac{\log(4n^2/\delta)}{d}} \cdot \log(T^*) \leq m^{\frac{1}{q}}\log(T^*)$. Similarly, for $t_{j,r,i} < t < \widetilde{T}$ and $y_i = j$, we can also upper bound $\langle \mathbf{w}_{j,r}^{(t)}, \boldsymbol{\xi}_i \rangle$ as follows,

$$\langle \mathbf{w}_{j,r}^{(t)}, \boldsymbol{\xi}_i \rangle \leq \overline{\rho}_{j,r,i}^{(t)} + 8nm^{\frac{1}{q}}\sqrt{\frac{\log(4n^2/\delta)}{d}} \cdot 4\log(T^*)$$

$$\leq 4m^{\frac{1}{q}}\log(T^*) + 32nm^{\frac{1}{q}}\sqrt{\frac{\log(4n^2/\delta)}{d}} \cdot \log(T^*)$$

$$\leq 8m^{\frac{1}{q}}\log(T^*),$$

where the first inequality is by Lemma C.15, the second inequality is by induction hypothesis $\overline{\rho}_{j,r,i}^{(t)} \leq 4m^{\frac{1}{q}}\log(T^*)$, the last inequality is by $32nm^{\frac{1}{q}}\sqrt{\frac{\log(4n^2/\delta)}{d}} \cdot \log(T^*) \leq m^{\frac{1}{q}}\log(T^*)$. Thus, plugging the upper and lower bounds of $\langle \mathbf{w}_{j,r}^{(t)}, \boldsymbol{\xi}_i \rangle$ into $I_2$ gives

$$|I_2| = \sum_{t_{j,r,i} < t < \widetilde{T}} \frac{\eta}{nm} \cdot |\ell_i'^{(t)}| \cdot \sigma'(\langle \mathbf{w}_{j,r}^{(t)}, \boldsymbol{\xi}_i \rangle) \cdot \mathbb{1}(y_i = j)\|\boldsymbol{\xi}_i\|_2^2$$

$$\leq \sum_{t_{j,r,i} < t < \widetilde{T}} \frac{\eta}{nm} \cdot \exp(-\frac{1}{m}\sigma(\langle \mathbf{w}_{j,r}^{(t)}, \boldsymbol{\xi}_i \rangle) + \widetilde{C}_0) \cdot \sigma'(\langle \mathbf{w}_{j,r}^{(t)}, \boldsymbol{\xi}_i \rangle) \cdot \mathbb{1}(y_i = j)\|\boldsymbol{\xi}_i\|_2^2$$

$$\leq \frac{e^{\widetilde{C}_0}\eta T^*}{nm} \exp(-\frac{(4m^{\frac{1}{q}}\log(T^*))^q}{m \cdot 4^q})q(4m^{\frac{1}{q}}\log(T^*))^{q-1}2^{q-1}\frac{3}{2}\sigma_p^2 d$$

$$\leq 0.25T^* \exp(-\frac{(4m^{\frac{1}{q}}\log(T^*))^q}{m \cdot 4^q}) \cdot 4m^{\frac{1}{q}}\log(T^*)$$

$$= 0.25T^* \exp(-\log(T^*)^q) \cdot 4m^{\frac{1}{q}}\log(T^*)$$

$$\leq m^{\frac{1}{q}}\log(T^*),$$

where the first inequality is by (C.33), the second inequality is by Lemma C.8 and upper and lower bound of $\langle \mathbf{w}_{j,r}^{(t)}, \boldsymbol{\xi}_i \rangle$ given above, the third inequality is by $\eta = O(nm/\{e^{\widetilde{C}_0}q2^{q+2}[4m^{\frac{1}{q}}\log(T^*)]^{q-2}\sigma_p^2 d\})$ in Condition 4.1, and the last inequality is due to the fact that $\log(T^*)^q \geq \log(T^*)$. Plugging the bound of $I_1, I_2$ into (C.34) completes the proof for $\overline{\rho}_{j,r,i}^{(t)}$.

3. Similarly, we can prove $\gamma_{j,r}^{(0)} \leq \gamma_{j,r}^{(t)} \leq 4m^{\frac{1}{q}}\log(T^*)$ in (C.16). By $|\gamma_{j,r}^{(0)}| \leq 4m^{\frac{1}{q}}\log(T^*)$ in Assumption C.1 and $\gamma_{j,r}^{(t)}$ is increasing, we have $\gamma_{j,r}^{(0)} \leq \gamma_{j,r}^{(t)}$ naturally holds. Therefore, we only need to prove $\gamma_{j,r}^{(t)} \leq 4m^{\frac{1}{q}}\log(T^*)$ holds for all $0 \leq t \leq T^*$: Assume there exists

$\gamma_{j,r}^{(t)} > 2m^{\frac{1}{q}}\log(T^*)$ for some $t \in [0, T^*]$, if this does not hold, $\gamma_{j,r}^{(t)} \leq 2m^{\frac{1}{q}}\log(T^*) \leq 4m^{\frac{1}{q}}\log(T^*)$ holds for all $t \in [0, T^*]$ indicates (C.16) holds. Thus, denote $\widetilde{t}_{j,r}$ to be the last time $t < T^*$ that $\gamma_{j,r}^{(t)} \leq 2m^{\frac{1}{q}}\log(T^*)$ hold. Then we have that

$$\gamma_{j,r}^{(\widetilde{T})} = \gamma_{j,r}^{(\widetilde{t}_{j,r})} - \underbrace{\frac{\eta}{nm} \cdot \sum_{i=1}^{n} \ell_i'^{(\widetilde{t}_{j,r})} \cdot \sigma'(\langle \mathbf{w}_{j,r}^{(\widetilde{t}_{j,r})}, y_i\boldsymbol{\mu}\rangle) \cdot \|\boldsymbol{\mu}\|_2^2}_{I_1'}$$

$$- \underbrace{\sum_{\widetilde{t}_{j,r} < t < \widetilde{T}} \frac{\eta}{nm} \cdot \sum_{i=1}^{n} \ell_i'^{(t)} \cdot \sigma'(\langle \mathbf{w}_{j,r}^{(t)}, y_i\boldsymbol{\mu}\rangle) \cdot \|\boldsymbol{\mu}\|_2^2}_{I_2'}. \tag{C.35}$$

We first bound $I_1'$ as follows,

$$|I_1'| \leq \eta n^{-1} m^{-1} nq \left(\gamma_{j,r}^{(\widetilde{t}_{j,r})}\right)^{q-1} \|\boldsymbol{\mu}\|_2^2 \leq qm^{-1}\eta(2m^{\frac{1}{q}}\log(T^*))^{q-1}\|\boldsymbol{\mu}\|_2^2 \leq m^{\frac{1}{q}}\log(T^*),$$

where the first inequality is by Lemma C.9 and C.10, the second inequality is by $\gamma_{j,r}^{(\widetilde{t}_{j,r})} \leq 2m^{\frac{1}{q}}\log(T^*)$, the last inequality is by $\eta \leq m \cdot 2^{q-3}/\{q[4m^{\frac{1}{q}}\log(T^*)]^{q-2}\|\boldsymbol{\mu}\|_2^2\}$ in Condition 4.1.

We then bound $I_2'$, we have

$$|I_2'| = \sum_{t_{j,r,i} < t < \widetilde{T}} \frac{\eta}{nm} \cdot \sum_{i=1}^{n} |\ell_i'^{(t)}| \cdot \sigma'(\langle \mathbf{w}_{j,r}^{(t)}, y_i\boldsymbol{\mu}\rangle) \cdot \|\boldsymbol{\mu}\|_2^2$$

$$= \sum_{t_{j,r,i} < t < \widetilde{T}} \frac{\eta}{nm} \cdot \left[ \sum_{i=1}^{n} |\ell_i'^{(t)}| \cdot \sigma'(\langle \mathbf{w}_{j,r}^{(t)}, y_i\boldsymbol{\mu}\rangle) \cdot \mathbb{1}(y_i = j)\|\boldsymbol{\mu}\|_2^2 \right.$$

$$\left. + \sum_{i=1}^{n} |\ell_i'^{(t)}| \cdot \sigma'(\langle \mathbf{w}_{j,r}^{(t)}, y_i\boldsymbol{\mu}\rangle) \cdot \mathbb{1}(y_i \neq j)\|\boldsymbol{\mu}\|_2^2 \right]$$

$$= \sum_{t_{j,r,i} < t < \widetilde{T}} \frac{\eta}{nm} \cdot \sum_{i=1}^{n} |\ell_i'^{(t)}| \cdot \sigma'(\langle \mathbf{w}_{j,r}^{(t)}, y_i\boldsymbol{\mu}\rangle) \cdot \mathbb{1}(y_i = j)\|\boldsymbol{\mu}\|_2^2 \tag{C.36}$$

where the third equality is by $\langle \mathbf{w}_{j,r}^{(t)}, y_i\boldsymbol{\mu}\rangle = -\gamma_{j,r}^{(t)} \leq 0$ in Lemma C.9. For $t_{j,r,i} < t < \widetilde{T}$, we upper bound $\langle \mathbf{w}_{j,r}^{(t)}, y_i\boldsymbol{\mu}\rangle, j = y_i$ (namely $\langle \mathbf{w}_{y_i,r}^{(t)}, y_i\boldsymbol{\mu}\rangle$) as

$$\langle \mathbf{w}_{j,r}^{(t)}, y_i\boldsymbol{\mu}\rangle = y_i \cdot j \cdot \gamma_{j,r}^{(t)} = \gamma_{j,r}^{(t)} \leq 4m^{\frac{1}{q}}\log(T^*)$$

where the equality is by Lemma C.10, the second inequality is by induction hypothesis $|\gamma_{j,r}^{(t)}| \leq 4m^{\frac{1}{q}}\log(T^*)$. For $\widetilde{t}_{j,r} < t < \widetilde{T}$ and $y_i = j$, we can also lower bound $\langle \mathbf{w}_{j,r}^{(t)}, y_i\boldsymbol{\mu}\rangle$ (namely $\langle \mathbf{w}_{y_i,r}^{(t)}, y_i\boldsymbol{\mu}\rangle$) as follows,

$$\langle \mathbf{w}_{j,r}^{(t)}, y_i\boldsymbol{\mu}\rangle = y_i \cdot j \cdot \gamma_{j,r}^{(t)} = \gamma_{j,r}^{(t)} \geq 2m^{\frac{1}{q}}\log(T^*)$$

where the inequality by $\gamma_{j,r}^{(t)} > 2m^{\frac{1}{q}}\log(T^*)$ due to the definition of $\widetilde{t}_{j,r}$. Thus, plugging the upper bound of $\langle \mathbf{w}_{j,r}^{(t)}, y_i\boldsymbol{\mu}\rangle$ and the lower bound of $\langle \mathbf{w}_{j,r}^{(t)}, y_i\boldsymbol{\mu}\rangle$ when $y_i = j$ into $|I_2'|$

(C.36) gives

$$|I_2'| = \sum_{t_{j,r,i} < t < \widetilde{T}} \frac{\eta}{nm} \cdot \sum_{i=1}^{n} |\ell_i'^{(t)}| \cdot \sigma'(\langle \mathbf{w}_{j,r}^{(t)}, y_i \boldsymbol{\mu} \rangle) \cdot \mathbb{1}(y_i = j) \cdot \|\boldsymbol{\mu}\|_2^2$$

$$\leq \sum_{t_{j,r,i} < t < \widetilde{T}} \frac{\eta}{nm} \cdot \sum_{i=1}^{n} \exp\left(-\frac{1}{m}\sigma(\langle \mathbf{w}_{j,r}^{(t)}, y_i \boldsymbol{\mu} \rangle) + \widetilde{C}_0\right) \cdot \sigma'(\langle \mathbf{w}_{j,r}^{(t)}, y_i \boldsymbol{\mu} \rangle) \cdot \mathbb{1}(y_i = j) \cdot \|\boldsymbol{\mu}\|_2^2$$

$$\leq \frac{e^{\widetilde{C}_0} \eta T^*}{m} \exp\left(-\frac{(4m^{\frac{1}{q}}\log(T^*))^q}{m \cdot 2^q}\right) q(4m^{\frac{1}{q}}\log(T^*))^{q-1} \|\boldsymbol{\mu}\|_2^2$$

$$\leq 0.25 T^* \exp\left(-\frac{(4m^{\frac{1}{q}}\log(T^*))^q}{m \cdot 2^q}\right) \cdot 4m^{\frac{1}{q}}\log(T^*)$$

$$\leq 0.25 T^* \exp\left(-\log(T^*)^q\right) \cdot 4m^{\frac{1}{q}}\log(T^*)$$

$$\leq m^{\frac{1}{q}}\log(T^*),$$

where the first inequality is by (C.33), the second inequality is by the upper bound of $\langle \mathbf{w}_{j,r}^{(t)}, y_i \boldsymbol{\mu} \rangle$ and the lower bound of $\langle \mathbf{w}_{j,r}^{(t)}, y_i \boldsymbol{\mu} \rangle$ ($y_i = j$) given above, the third inequality is by $\eta \leq O\big(m/\{4e^{\widetilde{C}_0} q[4m^{\frac{1}{q}}\log(T^*)]^{q-2}\|\boldsymbol{\mu}\|_2^2\}\big)$ in Condition 4.1, and the last inequality is due to the fact that $\log(T^*)^q \geq \log(T^*)$. Plugging the bound of $I_1', I_2'$ into (C.35) completes the proof for $\gamma_{j,r}^{(t)}$.

Therefore, Proposition C.11 holds for $t = \widetilde{T}$, which completes the induction. □

Finally, the following Lemma C.12, which is based on Proposition C.11, is proved.

*Proof of Lemma C.12.* Firstly, we prove that

$$-\ell'\big(y_i f(\mathbf{W}^{(t)}, \mathbf{x}_i)\big) \cdot \|\nabla f(\mathbf{W}^{(t)}, \mathbf{x}_i)\|_F^2 = O(\max\{\|\boldsymbol{\mu}\|_2^2, \sigma_p^2 d\}). \tag{C.37}$$

Without loss of generality, we suppose that $y_i = 1$ and $\mathbf{x}_i = [\boldsymbol{\mu}^\top, \boldsymbol{\xi}_i]$. Then we have that

$$\|\nabla f(\mathbf{W}^{(t)}, \mathbf{x}_i)\|_F \leq \frac{1}{m}\sum_{j,r}\left\|\left[\sigma'(\langle \mathbf{w}_{j,r}^{(t)}, \boldsymbol{\mu}\rangle)\boldsymbol{\mu} + \sigma'(\langle \mathbf{w}_{j,r}^{(t)}, \boldsymbol{\xi}_i\rangle)\boldsymbol{\xi}_i\right]\right\|_2$$

$$\leq \frac{1}{m}\sum_{j,r}\sigma'(\langle \mathbf{w}_{j,r}^{(t)}, \boldsymbol{\mu}\rangle)\|\boldsymbol{\mu}\|_2 + \frac{1}{m}\sum_{j,r}\sigma'(\langle \mathbf{w}_{j,r}^{(t)}, \boldsymbol{\xi}_i\rangle)\|\boldsymbol{\xi}_i\|_2$$

$$\leq 2q\left[F_{+1}(\mathbf{W}_{+1}^{(t)}, \mathbf{x}_i)\right]^{(q-1)/q} \max\{\|\boldsymbol{\mu}\|_2, 2\sigma_p\sqrt{d}\}$$

$$+ 2q\left[F_{-1}(\mathbf{W}_{-1}^{(t)}, \mathbf{x}_i)\right]^{(q-1)/q} \max\{\|\boldsymbol{\mu}\|_2, 2\sigma_p\sqrt{d}\}$$

$$\leq 2q\left\{\left[F_{+1}(\mathbf{W}_{+1}^{(t)}, \mathbf{x}_i)\right]^{(q-1)/q} + \left[1 + 4m^{\frac{1}{q}}\log(T^*)\right]^{(q-1)/q}\right\} \max\{\|\boldsymbol{\mu}\|_2, 2\sigma_p\sqrt{d}\},$$

where the first and second inequalities are by triangle inequality, the third inequality is by Jensen's inequality and Lemma C.8. The last inequality is by Lemma C.9, i.e., $\frac{1}{m}\sum_{j=-y, r\in[m]}\sigma(\langle \mathbf{w}_{j,r}^{(t)}, y\boldsymbol{\mu}\rangle) \leq \frac{1}{m}\sum_{j=-y, r\in[m]}\sigma(-\gamma_{j,r}^{(t)}) \leq \frac{1}{m}\sum_{j=-y, r\in[m]}\sigma(-\gamma_{j,r}^{(0)}) \leq \max_{j,r}\{0, (-\gamma_{j,r}^{(0)})^q\} \ll 4m^{\frac{1}{q}}\log(T^*)$, therefore, $F_{-1}(\mathbf{W}_{-1}^{(t)}, \mathbf{x}_i) \leq 1 + 4m^{\frac{1}{q}}\log(T^*)$ by

**Lemma C.9.** Denote $A = F_{+1}(\mathbf{W}_{+1}^{(t)}, \mathbf{x}_i)$, then we have $A \geq 0$. Thus,

$$- \ell'\big(y_i f(\mathbf{W}^{(t)}, \mathbf{x}_i)\big) \cdot \|\nabla f(\mathbf{W}^{(t)}, \mathbf{x}_i)\|_F^2$$

$$\leq -\ell'(A - 1 - 4m^{\frac{1}{q}} \log(T^*)) \cdot 4q^2 \left\{ A^{(q-1)/q} + \left[1 + 4m^{\frac{1}{q}} \log(T^*)\right]^{(q-1)/q} \right\}^2 \cdot \max\{\|\boldsymbol{\mu}\|_2, 2\sigma_p\sqrt{d}\}^2$$

$$= -4q^2 \ell'(A - 1 - 4m^{\frac{1}{q}} \log(T^*)) \left\{ A^{(q-1)/q} + \left[1 + 4m^{\frac{1}{q}} \log(T^*)\right]^{(q-1)/q} \right\}^2 \cdot \max\{\|\boldsymbol{\mu}\|_2^2, 4\sigma_p^2 d\}$$

$$\leq \left\{ \max_{z>0} -4q^2 \ell'(z - 1 - 4m^{\frac{1}{q}} \log(T^*))\{z^{(q-1)/q} + \left[1 + 4m^{\frac{1}{q}} \log(T^*)\right]^{(q-1)/q}\}^2 \right\} \cdot \max\{\|\boldsymbol{\mu}\|_2^2, 4\sigma_p^2 d\}$$

$$\overset{(i)}{=} O(\max\{\|\boldsymbol{\mu}\|_2^2, \sigma_p^2 d\}),$$

where (i) is by $\max_{z\geq 0} -4q^2 \ell'(z - 1 - 4m^{\frac{1}{q}} \log(T^*))(z^{(q-1)/q} + \left[1 + 4m^{\frac{1}{q}} \log(T^*)\right]^{(q-1)/q})^2 < \infty$ because $\ell'$ has an exponentially decaying tail. Now we can upper bound the gradient norm $\|\nabla L_S(\mathbf{W}^{(t)})\|_F$ as follows,

$$\|\nabla L_S(\mathbf{W}^{(t)})\|_F^2 \leq \left[\frac{1}{n} \sum_{i=1}^n \ell'\big(y_i f(\mathbf{W}^{(t)}, \mathbf{x}_i)\big) \|\nabla f(\mathbf{W}^{(t)}, \mathbf{x}_i)\|_F\right]^2$$

$$\leq \left[\frac{1}{n} \sum_{i=1}^n \sqrt{-O(\max\{\|\boldsymbol{\mu}\|_2^2, \sigma_p^2 d\})\ell'\big(y_i f(\mathbf{W}^{(t)}, \mathbf{x}_i)\big)}\right]^2$$

$$\leq O(\max\{\|\boldsymbol{\mu}\|_2^2, \sigma_p^2 d\}) \cdot \frac{1}{n} \sum_{i=1}^n -\ell'\big(y_i f(\mathbf{W}^{(t)}, \mathbf{x}_i)\big)$$

$$\leq O(\max\{\|\boldsymbol{\mu}\|_2^2, \sigma_p^2 d\}) L_S(\mathbf{W}^{(t)}),$$

where the first inequality is by triangle inequality, the second inequality is by (C.37), the third inequality is by Cauchy-Schwartz inequality and the last inequality is due to the property of the cross entropy loss $-\ell' \leq \ell$. $\qquad\square$

