# OpenReview forum: "Understanding the Benefits of SimCLR Pre-Training in Two-Layer Convolutional Neural Networks"
_ICLR.cc/2025/Conference — Submitted to ICLR 2025_

### Official Review · Reviewer_FqYo · 2024-10-28

**Soundness:** 3
**Presentation:** 2
**Contribution:** 3
**Rating:** 5
**Confidence:** 2

**Summary:**

This theoretical paper presents new contributions aimed at enhancing our understanding of how pre-training schemes improve fine-tuning performance. It introduces some theoretical results about SimCLR, one of the most popular contrastive learning methods for vision tasks. The paper demonstrates that, under specific conditions regarding the amount of labeled data, SimCLR pre-training coupled with supervised fine-tuning can achieve nearly optimal test loss. The main theoretical results are presented in Theorem 4.2 and 4.3. The paper and its appendix contain the proofs of these theorems. The paper also contains many novel analysis tools that enable the study of the SimCLR algorithm. The theoretical results are valid in a specific setting (simple binary classification task and a two-layers ConvNet), which is not a standard setting and may limit the potential impact of the contributions.

**Strengths:**

- This paper introduces new theoretical contributions to understand the advantage of SimCLR pre-training in fine-tuning stage. The paper demonstrates that, under specific conditions regarding the amount of labeled data, SimCLR pre-training coupled with supervised fine-tuning can achieve nearly optimal test loss for two-layers ConvNets. Under certain conditions related to the quantity of labeled and unlabeled data, as well as the signal-to-noise ratio (SNR), the convergence of training loss and a low test loss are assured. These theoretical results indicate that SimCLR pre-training during the fine-tuning stage can reduce label complexity, leading to a lower test loss.
- The paper and its appendix contain the proofs of the theorems 4.2 and 4.3, which are the main results of this paper.
- The paper contains many novel analysis tools that enable the study of the SimCLR algorithm.
- The appendix contains some experiments on synthetic and MNIST datasets to confirm the theoretical results.

**Weaknesses:**

**Impact of the contributions.** The paper studied a very specific setting: simple binary classification task and a two-layers ConvNet with $RELU^q$ output activation. In practice, this setting is not used often. For instance, SimCLR paper uses a ResNet-50 and was used on multiple multiclass classification benchmarks. There is a big gap between the theoretical setting used in this paper and the practical setting, so it is difficult to understand the potential impact of this paper contributions. It may be difficult to do, but it would be interesting to relax some of the assumptions to be able to get theoretical for a family of models which include ResNet-50.


**Lack of justification/motivation.** The paper has many assumptions that are not well motivated, or claims that are not well justified. Here are a few examples:
- L162: "this data model is particularly suitable to study SimCLR, which is originally proposed for vision tasks." This claim appears to lack sufficient justification. It may be beneficial to include an analysis to support this claim.
- L153: The patches are represented as vector rather than matrix in Definition 3.1. It seems to ignore the 2d nature of the patches. It could be valuable to justify this choice.
- L215: The paper focuses on two-layers ConvNet with $ReLU^q$ activation function with q > 2. This type of activation is not the most popular choice so it could be interesting to explain this choice.
- The assumptions regarding data augmentation for pre-training are not clearly stated.
- Definition 3.1 does not explain the assumptions on $\mu$


**Clarity.** The presentation and clarity of the paper could be enhanced. The paper includes a considerable number of notations, which can make it challenging to keep track of them all. Some notations appear to be undefined, while others are introduced later in the text, making it difficult to fully understand them until the subsequent paragraphs are read. Here are a few examples:
- $\Omega$ symbol is used L84, but it is defined at L106
- $W$ is used at L171 but it does not seem to be defined.
- $w_r$ is used at L172 but it is defined at L196
- $[F(W, x)]_r$ is used at L172 but it does not seem to be defined.
- $[n_0]$ is used L184 but it does not seem to be defined.
- L200: some variables like the unlabeled dataset are used to compute the loss, but they are not shown as input: $L(W, S_{unlabeled})$
- L158: The third bullet point in Definition 3.1 is not clear. It looks like a something is missing.


**Things to improve the paper that did not impact the score:**
- A lot of equations do not have number so it is difficult to reference them in the review.

**Questions:**

The questions are arranged in order of importance, with the first question being the most important.

- The paper should discuss the scope of the contributions, as the theoretical setting seems quite different from the practical setting. In particular, is it possible to remove or relax some assumptions to include a larger family of models?
- L279: "we remark that most of these assumptions are non-essential." It seems to be counter-intuitive to make some assumptions that are non-essential. How do the theoretical results change if these non-essential assumptions are removed?
- What are the assumptions about the data augmentation?
- Why does the paper focus $ReLU^q$ activation function for the theoretical analysis? Is it possible to include other activation functions?
- What are the assumptions about $\mu$? Can $\mu$ be the zero/null vector?

---

> ### Author Response · Authors · 2024-11-23
> **Response to Reviewer FqYo (Part a)**
>
> Thanks for your detailed review. Our answers to your question are as follows.
>
> ---
>
> **Weaknesses**:
>
> **W1&Q1**:
> - **W1**: Impact of the contributions. The paper studied a very specific setting: simple binary classification task and a two-layers ConvNet with $RELU^q$ output activation. In practice, this setting is not used often. For instance, SimCLR paper uses a ResNet-50 and was used on multiple multiclass classification benchmarks. There is a big gap between the theoretical setting used in this paper and the practical setting, so it is difficult to understand the potential impact of this paper contributions. It may be difficult to do, but it would be interesting to relax some of the assumptions to be able to get theoretical for a family of models which include ResNet-50.
> - **Q1**: The paper should discuss the scope of the contributions, as the theoretical setting seems quite different from the practical setting. In particular, is it possible to remove or relax some assumptions to include a larger family of models?
>
> **AW1&AQ1**: We admit that it requires some specific assumptions to provide an accurate theoretical analysis. We would also like to explain that we choose to study two-layer CNNs because it allows a direct comparison with the results of direct supervised learning on two-layer CNNs in [1].  However, even for this simple setting and two-layer CNN architecture, the influence of contrastive learning on supervised fine-tuning has not been analyzed in theory before. We believe that our current analysis provides a clear first step towards the theoretical understanding of the advantages of contrastive learning.
>
> In our theoretical analysis, we are following a signal learning framework. We believe that the insight behind this signal learning framework can help us understand many practical learning tasks as well.
>
> The idea of the signal learning framework is that, when training complicated neural networks on a finite training dataset, there may exist many (probably infinitely many) different parameter configurations that can all lead to $100\\%$ training accuracy. However, neural networks with different parameter configurations may perform drastically differently on test data. We can generally treat the “information in the data input that are related to the label” (characterized by the vector $\boldsymbol{\mu}$ in our theoretical study) as the “signal” in the data, and treat the “information in the data input that are not related to the label” (characterized by the Gaussian noise vector $\boldsymbol{\xi}$ in our theoretical study) as “noises”. Then clearly,  The signal is the label-dependent and generalizable part of the information, and noise is the label-independent and ungeneralizable part of the information, and whether the obtained neural network after training can achieve good prediction depends on whether it utilizes the “signal part” or the “noise part” to fit the training data, and this is exactly how we build our theory.
>
> Our analysis on SimCLR demonstrates that, although SimCLR does not utilize any labeled data, the knowledge that “data augmentation should not change the label” suffices to help the neural network to capture more signals in the pre-training stage. We believe this characterization is true for more practical scenarios as well. The idea it conveyed that SimCLR pre-training encourages signal learning in the supervised learning of CNNs also holds for more general cases. Although it is difficult to verify on real data because “signals” and “noises” and the level they are captured by a more complicated network is hard to quantify, we have provided additional real-data experiments to justify our claim (to a certain extent) in the revision. The added experiments can be found in Appendix A in the revised paper.
>
> For extension to more complicated architecture, currently there is a lack of tools and techniques to enable such a performance comparison with directly supervised learning in more modern models. Even some of the direct supervised learning has not been well studied in theory in these modern models. It is one of the important future directions to extend the current analysis of the influence of contrastive learning on supervised learning on CNNs to more complicated models.

---

> > ### Comment · Reviewer_FqYo · 2024-11-26
> >
> > Thank you for your responses and for providing the updated paper. While I appreciate that this is a theoretical paper and that certain assumptions are necessary, my primary concern remains with the setting of the analysis. Specifically, the use of a two-layer convolutional architecture with ReLU activation appears somewhat distant from current state-of-the-art models. I also observed that other reviewers have raised similar concerns regarding this aspect.

---

> ### Author Response · Authors · 2024-11-23
> **Response to Reviewer FqYo (Part b)**
>
> ---
>
> **W2**: Lack of justification/motivation. The paper has many assumptions that are not well motivated, or claims that are not well justified.
>
> **AW2**: The following are our explanations of the points mentioned in W2.
>
> - L162: SimCLR is one of the most popular contrastive learning methods for vision tasks. The data model adopted in this paper (Definition 3.1) is also inspired by the image data: the image inputs consist of different patches. Within these patches, only some of the patches are related to the class label of the image, while other patches are unrelated to the labels. Thus, the data model is designed to have a signal patch $y_i\cdot\boldsymbol{\mu}$ and a noise patch $\boldsymbol{\xi}$ that is irrelevant to the label [1]. Therefore, we claim that this data model with such design is suitable to study SimCLR.
>
> - L153:­­  Please note that we consider convolution filters and data patches that are both flattened into vectors. The image patches, which are 2-dim in nature, are reshaped into a vector, and the convolution filters are also reshaped into vectors of the same size, and the inner product between the vectorized filters and image patches are exactly equivalent to the desired computation when both the filters and the image patches are in 2-dim shapes. Therefore, our analysis exactly covers 2-dim images, or even 3-d tensors formed by 2-dim images with RGB channels.
>
>
> -  L215: $ReLU^q$ activation function with q > 2 is chosen here since the proof relies on the smoothness assumption.
>
> - The data augmentation assumptions &**Q3**: The data augmentation follows the following rules. For the data $\tilde{\boldsymbol{x}}$, an augmented data point $\tilde{\boldsymbol{x}}^{\text{pre-training}}$ is generated from the distribution $\mathbb{P}(\boldsymbol{x}|y= y_i)$. Given this distribution, one patch of $\tilde{\boldsymbol{x}}^{\text{pre-training}}$ is assigned as $y_i\cdot\boldsymbol{\mu}$ and the other patch of $\tilde{\boldsymbol{x}}^{\text{pre-training}}$ is assigned as $\tilde{\boldsymbol{\xi}}_i \sim \mathcal{N}(\mathbf{0}, \sigma_p^2\cdot(\boldsymbol{I} -\boldsymbol{\mu}\boldsymbol{\mu}^\top\cdot\| |\boldsymbol{\mu}\||_2^{-2}))$.
>
> - $\mu$: The vector $\mu$ refers to the signal and we do not have any extra assumption posed to it apart from the certain condition on signal-to-noise ratio ($\mathrm{SNR}$) mentioned in Condition 4.1.
>
> ---
>
> **W3**: Clarity. The presentation and clarity of the paper could be enhanced. The paper includes a considerable number of notations, which can make it challenging to keep track of them all. Some notations appear to be undefined, while others are introduced later in the text, making it difficult to fully understand them until the subsequent paragraphs are read.
>
> **AW3**: Thanks for pointing out this. We have revised the paper to make these notations clearer.
>
> - $\Omega$: $\Omega$ is introduced in the notation section. Due to the relatively fixed structure of the introduction section, the notation part could only be put at the end of the introduction section, around L106. We have added a footnote for $\Omega$ in L84 to explain it.
>
> - $\boldsymbol{W}$ refers to the matrix of filters, $\boldsymbol{w}_r$ refers to the r-th row of the matrix $W$, namely the r-th filter.
>
> - $[F(W,x)]_r$ refers to the r-th dimension of the output. Here,we adopt the way of first proposing the equation, and then explaining its mechanism and all the terms within it one by one.
>
> - $[n_0]$ refers to the set $\\{1,2,\dots,n_0\\}$. We have already added this explanation of it in the Notation section.
>
> - L200: We have revised this to specify the dataset used in the loss function computation $L_{S_{unlabeled}}(\boldsymbol{W})$.
>
> - L158: This data model definition has already been applied to a series of works [1,2,3]. We believe there isn’t any problem with it. We are happy to answer any questions if you still feel there are some specific problems with it.
>
> ---
>
> **W4**: Things to improve the paper that did not impact the score: A lot of equations do not have number so it is difficult to reference them in the review.
>
> **AW4**: Actually, we are following the style where equations in the paper are numbered only if they are referenced in the paper. We believe it is common to follow such a style for numbering equations.
>
> ---
>
> **Questions**:
>
> **Q2**: L279: "we remark that most of these assumptions are non-essential." It seems to be counter-intuitive to make some assumptions that are non-essential. How do the theoretical results change if these non-essential assumptions are removed?
>
> **AQ2**: The cited sentence here means most of these assumptions are easy to satisfy for most cases, and thus we only need to focus on the first two key conditions in Condition 4.1. We have revised this sentence to “we remark that most of these assumptions are easy to satisfy” in the paper, making it more precise.
>
> ---

---

> ### Author Response · Authors · 2024-11-23
> **Response to Reviewer FqYo (Part c)**
>
> ---
>
> **Q4**: Why does the paper focus $ReLU^q$ activation function for the theoretical analysis? Is it possible to include other activation functions?
>
> **AQ4**: In this paper, we analyze the advantage of SimCLR pre-training by comparing the performance of SimCLR pre-training followed by fine-tuning with direct supervised learning in [1]. Therefore, we adopt the same two-layer CNN model as that in [1] to allow a direct comparison.
>
> It is possible to conduct similar sets of analysis for CNNs with other activation functions. However, since CNNs with other activation functions may have different training dynamics, the corresponding conditions and assumptions may also vary.
>
> ---
>
> **Q5**: What are the assumptions about $\mu$? Can $\mu$ be the zero/null vector?
>
> **AQ5**: $\boldsymbol{\mu}$ here refers to the signal patch of the data in this problem. Apart from the requirement on the signal-to-noise ratio (SNR) in Condition 4.1, we do not pose any extra assumptions on the signal $\boldsymbol{\mu}$. Since SNR is defined as $\mathrm{SNR}=\||\boldsymbol{\mu}\||_2/(\sigma_p \sqrt{d})$ and we require $n_0\cdot\mathrm{SNR}^2=\tilde\Omega(1)$ in Condition 4.1 to achieve a small test loss, it implies that the signal $\boldsymbol{\mu}$ can not be a zero vector.
>
> ---
>
> Reference:
>
> [1] Yuan Cao, Zixiang Chen, Misha Belkin, and Quanquan Gu. Benign overfitting in two-layer convolutional neural networks. NeurIPS 35:25237–25250, 2022.
>
> [2] Samy Jelassi and Yuanzhi Li. Towards understanding how momentum improves generalization in deep learning. In ICML, pages 9965–10040. PMLR, 2022.
>
> [3] Yiwen Kou, Zixiang Chen, Yuanzhou Chen, and Quanquan Gu. Benign overfitting in two-layer relu convolutional neural networks. In ICML, pages 17615–17659. PMLR, 2023.

---

> ### Author Response · Authors · 2024-11-29
>
> ---
>
> Thank you for your response. We would like to further address your concern regarding the gap between the two-layer CNNs considered in this paper and practical applications as follows.
>
> - In [1], the authors study the direct supervised learning of two-layer CNNs. The goal of our paper is to establish comparable theoretical guarantees for SimCLR pre-training followed by supervised fine-tuning, allowing us to demonstrate how SimCLR enhances learning performance in comparison to [1]. Therefore, we believe that studying two-layer CNNs is a natural and appropriate choice that serves our purpose well.
>
> - Please note that even for direct supervised learning, results similar to those in [1] have not been established for deeper models, and similar studies [2,3,4] mostly focus on two-layer models. Compared to previous works, our research examines the more complicated procedure of “SimCLR pre-training -> remove projection head -> attach classification layer -> supervised fine-tuning”, which is more challenging to study. Therefore, while we agree that there is a big gap between our theory and the practice, we must emphasize that this gap exists in most existing theoretical studies. We sincerely hope that our technical contribution in developing a framework for the theoretical analysis of SimCLR is recognized, as we believe our novel analysis tools may help advance theoretical studies towards more practical scenarios.
>
>
> Although our paper focuses on two-layer CNNs, we believe the insights obtained from our theoretical study are applicable to more general settings. As demonstrated in the additional experiments on ViTs in Appendix A in the latest revision of the paper, the phenomenon that SimCLR pre-training enhances signal learning during supervised fine-tuning is also observed. We believe our theoretical analysis represents an important step towards theoretically understanding the benefits of SimCLR pre-training.
>
> ---
>
> Reference:
>
> [1] Yuan Cao, Zixiang Chen, Misha Belkin, and Quanquan Gu. Benign overfitting in two-layer convolutional neural networks. NeurIPS 35:25237–25250, 2022.
>
> [2] Spencer Frei, Niladri S. Chatterji, and Peter L. Bartlett. Benign overfitting without linearity: Neural network classifiers trained by gradient descent for noisy linear data. COLT, 2022.
>
> [3] Kaiyue Wen, Zhiyuan Li, and Tengyu Ma. Sharpness minimization algorithms do not only minimize sharpness to achieve better generalization. NeurIPS, 2023.
>
> [4] Yiwen Kou, Zixiang Chen, Yuan Cao, and Quanquan Gu. How does semi-supervised learning with pseudo-labelers work? a case study. ICLR, 2023.
>
> ---

---

> > ### Comment · Reviewer_HiBF · 2024-11-29
> >
> > Several other reviewers seem to share concerns about the simplified two-layer CNN setting. How do you plan to incorporate the discussion of ViT into the current paper? This appears to be a substantial change that would significantly alter the paper's current scope and focus.

---

> > > ### Author Response · Authors · 2024-11-29
> > >
> > > ---
> > >
> > > Thank you for your question. Through our discussions, we realized that we should include more details about the insights gained from our theoretical study. Therefore, we will certainly add more explanations about these insights. Following such explanations, it would be natural to discuss how these insights could help in understanding SimCLR when training other models, especially ViTs because of the partly comparable model structure. We believe that such discussions will not significantly alter the paper's current scope and focus. Instead, we think they will further strengthen our paper by more clearly demonstrating the value of our results.
> > >
> > > ---

---

### Official Review · Reviewer_HiBF · 2024-10-29

**Soundness:** 3
**Presentation:** 2
**Contribution:** 2
**Rating:** 5
**Confidence:** 3

**Summary:**

In this paper, the authors present a theoretical analysis to establish the benefits of SimCLR pretraining for downstream finetuning of classification tasks. Their analysis is based on several assumptions, including isotropic Gaussian distributions of the representations, the signal-to-noise ratio (SNR) of the training and testing data as well as number of labeled data. Focusing on a two-layer convolutional neural network (CNN) setting, the authors provide detailed proofs and explanations to support their main argument: SimCLR pretraining provably guarantees convergence of the training loss and enhances test-time robustness.

**Strengths:**

The paper is clearly and coherently written, allowing readers to easily follow the presented ideas. Despite being based on several simplified assumptions, the analysis and proofs provided are detailed and reasonable, making the work theoretically sound in the setting of a two-layer CNN.

**Weaknesses:**

There are several weaknesses in the paper that require clarification from the authors:

1) The paper aims to understand how SSL methods like SimCLR benefit downstream fine-tuning. However, the literature review provided is insufficient to comprehensively cover this topic, which has been widely studied both theoretically and empirically. A more thorough literature review on SSL's impact on fine-tuning performance is necessary.
2) The problem setting of 2-layer CNN presented appears overly simplified, potentially deviating from the practical deep learning problems that commonly employ SSL methods such as SimCLR. While deriving theoretical guarantees for modern architectures like transformers is challenging, a more realistic framing of the problem would better justify its relevance and importance.
3) The experiments in the appendix seem questionable in terms of their validity and relevance to the stated problem. Typically, SimCLR pretraining leverages large-scale unlabeled datasets. However, the experiments employ only 250 and 200 unlabeled data points for the synthetic data experiment and real data experiment, respectively. These small dataset sizes are atypical for SimCLR pretraining and may not represent the true behaviors of SimCLR pretraining combined with downstream fine-tuning tasks.

If the authors provide large-scale experiment results for the real data experiment and explain how the theorems can be applied to more modern backbone architectures, I would be willing to raise the score.

**Questions:**

1. How would the theorem transfer to more modern backbone architectures such as ViT? Can you provide more experiment ablation on more architectures?
2. Can you provide more ablation on the SNR rate and the number of the unlabeld data / labeled data?
3. What happens if the representation does not follow isotropic Gaussian distribution? Typically SimCLR type of loss results in non-isotropic representations, in that case, does the theorem still hold just by simply replacing the variance term in the SNR equation with the maximum variance across different embedding dimensions?

---

> ### Author Response · Authors · 2024-11-23
> **Response to Reviewer HiBF (Part a)**
>
> Thank you for your detailed comments and suggestions. We address your concerns as follows.
>
> ---
>
> **Weaknesses**:
>
> **W1**: The paper aims to understand how SSL methods like SimCLR benefit downstream fine-tuning. However, the literature review provided is insufficient to comprehensively cover this topic, which has been widely studied both theoretically and empirically. A more thorough literature review on SSL's impact on fine-tuning performance is necessary.
>
> **AW1**: Thanks for your comment. We have revised the paper to have a more thorough literature review about SSL's influence on fine-tuning performance. Please feel free to point out if you find any important papers missing here.
>
> ---
>
> **W2**: The problem setting of 2-layer CNN presented appears overly simplified, potentially deviating from the practical deep learning problems that commonly employ SSL methods such as SimCLR. While deriving theoretical guarantees for modern architectures like transformers is challenging, a more realistic framing of the problem would better justify its relevance and importance.
>
> **AW2**:  To the best of our knowledge, the influence of contrastive learning on supervised fine-tuning has not been well studied in theory even for the simple two-layer CNN model. As a theoretical paper, it inevitably requires specific simplification of models and assumptions to have an accurate and clear analysis of the advantage of SimCLR pre-training on the fine-tuning of two-layer CNNs. This model allows a direct comparison with the results of direct supervised learning on two-layer CNNs in [1]. We believe that our current analysis is a clear first step towards the theoretical understanding of the advantages of contrastive learning.
>
> ---
>
> **W3**: The experiments in the appendix seem questionable in terms of their validity and relevance to the stated problem. Typically, SimCLR pretraining leverages large-scale unlabeled datasets. However, the experiments employ only 250 and 200 unlabeled data points for the synthetic data experiment and real data experiment, respectively. These small dataset sizes are atypical for SimCLR pretraining and may not represent the true behaviors of SimCLR pretraining combined with downstream fine-tuning tasks.
>
> **AW3**: In the revised version of the paper, we have added experiments with large-scale unlabeled datasets for both synthetic-data and real-world data (MNIST) experiments in Appendix A. The synthetic-data experiment with large-scale unlabeled data ($n_0=100000$) is presented in Figure 5. The real-world data experiment with large-scale unlabeled data ($n_0=13800$) is presented in Figure 6. In these large-scale unlabeled datasets experiments, we could still observe similar results as before. Compared with direct supervised learning, SimCLR pre-training followed by supervised fine-tuning may require far less labeled data to achieve a high test accuracy.
>
> ---
>
> **Questions**:
>
> **Q1**: How would the theorem transfer to more modern backbone architectures such as ViT? Can you provide more experiment ablation on more architectures?
>
> **AQ1**: Our current analysis focuses on relatively simple two-layer CNNs, because SimCLR has not been well understood in theory even in training such simple models.  Therefore, it is natural to first consider our current setting, which is a relatively cleaner and more standard setting for theoretical studies that has been considered in a series of previous works [2,3,4].
>
> For more complicated architecture, currently there is a lack of tools and techniques to enable such a performance comparison with directly supervised learning in more modern models. Analysis of the influence of contrastive learning on supervised learning of more complicated models is an important future direction.
>
> In terms of experiments on more complicated architectures, we would like to point out that architectures like ViTs are orthogonal to our study. We believe the significance of theoretical analyses on CNNs should not be denied because the theory does not cover ViT.
>
> Moreover, we would also like to clarify that, unlike two-layer models where we can use the inner products between network weights and “signal” or “noise” vectors to characterize the level of “signal learning” or “noise memorization”, for more complicated models there may not be any convincing metric on the the level of “signal learning” or “noise memorization” by the model. Therefore, even if we add experiments on ViTs or other models, the observations would simply be like “the model makes worse predictions on more noisy data, and makes better predictions on less noisy data”. We believe such results may not be sufficiently meaningful.
>
> ---

---

> > ### Comment · Reviewer_HiBF · 2024-11-23
> >
> > Thank you for the detailed explanation. Most of my questions have been addressed, but I would still appreciate more in-depth insights on the ViT backbones. In particular, the use of a two-layer CNN feels less relevant in the current era, given the availability of advanced computational resources. For now, I have raised my score to just below the acceptance threshold. However, I would be open to increasing it if this point is thoroughly and convincingly justified.

---

> > > ### Author Response · Authors · 2024-11-29
> > >
> > > ---
> > >
> > > Thanks for your detailed feedback and for raising the score.
> > >
> > > In our latest revision, we have added experiments on ViTs. Specifically, we consider training a simple ViT whose input image patches are motivated by Definition 3.1 in our paper: one of the image patches is a 784-dimensional normalized MNIST image, while the other patch is a 784-dimensional Gaussian noise vector. The results are shown in Figure 10 in Appendix A in the latest revised version of the paper.
> > >
> > > The experiment results show that SimCLR pre-training combined with supervised fine-tuning achieves much higher test accuracy than direct supervised learning on the simple ViT. It demonstrates that our theoretical analysis in this paper on two-layer CNNs can also provide insights into how SimCLR enhances the learning of other models such as Vision Transformers.
> > >
> > > ---

---

> > > > ### Comment · Reviewer_HiBF · 2024-11-29
> > > >
> > > > Thank you for your response. However, the empirical experiments mentioned by the authors are widely recognized as common knowledge. I am seeking deeper theoretical insights instead.

---

> > > > > ### Author Response · Authors · 2024-11-29
> > > > >
> > > > > ---
> > > > >
> > > > > Thanks for your quick response. We would like to clarify that, what is important in our experiment result is the *drastic* difference in the performance of “SimCLR pre-training + supervised fine-tuning” and direct supervised learning.  Our experiments show that direct supervised learning can only achieve an accuracy slightly above $50\\%$, while “SimCLR pre-training + supervised fine-tuning” can achieve above $90\\%$ test accuracy. Please note that we are considering binary classification (zeros vs ones in MNIST), where random guess can give a $50\\%$ test accuracy. Therefore, in our experiment setting, direct supervised learning is barely better than random guess (demonstrating that the learning task we construct is highly noisy), while  “SimCLR pre-training + supervised fine-tuning” achieves pretty high accuracy.
> > > > >
> > > > > Our theory demonstrates that when learning two-layer CNNs, SimCLR may be drastically better than direct supervised learning when input images consist of “signal patches” and “noise patches”. And the drastic performance difference observed in our ViT experiments demonstrates that our conclusion is also true in training more complicated models such as ViTs. We believe that this phenomenon that "the performances of ViTs with/without SimCLR pre-training are drastically different in the case where images contain 'signal patches' and 'noise patches'" is not necessarily a common knowledge.
> > > > >
> > > > > In terms of deeper theoretical insights, we would like to quote some of our earlier responses to Reviewer FqYo and add some additional explanations below.
> > > > >
> > > > > In our theoretical analysis, we are following a signal learning framework. We believe that the insight behind this signal learning framework can help us understand many other learning tasks, including learning ViTs.
> > > > >
> > > > > The idea of the signal learning framework is that, when training complicated neural networks on a finite training dataset, there may exist many (probably infinitely many) different parameter configurations that can all lead to $100\\%$ training accuracy. However, neural networks with different parameter configurations may perform drastically differently on test data. We can generally treat the “information in the data input that are related to the label” (characterized by the vector $\boldsymbol{\mu}$ in our theoretical study) as the “signal” in the data, and treat the “information in the data input that are not related to the label” (characterized by the Gaussian noise vector $\boldsymbol{\xi}$ in our theoretical study) as “noises”. Then clearly,  The signal is the label-dependent and generalizable part of the information, and noise is the label-independent and ungeneralizable part of the information, and whether the obtained neural network after training can achieve good prediction depends on whether it utilizes the “signal part” or the “noise part” to fit the training data, and this is exactly how we build our theory.
> > > > >
> > > > >
> > > > > Following this idea, a natural question is:
> > > > >
> > > > > *How can SimCLR pre-training help encourage the model to learn more **label-relevant information**, if the pre-training process only uses **unlabeled data**?*
> > > > >
> > > > > Our analysis on SimCLR answers the above question by demonstrating that data augmentation is the key. Although SimCLR does not utilize any labeled data, the knowledge that “data augmentation should not change the label” suffices to help the neural network capture more signals in the pre-training stage. Our new experiments on ViTs show that this characterization is true for ViTs as well. Although the level of “signal learning” and “noise memorization” by a ViT is difficult to quantify, the drastic performance difference between  “SimCLR pre-training + supervised fine-tuning” and direct supervised learning observed in our ViT experiments justify our claim to a certain extent.
> > > > >
> > > > > We will make clarifications on the above points in the camera ready version of the paper. We hope that our response above answers your qeustion. Please let us know if you have any further comments or questions. Thank you.
> > > > >
> > > > > ---

---

> > > > > > ### Comment · Reviewer_HiBF · 2024-11-29
> > > > > >
> > > > > > Thank you very much for your response. I have also reviewed the responses provided to the other reviewers. While I genuinely appreciate efforts toward advancing theoretical development in this field—which I recognize is highly challenging given the complexity of modern architectures—I find it difficult to see how this setting is relevant to SimCLR or SSL pretraining in the current era. Therefore, I will maintain my current rating.

---

> > > > > > > ### Author Response · Authors · 2024-11-30
> > > > > > >
> > > > > > > ---
> > > > > > >
> > > > > > > Thank you for acknowledging our contribution to ​​advancing theoretical development.
> > > > > > >
> > > > > > > Regarding your comment that “I find it difficult to see how this setting is relevant to SimCLR or SSL pretraining in the current era”, please note that the same reason can be used to reject a lot of existing theoretical studies, including a significant portion of the published works we have cited in our discussion. We sincerely hope that you could re-evaluate our paper from a more objective perspective.
> > > > > > >
> > > > > > > Although currently, ViTs and other deep learning models are dominating in practical applications, self-supervised learning is a general learning paradigm that is arguably worth studying even without the context of deep learning. For example, recent work [1] studies feature learning of self-supervised learning methods on linear models. In our work, we make an effort to study a setting that is more related to the practice, establishing theoretical guarantees on the convergence and test accuracy of “SimCLR-pre-training + supervised fine-tuning” for two-layer CNNs. However, we feel that our effort is not appreciated. Instead, because we manage to make the setting more related to the practice, we are blamed for not being able to make it even more practical. We believe this is not fair.
> > > > > > >
> > > > > > > ---
> > > > > > >
> > > > > > > [1] Wenlong Ji, Zhun Deng, Ryumei Nakada, James Zou, and Linjun Zhang. "The power of contrast for feature learning: A theoretical analysis." Journal of Machine Learning Research 24, no. 330 (2023): 1-78.

---

> > > > > > > > ### Comment · Reviewer_HiBF · 2024-12-01
> > > > > > > >
> > > > > > > > Could you please explain how you "make the setting more related to the practice" in more detail?

---

> > > > > > > > > ### Author Response · Authors · 2024-12-01
> > > > > > > > >
> > > > > > > > > ---
> > > > > > > > >
> > > > > > > > > Thank you for your reply.
> > > > > > > > >
> > > > > > > > > As we mentioned, recent work [1] studies feature learning of self-supervised learning methods on **linear models**. In comparison, our work manages to analyze **two-layer CNNs**. This is why we commented that we “make the setting more related to the practice”.
> > > > > > > > >
> > > > > > > > > In more detail, please note that [1] addresses the problem of training linear models by minimizing a “linear contrastive loss” (see equation (3) and Definition 2 in [1]). In this framework, the minimizer of the contrastive loss can be explicitly solved (see Proposition 5 in [1]), allowing [1] to directly focus on such a solution rather than on any specific training algorithm.
> > > > > > > > >
> > > > > > > > > In contrast, our study examines the training of two-layer CNNs using a more practical loss function definition (incorporating log and exp functions) as employed in SinCLR. In our scenario, there is no explicit solution for the minimizer of the loss function. Consequently, we must analyze the training process using gradient descent, which poses a significant challenge due to the highly non-convex nature of the training objective.
> > > > > > > > >
> > > > > > > > > Furthermore, in terms of downstream tasks, [1] investigates the training of linear models, whereas we explore the fine-tuning of two-layer CNNs. Despite the highly non-convex landscape of the loss function in our setting, we successfully establish convergence and prediction guarantees, highlighting the strength of our study.
> > > > > > > > >
> > > > > > > > > We hope that our discussion above answers your question and convinces you that our setting is “more related to the practice” compared with the setting in [1] (published in 2023). We would also like to emphasize that despite the comparisons we made above, we consider [1] to be a very strong theory paper, because we believe
> > > > > > > > >
> > > > > > > > > *As a general learning paradigm, self-supervised learning is worth studying even without the context of ViTs and other deep learning models.*
> > > > > > > > >
> > > > > > > > > However, we sincerely hope that our efforts in studying two-layer CNNs will be recognized and treated as a strength, rather than a weakness, of our work. Thank you.
> > > > > > > > >
> > > > > > > > > ---
> > > > > > > > >
> > > > > > > > >
> > > > > > > > > [1] Wenlong Ji, Zhun Deng, Ryumei Nakada, James Zou, and Linjun Zhang. "The power of contrast for feature learning: A theoretical analysis." Journal of Machine Learning Research 24, no. 330 (2023): 1-78.

---

> > > > > > > > > > ### Comment · Reviewer_HiBF · 2024-12-02
> > > > > > > > > >
> > > > > > > > > > I will maintain my current score.

---

> ### Author Response · Authors · 2024-11-23
> **Response to Reviewer HiBF (Part b)**
>
> ---
>
> **Q2**: Can you provide more ablation on the SNR rate and the number of the unlabeld data / labeled data?
>
> **AQ2**: Thanks for your suggestions. We have added several groups of experiments with different signal-to-noise ratio ($\mathrm{SNR}$), the size of unlabeled data $n_0$ and size of labeled data $n$ in the Appendix A. The key takeaways from these experiments are summarized as follows.
>
> - The unlabeled data size $n_0$ in Figure 7: While all other conditions remain the same, larger size of unlabeled pre-training data $n_0$ achieves a better performance in the fine-tuning.
>
> - Signal-to-noise ratio ($\mathrm{SNR}$) in Figure 8: For experiments with smaller $\mathrm{SNR}$, it requires more (labeled or unlabeled) data to achieve a good test performance.
>
> - Labeled data size $n$ in Figure 9: In the SimCLR pre-training followed by supervised fine-tuning, given a satisfactory number of unlabeled data $n_0$, the condition on the size of labeled data $n$ to achieve a high test accuracy is mild.
>
> ---
>
> **Q3**: What happens if the representation does not follow isotropic Gaussian distribution? Typically SimCLR type of loss results in non-isotropic representations, in that case, does the theorem still hold just by simply replacing the variance term in the SNR equation with the maximum variance across different embedding dimensions?
>
> **AQ3**: It is possible to generalize our data model to the data distribution with different scales of Gaussian noise across different dimensions. In this setting, the signal-to-noise rate (SNR) will be defined as $\mathrm{SNR}=\||\boldsymbol{\mu}\||/\sqrt{\max_{k\in[d]}\mathbb{E}[\||\xi^{(k)}\||^2]\cdot d}$, where $\xi^{(k)}$ is the k-th dim of $\boldsymbol{\xi}$. The difference between this and our current definition of SNR is that the norm of noise is replaced with the maximum of the norm of noise entry.
>
> In addition, we would like to point out that, even in our current setting, the representations learned by SimCLR are non-isotropic representations (see the results of Theorem 5.3). Even if the data are isotropic, the first hidden layer outputs of the CNN trained by SimCLR are not isotropic.
>
> ---
>
> Reference:
>
> [1] Yuan Cao, Zixiang Chen, Misha Belkin, and Quanquan Gu. Benign overfitting in two-layer convolutional neural networks. NeurIPS 35:25237–25250, 2022.
>
> [2] Spencer Frei, Niladri S. Chatterji, and Peter L. Bartlett. Benign overfitting without linearity: Neural network classifiers trained by gradient descent for noisy linear data. COLT, 2022.
>
> [3] Kaiyue Wen, Zhiyuan Li, and Tengyu Ma. Sharpness minimization algorithms do not only minimize sharpness to achieve better generalization. NeurIPS, 2023.
>
> [4] Yiwen Kou, Zixiang Chen, Yuan Cao, and Quanquan Gu. How does semi-supervised learning with pseudo-labelers work? a case study. ICLR, 2023.

---

### Official Review · Reviewer_S5xt · 2024-11-02

**Soundness:** 3
**Presentation:** 2
**Contribution:** 3
**Rating:** 5
**Confidence:** 2

**Summary:**

The paper extends a prior analysis of Cao et al. (2022) (regarding supervised learning) with a particular focus of SimCLR self-supervised learning. Specifically, it revisits a toy binary classification problem designed by Cao et al. (2022), and consider a SimCLR training using a bespoke 2-layer convolutional neural network. As the result, the paper theoretically show that fine-tuning a SimCLR pre-trained two-layer convolutional neural network requires significantly less label complexity compared to the supervised counterpart (of Cao et al. (2022)) to achieve low training/test losses, in a sense that the complexity is independent to the signal-to-noise ratio of the data at fine-tuning stage provided that it is sufficiently accounted at the pre-training stage. Although the main body of the paper is focused on theoretical results, some supporting experimental results are also provided, e.g., on synthetic data and MNIST, in Appendix.

Cao et al., Benign overfitting in two-layer convolutional neural networks, NeurIPS 2022.

**Strengths:**

- The paper is well-motivated and clearly written.
- The paper presents solid theoretical results, which indeed provide insight about SimCLR.
- The area of study, i.e., theoretical understanding of recent contrastive learning methods, is still a timely topic in my opinion.
- Although not present in the main text, the paper also provides empirical supports as well.

**Weaknesses:**

- To my knowledge, there have been several works that study contrastive learning in theoretical aspects, although I could not find a discussion about them from the paper: e.g., [Bansal et al., 2021; HaoChen et al., 2021; Tan et al., 2024]. I think the paper should incorporate any discussions about such a line of research. For example, I am curious whether the paper’s observation about SimCLR as a matrix power method can be related to the spectral view of contrastive learning [HaoChen et al., 2021].
- I generally feel a lack of discussion about how the theory presented in this paper can be related to the real-world use of SimCLR. I think this is because the paper omits enough context about how can we really ensure that the simplifying assumptions made here are mild enough, so that it can be eventually connected to the real-world.
- The main body of the paper is quite theory-biased, and allocates its significant portion with the proof sketch. This may narrow down the potential audience, and I personally think the authors might consider to give more highlight on their empirical results as well as the theoretical ones.

Bansal et al., For self-supervised learning, Rationality implies generalization, provably, ICLR 2021.

HaoChen et al., Provable Guarantees for Self-Supervised Deep Learning with Spectral Contrastive Loss, NeurIPS 2021.

Tan et al., Contrastive Learning Is Spectral Clustering On Similarity Graph, ICLR 2024.

**Questions:**

- Is there any potential that the analysis presented in this paper can be generalized to more complex architecture?
- The overall setup assumes that the labeled data at fine-tuning stage is sampled from the same distribution where the pre-training data is sampled, considering a pure unsupervised learning scenario. But one important aspect of SimCLR like training is in its transferrability; as such, could the analysis can be generalized in any way if we relax the assumption of the labeled data to another distribution? For example, I am curious if the analysis could be extended when the labeled data is more noisy that the pre-training data.
- In my understanding, Theorem 4.3 implies that one still require the unlabeled sample size for SimCLR, $n_0$, at the similar rate of those for supervised learning to optimize the loss. But could this result be due to that the data considered is too easy to discriminate, even without labels, and perhaps the complexity of SimCLR can be much worse under a more challenging data assumption?

---

> ### Author Response · Authors · 2024-11-23
> **Response to Reviewer S5xt (Part a)**
>
> Thanks for your detailed comments and suggestions. We provide our responses to your questions as follows.
>
> ---
>
> **Weaknesses**:
>
> **W1**: To my knowledge, there have been several works that study contrastive learning in theoretical aspects, although I could not find a discussion about them from the paper: e.g., [Bansal et al., 2021; HaoChen et al., 2021; Tan et al., 2024]. I think the paper should incorporate any discussions about such a line of research. For example, I am curious whether the paper’s observation about SimCLR as a matrix power method can be related to the spectral view of contrastive learning [HaoChen et al., 2021].
>
> **AW1**: Thanks for pointing out this. We have revised the paper and added discussions about them. You can find the discussion in Section 2 (related work) in the revised paper. The following is the comparison between our paper and these papers.
>
> [1] presented a new upper bound of the generalization gap of classifiers by first performing self-supervised training to learn representations, followed by fitting a simple classifier such as linear classifier to the labels. Although both [1] and our paper focus on the theoretical analysis of self-supervised learning, but the assumptions and the problem settings are different. [1] assumed the training satisfies certain natural noise-robustness and rationality, while our paper poses assumptions to the training setting and the signal-to-noise ratio (SNR). In the fine-tuning stage, [1] adopted a simple classifier such as linear classifier, while more complex two-layer CNN architecture is adopted in our paper.
>
> [2] considered a spectral contrastive loss and performed spectral clustering on the population augmentation graph. We believe the setting in [2] is significantly different from our setting, and the spectral view discussed in [2] is not directly comparable to our study.
>
> [3] extended the idea proposed by [2] to general loss functions by showing the equivalence between InfoNCE loss and spectral clustering. Furthermore, [3] also extended this to more general settings, including multi-modal scenarios. But as mentioned above, the idea of spectral clustering on population augmentation graphs in [2,3] is still different from our idea of signal learning analysis for contrastive learning.
>
> In addition, we would like to point out that while [2,3] only focus on the analysis of contrastive learning (corresponding to the pre-training stage in our paper), our paper studies how contrastive learning influences the performance of the following fine-tuning stage.

---

> ### Author Response · Authors · 2024-11-23
> **Response to Reviewer S5xt (Part b)**
>
> ---
>
> **W2**: I generally feel a lack of discussion about how the theory presented in this paper can be related to the real-world use of SimCLR. I think this is because the paper omits enough context about how can we really ensure that the simplifying assumptions made here are mild enough, so that it can be eventually connected to the real-world.
>
> **AW2**: First of all, we would like to mention that the accurate theoretical analysis of the training process should be studied in some specific assumptions. Our assumptions are not stronger than the existing theoretical works such as [4]. We adopt the same data model to allow a direct comparison of the performance of SimCLR pre-training followed by supervised learning with that of directly supervised learning in [4].
>
> Regarding the real-world use, in our theoretical analysis, we are following a signal learning framework. We believe that the insight behind this signal learning framework can help us understand many practical learning tasks as well.
>
> The idea of the signal learning framework is that, when training complicated neural networks on a finite training dataset, there may exist many (probably infinitely many) different parameter configurations that can all lead to $100\\%$ training accuracy. However, neural networks with different parameter configurations may perform drastically differently on test data. We can generally treat the “information in the data input that are related to the label” (characterized by the vector $\boldsymbol{\mu}$ in our theoretical study) as the “signal” in the data, and treat the “information in the data input that are not related to the label” (characterized by the Gaussian noise vector $\boldsymbol{\xi}$ in our theoretical study) as “noises”. Then clearly,  The signal is the label-dependent and generalizable part of the information, and noise is the label-independent and ungeneralizable part of the information, and whether the obtained neural network after training can achieve good prediction depends on whether it utilizes the “signal part” or the “noise part” to fit the training data, and this is exactly how we build our theory.
>
> Our analysis on SimCLR demonstrates that, although SimCLR does not utilize any labeled data, the knowledge that “data augmentation should not change the label” suffices to help the neural network to capture more signals in the pre-training stage. We believe this characterization is true for more practical scenarios as well. The idea it conveyed that SimCLR pre-training encourages signal learning in the supervised learning of CNNs also holds for more general cases. Although it is difficult to verify on real data because “signals” and “noises” and the level they are captured by a more complicated network is hard to quantify, we have provided additional real-data experiments to justify our claim (to a certain extent) in the revision. The added experiments can be found in Appendix A in the revised paper.
>
> ---
>
> **W3**: The main body of the paper is quite theory-biased, and allocates its significant portion with the proof sketch. This may narrow down the potential audience, and I personally think the authors might consider to give more highlight on their empirical results as well as the theoretical ones.
>
> **AW3**: We would like to point out that this paper is exactly a theory paper. It aims to theoretically prove the advantage of contrastive learning on fine-tuning by providing a detailed overall theoretical analysis of the SimCLR pre-training followed by supervised fine-tuning.
>
> Empirical experiments on both synthetic and real-world datasets are conducted to verify our theoretical results. We have included the key takeaway of the experiments in Remark 4.4 in the main body of the paper. Due to the page limit, we include more experiment details in Appendix A. In the revised version of the paper, we have added more experiments for both synthetic-data and real-world dataset in Appendix A.
>
> ---
>
> **Questions**:
>
> **Q1**: Is there any potential that the analysis presented in this paper can be generalized to more complex architecture?
>
> **AQ1**: First of all, we would like to point out that, to the best of our knowledge, the influence of contrastive learning on supervised fine-tuning has not been analyzed in theory before even for this simple two-layer CNN architecture. We believe that our current analysis provides a clear and meaningful first step towards a better theoretical understanding of the advantages of contrastive learning.
>
> We would also like to explain that we choose to study two-layer CNNs because it allows a direct comparison with the results of direct supervised learning on two-layer CNNs in [4]. For more complicated architectures, even direct supervised learning has not been well studied in theory. We believe studying the performance of SimCLR (as well as direct supervised learning) on more complicated neural network architectures is an important future work direction.

---

> ### Author Response · Authors · 2024-11-23
> **Response to Reviewer S5xt (Part c)**
>
> ---
>
> **Q2**: The overall setup assumes that the labeled data at fine-tuning stage is sampled from the same distribution where the pre-training data is sampled, considering a pure unsupervised learning scenario. But one important aspect of SimCLR like training is in its transferrability; as such, could the analysis can be generalized in any way if we relax the assumption of the labeled data to another distribution? For example, I am curious if the analysis could be extended when the labeled data is more noisy that the pre-training data.
>
> **AQ2**: As you have mentioned, the setting where SimCLR pre-training and supervised fine-tuning use data from different distributions can be understood as combining our current setting with transfer learning. We do believe that considering the influence of pre-training on different downstream tasks is a good future research direction. However, we would like to point out that even the ‘pure’ contrastive learning setting studied in our paper has not been well understood in theory, and therefore, it is more natural to consider our current setting, which is arguably cleaner for theoretical studies.
>
> ---
>
> **Q3**: In my understanding, Theorem 4.3 implies that one still require the unlabeled sample size for SimCLR, $n_0$, at the similar rate of those for supervised learning to optimize the loss. But could this result be due to that the data considered is too easy to discriminate, even without labels, and perhaps the complexity of SimCLR can be much worse under a more challenging data assumption?
>
> **AQ3**:  We believe this is a misunderstanding. The condition on $n_0$ in our paper is $n_0\cdot \mathrm{SNR}^2 = \tilde\Omega(1)$ (in Condition 4.1), it is not the same as the condition $n\cdot\mathrm{SNR}^{q}=\tilde\Omega(1)$ (for $q>2$) (Condition 4.2 in [4]) for the number of labeled data $n$ in direct supervised learning in [4].
>
> We would also like to emphasize that the data model is the standard data adopted by a series of theoretical works [4,5,6]. We adopt the same data model to allow a direct comparison of the performance of SimCLR pre-training followed by supervised learning with that of directly supervised learning in [4].
>
> ---
>
> Reference：
>
> [1] Yamini Bansal, Gal Kaplun, and Boaz Barak. For self-supervised learning, Rationality implies generalization, provably, ICLR 2021.
>
> [2] Jeff Z. HaoChen, Colin Wei, Adrien Gaidon, and Tengyu Ma. Provable Guarantees for Self-Supervised Deep Learning with Spectral Contrastive Loss, NeurIPS 2021.
>
> [3] Zhiquan Tan, Yifan Zhang, Jingqin Yang, and Yang Yuan. Contrastive Learning Is Spectral Clustering On Similarity Graph, ICLR 2024.
>
> [4] Yuan Cao, Zixiang Chen, Misha Belkin, and Quanquan Gu. Benign overfitting in two-layer convolutional neural networks. NeurIPS 35:25237–25250, 2022.
>
> [5] Samy Jelassi and Yuanzhi Li. Towards understanding how momentum improves generalization in deep learning. In ICML, pages 9965–10040. PMLR,2022.
>
> [6] Yiwen Kou, Zixiang Chen, Yuanzhou Chen, and Quanquan Gu. Benign overfitting in two-layer relu convolutional neural networks. In ICML, pages 17615–17659. PMLR, 2023.

---

> ### Comment · Reviewer_S5xt · 2024-11-26
> **Official Comment by Reviewer S5xt**
>
> Thank you for the response. I appreciate that the authors have incorporated a discussion about related works.
>
> - The authors mentioned that Bansal et al. (2021) have also focused on the theoretical analysis of self-supervised learning. But if so, I think there should be a more in-depth discussions about the comparison of the conditions and how mild the current setup is and why it is significant. I don't think the difference between the use of linear vs. 2-layer CNN is a major point.
>
> - Also, if there is work such as Bansal et al. (2021), I don't think the authors' claim that the work is the first analysis on SimCLR  holds anymore, while the claim was used multiple times in the justifications throughout the response letters. I would also suggest the authors to further check if there is other theory papers that analyze contrastive learning beyond the list the reviewers' have suggested; I believe they will be just a partial subset of the full literature and I think these work should be adequately discussed in the manuscript.
>
> - Although I agree that having simplifying conditions is important in theory works, but I am not fully convinced that the current assumptions are mild enough to produce practical significance in SimCLR analysis; even while the assumptions may be useful for supervised setups.
>
> Overall, I found the response addresses some of my concerns; but I still feel lack of practical significance due to the unrealistic assumption of analysis, e.g., compared to Bansal et al. (2021). In this respect, I am keeping my original score of 5 this time.

---

> ### Author Response · Authors · 2024-11-29
>
> ---
> Thanks for your detailed response and the follow-up questions. Here we provide our answers to these questions.
>
> > **Q1**: The authors mentioned that Bansal et al. (2021) have also focused on the theoretical analysis of self-supervised learning. But if so, I think there should be a more in-depth discussions about the comparison of the conditions and how mild the current setup is and why it is significant. I don't think the difference between the use of linear vs. 2-layer CNN is a major point.
>
> **A1**: We would like to reemphasize that “[1] studies fine-tuning a simple (linear) model, while we consider fine-tuning a complicated two-layer CNN” is indeed a major and important difference between [1] and our work. In particular, the results in [1] are only meaningful when fine-tuning gives an “underfitting” model (training loss is strictly bounded away from zero). On the other hand, our work focuses on a setting where fine-tuning can give an “overfitting” model (training loss can be arbitrarily close to zero).
>
> Please note that, in the literature of generalization theory, “underfitting” models and “overfitting” models require completely different techniques to study. The type of result in [1] is the more classic approach focusing on bounding the “generalization gap” by characterizing the “complexity” of the classifier. However, in deep learning, people frequently find that such classic complexity-based generalization bounds cannot provide meaningful results for complicated neural networks - the upper bound of test accuracy given by these techniques are usually larger than $100\\%$, making the result “vacuous" (this is also why [1] has to focus on fine-tuning simple linear classifiers). This motivates a lot of works to develop new methods for generalization bounds of neural networks, and [2], which studies “benign overfitting” of two-layer CNNs, is a representative work along this new line of research, where the key to obtaining tighter generalization bounds is to accurately analyze the model training process. Our work, aiming to establish comparable results as [2] for SimCLR, also follows this more advanced line of research.
>
> For example, to see the difference between [1] and our work, please note that the generalization bound in [1] has the form $\sqrt{ C / n }$, where $C$ is the “complexity” of the simple classifier produced in the “simple fit” stage. When applied to the two-layer CNN considered in our work, this bound would be $\sqrt{ d / n }$. However, according to our assumptions, we have $d \gg n$, and this generalization bound $\sqrt{ d / n }$ is much larger than one, making the conclusion meaningless. In comparison, our work can give a tight generalization bound under the very mild assumption that $n = \tilde{\Omega}(1)$ (meaning that $n$ only needs to be poly-logarithmically large). This demonstrates the drastic difference between [1] and our work. However, we would like to clarify that such a drastic difference does not mean that [1] gives bad bounds, it is simply due to the fact that [1] and our work focuses on completely different settings, and trying to apply the result in [1] to our setting cannot give a fair comparison.
>
> Below, we list some other differences between our work and [1]:
>
> - [1] only studies the generalization gap. On the other hand, in our work, we do not only establish generalization bounds, but also demonstrate that “SimCLR pre-training + supervised fine-tuning” converges and the training loss function can be optimized arbitrarily close to zero.
>
> - [1] does not need to specify any particular model architecture, but instead, relies on assumptions that two terms appearing in the generalization bound, the “robustness gap” term and the “rationality gap”, are small, which are assumptions that are not easy to theoretically verify. In comparison, our work is of a different flavor – we focus on a specific CNN architecture, and our bounds are more concrete without terms like the “robustness gap” or “rationality gap”.
>
> Overall, we still believe that [1] and our work consider significantly different settings and our results are not directly comparable. We would also like to argue that in machine learning theory literature, it is very common that two theoretical works on the same topic are not directly comparable because they make completely different assumptions.
>
> ---

---

> ### Author Response · Authors · 2024-11-29
>
> ---
>
> > **Q2**: Also, if there is work such as Bansal et al. (2021), I don't think the authors' claim that the work is the first analysis on SimCLR holds anymore, while the claim was used multiple times in the justifications throughout the response letters. I would also suggest the authors to further check if there are other theory papers that analyze contrastive learning beyond the list the reviewers' have suggested; I believe they will be just a partial subset of the full literature and I think these works should be adequately discussed in the manuscript.
>
> **A2**: Thanks for your suggestion. What we meant is only that along the line of works that study the “signal learning” of neural networks (such as [2]), our work is the first to analyze SimCLR. We believe that we never make such claims in our paper, and we will make sure to tune down our claim and provide clearer context in our future discussions.
>
> We appreciate your reminder for us to give a more thorough literature review. We have also identified additional related works on theoretical analysis of self-supervised learning:
>
> - [6] introduced a theoretical framework for contrastive learning under a latent classes assumption and obtained the generalization bound for contrastive learning. The established generalization bound is also a complexity-based bound similar to [1].
>
> - [7] considered a certain class of reconstruction-based pretext tasks where reconstruction is achieved by linear regression, and analyzed sample complexity of such methods.
>
> The deadline for revising the manuscript during the discussion period has passed. But we will make sure to  include the references above in our camera-ready version of the paper,
>
> ---

---

> ### Author Response · Authors · 2024-11-29
>
> ---
> > **Q3**: Although I agree that having simplifying conditions is important in theory works, but I am not fully convinced that the current assumptions are mild enough to produce practical significance in SimCLR analysis; even while the assumptions may be useful for supervised setups.
>
> **A3**: We would like to further address your concern regarding the gap between the two-layer CNNs considered in this paper and practical applications as follows.
>
> - In [2], the authors study the direct supervised learning of two-layer CNNs. The goal of our paper is to establish comparable theoretical guarantees for SimCLR pre-training followed by supervised fine-tuning, allowing us to demonstrate how SimCLR enhances learning performance in comparison to [2]. Therefore, we believe that considering the same settings and assumptions as in [2] is a natural and appropriate choice that serves our purpose well.
>
> - Please note that even for direct supervised learning, recent related works [3,4,5] usually have similar assumptions as our paper. Compared to previous works, our research examines the more complicated procedure of “SimCLR pre-training -> remove projection head -> attach classification layer -> supervised fine-tuning”, which is more challenging to study. Therefore, while we agree that there is a big gap between our theory and the practice, we must emphasize that this gap exists in most existing theoretical studies. We sincerely hope that our technical contribution in developing a framework for the theoretical analysis of SimCLR is recognized, as we believe our novel analysis tools may help advance theoretical studies towards more practical scenarios.
>
> Although our paper makes a set of assumptions to enable precise theoretical analysis, we believe the insights obtained from our theoretical study are applicable to more general settings. As demonstrated in the additional experiments on ViTs in Appendix A in the latest revision of the paper, the phenomenon that SimCLR pre-training enhances signal learning during supervised fine-tuning is also observed. We believe our theoretical analysis represents an important step towards theoretically understanding the benefits of SimCLR pre-training.
>
>
> ---
>
> Reference:
>
>
> [1] Yamini Bansal, Gal Kaplun, and Boaz Barak. For self-supervised learning, Rationality implies generalization, provably, ICLR 2021.
>
> [2] Yuan Cao, Zixiang Chen, Misha Belkin, and Quanquan Gu. Benign overfitting in two-layer convolutional neural networks. NeurIPS 35:25237–25250, 2022.
>
> [3] Spencer Frei, Niladri S. Chatterji, and Peter L. Bartlett. Benign overfitting without linearity: Neural network classifiers trained by gradient descent for noisy linear data. COLT, 2022.
>
> [4] Kaiyue Wen, Zhiyuan Li, and Tengyu Ma. Sharpness minimization algorithms do not only minimize sharpness to achieve better generalization. NeurIPS, 2023.
>
> [5] Yiwen Kou, Zixiang Chen, Yuan Cao, and Quanquan Gu. How does semi-supervised learning with pseudo-labelers work? a case study. ICLR, 2023.
>
> [6] Sanjeev Arora, Hrishikesh Khandeparkar, Mikhail Khodak, Orestis Plevrakis, and Nikunj Saunshi. A theoretical analysis of contrastive unsupervised representation learning. ICML 2019.
>
> [7] Jason D. Lee, Qi Lei, Nikunj Saunshi, and Jiacheng Zhuo. Predicting what you already know helps: Provable self-supervised learning, NeurIPS, 2021.

---

### Official Review · Reviewer_uTEP · 2024-11-02

**Soundness:** 3
**Presentation:** 2
**Contribution:** 4
**Rating:** 6
**Confidence:** 4

**Summary:**

The paper presents a theoretical analysis of SimCLR pretraining, focusing on training a two-layer convolutional neural network to learn a signal-noise model, as explored in recent works like Cao et al., 2022. The authors derive conditions under which SimCLR pretraining and finetuning can achieve near-optimal test loss. Their findings indicate that SimCLR pretraining, followed by finetuning, can reach nearly optimal error rates with fewer labeled data than required for supervised learning.

**Strengths:**

- The theoretical results and proof techniques are novel. In particular, the connection between SimCLR and the matrix power method is especially interesting.

**Weaknesses:**

- Unlike other works focused on similar data distributions and network architectures, such as Cao et al., 2022, and Kou et al., 2023, the authors provide only a sufficient condition on the amount of unlabeled data, whereas previous studies offer tight necessary and sufficient conditions. I agree that providing only a sufficient amount of pretraining data effectively demonstrates SimCLR’s advantage over supervised learning. However, I believe that characterizing tight necessary and sufficient conditions would be valuable, as it may facilitate comparisons with other pretraining methods in future work.
- The data augmentation techniques introduced in the problem setting is toounrealistic. While I understand that defining data augmentation in an abstract manner within a signal-noise model is necessary, changing only the noise patch to new i.i.d. noise seems overly simplistic. This approach leads to augmented data from different pretraining samples with the same (unseen) label following the same distribution, which differs from practical scenarios. I believe a more practical definition for data augmentation could be used—such as applying a linear transformation to the noise patch.
- There are limited practical insights provided regarding the theoretical findings (see Question 1).

**Questions:**

- Could you provide some practical insights into the theoretical findings? Specifically, can the main proof technique—interpreting SimCLR as a matrix power method and characterizing top eigenvectors—be translated to real scenarios to offer intuition on why SimCLR is effective?
- Can the same proof technique be extended to ReLU networks and the 0-1 test loss considered in Kou et al.2023?

---

> ### Author Response · Authors · 2024-11-23
> **Response to Reviewer uTEP (Part a)**
>
> Thanks for your constructive review. We address your comments as follows.
>
> ---
>
> **Weaknesses**:
>
> **W1**: Unlike other works focused on similar data distributions and network architectures, such as Cao et al., 2022, and Kou et al., 2023, the authors provide only a sufficient condition on the amount of unlabeled data, whereas previous studies offer tight necessary and sufficient conditions. I agree that providing only a sufficient amount of pretraining data effectively demonstrates SimCLR’s advantage over supervised learning. However, I believe that characterizing tight necessary and sufficient conditions would be valuable, as it may facilitate comparisons with other pretraining methods in future work.
>
> **AW1**: Thanks for raising the question about matching lower bounds. Please note that the condition on $n$ for our upper bound is already very mild. As shown in Condition 4.1 for Theorem 4.2, the condition for the number of the labeled data is $n\geq\Omega(\log(1/\delta))$. Since it is obvious that we need at least one labeled data to perform supervised learning, this condition $n\geq\Omega(\log(1/\delta))$ for successful learning already matches the natural requirement $n\geq 1$ up to a logarithmic factor. In this sense, we believe no specific analysis on lower bounds are necessary in our study.
>
> In addition, we would like to point out that it is also quite common in the existing papers to compare only one side of the bound (only the upper bound or the lower bound) to obtain the conclusion, such as in [1,2].
> [1] compared the lower bound of Adam and with the upper bound of SGD, and demonstrated their convergence to different global solutions with significantly different generalization errors.
> [2] theoretically analyzed the upper bound to show how knowledge distillation improves the performance of neural networks in the learning and compared the result with lower bounds for learning without knowledge distillation.
>
> Therefore, by establishing an upper bound for SimCLR pre-training followed by supervised fine-tuning and comparing it with the lower bound of direct supervised learning, our paper follows the same nature as recent works [1,2].
> Therefore, it may not be necessary to give both the upper and lower bounds in order to obtain the above conclusions.
>
>
> ---
>
> **W2**: The data augmentation techniques introduced in the problem setting is too unrealistic. While I understand that defining data augmentation in an abstract manner within a signal-noise model is necessary, changing only the noise patch to new i.i.d. noise seems overly simplistic. This approach leads to augmented data from different pretraining samples with the same (unseen) label following the same distribution, which differs from practical scenarios. I believe a more practical definition for data augmentation could be used—such as applying a linear transformation to the noise patch.
>
> **AW2**: We agree that we consider an ideal case of data augmentation. As a theoretical paper, it is inevitable that we need to consider a certain idealized setting, and considering such an ideal version of data augmentation helps us to establish precise guarantees of the SimCLR algorithm. Moreover, to the best of our knowledge, the influence of contrastive learning on supervised fine-tuning has not been analyzed before even for this ideal setting. Therefore, we believe our paper can serve as a first step towards more realistic theoretical studies.

---

> ### Author Response · Authors · 2024-11-23
> **Response to Reviewer uTEP (Part b)**
>
> ---
>
> **W3&Q1**:  limited practical insights: Could you provide some practical insights into the theoretical findings? Specifically, can the main proof technique—interpreting SimCLR as a matrix power method and characterizing top eigenvectors—be translated to real scenarios to offer intuition on why SimCLR is effective?
>
> **AW3&AQ1**: Our theoretical analysis provides a signal learning framework. We believe that the insight behind this signal learning framework can help us understand many practical learning tasks as well.
>
> The idea of the signal learning framework is that, when training complicated neural networks on a finite training dataset, there may exist many (probably infinitely many) different parameter configurations that can all lead to $100\\%$ training accuracy. However, neural networks with different parameter configurations may perform drastically differently on test data. We can generally treat the “information in the data input that are related to the label” (characterized by the vector $\boldsymbol{\mu}$ in our theoretical study) as the “signal” in the data, and treat the “information in the data input that are not related to the label” (characterized by the Gaussian noise vector $\boldsymbol{\xi}$ in our theoretical study) as “noises”. Then clearly,  The signal is the label-dependent and generalizable part of the information, and noise is the label-independent and ungeneralizable part of the information, and whether the obtained neural network after training can achieve good prediction depends on whether it utilizes the “signal part” or the “noise part” to fit the training data, and this is exactly how we build our theory.
>
> Our analysis of SimCLR demonstrates that, although SimCLR does not utilize any labeled data, the knowledge that “data augmentation should not change the label” suffices to help the neural network to capture more signals in the pre-training stage. We believe this characterization is true for more practical scenarios as well. Although it is difficult to verify on real data because “signals” and “noises” and the level they are captured by a more complicated network is hard to quantify, we have provided additional real-data experiments to justify our claim (to a certain extent) in the revision. The added experiments can be found in Appendix A in the revised paper.
>
> ---
>
> **Q2**: Can the same proof technique be extended to ReLU networks and the 0-1 test loss considered in Kou et al.2023?
>
> **AQ2**: In this paper, the key focus is the analysis of the SimCLR pre-training stage. The results of the pre-training stage serve as the initialization of the fine-tuning stage, and therefore we could analyze how the pre-training stage influences the performance of the fine-tuning stage.
>
> We believe that it is a clear and concise way to illustrate the advantage of SimCLR pre-training on the fine-tuning stage in our setting by comparing it with direct supervised learning on two-layer CNN in [3].
>
> Analyzing how the pre-training influences the fine-tuning on ReLU networks is also an interesting future direction. Similar sets of analysis could be applied to the fine-tuning on ReLU networks, but due to the different training dynamics, the specific techniques and tools may be different.
>
> ---
>
> Reference:
>
> [1] Difan Zou, Yuan Cao, Yuanzhi Li, and Quanquan Gu. Understanding the Generalization of Adam in Learning Neural Networks with Proper Regularization. ICLR, 2023.
>
> [2] Zeyuan Allen-Zhu, Yuanzhi Li. Towards Understanding Ensemble, Knowledge Distillation and Self-Distillation in Deep Learning. ICLR 2023.
>
> [3] Yuan Cao, Zixiang Chen, Misha Belkin, and Quanquan Gu. Benign overfitting in two-layer convolutional neural networks. NeurIPS 35:25237–25250, 2022.

---

> ### Comment · Reviewer_uTEP · 2024-11-25
>
> Thank you for your response, which addresses most of my concerns. I raised my score to 6.
>
> However, I would like to clarify my first question. Specifically, can your proof technique using the matrix power method be interpreted as providing insight into the underlying mechanism that explains why SimCLR effectively learns meaningful signals in real-world scenarios? For example, recent works on feature learning, such as [1], have used their proof techniques to offer high-level intuition about the mechanisms behind methods like Cutout and CutMix.
>
> [1] Junsoo Oh and Chulhee Yun. Provable Benefit of Cutout and CutMix for Feature Learning. NeurIPS, 2024.

---

> > ### Author Response · Authors · 2024-11-29
> >
> > ---
> >
> > Thanks for your response and for raising the score.
> >
> > Regarding your question about how the matrix power method provides insights in real-world scenarios, we believe it's more accurate to say that our theoretical finding—that “although not using any labeled data, SimCLR pre-training can help the model learn more label-related information”—explains SimCLR's success in practical applications.
> >
> > The matrix power method corresponds to a precise iterative formula (See lemma 5.1 in the revised paper) related to the training of CNN filters, which is derived from the exact problem setting in our theoretical analysis. While we believe our analysis can be extended to other problem settings and model architectures, the learning dynamics may need to be characterized differently for each case.
> >
> > Although matrix power method is specific to our exact problem setting, it does reveal insights that are more generally applicable: we note that the matrix power method is based on the matrix $\mathbf{A}$, which is defined as
> >
> > $\boldsymbol{A} = \frac{\eta}{n_0^2\tau} \sum_{i=1}^{n_0} \sum_{i^{\prime} \neq i} ( \boldsymbol{z}_i\tilde{\boldsymbol{z}_i}^{\top} + \tilde{\boldsymbol{z}_i}\boldsymbol{z}_i^{\top} - \boldsymbol{z}_i \boldsymbol{z}\_{i\^{\prime}}\^{\top} - \boldsymbol{z}\_{i\^{\prime}}\boldsymbol{z}_i\^{\top})$
> >
> >
> > Recall that we denote $\mathbf{z}_i = y_i\cdot \boldsymbol{\mu} + \boldsymbol{\xi}_i$,  $\tilde{\mathbf{z}}_i =  y_i\cdot \boldsymbol{\mu} + \tilde{\boldsymbol{\xi}}_i$, where $\boldsymbol{\mu}$ is the “signal vector” while $\boldsymbol{\xi}_i, \tilde{\boldsymbol{\xi}}_i$ are independent “noise vectors”.
> >
> > We observe that, thanks to data augmentation, in $\mathbf{A}$, there are no “self-correlation” terms of noise vectors: there are no terms like $\boldsymbol{ \xi }_i \boldsymbol{ \xi }_i^\top$. Instead, there are only terms like $\boldsymbol{ \xi }_i\tilde{\boldsymbol{ \xi }}_i^\top$, or $\boldsymbol{ \xi }_i\boldsymbol{ \xi }\_{i’}^\top$, $(i’ \neq i)$. This is the key reason why we can show that the leading eigenvector of $\mathbf{A}$ is well-aligned with the “signal vector” $\boldsymbol{\mu}$ in Lemma 5.2. This shows how an ideal data augmentation can effectively enhance signal learning in SimCLR pre-training, although no labeled data is used.
> >
> > Therefore, we emphasize that it is the conclusion that “although not using any labeled data, SimCLR pre-training can help the model learn more label-related information”, rather than the specific “matrix power method”, that provides insights into real-world scenarios. Since we can no longer upload revised papers during the rebuttal period, we will add more discussions in the camera-ready.
> >
> > We also appreciate your help pointing out the related work [1]. It is a very relevant result also focusing on feature learning, and we will add and discuss it in the camera-ready.
> >
> >
> > [1] Junsoo Oh and Chulhee Yun. Provable Benefit of Cutout and CutMix for Feature Learning. NeurIPS, 2024.
> >
> > ---

---

### Meta-Review · Area_Chair_ySKR · 2024-12-21

**Metareview:**

This paper was reviewed by four experts in the field and received 6, 5, 5, 5 as the final ratings. The reviewers agreed that the paper studies the interesting problem of theoretically understanding recent contrastive learning methods, the introduced proof techniques are novel, and that the paper is well-motivated and clearly written.

The main concern raised by the reviewers is that the paper studies the very simplistic setting of binary classification tasks with two-layer convolutional neural networks (CNNs). This setting is less relevant in the current era, where more sophisticated deep learning architectures are used to employ contrastive learning methods such as SimCLR. In the rebuttal, the authors have explained that a theoretical paper necessitates specific simplification of models; they have also mentioned that SimCLR has not been well-understood theoretically even for simple two-layer CNNs and that this paper provides a first step towards the theoretical understanding of the advantages of contrastive learning. While the reviewers have appreciated the authors’ efforts in advancing the theoretical understanding in this field, they have expressed concerns about the practical relevance of simple two-layer CNNs for SSL or SimCLR pre-training in the current era. This issue has been discussed extensively between the authors and the reviewers and also between the AC and the reviewers during the post-rebuttal discussion period; the reviewers have not been convinced about the practical usefulness of the contributions.

The authors have stated in the rebuttal that their theoretical analysis with two-layer CNNs can also provide insights into how SimCLR enhances the learning of more advanced models such as vision transformers; however, further detailed analyses and deeper theoretical insights are necessary to appropriately validate this claim.

Another concern was raised stating that the data augmentation techniques introduced in the problem setting are too unrealistic. While the authors have explained that they need to consider certain idealized settings to establish theoretical guarantees, such overly simplistic assumptions may be less relevant from a practical standpoint.

We appreciate the authors' efforts in meticulously responding to each reviewer's comments. We also appreciate their efforts in conducting additional experiments to address some of the reviewers' concerns (such as the experiments with large-scale data, experiments with different SNR values, and experiments using vision transformers). However, in light of the above discussions, we conclude that the paper may not be ready for an ICLR publication in its current form. While the paper clearly has merit, the decision is not to recommend acceptance. The authors are encouraged to consider the reviewers’ comments when revising the paper for submission elsewhere.

**Additional Comments On Reviewer Discussion:**

Please see my comments above.

---

### Decision · Program_Chairs · 2025-01-22

Reject